# Regret-Optimal Q-Learning with Low Cost for Single-Agent and Federated Reinforcement Learning

**Haochen Zhang**[1,*]**, Zhong Zheng**[1,*]**, Lingzhou Xue**[1,†]
[1]Department of Statistics, The Pennsylvania State University
{hqz5340,zvz5337,lzxue}@psu.edu

## Abstract

Motivated by real-world settings where data collection and policy deployment—whether for a single agent or across multiple agents—are costly, we study the problem of on-policy single-agent reinforcement learning (RL) and federated RL (FRL) with a focus on minimizing burn-in costs (the sample sizes needed to reach near-optimal regret) and policy switching or communication costs. In parallel finite-horizon episodic Markov Decision Processes (MDPs) with $S$ states and $A$ actions, existing methods either require superlinear burn-in costs in $S$ and $A$ or fail to achieve logarithmic switching or communication costs. We propose two novel model-free RL algorithms—Q-EarlySettled-LowCost and FedQ-EarlySettled-LowCost—that are the first in the literature to simultaneously achieve: (i) the best near-optimal regret among all known model-free RL or FRL algorithms, (ii) low burn-in cost that scales linearly with $S$ and $A$, and (iii) logarithmic policy switching cost for single-agent RL or communication cost for FRL. Additionally, we establish gap-dependent theoretical guarantees for both regret and switching/communication costs, improving or matching the best-known gap-dependent bounds.

## 1 Introduction

Reinforcement Learning (RL) [78] is a subfield of machine learning focused on sequential decision-making. Often modeled as a Markov Decision Process (MDP), RL tries to obtain an optimal policy through sequential interactions with the environment. It finds applications in various fields, such as games [73, 74, 75, 82], robotics [32, 45], and autonomous driving [101]. We assume the presence of a central server and $M$ local agents in the system. Each agent interacts independently with an episodic MDP consisting of $S$ states, $A$ actions, and $H$ steps per episode.

In this paper, we focus on model-free online RL and federated RL for tabular episodic Markov Decision Processes (MDPs) with inhomogeneous transition kernels, consisting of $S$ states, $A$ actions, and $H$ steps per episode. It is known that the regret information-theoretic lower bound for any tabular MDP and any learning algorithm is $\Omega(\sqrt{H^2SAT})$, where $T$ denotes the total number of steps [37]. The model-based algorithm UCBVI [9] first reaches this lower bound up to a logarithmic factor. Model-free algorithms—commonly called Q-learning—are widely used in practice due to their simplicity of implementation and lower memory requirements [37]. Specifically, model-based methods typically require memory that scales quadratically with the number of states $S$ for storing the estimated transition kernel. Model-free methods require memory that only scales linearly with $S$ but generally face greater challenges in achieving comparable regret.

[37] proposed the first two model-free algorithms with theoretical guarantees: both attaining suboptimal regrets compared with the information-theoretic lower bound. [10] modified their algorithms

---

*Equal Contribution.
†Corresponding Author.

and further reduced the number of policy updates, also known as the switching cost, to a logarithmic dependency on $T$. Later, [111] proposed UCB-Advantage that reaches the near-optimal regret of $\tilde{O}(\sqrt{H^2SAT})$ and a logarithmic switching cost, but it comes with a large burn-in cost: the regret upper bound is valid only when $T \geq \tilde{O}(S^6A^4H^{28})$. Here, $\tilde{O}$ hides logarithmic factors. To mitigate this, [50] introduced the near-optimal Q-EarlySettled-Advantage algorithm, which significantly reduces the burn-in cost to $\tilde{O}(SAH^{10})$, scaling linearly with $S$ and $A$. However, this improvement comes at the expense of a high switching cost that scales linearly with $T$. Thus, UCB-Advantage and Q-EarlySettled-Advantage suffer notable limitations: the former requires a large burn-in cost, and the latter fails to achieve logarithmic switching cost. This raises the following open question:

*Is it possible that a model-free RL algorithm achieves the near-optimal regret $\tilde{O}(\sqrt{H^2SAT})$ with a burn-in cost that scales linearly with $S, A$ and a logarithmic switching cost simultaneously?*

In many real-world scenarios, an individual agent faces significant limitations in data collection, and agents can jointly learn an optimal policy, thereby improving the sample efficiency. This naturally motivates the framework of Federated Reinforcement Learning (FRL) that leverages parallel explorations across multiple agents coordinated by a central server. FRL enables faster learning while preserving data privacy and maintaining low communication costs, defined as the total number of scalars shared among the central server and the local agents. The regret information-theoretic lower bound for any tabular MDP and any FRL algorithm with $M$ agents naturally extends to $\Omega(\sqrt{MH^2SAT})$, where $T$ denotes the average number of steps per agent. Among existing methods, the only model-based FRL algorithm that matches this lower bound (up to logarithmic factors) is Fed-UCBVI [46]. In the following, we review model-free algorithms for which the communication cost scales logarithmically with $T$. [113] proposed the first two model-free FRL algorithms with suboptimal regrets. [114] introduced FedQ-Advantage that attains the near-optimal regret bound of $\tilde{O}(\sqrt{MH^2SAT})$ with a high burn-in cost of $\tilde{O}(MS^3A^2H^{12})$. Thus, it is natural to ask the following question for the federated setting:

*Is it possible that a model-free FRL algorithm attains the near-optimal regret $\tilde{O}(\sqrt{MH^2SAT})$ with a burn-in cost that scales linearly with $S, A$ and a logarithmic communication cost simultaneously?*

These two questions are challenging due to several non-trivial difficulties. First, the Q-EarlySettled-Advantage algorithm [50] updates its policy after each episode, incurring a switching cost that scales linearly with $T$. While this algorithm demonstrates low burn-in cost in single-agent scenarios, its effectiveness in federated learning settings remains unknown in the literature. Second, while UCB-Advantage [111] and its federated extension FedQ-Advantage [114] leverage reference-advantage decomposition to reach near-optimal regrets, neither incorporates Lower Confidence Bounds (LCB) to settle the reference function like Q-EarlySettled-Advantage. Thus, their burn-in costs exhibit a superlinear dependence on $S$ and $A$.

To simultaneously achieve logarithmic switching/communication costs while maintaining low burn-in costs, an algorithm must satisfy two requirements: (1) infrequent policy updates rather than per-episode updates, and (2) proper incorporation of LCB methods. This creates a fundamental trade-off: while delayed updates reduce switching and communication costs, their combination with LCB methods inevitably introduces additional regret and reference function settling errors. Bounding them with the reference functions introduced in [50, 111] involves controlling a weighted sum of a sequence of random variables, where neither the weights nor the random variables adapt to the data generation process. As a result, standard concentration inequalities cannot be directly applied to this type of non-martingale sum, presenting a key challenge in extending the framework to simultaneously achieve low burn-in costs and logarithmic switching/communication costs. Prior techniques, such as the empirical process [50] that accommodates non-adaptive random variables and round-wise approximation methods [105, 113, 114] that handle non-adaptive weights, are insufficient when both forms of non-adaptiveness coexist.

**Summary of Our Contributions.** We answer the two open questions affirmatively by proposing the FRL algorithm **FedQ-EarlySettled-LowCost** and its single-agent counterpart **Q-EarlySettled-LowCost** for the case when $M = 1$. Our main contributions are summarized as follows:

(i) **Algorithm Design:** We propose the first round-based algorithm for single-agent RL that achieves logarithmic switching cost, advancing beyond traditional per-episode updates. For FRL, we introduce the LCB technique for the first time to attain a low burn-in cost. While the logarithmic

switching/communication cost entails a trade-off that slightly increases regret, our use of a refined bonus term—while maintaining optimism—yields improved regret performance over Q-EarlySettled-Advantage [50] and FedQ-Advantage [114], the current state-of-the-art algorithms for provable model-free single-agent RL and FRL, respectively.

(ii) **Best Regret Performance:** In both single-agent RL and FRL scenarios, our algorithms achieve the best-known regret bounds among existing model-free approaches. In the single-agent RL setting, Q-EarlySettled-LowCost improves upon Q-EarlySettled-Advantage—the best method in the literature—by a factor of $\log(SAT)$. This is a significant advancement, as logarithmic factors in $T$ are known to be crucial for practical performance [68, 109]. For the FRL setting, compared with the existing state-of-the-art algorithm FedQ-Advantage, FedQ-EarlySettled-LowCost eliminates superlinear dependence on $S$ and $A$. It is significant for large-scale applications such as text-based games [13] and recommender systems [19]. Numerical results in Appendix B demonstrate that our algorithms consistently achieve the lowest regret.

(iii) **Simultaneous Low Burn-in Costs and Logarithmic Switching/Communication Costs:** Our algorithms achieve low burn-in costs that scale linearly with $S$ and $A$, while maintaining logarithmic switching/communication costs. In single-agent RL, Q-EarlySettled-LowCost simultaneously (1) reduces the burn-in cost to $\tilde{O}(SAH^{10})$, which linearly depends on $S$ and $A$, representing a significant improvement over the burn-in cost $\tilde{O}(S^6A^3H^{28})$ of UCB-Advantage; and (2) maintains a logarithmic switching cost that outperforms the linearly scaling cost of Q-EarlySettled-Advantage. Similarly, in the FRL setting, FedQ-EarlySettled-LowCost (1) reduces the burn-in cost to $O(MSAH^{10})$ compared with $O(MS^3A^2H^{12})$ for FedQ-Advantage; and (2) maintains a logarithmic communication cost.

In Table 1 and Table 2, we compare Q-EarlySettled-LowCost with existing model-free single-agent RL algorithms, and FedQ-EarlySettled-LowCost with other model-free FRL approaches. The results further demonstrate that our algorithms are the first to simultaneously achieve the near-optimal regret, low burn-in costs, and logarithmic switching/communication costs in both single-agent RL and FRL.

Table 1: Comparison of model-free single-agent RL algorithms.

| Algorithm (Reference) | Near-optimal regret | Logarithmic switching cost | Low burn-in cost |
|---|---|---|---|
| UCB-Hoeffding [37] | ✗ | ✗ | ✗ |
| UCB-Bernstein [37] | ✗ | ✗ | ✗ |
| UCB2-Hoeffding [10] | ✗ | ✓ | ✗ |
| UCB2-Bernstein [10] | ✗ | ✓ | ✗ |
| UCB-Advantage [111] | ✓ | ✓ | ✗ |
| Q-EarlySettled-Advantage [50] | ✓ | ✗ | ✓ |
| Q-EarlySettled-LowCost (**this work**) | ✓ | ✓ | ✓ |

Table 2: Comparison of model-free FRL algorithms.

| Algorithm (Reference) | Near-optimal regret | Logarithmic communication cost | Low burn-in cost |
|---|---|---|---|
| FedQ-Hoeffding [113] | ✗ | ✓ | ✗ |
| FedQ-Bernstein [113] | ✗ | ✓ | ✗ |
| FedQ-Advantage [114] | ✓ | ✓ | ✗ |
| FedQ-EarlySettled-LowCost (**this work**) | ✓ | ✓ | ✓ |

(iv) **Gap-Dependent Results:** We present gap-dependent analyses in both single-agent RL and FRL settings for MDPs with positive suboptimality gaps [84, 102]. For the single-agent RL setting, we establish the first gap-dependent switching cost bound for algorithms employing LCB techniques, while simultaneously achieving the best gap-dependent regret matching that of Q-EarlySettled-Advantage [115]. In the FRL setting, our algorithm not only matches the best known communication cost bound of FedQ-Hoeffding [105], but also provides improved gap-dependent regret guarantees, advancing beyond the only existing results in [105].

(v) **Technical Novelty**: To integrate the LCB technique with the round-based design for achieving simultaneous low burn-in costs and logarithmic switching/communication costs, we address the challenge of non-adaptiveness in controlling the weighted sum through the surrogate reference function used in Lemma I.3 when approximating the non-adaptive random variables. It simplifies the problem to the case that only the weights show non-adaptiveness, allowing us to apply round-wise approximations to bound the weighted sum. It facilitates the proof of the regret theoretical guarantees while yielding strictly improved regret bounds. Details can be found in Appendix C.

## 2 Background and Problem Formulation

### 2.1 Preliminaries

**Tabular Episodic Markov Decision Process (MDP).** A tabular episodic MDP is denoted as $\mathcal{M} := (\mathcal{S}, \mathcal{A}, H, \mathbb{P}, r)$, where $\mathcal{S}$ is the set of states with $|\mathcal{S}| = S$, $\mathcal{A}$ is the set of actions with $|\mathcal{A}| = A$, $H$ is the number of steps in each episode, $\mathbb{P} := \{\mathbb{P}_h\}_{h=1}^H$ is the heterogeneous transition kernel so that $\mathbb{P}_h(\cdot \mid s, a)$ characterizes the distribution over the next state given the state action pair $(s, a)$ at step $h$ and $r := \{r_h\}_{h=1}^H$ collects deterministic reward functions on $\mathcal{S} \times \mathcal{A}$ with each bounded by $[0, 1]$.

In each episode, an initial state $s_1$ is selected arbitrarily by an adversary. At each step $h \in [H] = \{1, 2, ..., H\}$, an agent observes a state $s_h \in \mathcal{S}$, picks an action $a_h \in \mathcal{A}$, receives the reward $r_h = r_h(s_h, a_h)$ and then transits to the next state $s_{h+1}$. The episode ends when an absorbing state $s_{H+1}$ is reached. For convenience, we denote $\mathbb{P}_{s,a,h} f = \mathbb{E}_{s_{h+1} \sim \mathbb{P}_h(\cdot|s,a)}(f(s_{h+1})|s_h = s, a_h = a)$, $\mathbb{1}_s f = f(s)$ and $\mathbb{V}_{s,a,h}(f) = \mathbb{P}_{s,a,h} f^2 - (\mathbb{P}_{s,a,h} f)^2$ for any function $f : \mathcal{S} \to \mathbb{R}$ and triple $(s, a, h)$.

**Policies and Value Functions.** A policy $\pi$ is a collection of $H$ functions $\left\{\pi_h : \mathcal{S} \to \Delta^{\mathcal{A}}\right\}_{h \in [H]}$, where $\Delta^{\mathcal{A}}$ is the set of probability distributions over $\mathcal{A}$. A policy is deterministic if for any $s \in \mathcal{S}$, $\pi_h(s)$ concentrates all the probability mass on an action $a \in \mathcal{A}$. In this case, we denote $\pi_h(s) = a$.

Denote state value functions $V_h^\pi : \mathcal{S} \to \mathbb{R}$ by

$$V_h^\pi(s) := \sum_{h'=h}^H \mathbb{E}_{(s_{h'}, a_{h'}) \sim (\mathbb{P}, \pi)} \left[r_{h'}(s_{h'}, a_{h'}) \mid s_h = s\right]$$

and state-action value functions $Q_h^\pi : \mathcal{S} \times \mathcal{A} \to \mathbb{R}$ by

$$Q_h^\pi(s, a) := r_h(s, a) + \sum_{h'=h+1}^H \mathbb{E}_{(s_{h'}, a_{h'}) \sim (\mathbb{P}, \pi)} \left[r_{h'}(s_{h'}, a_{h'}) \mid s_h = s, a_h = a\right].$$

For tabular episodic MDP, there exists an optimal policy $\pi^\star$ such that $V_h^\star(s) := \sup_\pi V_h^\pi(s) = V_h^{\pi^\star}(s)$ for all $(s, h) \in \mathcal{S} \times [H]$ [9]. Then for any $(s, a, h) \in \mathcal{S} \times \mathcal{A} \times [H]$, the Bellman Equation and the Bellman Optimality Equation can be expressed as:

$$\begin{cases} V_h^\pi(s) = \mathbb{E}_{a' \sim \pi_h(s)}[Q_h^\pi(s, a')] \\ Q_h^\pi(s, a) := r_h(s, a) + \mathbb{P}_{s,a,h} V_{h+1}^\pi \\ V_{H+1}^\pi(s) = 0, \forall (s, a, h) \end{cases} \text{ and } \begin{cases} V_h^\star(s) = \max_{a' \in \mathcal{A}} Q_h^\star(s, a') \\ Q_h^\star(s, a) := r_h(s, a) + \mathbb{P}_{s,a,h} V_{h+1}^\star \\ V_{H+1}^\star(s) = 0, \forall (s, a, h). \end{cases} \quad (1)$$

**Suboptimality Gap.** For any given MDP, we can formally define the suboptimality gap as follows.

**Definition 2.1.** *For any $(s, a, h)$, the suboptimality gap is defined as $\Delta_h(s, a) := V_h^\star(s) - Q_h^\star(s, a)$.*

Equation (1) implies that for any $(s, a, h)$, $\Delta_h(s, a) \geq 0$. We then define the minimum gap:

**Definition 2.2.** *We define the **minimum gap** as $\Delta_{\min} := \inf\{\Delta_h(s, a) \mid \Delta_h(s, a) > 0, \forall (s, a, h)\}$.*

We remark that if $\{\Delta_h(s, a) \mid \Delta_h(s, a) > 0, \forall (s, a, h)\} = \emptyset$, then all actions are optimal, leading to a degenerate MDP. Therefore, we assume that the set is nonempty and $\Delta_{\min} > 0$. Definitions 2.1 and 2.2 and the non-degeneration are standard in the literature on gap-dependent analysis [76, 96, 98].

**Switching Cost.** Similar to [68], the switching cost[3] is defined as follows:

**Definition 2.3.** *The switching cost for an algorithm with $U$ episodes is $N_{\text{switch}} := \sum_{k=1}^{U-1} \mathbb{I}[\pi^{u+1} \neq \pi^u]$. Here, $\pi^u$ is the implemented policy for generating the $u-$th episode.*

---

[3]Some works name it global switching cost and also analyzes the local switching cost defined as $\tilde{N}_{\text{switch}} := \sum_{u=1}^{U-1} \sum_{s,h} \mathbb{I}[\pi_h^{u+1}(s) \neq \pi_h^u(s)]$. [10, 111] proved the same cost upper bound under both definitions.

## 2.2 The Federated Reinforcement Learning (FRL) Framework

We consider an FRL setting similar to [113, 114], where a central server coordinates $M$ agents, each interacting with an independent MDP. For agent $m$, let $U_m$ be the number of episodes, $\pi^{m,u}$ the policy used in episode $u$, and $s_1^{m,u}$ the initial state. The regret over $\hat{T} = H \sum_{m=1}^M U_m$ total steps is

$$\text{Regret}(T) = \sum_{m=1}^{M} \sum_{u=1}^{U_m} \left( V_1^\star(s_1^{m,u}) - V_1^{\pi^{m,u}}(s_1^{m,u}) \right). \tag{2}$$

Here, $T := \hat{T}/M$ is the average total steps for $M$ agents. When $M = 1$, Equation (2) also defines the regret for single-agent RL, where $T$ represents the total number of steps in the learning process.

The communication cost of an FRL algorithm is defined as the number of scalars (integers or real numbers) communicated between the server and agents.

# 3 Algorithm Design

## 3.1 Algorithm Details

Now we present FedQ-EarlySettled-LowCost, our model-free FRL algorithm with $M$ agents, along with its single-agent variant (when $M = 1$), Q-EarlySettled-LowCost. FedQ-EarlySettled-LowCost runs in rounds indexed by $k \in \{1, 2, ..., K\}$, where each agent $m$ performs $n^{m,k}$ episodes in round $k$ (to be defined later). This formulation naturally accommodates a common form of system heterogeneity [52], referred to as heterogeneous exploration speed. It allows agents to explore the environment at different rates and generate varying numbers of episodes in each round. This type of heterogeneity is commonly considered in model-free FRL studies [113, 114], which is distinct from the environment heterogeneity considered in [46].

For episode $j$ in round $k$, define the trajectory collected by agent $m$ as $\{(s_h^{m,k,j}, a_h^{m,k,j}, r_h^{m,k,j})\}_{h=1}^H$. Let $n_h^{m,k}(s,a)$ denote the number of times that agent $m$ visits $(s,a)$ at step $h$ in round $k$, $n_h^k(s,a) = \sum_{m=1}^M n_h^{m,k}(s,a)$ and $N_h^k(s,a) = \sum_{k'=1}^{k-1} n_h^{k'}(s,a)$. We omit $(s,a)$ when there is no ambiguity.

Define $V_h^k, Q_h^k, V_h^{L,k}$ and $V_h^{R,k}$ as the estimated $V-$function, the estimated $Q-$function, the lower bound function and the reference function at step $h$ at the beginning of round $k$. Specifically, $Q_{H+1}^k, V_{H+1}^k, V_{H+1}^{L,k}, V_{H+1}^{R,k} = 0$. We also define the advantage function as $V_h^{A,k} = V_h^k - V_h^{R,k}$. At the beginning of round $k$, the central server maintains $N_h^k$, policy $\pi^k = \{\pi_h^k\}_{h=1}^H$, and four other quantities for any $(s,a,h)$: $\mu_h^{R,k}(s,a), \sigma_h^{R,k}(s,a), \mu_h^{A,k}(s,a)$ and $\sigma_h^{A,k}(s,a)$ (all zero-initialized when $k = 1$), which will be explained later. We then specify each component of the algorithms as follows.

**Coordinated Exploration.** At the beginning of round $k$, the server broadcasts $\pi^k$, along with $\{N_h^k(s, \pi_h^k(s)), V_h^k(s), V_h^{L,k}(s), V_h^{R,k}(s)\}_{s,h}$ to all agents. Here, $Q_h^1 = V_h^1 = V_h^{R,1} = H, V_h^{L,1} = N_h^1 = 0$ for any $(s,a,h)$ and $\pi^1$ is an arbitrary deterministic policy. Each agent $m$ will then collect $n^{m,k}$ trajectories under the policy $\pi^k$.

**Event-Triggered Termination of Exploration.** Similar to [113], in round $k$, for any agent $m$, at the end of each episode, if any $(s,a,h)$ has been visited by $c_h^k(s,a)$ times, then the exploration for all agents will be terminated. This trigger condition guarantees

$$n_h^{m,k}(s,a) \le c_h^k(s,a) := \max\left\{1, \left\lfloor \frac{N_h^k(s,a)}{MH(H+1)} \right\rfloor\right\}, \forall(s,a,h,m) \tag{3}$$

and there exists at least one tuple $(s,a,h,m)$ such that the equality holds.

**Local Aggregation.** For any visited $(s,a,h)$ with $a = \pi_h^k(s)$, agent $m$ computes the following six local sums over all next states of visits to $(s,a,h)$ and send these local sums along with $\{r_h(s, \pi_h^k(s)), n_h^{m,k}(s, \pi_h^k(s))\}_{s,h}$ to the central server at the end of round $k$.

$$\left[ v_h^{m,k}, v_{h,l}^{m,k}, \mu_{h,r}^{m,k}, \sigma_{h,r}^{m,k}, \mu_{h,a}^{m,k}, \sigma_{h,a}^{m,k} \right](s,a)$$

$$= \sum_{j=1}^{n^{m,k}} \left[ V_{h+1}^k, V_{h+1}^{L,k}, V_{h+1}^{R,k}, (V_{h+1}^{R,k})^2, V_{h+1}^{A,k}, (V_{h+1}^{A,k})^2 \right] (s_{h+1}^{m,k,j}) \cdot \mathbb{I}\left[ (s_h^{m,k,j}, a_h^{m,k,j}) = (s,a) \right]. \tag{4}$$

**Central Aggregation.** After receiving the information, for any visited $(s, a, h)$ with $a = \pi_h^k(s)$, the central server computes $n_h^k = \sum_{m=1}^M n_h^{m,k}$, $N_h^{k+1} = N_h^k + n_h^k$ and six round-wise means:

$$\left[ v_h^k, v_h^{l,k}, \mu_h^{r,k}, \sigma_h^{r,k}, \mu_h^{a,k}, \sigma_h^{a,k} \right](s,a) = \sum_{m=1}^M \left[ v_h^{m,k}, v_{h,l}^{m,k}, \mu_{h,r}^{m,k}, \sigma_{h,r}^{m,k}, \mu_{h,a}^{m,k}, \sigma_{h,a}^{m,k} \right] / n_h^k(s,a). \quad (5)$$

It also updates two global means, $\mu_h^{R,k+1}(s,a)$ and $\sigma_h^{R,k+1}(s,a)$, as

$$\left( \mu_h^{R,k+1}, \sigma_h^{R,k+1} \right)(s,a) = \left[ N_h^k \cdot \left( \mu_h^{R,k}, \sigma_h^{R,k} \right)(s,a) + n_h^k \cdot \left( \mu_h^{r,k}, \sigma_h^{r,k} \right)(s,a) \right] / N_h^{k+1}(s,a), \quad (6)$$

which is the historical mean of the reference function and the squared reference function over all next states of visits to $(s, a, h)$ in the first $k$ rounds.

Define $\eta_t = \frac{H+1}{H+t}$ and $\eta_i^t = \eta_i \prod_{j=i+1}^t (1 - \eta_j)$ for any $1 \le i \le t \in \mathbb{N}_+$, with $\eta_0^0 = 1$ and $\eta_0^t = 0$. We also define $\eta^c(n_1, n_2) = \prod_{t=n_1}^{n_2} (1 - \eta_t)$ for any $n_1 \le n_2 \in \mathbb{N}_+$ and the learning rate $\eta_\alpha = 1 - \eta^c(N_h^k + 1, N_h^{k+1})$. Here, $\eta_\alpha$ is a simplified notation depending on $(s, a, h, k)$. Then, for any visited $(s, a, h)$ with $a = \pi_h^k(s)$, the central server updates the estimated $Q-$function as follows:

$$Q_h^{k+1}(s,a) = \min \left\{ Q_h^{U,k+1}(s,a), Q_h^{R,k+1}(s,a), Q_h^k(s,a) \right\}. \quad (7)$$

Here, for each $(s, a, h)$, the Hoeffding-type $Q-$estimate $Q_h^{U,k+1}$ [37, 113] and the Reference-Advantage-type $Q-$estimate $Q_h^{R,k+1}$ [50, 111] are updated according to the following two cases:

**Case 1:** $N_h^k(s,a) < 2MH(H+1) =: i_0$. In this case, Equation (3) implies that each agent can visit $(s, a, h)$ at most once. Denote $1 \le m_1 < \ldots < m_{n_h^k} \le M$ as the agent indices with $n_h^{m,k}(s,a) = 1$. The central server first updates the two global weighted means of the advantage function $V_{h+1}^{A,k}$ and the squared advantage function $(V_{h+1}^{A,k})^2$ over all next states of visits to $(s, a, h)$ as:

$$\left( \mu_h^{A,k+1}, \sigma_h^{A,k+1} \right)(s,a) = (1 - \eta_\alpha) \left( \mu_h^{A,k}, \sigma_h^{A,k} \right)(s,a) + \sum_{t=1}^{n_h^k} \eta_{N_h^k+t}^{N_h^{k+1}} \left( \mu_{h,a}^{m_t,k}, \sigma_{h,a}^{m_t,k} \right)(s,a). \quad (8)$$

The UCB-type, LCB-type [50] and the reference-advantage-type $Q-$estimates are updated as follows:

$$Q_h^{U,k+1}(s,a) = (1 - \eta_\alpha) Q_h^{U,k}(s,a) + \eta_\alpha r_h(s,a) + \sum_{t=1}^{n_h^k} \eta_{N_h^k+t}^{N_h^{k+1}} v_h^{m_t,k}(s,a) + B_h^{k+1}(s,a). \quad (9)$$

$$Q_h^{L,k+1}(s,a) = (1 - \eta_\alpha) Q_h^{L,k}(s,a) + \eta_\alpha r_h(s,a) + \sum_{t=1}^{n_h^k} \eta_{N_h^k+t}^{N_h^{k+1}} v_{h,l}^{m_t,k}(s,a) - B_h^{k+1}(s,a). \quad (10)$$

$$Q_h^{R,k+1}(s,a) = (1 - \eta_\alpha) Q_h^{R,k} + \eta_\alpha \left( r_h + \mu_h^{R,k+1} \right) + \sum_{t=1}^{n_h^k} \eta_{N_h^k+t}^{N_h^{k+1}} \left( v_h^{m_t,k} - \mu_{h,r}^{m_t,k} \right) + B_h^{R,k+1}(s,a). \quad (11)$$

**Case 2:** $N_h^k(s,a) \ge i_0$. In this case, the server updates the two global weighted means as

$$\left( \mu_h^{A,k+1}, \sigma_h^{A,k+1} \right)(s,a) = (1 - \eta_\alpha) \left( \mu_h^{A,k}, \sigma_h^{A,k} \right)(s,a) + \eta_\alpha \left( \mu_h^{a,k}, \sigma_h^{a,k} \right)(s,a). \quad (12)$$

Now the three $Q-$estimates are updated as follows:

$$Q_h^{U,k+1}(s,a) = (1 - \eta_\alpha) Q_h^{U,k}(s,a) + \eta_\alpha \left( r_h(s,a) + v_h^k(s,a) \right) + B_h^{k+1}(s,a). \quad (13)$$

$$Q_h^{L,k+1}(s,a) = (1 - \eta_\alpha) Q_h^{L,k}(s,a) + \eta_\alpha \left( r_h(s,a) + v_h^{l,k}(s,a) \right) - B_h^{k+1}(s,a). \quad (14)$$

$$Q_h^{R,k+1}(s,a) = (1 - \eta_\alpha) Q_h^{R,k}(s,a) + \eta_\alpha \left( r_h + \mu_h^{R,k+1} + v_h^k - \mu_h^{r,k} \right)(s,a) + B_h^{R,k+1}(s,a). \quad (15)$$

In both cases, the cumulative bonuses are given as:

$$B_h^{k+1}(s,a) = \sum_{t=N_h^k+1}^{N_h^{k+1}} \eta_t^{N_h^{k+1}} b_t, \quad B_h^{R,k+1}(s,a) = \sum_{t=N_h^k+1}^{N_h^{k+1}} \eta_t^{N_h^{k+1}} b_{h,t}^R(s,a), \quad (16)$$

where $b_t = c_b\sqrt{H^3\iota/t}$ for a sufficiently large constant $c_b$ and a positive constant $\iota$ determined later, and $b_{h,t}^{\mathrm{R}}(s,a)$ is computed as follows. For a sufficiently large constant $c_b^{\mathrm{R}}$, the central server calculates

$$\beta_h^{\mathrm{R},k+1}(s,a) = c_b^{\mathrm{R}}\sqrt{\frac{\iota}{N_h^{k+1}}}\left(\sqrt{\sigma_h^{\mathrm{R},k+1} - \left(\mu_h^{\mathrm{R},k+1}\right)^2} + \sqrt{H\left(\sigma_h^{\mathrm{A},k+1} - \left(\mu_h^{\mathrm{A},k+1}\right)^2\right)}\right).$$

Then for a sufficiently large constant $c_b^{\mathrm{R},2} > 0$ and $t \in (N_h^k, N_h^{k+1})$, let $b_{h,t}^{\mathrm{R}} = \beta_h^{\mathrm{R},k} + c_b^{\mathrm{R},2}H^2\iota/t$ and

$$b_{h,N_h^{k+1}}^{\mathrm{R}} = \left(1 - 1/\eta_{N_h^{k+1}}\right)\beta_h^{\mathrm{R},k} + \beta_h^{\mathrm{R},k+1}/\eta_{N_h^{k+1}} + c_b^{\mathrm{R},2}H^2\iota/N_h^{k+1}.$$

After updating the estimated $Q-$function, the central server proceeds to update $V_h^{k+1}(s)$, $V_h^{\mathrm{L},k+1}(s)$, and $\pi_h^{k+1}(s)$ for each $(s,h) \in \mathcal{S} \times [H]$ as follows:

$$V_h^{k+1}(s) = \max_{a'\in\mathcal{A}} Q_h^{k+1}(s,a'), \ V_h^{\mathrm{L},k+1}(s) = \max\left\{\max_{a'\in\mathcal{A}} Q_h^{\mathrm{L},k+1}(s,a'), V_h^{\mathrm{L},k}(s)\right\}, \tag{17}$$

$$\pi_h^{k+1}(s) = \arg\max_{a'\in\mathcal{A}} Q_h^{k+1}(s,a'). \tag{18}$$

Finally, for any state-step pair $(s,h)$, the central server updates the reference function as $V_h^{\mathrm{R},k+1}(s) = V_h^{k+1}(s)$ if either: (1) $V_h^{k+1}(s) - V_h^{\mathrm{L},k+1}(s) > \beta$, or (2) it is the first round where $V_h^{k+1}(s) - V_h^{\mathrm{L},k+1}(s) \le \beta$ for predefined $\beta \in (0, H]$. Otherwise, the server settles the reference function by $V_h^{\mathrm{R},k+1}(s) = V_h^{\mathrm{R},k}(s)$. In this case, the settlement is triggered after the condition $V_h^{k+1}(s) - V_h^{\mathrm{L},k+1}(s) \le \beta$ first holds for some round $k$, as guaranteed by the monotonically non-increasing property of $V_h^{k+1}(s) - V_h^{\mathrm{L},k+1}(s)$ established in Equation (7) and Equation (17). The algorithm then proceeds to round $k + 1$. Algorithm 1 and Algorithm 2 formally present the algorithms. For reader's convenience, we provide graphical illustrations and two notation tables in Appendix D.

---

**Algorithm 1** FedQ-EarlySettled-LowCost (Central Server)

1: **Input:** $T_0 \in \mathbb{N}_+$.
2: **Initialize** $k = 1, Q_h^{\mathrm{U},1}(s,a) = Q_h^{\mathrm{R},1}(s,a) = Q_h^1(s,a) = V_h^1(s) = V_h^{\mathrm{R},1}(s) = H, Q_h^{\mathrm{L},1}(s,a) = V_h^{\mathrm{L},1}(s) = N_h^1(s,a) = 0, u_h^{\mathrm{R},1}(s) = \text{True}, \forall(s,a,h) \in \mathcal{S} \times \mathcal{A} \times [H]$ and an arbitrary policy $\pi_1$.

3: **while** $\sum_{h=1}^H \sum_{s,a} N_h^k(s,a) < T_0$ **do**
4:      Broadcast $\pi^k, \{N_h^k(s, \pi_h^k(s))\}_{s,h}, \{V_h^k(s)\}_{s,h}, \{V_h^{\mathrm{L},k}(s)\}_{s,h}$ and $\{V_h^{\mathrm{R},k}(s)\}_{s,h}$ to all agents.
5:      Wait until receiving a termination signal and send the signal to all agents.
6:      Receive the information from clients and compute round-wise means in Equation (5).
7:      **for** any $(s,a,h) \in \mathcal{S} \times \mathcal{A} \times [H]$ **do**
8:          **if** $n_h^k(s,a) = 0$, **then** $Q_h^{k+1}(s,a) \leftarrow Q_h^k(s,a)$
         **else** Update $Q_h^{k+1}(s,a)$ via Equation (7)
9:      **end for**
10:     **for** any $(s,h) \in \mathcal{S} \times [H]$ **do**
11:        Update $V_h^{k+1}(s), V_h^{\mathrm{L},k+1}(s)$ and $\pi_h^{k+1}(s)$ via Equation (17) and Equation (18).
12:        **if** $V_h^{k+1}(s) - V_h^{\mathrm{L},k+1}(s) > \beta$, **then** $V_h^{\mathrm{R},k+1}(s) = V_h^{k+1}(s)$.
         **else if** $u_h^{\mathrm{R},k}(s) = \text{True}$, **then** $V_h^{\mathrm{R},k+1}(s) = V_h^{k+1}(s), u_h^{\mathrm{R},k+1}(s) = \text{False}$.
         **end if**
13:     **end for**
14:     $k \leftarrow k + 1$.
15: **end while**

---

### 3.2 Intuition behind the Algorithm Design

**UCB and Reference-Advantage Decomposition with Refined Bonus.** Similar to [50, 114], we adopt two techniques—upper confidence bound (UCB) exploration with the bonuses in the estimated $Q-$function and reference-advantage decomposition—to attain the near-optimal regret bound. To further improve regret performance, we refine the bonus term $B_h^{\mathrm{R},k}$ used to update the estimated $Q-$function by removing its dependence on $(N_h^k)^{3/4}$ [50, 114]. This refinement enables

---

**Algorithm 2** FedQ-EarlySettled-LowCost (Agent $m$ in Round $k$)

---

1: **Initialize** $n_h^m = v_h^m = v_{h,l}^m = \mu_{h,\mathrm{r}}^m = \sigma_{h,\mathrm{r}}^m = \mu_{h,\mathrm{a}}^m = \sigma_{h,\mathrm{a}}^m = 0, \forall (s,a,h) \in \mathcal{S} \times \mathcal{A} \times [H]$.

2: Receive $\pi^k$, $\{N_h^k(s, \pi_h^k(s))\}_{s,h}$, $\{V_h^k(s)\}_{s,h}$, $\{V_h^{\mathrm{L},k}(s)\}_{s,h}$ and $\{V_h^{\mathrm{R},k}(s)\}_{s,h}$.

3: **while** no termination signal from the central server **do**

4:     **while** $n_h^m(s,a) < \max\left\{1, \left\lfloor \frac{N_h^k(s,a)}{MH(H+1)} \right\rfloor\right\}$, $\forall (s,a,h) \in \mathcal{S} \times \mathcal{A} \times [H]$ **do**

5:         Collect a new trajectory $\{(s_h, a_h, r_h)\}_{h=1}^H$ with $a_h = \pi_h^k(s_h)$.

6:         For any $h \in [H]$, $n_h^m(s_h, a_h) \stackrel{+}{=} 1$ and $(v_h^m, v_{h,l}^m, \mu_{h,\mathrm{r}}^m, \sigma_{h,\mathrm{r}}^m, \mu_{h,\mathrm{a}}^m, \sigma_{h,\mathrm{a}}^m)(s_h, a_h) \stackrel{+}{=}$
        $(V_{h+1}^k, V_{h+1}^{\mathrm{L},k}, V_{h+1}^{\mathrm{R},k}, (V_{h+1}^{\mathrm{R},k})^2, V_{h+1}^{\mathrm{A},k}, (V_{h+1}^{\mathrm{A},k})^2)(s_{h+1})$

7:     **end while**

8:     Send a termination signal to the central server.

9: **end while**

10: For any $(s,h) \in \mathcal{S} \times [H]$ with $a = \pi_h^k(s)$,
    $(n_h^{m,k}, v_h^{m,k}, v_{h,l}^{m,k}, \mu_{h,\mathrm{r}}^{m,k}, \sigma_{h,\mathrm{r}}^{m,k}, \mu_{h,\mathrm{a}}^{m,k}, \sigma_{h,\mathrm{a}}^{m,k})(s,a) \leftarrow (n_h^m, v_h^m, v_{h,l}^m, \mu_{h,\mathrm{r}}^m, \sigma_{h,\mathrm{r}}^m, \mu_{h,\mathrm{a}}^m, \sigma_{h,\mathrm{a}}^m)(s,a)$.

11: For any $(s,h) \in \mathcal{S} \times [H]$, send $\{(r_h, n_h^{m,k}, v_h^{m,k}, v_{h,l}^{m,k}, \mu_{h,\mathrm{r}}^{m,k}, \sigma_{h,\mathrm{r}}^{m,k}, \mu_{h,\mathrm{a}}^{m,k}, \sigma_{h,\mathrm{a}}^{m,k})(s, \pi_h^k(s))\}$.

---

our algorithms to outperform both Q-EarlySettled-Advantage in the single-agent RL setting and FedQ-Advantage in the FRL setting.

**LCB for Early Settlement of the Reference Function.** Compared with UCB-Advantage and FedQ-Advantage, our algorithms incorporate a Lower Confidence Bound (LCB)-type estimate $Q_h^{\mathrm{L},k}$. $V_h^{\mathrm{L},k}$ derived accordingly serves as a lower bound of $V_h^\star$, while $V_h^k$ is an upper bound for $V_h^\star$ since $Q_h^k \geq Q_h^\star$ by the UCB-design. To obtain an accurate reference function $V_h^{\mathrm{R}}$, we aim to settle the reference function $V_h^{\mathrm{R},k}$ by $V_h^k$ when $V_h^k - V_h^\star \leq \beta$ for the first time. Both UCB-Advantage and FedQ-Advantage settle the reference function at a given $(s,h)$ after it has been visited sufficiently often—when the number of visits reaches a threshold $N_0(\beta)$. This is a rather conservative condition, resulting in a large burn-in cost. In contrast, the LCB technique guarantees that $V_h^\star \in [V_h^{\mathrm{L},k}, V_h^k]$, enabling an early settlement when $V_h^k - V_h^{\mathrm{L},k} \leq \beta$, which consequently achieves a low burn-in cost.

**Event-Triggered Termination and Infrequent Policy Updates.** Our algorithms switch policies infrequently, as estimated $Q-$function and policies are updated only after each round ends due to condition (3). This design ensures that visits to each $(s,a,h)$ grow at a controlled exponential rate across rounds, enabling logarithmic bounds on switching/communication costs.

## 4 Theoretical Guarantees

When $M = 1$, the FedQ-EarlySettled-LowCost algorithm reduces to its single-agent variant, Q-EarlySettled-LowCost, by eliminating the central server and the agent-server communication process. In this section, we present the theoretical performance of our algorithms in both single-agent RL and FRL settings. We first set the constant $\iota = \log(28SAT_1/p)$, where $p \in (0,1)$ is the failure rate and $T_1$ is a known upper bound of the total steps $\hat{T}$ as defined in (b) of Lemma F.1.

### 4.1 Worst-Case Guarantees of Q-EarlySettled-LowCost

We now present the worst-case results for Q-EarlySettled-LowCost. It achieves the best regret among all model-free single-agent RL algorithms with a low burn-in cost and a logarithmic switching cost.

**Theorem 4.1.** *For any $p \in (0,1)$, let $\iota_0 = \log(SAT/p)$. Then for Q-EarlySettled-LowCost (Algorithms 1 and 2 with $M = 1$ and $\beta \in (0, H]$), with probability at least $1 - p$, we have*

$$\mathrm{Regret}(T) \leq O\left((1+\beta)\sqrt{H^2 SAT\iota_0^2} + H^6 SA\iota_0^2/\beta\right).$$

Setting $\beta = \Theta(1)$, when $T > \tilde{O}(SAH^{10})$, the regret bound matches the lower bound $O(\sqrt{H^2 SAT})$ up to logarithmic factors. Next, we compare our algorithm's performance with two near-optimal algorithms: UCB-Advantage [111] and Q-EarlySettled-Advantage [50]. UCB-Advantage has a

regret of $\tilde{O}(\sqrt{H^2SAT} + H^8S^2A^{3/2}T^{1/4})$ and a burn-in cost of $\tilde{O}(S^6A^3H^{28})$, while our algorithm achieves a lower regret with only linear dependence on $S, A$ and a better dependence on $H$, and a much smaller burn-in cost with only linear dependence on $S, A$. Compared with Q-EarlySettled-Advantage, our algorithm further improves the regret bound by a factor of $\log(SAT/p)$ and shows better regret in the numerical experiments in Appendix B.1 due to the refinement of the cumulative bonus $B_h^{\text{R},k+1}$ in Equation (16), and the use of the surrogate reference function in the proof.

**Theorem 4.2.** *Let $\tilde{C} = H^2(H+1)SA$. For Q-EarlySettled-LowCost (Algorithms 1 and 2 with $M = 1$ and $\beta \in (0, H]$), the switching cost is bounded by $\max\{2\tilde{C} + 4\tilde{C}\log(T/\tilde{C}), 3\tilde{C}\}$.*

When $T > e^{\frac{1}{4}}\tilde{C}$, our algorithm achieves a logarithmic switching cost of $O(H^3SA\log(T/(HSA)))$.

### 4.2 Worst-Case Guarantees of FedQ-EarlySettled-LowCost

We now discuss the worst-case results for FedQ-EarlySettled-LowCost. It achieves the best regret among all model-free FRL algorithms with a low burn-in cost and a logarithmic communication cost.

**Theorem 4.3.** *For any $p \in (0, 1)$, let $\iota_1 = \log(MSAT/p)$. Then for FedQ-EarlySettled-LowCost (Algorithms 1 and 2 with $\beta \in (0, H]$), with probability at least $1 - p$, we have*

$$\text{Regret}(T) \le O\left((1+\beta)\sqrt{MH^2SAT\iota_1^2} + \frac{H^6SA\iota_1^2}{\beta} + MH^5SA\iota_1^2\right).$$

Appendix I provides a unified proof for Theorem 4.1 and Theorem 4.3. Setting $\beta = \Theta(1)$, when $T > \tilde{O}(MSAH^{10})$, the result becomes $\tilde{O}(\sqrt{MH^2SAT})$, matching the lower bound with a total of $MT$ steps. Compared with FedQ-Advantage [114] with a near-optimal regret bound $\tilde{O}(\sqrt{MH^2SAT} + M^{\frac{1}{4}}H^{\frac{11}{4}}SAT^{\frac{1}{4}} + MH^7S^2A^{\frac{3}{2}})$, our method achieves lower regret with milder dependence on $H, S, A$. Furthermore, FedQ-Advantage requires $\tilde{O}(MS^3A^2H^{12})$ samples to reach near-optimality, while our method only needs $\tilde{O}(MSAH^{10})$, with a burn-in cost scaling linearly in $S, A$. Numerical experiments in Appendix B.2 also demonstrate that FedQ-EarlySettled-LowCost achieves the best regret performance among all model-free FRL algorithms.

Proving the worst-case regret bounds in Theorems 4.1 and 4.3 is challenging due to the technical difficulty of double non-adaptiveness, and we overcome it by the novel use of a technical tool, the surrogate reference function. Please refer to Appendix C for more details. Next, we discuss the worst-case communication cost results of FedQ-EarlySettled-Advantage.

**Theorem 4.4.** *For FedQ-EarlySettled-LowCost (Algorithm 1 and Algorithm 2 with $\beta \in (0, H]$), the number of rounds $K$ is bounded by $\max\{2M\tilde{C} + 4M\tilde{C}\log(T/\tilde{C}), 3M\tilde{C}\}$.*

Appendix J presents a unified proof for Theorem 4.2 and Theorem 4.4. When $T > e^{\frac{1}{4}}\tilde{C}$, we have $K \le O(MH^3SA\log(T))$. As each round incurs $O(MHS)$ communication cost, the total cost is $O(M^2H^4S^2A\log(T))$, growing logarithmically with $T$. If agent waiting is permitted, synchronization in each round can be delayed until all agents satisfy the trigger condition in Equation (3), similar to the setting in [114] when the optimal forced synchronization is disabled. In this case, the number of rounds $K$ can be bounded by $\max\{2\tilde{C} + 4\tilde{C}\log(T/\tilde{C}), 3M\tilde{C}\}$. which is independent of $M$.

### 4.3 Gap-Dependent Guarantees

This section provides gap-dependent results under both single-agent and federated settings. We define the maximal conditional variance $\mathbb{Q}^\star := \max_{s,a,h}\{\mathbb{V}_{s,a,h}(V_{h+1}^\star)\} \in [0, H^2]$ [102]. Theorem 4.5 establishes the best-known gap-dependent regret for model-free RL, matching that of Q-EarlySettled-Advantage in [115], while maintaining a logarithmic switching cost.

**Theorem 4.5.** *For Q-EarlySettled-LowCost (Algorithms 1 and 2 with $M = 1$ and $\beta \in (0, H]$),*

$$\mathbb{E}\left(\text{Regret}(T)\right) \le O\left(\frac{(\mathbb{Q}^\star + \beta^2H)H^3SA\log(SAT)}{\Delta_{\min}} + \frac{H^7SA\log^2(SAT)}{\beta}\right).$$

Next, we present the gap-dependent switching cost results under the same assumptions as [105]: full synchronization, random initialization, and G-MDPs. We first review the initial two assumptions:

(I) Full synchronization. Similar to [113], we assume that there is no latency during communications, and the agents and server are fully synchronized [60]. This means $n^{m,k} = n^k$ for each agent $m$.

(II) Random initializations. We assume that the initial states $\{s_1^{k,j,m}\}_{k,j,m}$ are randomly generated with some distribution on $\mathcal{S}$, and the generation is not affected by any result in the learning process.

Assumption (I) implies that all agents have the same exploration speed. We now introduce G-MDPs.

**Definition 4.6.** *A G-MDP satisfies the following two conditions:*

*(a) The stationary visiting probabilities under optimal policies are unique: if both $\pi^{*,1}$ and $\pi^{*,2}$ are optimal policies, then we have $\mathbb{P}\left(s_h = s | \pi^{*,1}\right) = \mathbb{P}\left(s_h = s | \pi^{*,2}\right) =: \mathbb{P}_{s,h}^{\star}$.*

*(b) Let $\mathcal{A}_h^{\star}(s) = \{a \mid a = \arg\max_{a'} Q_h^{\star}(s, a')\}$. For any $(s, h) \in \mathcal{S} \times [H]$, if $\mathbb{P}_{s,h}^{\star} > 0$, then $|\mathcal{A}_h^{\star}(s)| = 1$, which means that the optimal action is unique.*

G-MDPs represent MDPs with generally unique optimal policies. Compared to requiring a unique optimal policy, G-MDPs allow the optimal actions to vary outside the support under optimal policies, i.e., the state-step pairs with $\mathbb{P}_{s,h}^{\star} = 0$.

For any G-MDP, we define $C_{st} = \min\{\mathbb{P}_{s,h}^{\star} \mid s \in \mathcal{S}, h \in [H], \mathbb{P}_{s,h}^{\star} > 0\}$, which reflects the minimum visiting probability over the support of the optimal policy.

With these assumptions, we now present a gap-dependent switching cost bound for the Q-EarlySettled-LowCost algorithm. This result fills a notable gap by providing the first such bound for LCB-based algorithms, and it matches the best-known gap-dependent switching cost guarantee of the single-agent FedQ-Hoeffding algorithm [105], which, however, suffers from a higher and suboptimal regret.

**Theorem 4.7.** *For any $p \in (0,1)$, let $\iota_0 = \log(\frac{SAT}{p})$. Then for Q-EarlySettled-LowCost (Algorithms 1 and 2 with $M = 1$ and $\beta \in (0, H]$), under the random initialization assumption and a G-MDP, with probability at least $1 - p$, the switching cost is bounded by*

$$O\left(H^3 SA \log\left(\frac{H^4 SA \iota_0}{\beta \Delta_{\min}^2}\right) + H^3 S \log\left(\frac{1}{C_{st}}\right) + H^2 \log\left(\frac{T}{HSA}\right)\right).$$

Theorem 4.8 and Theorem 4.9 present gap-dependent results for FedQ-EarlySettled-LowCost.

**Theorem 4.8.** *For FedQ-EarlySettled-LowCost (Algorithms 1 and 2 with $\beta \in (0, H]$), let $\iota_2 = \log(MSAT)$, then we have*

$$\mathbb{E}\left(\text{Regret}(T)\right) \leq O\left(\frac{(\mathbb{Q}^{\star} + \beta^2 H) H^3 SA \iota_2}{\Delta_{\min}} + \frac{H^7 SA \iota_2^2}{\beta} + MH^6 SA \iota_2^2\right).$$

Appendix K gives a unified proof for Theorem 4.5 and Theorem 4.8. Compared with the only federated gap-dependent regret bound $O(H^6 SA \iota_1/\Delta_{\min} + MH^5 SA\sqrt{\iota})$ established for FedQ-Hoeffding in [105], Theorem 4.8 improves the dependence on $\Delta_{\min}$ by a factor of $H$ for the worst scenario, where $\mathbb{Q}^{\star} = \Theta(H^2)$. Furthermore, in the best scenario when the MDP is deterministic and $\mathbb{Q}^{\star} = 0$, our bound scales as $\tilde{O}(\Delta_{\min}^{-\frac{1}{3}})$ for specific $\beta$, improving upon the linear dependency.

**Theorem 4.9.** *For any $p \in (0,1)$, let $\iota_1 = \log(\frac{MSAT}{p})$. Then for FedQ-EarlySettled-LowCost (Algorithms 1 and 2 with $\beta \in (0, H]$), under a G-MDP and the assumptions of full synchronization and random initialization, with probability at least $1 - p$, the number of rounds $K$ is bounded by:*

$$O\left(MH^3 SA \log\left(MH\iota_1\right) + H^3 SA \log\left(\frac{H^4 SA}{\beta \Delta_{\min}^2}\right) + H^3 S \log\left(\frac{1}{C_{st}}\right) + H^2 \log\left(\frac{T}{HSA}\right)\right).$$

Appendix L gives a unified proof for Theorem 4.7 and Theorem 4.9. This result matches the only gap-dependent upper bound on the number of communication rounds, established for FedQ-Hoeffding [105], while our algorithm simultaneously achieves a better and near-optimal regret.

## 5 Conclusion

We propose two novel model-free algorithms, Q-EarlySettled-LowCost and FedQ-EarlySettled-LowCost, that simultaneously achieves the near-optimal regret, a low burn-in cost that scales linearly with $S$ and $A$, and a logarithmic switching/communication cost. Technically, we combine LCB and UCB with reference-advantage decomposition for more efficient reference function learning.

## Acknowledgment

The work of H. Zhang, Z. Zheng, and L. Xue was supported by the U.S. National Science Foundation under the grants DMS-1953189 and CCF-2007823 and by the U.S. National Institutes of Health under the grant 1R01GM152812.

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

**Organization of the appendix.** Appendix A reviews related works. Appendix B presents the results of our numerical experiments, demonstrating the best regret performance and showing the $\log T-$type switching/communication cost. Appendix C analyzes our technical novelty that leads to the regret improvement. Appendix D provides graphical illustrations and two notation tables for our algorithms. Appendix E and Appendix F include some useful theorems and lemmas. Appendix G proves some probability events and Appendix H explores the properties of estimated value functions. Appendix I contains the proof of the worst-case regret bounds (Theorem 4.1 and Theorem 4.3). Appendix J contains the proof of the worst-case switching/communication cost bounds (Theorem 4.2 and Theorem 4.4). Appendix K provides the proof of the gap-dependent regret bounds (Theorem 4.5 and Theorem 4.8). Appendix L presents the proof of the gap-dependent switching/communication cost bounds (Theorem 4.7 and Theorem 4.9).

# A  Related Work

**On-Policy RL for Finite-Horizon Tabular MDPs with Worst-Case Regret.** There are mainly two types of algorithms for reinforcement learning: model-based and model-free learning. Model-based algorithms learn a model from past experience and make decisions based on this model, while model-free algorithms only maintain a group of value functions and take the induced optimal actions. Due to these differences, model-free algorithms are usually more space-efficient and time-efficient compared with model-based algorithms. However, model-based algorithms may achieve better learning performance by leveraging the learned model.

Next, we discuss the literature on model-based and model-free algorithms for finite-horizon tabular MDPs with worst-case regret. [1, 3, 7, 9, 20, 42, 102, 107, 108, 116] focus on model-based algorithms. Notably, [107] provide an algorithm that achieves a regret of $\tilde{O}(\min\{\sqrt{SAH^2T}, T\})$, which matches the information-theoretic lower bound. [37, 50, 61, 98, 111] focus on model-free algorithms. Three of them [50, 61, 111] achieve the near-optimal regret of $\tilde{O}(\sqrt{SAH^2T})$.

**Suboptimality Gap.** When there is a strictly positive suboptimality gap, it is possible to achieve logarithmic regret bounds. In RL, earlier work obtains asymptotic logarithmic regret bounds [8, 79]. Recently, non-asymptotic logarithmic regret bounds are obtained [34, 36, 66, 76]. Specifically, [36] develops a model-based algorithm, and their bound depends on the policy gap instead of the action gap studied in this paper. [66] derives problem-specific logarithmic type lower bounds for both structured and unstructured MDPs. [76] extends the model-based algorithm proposed by [102] and obtains logarithmic regret bounds. More recently, [14] further improves model-based gap-dependent results. Logarithmic regret bounds are also established in the linear function approximation setting [34], and [65] provides gap-dependent guarantees for offline RL with linear function approximation.

Specifically, for model-free algorithms, [98] shows that the optimistic $Q$-learning algorithm in [37] enjoys a logarithmic regret $O(\frac{H^6SAT}{\Delta_{\min}})$, which is subsequently refined by [95]. In their work, [95] introduces the Adaptive Multi-step Bootstrap (AMB) algorithm. [115] further improves the logarithmic regret bound by leveraging the analysis of the UCB-Advantage algorithm [111] and the Q-EarlySettled-Advantage algorithm [50]. [106] provides the first fine-grained, gap-dependent regret upper bound for a UCB-based algorithm, specifically UCB-Hoeffding. In the federated setting, [105] further establishes the first gap-dependent bounds for both regret and communication cost.

Several other studies also investigate gap-dependent sample complexity bounds [4, 41, 59, 80, 81, 83, 85, 91].

**Variance Reduction in RL.** The reference-advantage decomposition used in [50] and [111] is a technique of variance reduction that is originally proposed for finite-sum stochastic optimization [31, 40, 64]. Later, model-free RL algorithms also use variance reduction to improve sample efficiency. For example, it is used in learning with generative models [71, 72, 88], policy evaluation [23, 43, 87, 96], offline RL [70, 100], and $Q$-learning [50, 51, 97, 111].

**RL with Low Switching Costs and Batched RL.** Research in RL with low switching costs aims to minimize the number of policy switches while maintaining comparable regret bounds to fully adaptive counterparts, and it applies to federated RL. In batched RL [29, 67], the agent sets the number of batches and the length of each batch upfront, implementing an unchanged policy in a batch and aiming for fewer batches and lower regret. [10] first introduces the problem of RL with low switching cost and proposes a $Q$-learning algorithm with lazy updates, achieving $\tilde{O}(H^3SA\log T)$ switching

cost. This work is advanced by [111], which improves the regret upper bound and the switching cost simultaneously. Additionally, [90] studies RL under the adaptivity constraint. Recently, [68] proposes a model-based algorithm with $\tilde{O}(\log \log T)$ switching cost. [110] proposes a batched RL algorithm that is well-suited for the federated setting.

**Multi-Agent RL (MARL) with Event-Triggered Communications.** We review a few recent works on on-policy MARL with linear function approximations. [24] introduces Coop-LSVI for cooperative MARL. [62] proposes an asynchronous version of LSVI-UCB that originates from [38], matching the same regret bound with improved communication complexity compared with [24]. [35] develops two algorithms that incorporate randomized exploration, achieving the same regret and communication complexity as [62]. [24, 35, 62] employ event-triggered communication conditions based on determinants of certain quantities. Different from our federated algorithm, during the synchronization in [24] and [62], local agents share original rewards or trajectories with the server. On the other hand, [35] reduces communication cost by sharing compressed statistics in the non-tabular setting with linear function approximation.

**Federated and Distributed RL.** Existing literature on federated and distributed RL algorithms highlights various aspects. For value-based algorithms, [33], [92], and [113] focus on linear speedup. [2] proposes a parallel RL algorithm with low communication cost. [92] and [93] discuss the improved covering power of heterogeneity. [16] and [94] work on robustness. Particularly, [16] proposes algorithms in both offline and online settings, obtaining near-optimal sample complexities and achieving superior robustness guarantees. In addition, several works investigate value-based algorithms such as $Q$-learning in different settings, including [5, 12, 27, 39, 44, 92, 93, 99, 104, 112]. The convergence of decentralized temporal difference algorithms is analyzed by [18, 21, 22, 53, 77, 86, 89, 103].

Some other works focus on policy gradient-based algorithms. Communication-efficient policy gradient algorithms are studied by [15] and [26]. [48] further reduces the communication complexity and also demonstrates a linear speedup in the synchronous setting. Optimal sample complexity for global convergence in federated RL, even in the presence of adversaries, is studied in [28]. [47] proposes an algorithm to address the challenge of lagged policies in asynchronous settings.

The convergence of distributed actor-critic algorithms is analyzed by [17, 69]. Federated actor-learner architectures are explored by [6, 25, 63]. Distributed inverse reinforcement learning is examined by [11, 30, 54, 55, 56, 57, 58].

# B    Numerical Experiments

In this section, we present experiments conducted in a synthetic environment to demonstrate the following two conclusions:

- When $M = 1$, Q-EarlySettled-LowCost achieves better regret compared with all other single-agent model-free algorithms: UCB-Hoeffding and UCB-Bernstein [37], UCB2-Hoeffding and UCB2B [10], UCB-Advantage [111] and Q-EarlySettled-Advantage [50], while maintaining logarithmic switching cost.

- FedQ-EarlySettled-LowCost achieves the best regret performance compared with other federated model-free algorithms, including FedQ-Hoeffding and FedQ-Bernstein[113] and FedQ-Advantage [114], while also maintaining logarithmic communication cost.

To evaluate the proposed algorithms, we simulate a synthetic tabular episodic Markov Decision Process. Specifically, we consider two cases with $(H, S, A) = (5, 3, 2)$ and $(7, 10, 5)$. The reward $r_h(s, a)$ for each $(s, a, h)$ is generated independently and uniformly at random from $[0, 1]$. $\mathbb{P}_h(\cdot \mid s, a)$ is generated on the $S$-dimensional simplex independently and uniformly at random for $(s, a, h)$. Then we will discuss the experiment results for each conclusion separately.

## B.1    Comparison of Single-Agent RL Algorithms

Under the given MDP, we set $M = 1$ and generate $3 * 10^5$ episodes for $(H, S, A) = (5, 3, 2)$ and $2 * 10^6$ episodes for $(H, S, A) = (7, 10, 5)$. For each episode, we randomly choose the initial state

uniformly from the $S$ states[4]. For the other six single-agent algorithms, we use their hyperparameter settings based on the publicly available code[5] in [115]. For FedQ-EarlySettled-LowCost algorithm, we similarly set $\iota = 1$, the hyper-parameter $c_b = \sqrt{2}$ in the bonus $b_t$, $c_b^{\text{R}} = 2$ in the cumulative bonus $\beta_h^{\text{R},k}$, $c_b^{\text{R},2} = 1$ in the bonus $b_{h,t}^{\text{R}}$ and $\beta = 0.05$.

To show error bars, we collect 10 sample paths for all algorithms under the same MDP environment and show the relationship between $\text{Regret}(T)/\log(T/H + 1)$ and the total number of episodes for each agent $T/H$ in Figure 1. For both panels, the solid line represents the median of the 10 sample paths, while the shaded area shows the 10th and 90th percentiles.

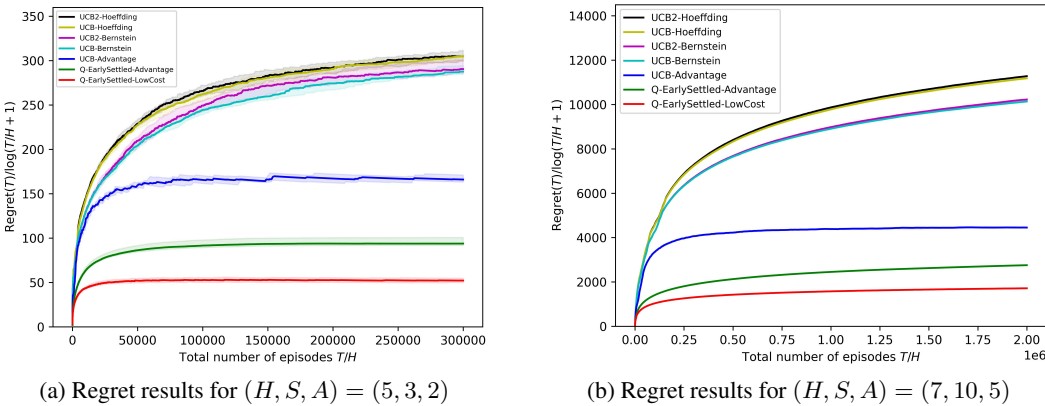

(a) Regret results for $(H, S, A) = (5, 3, 2)$      (b) Regret results for $(H, S, A) = (7, 10, 5)$

Figure 1: Numerical comparison of regrets for single-agent model-free algorithms

From the two figures, we observe that when $M = 1$, our Q-EarlySettled-LowCost algorithm enjoy the best regret compared with the other six single-agent model-free algorithms. We also note that the red curves for the Q-EarlySettled-LowCost algorithm approach horizontal lines as the total number of episodes $T/H$ becomes sufficiently large. Since the y-axis is $\text{Regret}(T)/\log(T/H + 1)$, this suggests that the regret grows logarithmically with $T$, which matches our gap-dependent regret bound result in Theorem 4.5. We also show the logarithmic switching cost results in the following Figure 2.

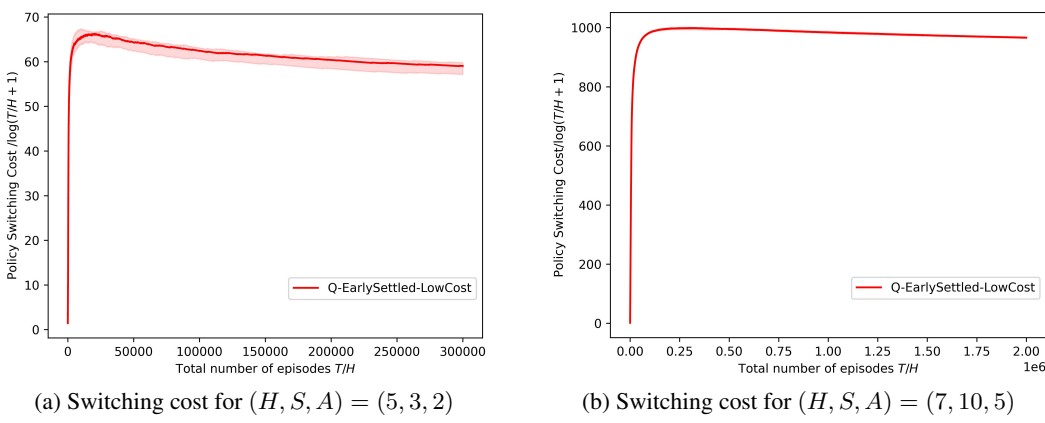

(a) Switching cost for $(H, S, A) = (5, 3, 2)$      (b) Switching cost for $(H, S, A) = (7, 10, 5)$

Figure 2: Switching cost results for Q-EarlySettled-LowCost when $M = 1$

From Figure 2, we note that the red curves for Q-EarlySettled-LowCost algorithm also approach horizontal lines as the total number of episodes $T/H$ becomes sufficiently large. This suggests that the switching cost grows logarithmically with $T$, which matches our logarithmic switching cost bound result in Theorem 4.2 and Theorem 4.7.

---

[4]All the experiments in this subsection are run on a server with Intel Xeon E5-2650v4 (2.2GHz) and 100 cores. Each replication is limited to a single core and 8GB of RAM. The total execution time is about 5 hours. The code for the numerical experiments is included in the supplementary materials along with the submission.

[5]https://openreview.net/attachment?id=6tyPSkshtF&name=supplementary_material

## B.2 Comparison of FRL Algorithms

Under the given MDP, we set $M = 10$ and generate $3 * 10^5$ episodes for $(H, S, A) = (5, 3, 2)$ and $2 * 10^6$ episodes for $(H, S, A) = (7, 10, 5)$[6]. For each episode, we randomly choose the initial state uniformly from the $S$ states. For the other three federated model-free algorithms, FedQ-Hoeffding, FedQ-Bernstein, and FedQ-Advantage, we use their hyperparameter settings based on the publicly available code[7] in [114]. For the FedQ-EarlySettled-LowCost algorithm, we use the same hyperparameter setting as specified in Appendix B.1.

To show error bars, we also collect 10 sample paths for all algorithms under the same MDP environment and show the relationship between $\text{Regret}(T)/\log(T/H + 1)$ and the total number of episodes for each agent $T/H$ in Figure 3. For both panels, the solid line represents the median of the 10 sample paths, while the shaded area shows the 10th and 90th percentiles.

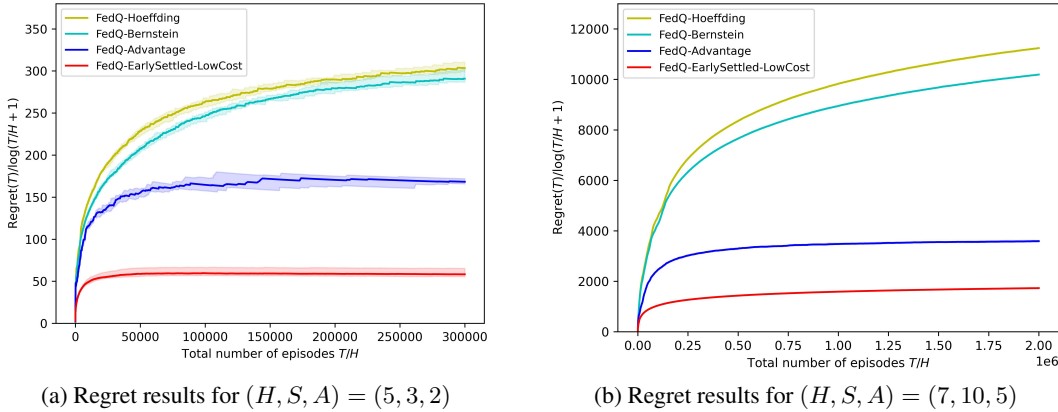

(a) Regret results for $(H, S, A) = (5, 3, 2)$      (b) Regret results for $(H, S, A) = (7, 10, 5)$

Figure 3: Numerical comparison of regrets for federated model-free algorithms

From the two figures, we observe that our proposed FedQ-EarlySettled-LowCost algorithm enjoy the best regret compared with the other three federated model-free algorithms. We also note that the red curves for the FedQ-EarlySettled-LowCost algorithm approach horizontal lines as the total number of episodes $T/H$ becomes sufficiently large. This suggests that the regret grows logarithmically with $T$, which matches our gap-dependent regret bound result in Theorem 4.8.

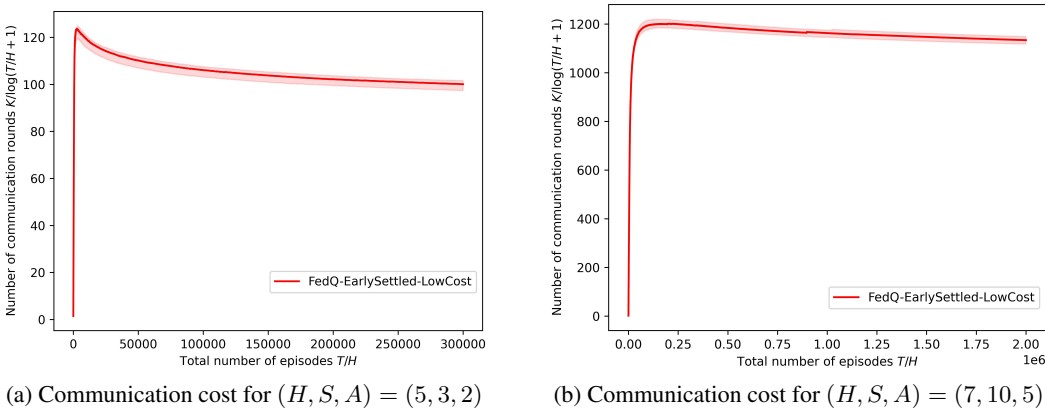

(a) Communication cost for $(H, S, A) = (5, 3, 2)$      (b) Communication cost for $(H, S, A) = (7, 10, 5)$

Figure 4: Number of communication rounds for FedQ-EarlySettled-LowCost

---

[6]All the experiments in this subsection are run on a server with Intel Xeon E5-2650v4 (2.2GHz) and 100 cores. Each replication is limited to five cores and 15GB of RAM. The total execution time is about 15 hours. The code for the numerical experiments is included in the supplementary materials along with the submission.

[7]`https://openreview.net/attachment?id=FoUpv84hMw&name=supplementary_material`

From Figure 4, we find that the number of communication rounds curves for the FedQ-EarlySettled-LowCost algorithm approach horizontal lines as the total number of episodes $T/H$ becomes sufficiently large. This suggests that the number of communication rounds grows logarithmically with $T$, which matches our logarithmic gap-dependent communication cost bound result in Theorem 4.4 and Theorem 4.9.

## C   Technical Novelty in Depth

In this section, we highlight our technical novelty for proving the worst-case regrets for both Q-EarlySettled-LowCost and FedQ-EarlySettled-LowCost, where we overcome the challenge of **simultaneous non-adaptiveness** by using the **surrogate reference function**.

We prove the worst-case regret bounds in Theorem 4.1 and Theorem 4.3 by relating the regret to the estimation error of the optimal $Q-$functions that takes the form

$$\sum_{k,j,m} (Q_h^k - Q_h^\star)(s_h^{k,j,m}, a_h^{k,j,m}).$$

It is common in the literature to bound the summation by recursions on $h$ in model-free algorithms [50, 98, 111, 113]. In more detail, for each individual error $(Q_h^k - Q_h^\star)(s_h^{k,j,m}, a_h^{k,j,m})$, by the update rule in Equation (7), it can be upper bounded by $(Q_h^R - Q_h^\star)(s_h^{k,j,m}, a_h^{k,j,m})$. Furthermore, based on the reference-advantage-type update given in Equation (11) and Equation (15), Equation (77) in Appendix I shows that with high probability, $(Q_h^{R,k} - Q_h^\star)(s_h^{k,j,m}, a_h^{k,j,m}) \leq \mathcal{G}_0$, where

$$\mathcal{G}_0 = \sum_{n=1}^{N_h^k} \tilde{\eta}_n^{N_h^k} \left( \left(V_{h+1}^{k^n} - V_{h+1}^{R,k^n}\right)(s_{h+1}^{k^n,j^n,m^n}) + \mu_h^{R,k^n+1} - \mathbb{P}_{s_h^{k,j,m}, a_h^{k,j,m}, h} V_{h+1}^\star \right) + C_B.$$

Here, $k^n$ represents the round index that the $n-$th visit to $(s_h^{k,j,m}, a_h^{k,j,m}, h)$ happens. $\tilde{\eta}_n^{N_h^k}$ denotes the cumulative weight for the $n-$th visit to this state-action-step triple under our delayed-policy-switching scheme (see Equation (19) in Appendix F for the definition), which approximates $\eta_n^{N_h^k}$ (the exact weight in high burn-in cost algorithms requiring frequent policy updates [37, 50]). $C_B$ collects some constants and the cumulative bonuses. All three notations hide their dependency on $(k, j, m, h)$. We start from the decomposition of $\mathcal{G}_0$, following the design in [50] for Q-EarlySettled-Advantage with frequent policy switching, and then explain the challenges of non-adaptiveness therein and our solution of surrogate reference function.

**Decomposition.** We use $V_h^{R,K+1}$ to denote the settled reference function. By applying the inequality $V_{h+1}^{k^n} - V_{h+1}^{R,k^n} \leq V_{h+1}^{k^n} - V_{h+1}^{R,K+1}$, which follows directly from the monotonically non-increasing property of $V_{h+1}^{R,k}$ as guaranteed by the reference function settling rule in line 12 of Algorithm 1, $\mathcal{G}_0$ can be further upper bounded by the summation of $C_B$ and the following terms:

$$\mathcal{G}_1 := \sum_{n=1}^{N_h^k} \tilde{\eta}_n^{N_h^k} \left(V_{h+1}^{k^n} - V_{h+1}^\star\right)(s_{h+1}^{k^n,j^n,m^n}),$$

$$\mathcal{G}_2 := \sum_{n=1}^{N_h^k} \tilde{\eta}_n^{N_h^k} \left( \mathbb{1}_{s_{h+1}^{k^n,j^n,m^n}} - \mathbb{P}_{s_h^{k,j,m}, a_h^{k,j,m}, h} \right) \left( V_{h+1}^\star - V_{h+1}^{R,K+1} \right),$$

$$\mathcal{G}_3 := \sum_{n=1}^{N_h^k} \tilde{\eta}_n^{N_h^k} \frac{1}{N_h^{k^n+1}} \sum_{i=1}^{N_h^{k^n+1}} \left( V_{h+1}^{R,k^i}(s_{h+1}^{k^i,j^i,m^i}) - V_{h+1}^{R,K+1}(s_{h+1}^{k^i,j^i,m^i}) \right),$$

$$\mathcal{G}_4 := \sum_{n=1}^{N_h^k} \tilde{\eta}_n^{N_h^k} \frac{1}{N_h^{k^n+1}} \sum_{i=1}^{N_h^{k^n+1}} \left( V_{h+1}^{R,K+1}(s_{h+1}^{k^i,j^i,m^i}) - \mathbb{P}_{s_h^{k,j,m}, a_h^{k,j,m}, h} V_{h+1}^{R,K+1} \right).$$

This decomposition follows the structure of the reference-advantage decomposition, where $\mathcal{G}_1$ reflects the $V$-value function estimation error, $\mathcal{G}_2$ and $\mathcal{G}_4$ reflect the empirical estimation error for the settled

advantage function and reference function respectively, and $\mathcal{G}_3$ reflects the reference settling error. Thus, the estimation error $\sum_{k,j,m}(Q_h^k - Q_h^\star)(s_h^{k,j,m}, a_h^{k,j,m})$ can be upper bounded by the sum of five corresponding summations with regard to $C_B, \mathcal{G}_1, \mathcal{G}_2, \mathcal{G}_3, \mathcal{G}_4$.

**Challenges: Simultaneous Non-Adaptiveness.** However, the summations involving $\mathcal{G}_2$ and $\mathcal{G}_4$ cannot be directly bounded using standard concentration inequalities. This limitation arises from the non-adaptive nature of **both** the **weights** $\{\tilde{\eta}_n^{N_h^k}\}$ and the **settled reference function** $V_{h+1}^{\mathrm{R},K+1}$ at the next steps of historical visits to $(s_h^{k,j,m}, a_h^{k,j,m}, h)$.

The weights $\{\tilde{\eta}_n^{N_h^k}\}$ exhibit non-adaptiveness due to our special aggregation scheme used in Case 2 of the estimated $Q-$function updates (Equation (7)), which assigns identical weights to all visits within a given round. Each weight depends on the total number of visits to the corresponding state-action-step triple in that round, a quantity that is unknown during visitation due to our event-triggered termination condition specified in Equation (3). This uncertainty results in the non-adaptiveness of the weights. The non-adaptiveness of $V_{h+1}^{\mathrm{R},K+1}$ is because the reference settling depends on all the historical information in the learning process (see line 12 in Algorithm 1 for the reference function's update rule).

[113] addresses the non-adaptiveness of the weights through round-wise approximations, and [50] tackles the non-adaptiveness related to the settled reference function via the empirical process. **However, neither approach simultaneously resolves both forms of non-adaptiveness**, and their direct combination presents significant technical challenges. The empirical process employed in [50] introduces **an additional logarithmic factor** of $T$ in the error bound when constructing $\epsilon-$nets to cover the function space induced by $V_{h+1}^{\mathrm{R},K+1}$.

**Our Solution: Surrogate Reference Functions.** To combine the round-based design (with equal weight assignments in each round) and the LCB technique (used in the reference-advantage decomposition) for simultaneous low burn-in costs and logarithmic switching/communication costs, we adapt the **surrogate reference function** technique from [115]—originally developed for gap-dependent analysis—and successfully incorporate it into our **worst-case regret framework**. There are two benefits of this adaptation: (i) It resolves the fundamental challenge of simultaneous non-adaptiveness in the learning process. (ii) By replacing the empirical process technique from [50], it eliminates the additional logarithmic factor in the error bound, yielding tighter regret guarantees.

The surrogate reference function $\hat{V}_h^{\mathrm{R},k}$ in our framework is defined as:

$$\hat{V}_h^{\mathrm{R},k}(s) := \max\left\{V_h^\star(s), \min\left\{V_h^\star(s) + \beta, V_h^{\mathrm{R},k}(s)\right\}\right\}, \forall(s,h,k).$$

It naturally adapts to the learning process. In addition, building on the optimistic property demonstrated in Lemma H.1, we can show that

$$\hat{V}_h^{\mathrm{R},k}(s) = \min\left\{V_h^\star(s) + \beta, V_h^{\mathrm{R},k}(s)\right\}$$

with high probability. Thus, it maintains the same monotonically non-increasing property as the reference function $V_h^{\mathrm{R},k}(s)$ and coincides with the settled reference function $V_h^{\mathrm{R},K+1}$ once the reference function is settled (see line 12 in Algorithm 1 for the update rule of reference functions). By applying the inequality $V_{h+1}^{k^n} - V_{h+1}^{\mathrm{R},k^n} \le V_{h+1}^{k^n} - \hat{V}_{h+1}^{\mathrm{R},k^n}$, $\mathcal{G}_0$ can be upper bounded by the summation of $C_B, \mathcal{G}_1, \mathcal{G}_2, \mathcal{G}_3, \mathcal{G}_4$, where all the $V_{h+1}^{\mathrm{R},K+1}$s are replaced by the surrogate reference function $\hat{V}_{h+1}^{\mathrm{R},k^n}$.

In the new decomposition, the summation over $C_B$ is well-controlled in Lemma I.1, following a similar approach to [50, Appendix E.2]. The summation over $\mathcal{G}_1$ can be bounded by

$$\exp\left(\frac{3}{H}\right) \sum_{k,j,m} \left(Q_{h+1}^k - Q_{h+1}^\star\right)(s_{h+1}^{k,j,m}, a_{h+1}^{k,j,m})$$

and an additional constant term as shown in Equation (82) of Appendix I, establishing the error recursion for step $h$. This step relies on a double-sum rearrangement, a standard technique in model-free RL analysis [37, 50, 113]. The summation for $\mathcal{G}_3$ can be bounded in the term $R_{3,3}$ of Lemma I.3, using the technique from [50, Lemma 3] for controlling the reference settling error. Furthermore, the revised $\mathcal{G}_2$ and $\mathcal{G}_4$ introduce non-adaptiveness only through their weights, allowing us to apply

round-wise approximation methods [113] to bound them. For the revised $\mathcal{G}_2$, we approximate it with

$$\sum_{n=1}^{N_h^k} \eta_n^{N_h^k} \left( \mathbb{1}_{s_{h+1}^{k^n,j^n,m^n}} - \mathbb{P}_{s_h^{k,j,m},a_h^{k,j,m},h} \right) \left( V_{h+1}^\star - \hat{V}_{h+1}^{\mathrm{R},k^n} \right),$$

where all elements adapt to the data generation process. The approximation error can be controlled via round-wise concentration inequalities (see Lemma F.5). The summation with regard to the revised $\mathcal{G}_4$ can be bounded through decomposition into four terms $R_{3,1}$, $R_{3,2}$, $R_{3,4}$ and $R_{3,5}$ in Lemma I.3 following the same idea. For more technical details, we refer readers to Lemma I.2 for the analysis of the summation of $\mathcal{G}_2$ and to Lemma I.3 for the detailed treatment of the summation of $\mathcal{G}_4$.

Using the surrogate reference functions not only solves the challenge of simultaneous non-adaptiveness but also yields a $\log(SAT/p)$ improvement over the current state-of-the-art Q-EarlySettled-Advantage [50] in single-agent RL by avoiding using the empirical process.

To our knowledge, this work represents the first successful integration of the surrogate reference functions and round-wise approximation to handle both forms of non-adaptiveness of the weights and the settled reference function. This methodological advancement significantly expands the analytical tools available for federated reinforcement learning and overcomes limitations present in previous approaches.

## D    Graphical Illustrations and Notation Tables

In this section, we provide graphical illustrations and notational reference tables to enhance comprehension of our algorithms.

### D.1    Graphical Illustrations

In this subsection, we present some graphical illustrations of our FRL framework and round-based design. Figure 5 presents the central server's initialization phase, where the central server broadcast key parameters to all agents at the start of each training round.

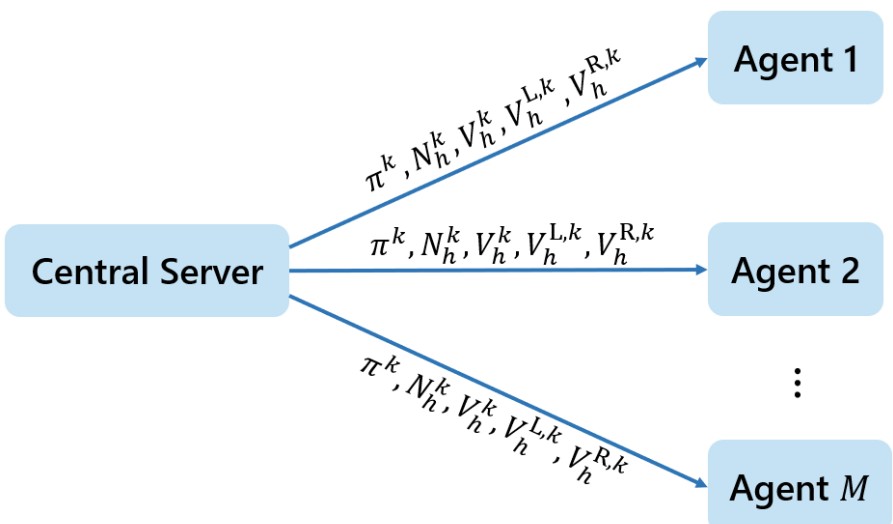

Figure 5: Central server broadcast protocol. At the beginning of round $k$, for any state-step pair $(s,h) \in \mathcal{S} \times [H]$, the central server broadcasts the current policy $\pi^k$, the total number of visits before round $k$ $N_h^k(s, \pi_h^k(s))$, the $V-$estimates $V_h^k(s, \pi_h^k(s))$, the lower bound function $V_h^{\mathrm{L},k}(s, \pi_h^k(s))$ and the reference function $V_h^{\mathrm{R},k}(s, \pi_h^k(s))$ to each agent.

The following Figure 6 illustrates the agent-to-server data transmission phase. After exploration terminates in round $k$ (when condition (3) is met), each agent transmits the rewards and some local summary statistics to the central server.

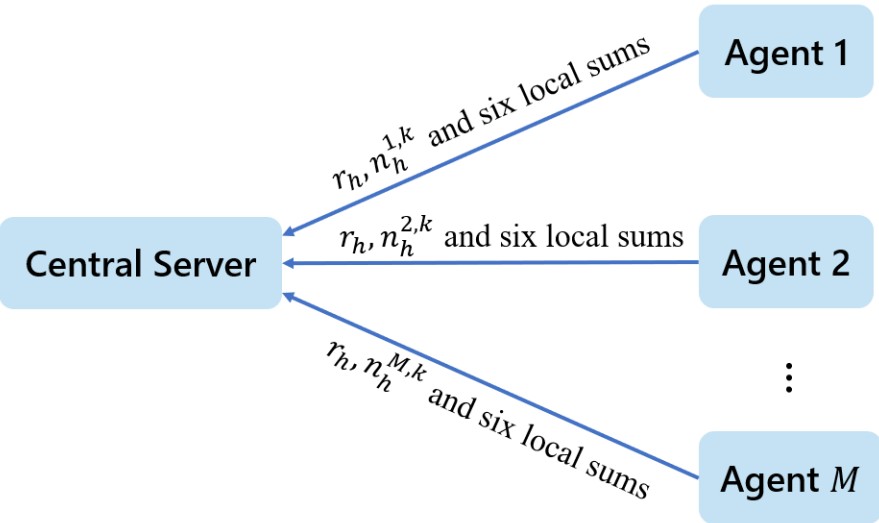

Figure 6: Agent-to-server data transmission. At the end of each round $k$, for any state-step pair $(s, h) \in \mathcal{S} \times [H]$, agent $m$ sends the reward $r_h(s, \pi_h^k(s))$, the number of visits in round $k$ $n_h^{m,k}(s, \pi_h^k(s))$ and six local sums in Equation (4) to the central server.

The following Figure 7 explains our round-based design for infrequent policy and value function estimate updates. Unlike conventional per-episode updates, our algorithms accumulate trajectory data across multiple episodes within each round, updating both the value function estimates and policy only at the end of each round.

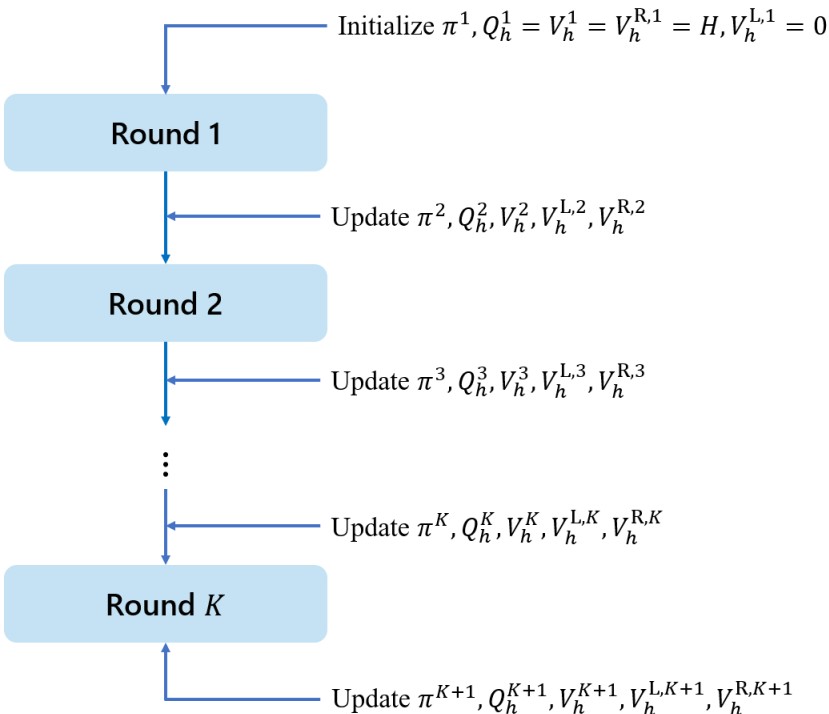

Figure 7: Round-based design. At the beginning of the learning process, the central server initializes $Q_h^1 = V_h^1 = V_h^{\mathrm{R},1} = H$ and $V_h^{\mathrm{L},1} = 0$ and chooses an arbitrary policy $\pi^1$. At the end of round $k$, the central server updates the policy $\pi^{k+1}$ and $(Q_h^{k+1}, V_h^{k+1}, V_h^{\mathrm{L},k+1}, V_h^{\mathrm{R},k+1})$ for any visited state-action-step triple $(s, a, h) \in \mathcal{S} \times \mathcal{A} \times [H]$.

## D.2 Notation Tables

In this subsection, we provide two notation tables of FedQ-EarlySettled-LowCost to enhance the readability of the paper. One of the table consists of global variables utilized for central server aggregation, while the other table presents local variables employed for agent local training.

Table 3: Global Variables

| Variable | Definition |
|---|---|
| $Q_h^k$ | the estimated $Q-$value function of step $h$ at the beginning of round $k$ |
| $Q_h^{\mathrm{U},k}$ | the UCB-type $Q-$estimates of step $h$ at the beginning of round $k$ |
| $Q_h^{\mathrm{L},k}$ | the LCB-type $Q-$estimates of step $h$ at the beginning of round $k$ |
| $Q_h^{\mathrm{R},k}$ | the reference-advantage-type $Q-$estimates of step $h$ at the beginning of round $k$ |
| $V_h^k$ | the estimated $V-$value function of step $h$ at the beginning of round $k$ |
| $V_h^{\mathrm{L},k}$ | the lower bound function of step $h$ at the beginning of round $k$ |
| $V_h^{\mathrm{R},k}$ | the reference function of step $h$ at the beginning of round $k$ |
| $V_h^{\mathrm{A},k}$ | the advantage function $V_h^k - V_h^{\mathrm{R},k}$ of step $h$ at the beginning of round $k$ |
| $B_h^k$ | the Hoeffding-type cumulative bonus in round $k$ |
| $B_h^{\mathrm{R},k}$ | the reference-advantage-type cumulative bonus in round $k$ |
| $N_h^k(s,a)$ | the total number of visits to $(s,a,h)$ before round $k$ |
| $n_h^k(s,a)$ | the total number of visits to $(s,a,h)$ in round $k$ |
| $\mu_h^{\mathrm{R},k}(s,a)$ | the mean of the reference function at all next states of the visits to $(s,a,h)$ before round $k$ |
| $\sigma_h^{\mathrm{R},k}(s,a)$ | the mean of the squared reference function at all next states of the visits to $(s,a,h)$ before round $k$ |
| $\mu_h^{\mathrm{A},k}(s,a)$ | the weighted sum of the advantage function at all next states of the visits to $(s,a,h)$ before round $k$ |
| $\sigma_h^{\mathrm{A},k}(s,a)$ | the weighted sum of the squared advantage function at all next states of the visits to $(s,a,h)$ before round $k$ |
| $v_h^k(s,a)$ | the mean of $V_h^k$ at all next states of the visits to $(s,a,h)$ in round $k$ |
| $v_h^{l,k}(s,a)$ | the mean of $V_h^{\mathrm{L},k}$ at all next states of the visits to $(s,a,h)$ in round $k$ |
| $\mu_h^{\mathrm{r},k}(s,a)$ | the mean of $V_h^{\mathrm{R},k}$ at all next states of the visits to $(s,a,h)$ in round $k$ |
| $\sigma_h^{\mathrm{r},k}(s,a)$ | the mean of $(V_h^{\mathrm{R},k})^2$ at all next states of the visits to $(s,a,h)$ in round $k$ |
| $\mu_h^{\mathrm{a},k}(s,a)$ | the mean of $V_h^{\mathrm{A},k}$ at all next states of the visits to $(s,a,h)$ in round $k$ |
| $\sigma_h^{\mathrm{a},k}(s,a)$ | the mean of $(V_h^{\mathrm{A},k})^2$ at all next states of the visits to $(s,a,h)$ in round $k$ |
| $u_h^{\mathrm{R}}$ | the indicator used to terminate the reference function update. |

Table 4: Local Variables

| Variable | Definition |
|---|---|
| $n_h^{m,k}(s,a)$ | the total number of visits to $(s,a,h)$ of agent $m$ in round $k$ |
| $v_h^{m,k}(s,a)$ | the mean of $V_h^k$ at all next states of the visits to $(s,a,h)$ of agent $m$ in round $k$ |
| $v_{h,l}^{m,k}(s,a)$ | the mean of $V_h^{\mathrm{L},k}$ at all next states of the visits to $(s,a,h)$ of agent $m$ in round $k$ |
| $\mu_{h,\mathrm{r}}^{m,k}(s,a)$ | the mean of $V_h^{\mathrm{R},k}$ at all next states of the visits to $(s,a,h)$ of agent $m$ in round $k$ |
| $\sigma_{h,\mathrm{r}}^{m,k}(s,a)$ | the mean of $(V_h^{\mathrm{R},k})^2$ at all next states of the visits to $(s,a,h)$ of agent $m$ in round $k$ |
| $\mu_{h,\mathrm{a}}^{m,k}(s,a)$ | the mean of $V_h^{\mathrm{A},k}$ at all next states of the visits to $(s,a,h)$ of agent $m$ in round $k$ |
| $\sigma_{h,\mathrm{a}}^{m,k}(s,a)$ | the mean of $(V_h^{\mathrm{A},k})^2$ at all next states of the visits to $(s,a,h)$ of agent $m$ in round $k$ |

# E  General Theorems

**Theorem E.1.** (Azuma-Hoeffding Inequality). *Suppose $\{X_k\}_{k=0}^{\infty}$ is a martingale and $|X_k - X_{k-1}| \leq c_k$, $\forall k \in \mathbb{N}_+$, almost surely. Then for any $N \in \mathbb{N}_+$ and $\epsilon > 0$, it holds that:*

$$\mathbb{P}\left(|X_N - X_0| \geq \epsilon\right) \leq 2 \exp\left(-\frac{\epsilon^2}{2\sum_{k=1}^{N} c_k^2}\right).$$

**Theorem E.2.** (Freedman's inequality, [49, Theorem EC.1]). *Consider a filtration $\mathcal{F}_0 \subset \mathcal{F}_1 \subset \mathcal{F}_2 \subset \cdots$, and let $\mathbb{E}_k$ stand for the expectation conditioned on $\mathcal{F}_k$. Suppose that $Y_n = \sum_{k=1}^{n} X_k \in \mathbb{R}$, where $\{X_k\}$ is a real-valued scalar sequence obeying*

$$|X_k| \leq R \quad and \quad \mathbb{E}_{k-1}[X_k] = 0 \quad for\ all\ k \geq 1$$

*for some quantity $R < \infty$. We also define $W_n := \sum_{k=1}^{n} \mathbb{E}_{k-1}[X_k^2]$. In addition, suppose that $W_n \leq \sigma^2$ holds deterministically for some given quantity $\sigma^2 < \infty$. Then for any positive integer $m \geq 1$, with probability at least $1 - \delta$, one has*

$$|Y_n| \leq \sqrt{8 \max\left\{W_n, \frac{\sigma^2}{2^m}\right\} \log \frac{2m}{\delta}} + \frac{4}{3} R \log \frac{2m}{\delta}.$$

**Theorem E.3.** ([109, Lemma 10]). *Let $X_1, X_2, \ldots$ be a sequence of random variables taking value in $[0, l]$. Define $\mathcal{F}_k = \sigma(X_1, X_2, \ldots, X_k)$ and $Y_k = \mathbb{E}[X_k | \mathcal{F}_{k-1}]$ for $k \geq 1$. For any $\delta > 0$, we have that*

$$\mathbb{P}\left[\exists n, \sum_{k=1}^{n} X_k \geq 3 \sum_{k=1}^{n} Y_k + l \log(1/\delta)\right] \leq \delta$$

*and*

$$\mathbb{P}\left[\exists n, \sum_{k=1}^{n} Y_k \geq 3 \sum_{k=1}^{n} X_k + l \log(1/\delta)\right] \leq \delta.$$

# F  Key Lemmas

In this section, we introduce some useful lemmas. Before starting, we rank the visits to any given triple $(s, a, h)$ based on the "round first, episode second, step third, agent fourth" rule (time order) and use $k^i(s, a, h)$, $j^i(s, a, h)$, $m^i(s, a, h)$ to denote the the round, episode and agent index for the $i-$th visit, respectively. When there is no ambiguity, we use $k^i$, $m^i$, and $j^i$ for short.

Let $\mathcal{X} = (\mathcal{S}, \mathcal{A}, H, T, \iota)$. The notation $f(\mathcal{X}) \lesssim g(\mathcal{X})$ means that there exists a universal constant $C_1 > 0$ such that $f(\mathcal{X}) \leq C_1 g(\mathcal{X})$. In the following sections, we assume that $0/0 = 0$. For any $C \in \mathbb{N}_+$, we use $[C]$ to denote the set $\{1, 2, ..., C\}$. We also use $\mathbb{I}[x]$ to denote the indicator function, which equals 1 when the event $x$ is true and 0 otherwise.

We now begin to introduce lemmas. Lemma F.1 establishes key relationships between certain quantities used in Algorithm 1 and Algorithm 2.

**Lemma F.1.** *The following relationships hold for Algorithm 1 and Algorithm 2.*

*(a) For any $(s, a, h, k) \in \mathcal{S} \times \mathcal{A} \times [H] \times [K]$, we have*

$$n_h^{m,k}(s, a) \leq \max\left\{1, \frac{N_h^k(s, a)}{MH(H+1)}\right\}, \forall m \in [M].$$

*and*

$$n_h^k(s, a) \leq \max\left\{M, \frac{N_h^k(s, a)}{H(H+1)}\right\}.$$

*If $N_h^k(s, a) \geq i_0$,*

$$n_h^k(s, a) \leq \frac{N_h^k(s, a)}{H(H+1)}.$$

*(b) Let $T_1 = \left(1 + \frac{1}{H(H+1)}\right) T_0 + MSAH$, then $\hat{T} \leq T_1 \leq 2\hat{T} + MSAH$.*

*(c)* $K \leq \frac{T_1}{H}$.

*(d) For any $(s, a, h) \in \mathcal{S} \times \mathcal{A} \times [H]$, $N_h^{K+1}(s, a) \leq \frac{T_1}{H}$.*

*Proof.* (a) is obvious given Equation (3).

(b) According to (a) and the definition of $T_0$ in Algorithm 2, we have:

$$\hat{T} = \sum_{s,a,h} N_h^{K+1}(s, a) \leq \sum_{s,a,h} N_h^K(s, a) + \sum_{s,a,h} n_h^K(s, a)$$

$$\leq T_0 + \sum_{s,a,h} \left( \frac{1}{H(H+1)} N_h^K(s, a) + M \right) = T_1.$$

The right side is obvious by noting that $T_0 < \hat{T}$ given the Algorithm 1.

(c) It is proved by $K \leq \frac{\hat{T}}{H} \leq \frac{T_1}{H}$.

(d) It is because $N_h^{K+1}(s, a) \leq \sum_{s,a} N_h^{K+1}(s, a) = \frac{\hat{T}}{H} \leq \frac{T_1}{H}$. $\qquad\square$

In the following Lemma F.2, we will present some properties of the weight $\eta_i^t$.

**Lemma F.2.** *The following properties hold for all $t \in \mathbb{N}_+$:*

*(a) For $\frac{1}{2} \leq \alpha \leq 1$, $1/t^\alpha \leq \sum_{i=1}^t \eta_i^t/i^\alpha \leq 2/t^\alpha$.*

*(b) $\max_{i \in [t]} \eta_i^t \leq 2H/t$.*

*(c) $\sum_{t=i}^\infty \eta_i^t = 1 + 1/H$.*

*(d) For any $t \in \mathbb{N}_+$ and $i \in [t-1]$, $\eta_{i+1}^t / \eta_i^t = 1 + H/i > 1$.*

*(e) For any $t \in \mathbb{N}_+$ and any $(s, a, h) \in \mathcal{S} \times \mathcal{A} \times [H]$, if $t < i$, $k^i(s, a, h) = k$ and $N_h^k(s, a) \geq i_0$, we have that $\eta_t^{N_h^k} / \eta_t^i \leq \exp(1/H)$.*

*Proof.* Here (a) is proved in [50, Lemma 1]. (b), (c) and (d) are directly from [113, Lemma B.2], so here we need to prove property (e). Note that

$$\frac{\eta_t^{N_h^k}}{\eta_t^i} = \prod_{q=N_h^k+1}^i (1 - \eta_q)^{-1} \overset{(I)}{\leq} \left( 1 - \eta_{N_h^k+1} \right)^{-(i - N_h^k)} \overset{(II)}{\leq} \left( 1 - \eta_{N_h^k+1} \right)^{-\frac{N_h^k}{H(H+1)}} \leq \exp(1/H).$$

Here (I) is because $\eta_q$ is monotonically decreasing. (II) is because $i - N_h^k(s, a) \leq n_h^k(s, a) \leq \frac{N_h^k(s,a)}{H(H+1)}$ for $N_h^k(s, a) \geq i_0$ by (a) of Lemma F.1. The last inequality uses the definition of $\eta_{N_h^k+1}$. $\quad\square$

Next, we define the new weights $\tilde{\eta}_i^t(s, a, h)$ for any $n_1 < i \leq n_2 \leq t \in \mathbb{N}_+$ and any $(s, a, h) \in \mathcal{S} \times \mathcal{A} \times [H]$, where $n_1 = N_h^{k^i}(s, a)$ and $n_2 = N_h^{k^i+1}(s, a)$:

$$\tilde{\eta}_i^t(s, a, h) = \eta_i^t \mathbb{I}[n_1 < i_0] + \frac{1 - \eta^c(n_1 + 1, n_2)}{n_2 - n_1} \eta^c(n_2 + 1, t) \mathbb{I}[n_1 \geq i_0]. \tag{19}$$

We will use the simplified notation $\tilde{\eta}_i^t$ when there is no ambiguity. In Lemma F.3, we present some key properties of the new weights $\tilde{\eta}_i^t$ and their relationship with the original weights $\eta_i^t$.

**Lemma F.3.** *The following relationships hold for any $(s, a, h, k) \in \mathcal{S} \times \mathcal{A} \times [H] \times [K]$ with $t = N_h^k(s, a)$.*

*(a) For any $i_1, i_2 \in [t]$, if $k^{i_1}(s, a, h) = k^{i_2}(s, a, h)$ and $N_h^{k^{i_1}(s,a,h)}(s, a) \geq i_0$, we have that $\tilde{\eta}_{i_1}^t(s, a, h) = \tilde{\eta}_{i_2}^t(s, a, h)$.*

(b) *For any $k' < k$ and $k' \in [k]$, we have that*

$$\sum_{i'=N_h^{k'}+1}^{N_h^{k'+1}} \tilde{\eta}_{i'}^t(s,a,h) = \sum_{i'=N_h^{k'}+1}^{N_h^{k'+1}} \eta_{i'}^t,$$

*which indicates that $\sum_{i=1}^t \tilde{\eta}_i^t = \mathbb{I}[t > 0]$.*

(c) *For any $i \in [t]$, we have that*
$$\tilde{\eta}_i^t / \eta_i^t \le \exp(1/H).$$

(d) *For $t_1 = N_h^{k+1}(s,a)$,*

$$\sqrt{\sum_{i=t+1}^{t_1} (\tilde{\eta}_i^{t_1} - \eta_i^{t_1})^2} \lesssim \sum_{i=t+1}^{t_1} \eta_i^{t_1} / \sqrt{i}.$$

*Proof.* Here, (a) and (b) are directly from [113, Lemma B.3] and (d) is from [113, Equation (19)], so we only prove (c) here. (d) of [113, Lemma B.3] proves that $\tilde{\eta}_i^t \le \exp(1/H)\eta_i^t$ when $N_h^{k^i(s,a,h)}(s,a) \ge i_0$. Additionally, by the definition of $\tilde{\eta}_i^t$ (see Equation (19)), we have $\tilde{\eta}_i^t = \eta_i^t$ when $N_h^{k^i(s,a,h)}(s,a) < i_0$. Combining both cases, the proof is complete. $\square$

Next, we develop an immediate corollary of Freedman's inequality.

**Lemma F.4.** *Let $\iota = \log(\frac{2SAT_1}{\delta})$ for any $\delta \in (0,1)$. For any $(s,a,h,k) \in \mathcal{S} \times \mathcal{A} \times [H] \times [K]$, let $\{W_{h+1}^{k^i} \in \mathbb{R}^S \mid 1 \le i \le N_h^k(s,a)\}$ and $\{u_h^i(s,a,N_h^k(s,a)) \in \mathbb{R} \mid 1 \le i \le N_h^k(s,a)\}$ be collections of vectors and scalars, which obey the following properties:*

- $W_{h+1}^{k^i}$ *is fully determined by the samples collected up to $h-$th step of $j^i-$th episode of agent $m^i$ in round $k^i$, where the samples is ordered based on the "round first, episode second, step third, agent fourth" rule (time order).*

- $\|W_{h+1}^{k^i}\|_\infty \le C_w$;

- $u_h^i(s,a,N_h^k)$ *is fully determined by the episodes collected up to the $h-$th step of $j^i-$th episode of agent $m^i$ in round $k^i$, and a positive integer $N_h^k \in [T_1/H]$. The samples is ordered based on the "round first, episode second, step third, agent fourth" rule.*

- $0 \le u_h^i(s,a,N_h^k) \le C_u(N_h^k)$.

- $0 \le \sum_{i=1}^{N_h^k(s,a)} u_h^i(s,a,N_h^k) \le 2$.

*With probability at least $1 - \delta$, the following inequality hold simultaneously for all $(s,a,h,k) \in \mathcal{S} \times \mathcal{A} \times [H] \times [K]$.*

$$\left| \sum_{i=1}^{N_h^k} u_h^i(s,a,N_h^k) \left( \mathbb{1}_{s_{h+1}^{k^i,j^i,m^i}} - \mathbb{P}_{s,a,h} \right) W_{h+1}^{k^i} \right|$$

$$\le 5\sqrt{C_u(N_h^k)\iota \sum_{i=1}^{N_h^k} u_h^i(s,a,N_h^k)\mathbb{V}_{s,a,h}(W_{h+1}^{k^i})} + 8\left( \sqrt{\frac{C_u(N_h^k)C_w^2}{N_h^k}} + C_u(N_h^k)C_w \right)\iota.$$

*Proof.* For any $N_h^k = N \in [T_1/H]$, it holds that

$$\sum_{i=1}^{N_h^k} u_h^i(s,a,N_h^k) \left( \mathbb{1}_{s_{h+1}^{k^i,j^i,m^i}} - \mathbb{P}_{s,a,h} \right) W_{h+1}^{k^i} = \sum_{i=1}^{N} u_h^i(s,a,N) \left( \mathbb{1}_{s_{h+1}^{k^i,j^i,m^i}} - \mathbb{P}_{s,a,h} \right) W_{h+1}^{k^i}$$

$$= \sum_{(k',j',h,m') \le (k^N,j^N,h,m^N)} X_h^{k',j',m'},$$

where

$$X_h^{k',j',m'} = u_h^{i^{k',j',m'}}(s,a,N)\left(\mathbb{1}_{s_{h+1}^{k',j',m'}} - \mathbb{P}_{s,a,h}\right)W_{h+1}^{k'}(s,a)\mathbb{I}\left[(s_h^{k',j',m'},a_h^{k',j',m'}) = (s,a)\right].$$

Here, $(k',j',h,m') \leq (k^N,j^N,h,m^N)$ means the sample at $(k',j',h,m')$ is collected before the sample collected at $(k^N,j^N,h,m^N)$ with the order "round first, episode second, step third, agent fourth". $i^{k',j',m'}$ is the number of visits to $(s,a)$ at step $h$ before the sample $(k',j',h,m')$. We order $X_h^{k',j',m'}$ to be $\{Y_1,Y_2,...,Y_n\}$. Then given $(s,a,h,k,N) \in \mathcal{S} \times \mathcal{A} \times [H] \times [T_1/H] \times [T_1/H]$, we have $\mathbb{E}_{r-1}[Y_r] = 0$. It also holds that:

$$|Y_r| \leq \|u_{h+1}^i\|_\infty \cdot 2\|W_{h+1}^{k^i}\|_\infty \leq 2C_wC_u(N),$$

and

$$\sum_{r=1}^N \mathbb{E}_{r-1}Y_r^2 = \sum_{i=1}^N (u_h^i(s,a,N))^2\mathbb{V}_{s,a,h}(W_{h+1}^{k^i}) \leq 2C_u(N)C_w^2.$$

Using Theorem E.2, with $R = 2C_u(N)C_w$, $m = \log_2(N)$, $\sigma^2 = 2C_u(N)C_w^2$, we have, with probability at least $1 - \delta/SAT_1^2$:

$$\left|\sum_{i=1}^N u_h^i(s,a,N)\left(\mathbb{1}_{s_{h+1}^{k^i,j^i,m^i}} - \mathbb{P}_{s,a,h}\right)W_{h+1}^{k^i}\right|$$

$$\leq \sqrt{24\max\left\{(u_h^i(s,a,N))^2\mathbb{V}_{s,a,h}(W_{h+1}^{k^i}), \frac{2C_u(N)C_w^2}{N}\right\}\iota} + 8C_u(N)C_w\iota$$

$$\leq 5\sqrt{C_u(N)\iota\sum_{i=1}^N u_h^i(s,a,N)\mathbb{V}_{s,a,h}(W_{h+1}^{k^i})} + 8\left(\sqrt{\frac{C_u(N)C_w^2}{N}} + C_u(N)C_w\right)\iota.$$

Consider all combinations of $(s,a,h,k,N) \in \mathcal{S} \times \mathcal{A} \times [H] \times [\frac{T_1}{H}] \times [\frac{T_1}{H}]$, we finish the proof. $\qquad\square$

Using Lemma F.4, we derive an upper bound for the summation of non-martingale differences weighted by $\tilde{\eta}_i^t$. The results are presented in the following Lemma F.5.

**Lemma F.5.** *For any $(s,a,h,k) \in \mathcal{S} \times \mathcal{A} \times [H] \times [K]$, let $\{W_{h+1}^{k^i} \in \mathbb{R}^S \mid 1 \leq i \leq N_h^k(s,a)\}$ be collections of vectors, which obey the following properties:*

- *$W_{h+1}^{k^i}$ is fully determined by the samples collected up to $h-$th step of $j^i-$th episode of agent $m^i$ in round $k^i$, where the samples is ordered based on the "round first, episode second, step third, agent fourth" rule.*

- *$\|W_{h+1}^{k^i}\|_\infty \leq C_w$.*

*Then with probability at least $1 - \delta$, the following inequality holds:*

$$\left|\sum_{i=1}^{N_h^k} \tilde{\eta}_i^{N_h^k}\left(\mathbb{1}_{s_{h+1}^{k^i,j^i,m^i}} - \mathbb{P}_{s,a,h}\right)W_{h+1}^{k^i}\right| \lesssim \sqrt{\frac{H\iota}{N_h^k}\sum_{i=1}^{N_h^k}\eta_i^{N_h^k}\mathbb{V}_{s,a,h}(W_{h+1}^{k^i})} + \frac{HC_w\iota}{N_h^k}$$

$$= \sqrt{\frac{H\iota}{N_h^k}\sum_{i=1}^{N_h^k}\tilde{\eta}_i^{N_h^k}\mathbb{V}_{s,a,h}(W_{h+1}^{k^i})} + \frac{HC_w\iota}{N_h^k}.$$

*Proof.*

$$\sum_{i=1}^{N_h^k}\tilde{\eta}_i^{N_h^k}\left(\mathbb{1}_{s_{h+1}^{k^i,j^i,m^i}} - \mathbb{P}_{s,a,h}\right)W_{h+1}^{k^i} = \sum_{i=1}^{N_h^k}\left[\eta_i^{N_h^k} + \left(\tilde{\eta}_i^{N_h^k} - \eta_i^{N_h^k}\right)\right]\left(\mathbb{1}_{s_{h+1}^{k^i,j^i,m^i}} - \mathbb{P}_{s,a,h}\right)W_{h+1}^{k^i}.$$

We know $\max_{i \in [N_h^k]} \eta_i^{N_h^k} \leq 2H/N_h^k$ and $\sum_{i=1}^{N_h^k} \eta_i^{N_h^k} \leq 1$ from (b) in Lemma F.2. Using Lemma F.4 with $C_u(N_h^k) = 2H/N_h^k$, with probability at least $1 - \delta/2$, it holds that:

$$\left| \sum_{i=1}^{N_h^k} \eta_i^{N_h^k} \left( \mathbb{1}_{s_{h+1}^{k^i,j^i,m^i}} - \mathbb{P}_{s,a,h} \right) W_{h+1}^{k^i} \right| \lesssim \sqrt{\frac{H\iota}{N_h^k} \sum_{i=1}^{N_h^k} \eta_i^{N_h^k} \mathbb{V}_{s,a,h}(W_{h+1}^{k^i})} + \frac{HC_w\iota}{N_h^k}. \quad (20)$$

For the second term, let $\{k_1 < k_2 < ... < k_t \leq K\}$ be the collection of round indices that $n_h^{k_i}(s,a) > 0$ for any $i \in [t]$. Then we have:

$$\sum_{i=1}^{N_h^k} \left( \tilde{\eta}_i^{N_h^k} - \eta_i^{N_h^k} \right) \left( \mathbb{1}_{s_{h+1}^{k^i,j^i,m^i}} - \mathbb{P}_{s,a,h} \right) W_{h+1}^{k^i}$$

$$= \sum_{n=1}^{t} \eta^c(N_h^{k_n+1} + 1, N_h^k) \sum_{i=N_h^{k_n}+1}^{N_h^{k_n+1}} \left( \tilde{\eta}_i^{N_h^{k_n+1}} - \eta_i^{N_h^{k_n+1}} \right) \left( \mathbb{1}_{s_{h+1}^{k_n,j^i,m^i}} - \mathbb{P}_{s,a,h} \right) W_{h+1}^{k_n}. \quad (21)$$

For $N_h^{k_n+1} < i_0$, since $\tilde{\eta}_i^{N_h^{k_n+1}} = \eta_i^{N_h^{k_n+1}}$,

$$\sum_{i=N_h^{k_n}+1}^{N_h^{k_n+1}} \left( \tilde{\eta}_i^{N_h^{k_n+1}} - \eta_i^{N_h^{k_n+1}} \right) \left( \mathbb{1}_{s_{h+1}^{k_n,j^i,m^i}} - \mathbb{P}_{s,a,h} \right) W_{h+1}^{k_n} = 0.$$

For $N_h^{k_n+1} \geq i_0$ and $i \in [N_h^{k_n} + 1, N_h^{k_n+1}]$, by (d) of Lemma F.2 and (a) and (b) in Lemma F.3, we know

$$\tilde{\eta}_i^{N_h^{k_n+1}}, \eta_i^{N_h^{k_n+1}} \in \left[ \eta_{N_h^{k_n}+1}^{N_h^{k_n+1}}, \eta_{N_h^{k_n+1}}^{N_h^{k_n+1}} \right] \text{ and thus } \left| \tilde{\eta}_i^{N_h^{k_n+1}} - \eta_i^{N_h^{k_n+1}} \right| \leq \eta_{N_h^{k_n+1}}^{N_h^{k_n+1}} - \eta_{N_h^{k_n}+1}^{N_h^{k_n+1}}.$$

Then for given $N_h^{k_n+1} = N_2 \in [i_0, T_1/H]$, $N_h^{k_n} = N_1 \in [0, N_2]$, and $(s,a,h,k_n,n) \in \mathcal{S} \times \mathcal{A} \times [H] \times [T_1/H] \times [T_1/H]$, using Theorem E.2 with $R = 2C_w(\eta_{N_2}^{N_2} - \eta_{N_1+1}^{N_2})$, $m = \lceil \log_2(N_2) \rceil$, $\sigma^2 = 4N_2(\eta_{N_2}^{N_2} - \eta_{N_1+1}^{N_2})^2 C_w^2$, with probability at least $1 - \delta/2SAT_1^4$, it holds that:

$$\left| \sum_{i=N_h^{k_n}+1}^{N_h^{k_n+1}} \left( \tilde{\eta}_i^{N_h^{k_n+1}} - \eta_i^{N_h^{k_n+1}} \right) \left( \mathbb{1}_{s_{h+1}^{k_n,j^i,m^i}} - \mathbb{P}_{s,a,h} \right) W_{h+1}^{k_n} \right|$$

$$\lesssim \sqrt{\sum_{i=N_1+1}^{N_2} \left( \tilde{\eta}_i^{N_2} - \eta_i^{N_2} \right)^2 \mathbb{V}_{s,a,h}(W_{h+1}^{k_n})\iota} + \left( \eta_{N_2}^{N_2} - \eta_{N_1+1}^{N_2} \right) C_w\iota$$

Consider all the possible combinations of $N_h^{k_n} = N_1 \in [T_1/H]$, $N_h^{k_n+1} = N_2 \in [T_1/H]$ and $(s,a,h,k_n) \in \mathcal{S} \times \mathcal{A} \times [H] \times [T_1/H]$, with probability at least $1 - \delta/2$, it simultaneously hold for all $(s,a,h,k_n,n) \in \mathcal{S} \times \mathcal{A} \times [H] \times [T_1/H] \times [T_1/H]$ that

$$\left| \sum_{i=N_h^{k_n}+1}^{N_h^{k_n+1}} \left( \tilde{\eta}_i^{N_h^{k_n+1}} - \eta_i^{N_h^{k_n+1}} \right) \left( \mathbb{1}_{s_{h+1}^{k_n,j^i,m^i}} - \mathbb{P}_{s,a,h} \right) W_{h+1}^{k_n} \right|$$

$$\lesssim \sqrt{\sum_{i=N_h^{k_n}+1}^{N_h^{k_n+1}} \left( \tilde{\eta}_i^{N_h^{k_n+1}} - \eta_i^{N_h^{k_n+1}} \right)^2 \mathbb{V}_{s,a,h}(W_{h+1}^{k_n})\iota} + \left( \eta_{N_h^{k_n+1}}^{N_h^{k_n+1}} - \eta_{N_h^{k_n}+1}^{N_h^{k_n+1}} \right) C_w\iota$$

$$\lesssim \sum_{i=N_h^{k_n}+1}^{N_h^{k_n+1}} \frac{\eta_i^{N_h^{k_n+1}}}{\sqrt{i}} \sqrt{\mathbb{V}_{s,a,h}(W_{h+1}^{k_n})\iota} + \left( \eta_{N_h^{k_n+1}}^{N_h^{k_n+1}} - \eta_{N_h^{k_n}+1}^{N_h^{k_n+1}} \right) C_w\iota.$$

Here, the last inequality is by (d) of Lemma F.3. Applying this inequality to Equation (21), then

$$
\left| \sum_{i=1}^{N_h^k} \left( \tilde{\eta}_i^{N_h^k} - \eta_i^{N_h^k} \right) \left( \mathbb{1}_{s_{h+1}^{k^i,j^i,m^i}} - \mathbb{P}_{s,a,h} \right) W_{h+1}^{k^i} \right|
$$

$$
\lesssim \sum_{n=1}^{t} \eta^c(N_h^{k_n+1} + 1, N_h^k) \left( \sum_{i=N_h^{k_n}+1}^{N_h^{k_n+1}} \frac{\eta_i^{N_h^{k_n+1}}}{\sqrt{i}} \cdot \sqrt{\mathbb{V}_{s,a,h}(W_{h+1}^{k_n})\iota} + \left( \eta_{N_h^{k_n+1}}^{N_h^{k_n+1}} - \eta_{N_h^{k_n}+1}^{N_h^{k_n+1}} \right) C_w \iota \right)
$$

$$
= \sum_{n=1}^{t} \sum_{i=N_h^{k_n}+1}^{N_h^{k_n+1}} \frac{\eta_i^{N_h^k}}{\sqrt{i}} \cdot \sqrt{\mathbb{V}_{s,a,h}(W_{h+1}^{k_n})\iota} + \sum_{n=2}^{t} \left( \eta_{N_h^{k_{n-1}+1}}^{N_h^k} - \eta_{N_h^{k_n}+1}^{N_h^k} \right) C_w \iota
$$

$$
+ \left( \eta_{N_h^{k_t+1}}^{N_h^k} - \eta_{N_h^{k_1}+1}^{N_h^k} \right) C_w \iota
$$

$$
\leq \sqrt{ \left( \sum_{i=1}^{N_h^k} \frac{\eta_i^{N_h^k}}{i} \right) \left( \sum_{i=1}^{N_h^k} \eta_i^{N_h^k} \mathbb{V}_{s,a,h}(W_{h+1}^{k^i}) \right) } \, \iota + \eta_{N_h^k}^{N_h^k} C_w \iota
$$

$$
\lesssim \sqrt{ \frac{\iota}{N_h^k} \left( \sum_{i=1}^{N_h^k} \eta_i^{N_h^k} \mathbb{V}_{s,a,h}(W_{h+1}^{k^i}) \right) } + \frac{H C_w \iota}{N_h^k}. \tag{22}
$$

In the second last inequality, we use the Cauchy-Schwarz inequality and the monotonicity of $\eta_i^{N_h^k}$. We also use $N_h^{k_t+1} = N_h^k$ because there is no visit to $(s, a, h)$ from round $k_t + 1$ to round $k - 1$. The last inequality is by (a) and (b) in Lemma F.2. Combining Equation (20) and Equation (22), we prove the first conclusion. Moreover, note that:

$$
\sum_{i=1}^{N_h^k} \eta_i^{N_h^k} \mathbb{V}_{s,a,h}(W_{h+1}^{k^i}) = \sum_{n=1}^{t} \left( \sum_{i=N_h^{k_n}+1}^{N_h^{k_n+1}} \eta_i^{N_h^k} \right) \mathbb{V}_{s,a,h}(W_{h+1}^{k_n})
$$

$$
= \sum_{n=1}^{t} \left( \sum_{i=N_h^{k_n}+1}^{N_h^{k_n+1}} \tilde{\eta}_i^{N_h^k} \right) \mathbb{V}_{s,a,h}(W_{h+1}^{k_n}) = \sum_{i=1}^{N_h^k} \tilde{\eta}_i^{N_h^k} \mathbb{V}_{s,a,h}(W_{h+1}^{k^i}).
$$

The second equality is because of (b) in Lemma F.3. Then we finish the proof. $\square$

**Lemma F.6.** *For any non-negative weight sequence $\{\omega_h^{k,j,m} | k \in [K], m \in [M], j \in [n^{m,k}]\}$ and $\alpha \in [0, 1)$, it holds for any $h \in [H]$ that:*

$$
\sum_{k,j,m,N_h^k>0} \frac{\omega_h^{k,j,m}}{N_h^k(s_h^{k,j,m}, a_h^{k,j,m})^\alpha}
$$

$$
\leq \sum_{k,j,m,N_h^k>0} \omega_h^{k,j,m} \frac{\mathbb{I}\left[0 < N_h^k(s_h^{k,j,m}, a_h^{k,j,m}) < M\right]}{N_h^k(s_h^{k,j,m}, a_h^{k,j,m})^\alpha} + \frac{2^\alpha}{1-\alpha}(SA\|\omega\|_{\infty,h})^\alpha \|\omega\|_{1,h}^{1-\alpha}
$$

$$
\leq 2MSA\|\omega\|_{\infty,h} + \frac{2^\alpha}{1-\alpha}(SA\|\omega\|_{\infty,h})^\alpha \|\omega\|_{1,h}^{1-\alpha}.
$$

*Here, $\|\omega\|_{\infty,h} = \max_{k,j,m}\{\omega_h^{k,j,m}\}$ and $\|\omega\|_{1,h} = \sum_{k,j,m} \omega_h^{k,j,m}$.*
*For $\alpha = 1$, for any $h \in [H]$, we have the following conclusion:*

$$
\sum_{k,j,m,N_h^k>0} \frac{1}{N_h^k(s_h^{k,j,m}, a_h^{k,j,m})} \leq 2MSA + 2SA\iota.
$$

*Proof.*

$$\sum_{k,j,m,N_h^k>0} \frac{\omega_h^{k,j,m}}{N_h^k(s_h^{k,j,m},a_h^{k,j,m})^\alpha} = \sum_{s,a} \sum_{k,j,m,N_h^k>0} \frac{\omega_h^{k,j,m}}{(N_h^k(s,a))^\alpha} \mathbb{I}[(s_h^{k,j,m},a_h^{k,j,m})=(s,a)].$$

We decompose the summation into two terms

$$\sum_{s,a} \sum_{k,j,m} \frac{\omega_h^{k,j,m}}{(N_h^k(s,a))^\alpha} \mathbb{I}[(s_h^{k,j,m},a_h^{k,j,m})=(s,a)] \left( \mathbb{I}\left[0<N_h^k<M\right] + \mathbb{I}\left[N_h^k \ge M\right] \right).$$

Let $k_0(s,a) = \max\{k \mid 1 \le k \le K, N_h^k(s,a) < M\}$. Then for the first term, it holds that

$$\sum_{k,j,m} \frac{\omega_h^{k,j,m}}{(N_h^k(s,a))^\alpha} \mathbb{I}[(s_h^{k,j,m},a_h^{k,j,m})=(s,a)]\mathbb{I}\left[0<N_h^k(s,a)<M\right]$$

$$\le \|\omega\|_{\infty,h} \sum_{k,j,m} \mathbb{I}[(s_h^{k,j,m},a_h^{k,j,m})=(s,a)]\mathbb{I}\left[0<N_h^k(s,a)<M\right]$$

$$= \|\omega\|_{\infty,h} \sum_{k=1}^{k_0}\sum_{j,m} \mathbb{I}[(s_h^{k,j,m},a_h^{k,j,m})=(s,a)]$$

$$= \|\omega\|_{\infty,h} N_h^{k_0+1}(s,a) \le 2M\|\omega\|_{\infty,h}, \tag{23}$$

and thus

$$\sum_{s,a} \sum_{k,j,m} \frac{\omega_h^{k,j,m}}{(N_h^k(s,a))^\alpha} \mathbb{I}[(s_h^{k,j,m},a_h^{k,j,m})=(s,a)]\mathbb{I}\left[0<N_h^k<M\right] \le 2MSA\|\omega\|_{\infty,h}. \tag{24}$$

For the second term, let

$$c_h(s,a) = \sum_{k,j,m} \omega_h^{k,j,m}\mathbb{I}[(s_h^{k,j,m},a_h^{k,j,m})=(s,a)]\mathbb{I}\left[N_h^k(s,a) \ge M\right]$$

$$= \sum_{k=k_0+1}^{K}\sum_{j,m} \omega_h^{k,j,m}\mathbb{I}[(s_h^{k,j,m},a_h^{k,j,m})=(s,a)].$$

Then we have $\sum_{s,a} c_h(s,a) \le \sum_{k,j,m} \omega_h^{k,j,m} = \|\omega\|_{1,h}$. Given the term

$$\sum_{k,j,m} \frac{\omega_h^{k,j,m}}{(N_h^k(s,a))^\alpha} \mathbb{I}[(s_h^{k,j,m},a_h^{k,j,m})=(s,a)],$$

when the weights $\omega_h^{k,j,m}$ concentrates on smallest round indices with largest values of $\frac{1}{(N_h^k(s,a))^\alpha}$, we can obtain the largest value. Let $k_0(s,a) < k_1 < k_2 < ... < k_t \le K$ be all round indices that satisfy $n_h^{k_i}(s,a) > 0$. Then we have:

$$c_h(s,a) \le \|\omega\|_{\infty,h} \sum_{k=k_0+1}^{K}\sum_{j,m} \mathbb{I}[(s_h^{k,j,m},a_h^{k,j,m})=(s,a)] = \|\omega\|_{\infty,h} \sum_{i=1}^{t} n_h^{k_i}(s,a).$$

Let

$$q = \max \left\{ q \mid 0 \le q \le t, \|\omega\|_{\infty,h} \sum_{i=1}^{q} n_h^{k_i}(s,a) \le c_h(s,a) \right\},$$

and

$$d = c_h(s,a) - \|\omega\|_{\infty,h} \sum_{i=1}^{q} n_h^{k_i}(s,a).$$

Then for $q < t$, we have the following inequality:

$$\sum_{k,j,m} \frac{\omega_h^{k,j,m}}{(N_h^k(s,a))^\alpha} \mathbb{I}[(s_h^{k,j,m}, a_h^{k,j,m}) = (s,a)] \mathbb{I}\left[N_h^k(s,a) \geq M\right]$$

$$\leq \sum_{i=1}^q \|\omega\|_{\infty,h} \frac{n_h^{k_i}(s,a)}{(N_h^{k_i}(s,a))^\alpha} + \frac{d}{(N_h^{k_{q+1}}(s,a))^\alpha}. \tag{25}$$

Note that for any $0 < y < x$ and $\alpha \in [0,1)$, we have:

$$\frac{x-y}{x^\alpha} \leq \frac{1}{1-\alpha}(x^{1-\alpha} - y^{1-\alpha}). \tag{26}$$

Then, for any $i \in [t]$, let $x = N_h^{k_i}(s,a)$ and $y = N_h^{k_i+1}(s,a)$, it holds that:

$$\frac{n_h^{k_i}(s,a)}{(N_h^{k_i}(s,a))^\alpha} \leq 2^\alpha \frac{n_h^{k_i}(s,a)}{(N_h^{k_i+1}(s,a))^\alpha} \leq 2^\alpha \left(\frac{(N_h^{k_i+1}(s,a))^{1-\alpha} - (N_h^{k_i}(s,a))^{1-\alpha}}{1-\alpha}\right). \tag{27}$$

Here the first inequality is because $N_h^{k_i+1}(s,a) = N_h^{k_i}(s,a) + n_h^{k_i}(s,a) \leq 2N_h^{k_i}(s,a)$ by (a) of Lemma F.1. Summing up Equation (27) from 1 to $q$, we know

$$\sum_{i=1}^q \frac{n_h^{k_i}(s,a)}{(N_h^{k_i}(s,a))^\alpha} \leq 2^\alpha \sum_{i=1}^q \frac{(N_h^{k_i+1}(s,a))^{1-\alpha} - (N_h^{k_i}(s,a))^{1-\alpha}}{1-\alpha}$$

$$\leq 2^\alpha \sum_{i=1}^q \frac{(N_h^{k_{i+1}}(s,a))^{1-\alpha} - (N_h^{k_i}(s,a))^{1-\alpha}}{1-\alpha}$$

$$= 2^\alpha \left(\frac{(N_h^{k_{q+1}}(s,a))^{1-\alpha}}{1-\alpha} - \frac{(N_h^{k_1}(s,a))^{1-\alpha}}{1-\alpha}\right)$$

$$\leq 2^\alpha \frac{\left(\sum_{i=1}^q n_h^{k_i}(s,a)\right)^{1-\alpha}}{1-\alpha}. \tag{28}$$

The second inequality is because $k_i + 1 \leq k_{i+1}$ and thus $N_h^{k_i+1}(s,a) \leq N_h^{k_{i+1}}(s,a)$. The last inequality is because for any $x > 1$ and $0 \leq \alpha < 1$, we have the following inequality

$$x^{1-\alpha} \leq (x-1)^{1-\alpha} + 1,$$

and we can let $x = N_h^{k_{q+1}}(s,a)/N_h^{k_1}(s,a)$. Applying Equation (28) to Equation (25), for $q < t$, we have

$$\sum_{k,j,m} \frac{\omega_h^{k,j,m}}{(N_h^k(s,a))^\alpha} \mathbb{I}[(s_h^{k,j,m}, a_h^{k,j,m}) = (s,a)] \mathbb{I}\left[N_h^k(s,a) \geq M\right]$$

$$\leq 2^\alpha \|\omega\|_{\infty,h} \frac{\left(\sum_{i=1}^q n_h^{k_i}(s,a)\right)^{1-\alpha}}{1-\alpha} + \frac{d}{(N_h^{k_{q+1}}(s,a))^\alpha}$$

$$\leq (2\|\omega\|_{\infty,h})^\alpha \left(\frac{\left(\|\omega\|_{\infty,h} \sum_{i=1}^q n_h^{k_i}(s,a)\right)^{1-\alpha}}{1-\alpha} + \frac{d}{(N_h^{k_{q+1}+1}(s,a)\|\omega\|_{\infty,h})^\alpha}\right)$$

$$\leq (2\|\omega\|_{\infty,h})^\alpha \left(\frac{\left(\|\omega\|_{\infty,h} \sum_{i=1}^q n_h^{k_i}(s,a)\right)^{1-\alpha}}{1-\alpha} + \frac{d}{(c_h(s,a))^\alpha}\right)$$

$$\leq (2\|\omega\|_{\infty,h})^\alpha \frac{(c_h(s,a))^{1-\alpha}}{1-\alpha}. \tag{29}$$

Here the second inequality is because $N_h^{k_{q+1}+1}(s,a) \leq 2N_h^{k_{q+1}}(s,a)$ for $q < t$ and $N_h^{k_{q+1}}(s,a) \geq M$. the second last inequality is because $c_h(s,a) \leq N_h^{k_{q+1}+1}(s,a)\|\omega\|_{\infty,h}$ by the definition of $q$. The last inequality is by Equation (26) with $x = c_h(s,a)$ and $y = \|\omega\|_{\infty,h}\sum_{i=1}^{q} n_h^{k_i}(s,a)$.

We can also prove the Equation (29) directly by Equation (28) for $q = t$ with $d = 0$. Therefore, with Equation (29), we can conclude that

$$\sum_{s,a}\sum_{k,j,m}\frac{\omega_h^{k,j,m}}{(N_h^k(s,a))^\alpha}\mathbb{I}[(s_h^{k,j,m},a_h^{k,j,m}) = (s,a)]\mathbb{I}\left[N_h^k(s,a) \geq M\right]$$

$$\leq \frac{2^\alpha\|\omega\|_{\infty,h}^\alpha}{1-\alpha}\sum_{s,a}(c_h(s,a))^{1-\alpha} \leq \frac{2^\alpha}{1-\alpha}(SA)^\alpha\|\omega\|_{\infty,h}^\alpha\|\omega\|_{1,h}^{1-\alpha}. \tag{30}$$

The last inequality is by Hölder's inequality, as $\sum_{s,a}c_h(s,a)^{1-\alpha} \leq (SA)^\alpha\|\omega\|_{1,h}^{1-\alpha}$. Combining the results of Equation (23) and Equation (30), we prove the following conclusion:

$$\sum_{k,j,m,N_h^k>0}\frac{\omega_h^{k,j,m}}{N_h^k(s_h^{k,j,m},a_h^{k,j,m})^\alpha} \leq 2MSA\|\omega\|_{\infty,h} + \frac{2^\alpha}{1-\alpha}(SA)^\alpha\|\omega\|_{\infty,h}^\alpha\|\omega\|_{1,h}^{1-\alpha}.$$

For $\alpha = 1$, it holds that:

$$\sum_{k,j,m}\frac{1}{N_h^k(s_h^{k,j,m},a_h^{k,j,m})} = \sum_{s,a}\sum_{k=1}^{K}\frac{n_h^k(s,a)}{N_h^k(s,a)} = \sum_{s,a}\sum_{k=1}^{k_0}\frac{n_h^k(s,a)}{N_h^k(s,a)} + \sum_{s,a}\sum_{k=k_0+1}^{K}\frac{n_h^k(s,a)}{N_h^k(s,a)}.$$

Let $\omega_h^{k,j,m} = 1$. By Equation (23), we know

$$\sum_{k=1}^{k_0}\frac{n_h^k(s,a)}{N_h^k(s,a)} \leq 2M, \quad \sum_{s,a}\sum_{k=1}^{k_0}\frac{n_h^k(s,a)}{N_h^k(s,a)} \leq 2MSA. \tag{31}$$

We also have

$$\sum_{k=k_0+1}^{K}\frac{n_h^k(s,a)}{N_h^k(s,a)} \leq 2\sum_{k=k_0+1}^{K}\frac{n_h^k(s,a)}{N_h^{k+1}(s,a)}$$

$$\leq 2\sum_{k=k_0+1}^{K}\left(\log(N_h^{k+1}(s,a)) - \log(N_h^k(s,a))\right)$$

$$\leq 2\log(T_1). \tag{32}$$

Here the first inequality is because for $k > k_0(s,a)$, $N_h^k(s,a) \geq M$, we have $N_h^{k+1}(s,a) \leq 2N_h^k(s,a)$. The second inequality is because for $0 < y < x$,

$$\frac{x-y}{x} \leq \log(x) - \log(y).$$

The last inequality is because $N_h^{K+1}(s,a) \leq T_1$. To summarize, combining the results of Equation (31) and Equation (32), we then prove that

$$\sum_{k,j,m}\frac{1}{N_h^k(s_h^{k,j,m},a_h^{k,j,m})} \leq MSA + 2SA\log(T_1) \leq MSA + 2SA\iota.$$

$\square$

# G   Probability Events

In this section, we provide some probability events for FedQ-EarlySettled-LowCost.

**Lemma G.1.** *Define* $\hat{V}_h^{R,k}(s) := \max\left\{V_h^\star(s), \min\{V_h^\star(s) + \beta, V_h^{R,k}(s)\}\right\}$ *for any* $(s,h,k) \in \mathcal{S} \times [H] \times [K]$. *Using* $\forall(s,a,h,k)$ *as the simplified notation for* $\forall(s,a,h,k) \in \mathcal{S} \times \mathcal{A} \times [H] \times [K]$. *Then we have the following conclusions.*

*(a)* ([113, Lemma C.1]) *With probability at least $1 - \delta$, the following event holds:*

$$\mathcal{E}_1 = \left\{ Q_h^{\mathrm{U},k}(s,a) \geq Q_h^\star(s,a), \forall (s,a,h,k) \right\}.$$

*(b) With probability at least $1 - \delta$, the following event holds:*

$$\mathcal{E}_2 = \left\{ \left| \sum_{n=1}^{N_h^k} \tilde{\eta}_n^{N_h^k} \left( \mathbb{1}_{s_{h+1}^{k^n,j^n,m^n}} - \mathbb{P}_{s,a,h} \right) \left( V_{h+1}^{k^n} - V_{h+1}^{\mathrm{R},k^n} \right) \right| \right.$$
$$\left. \lesssim \sqrt{ \frac{H\iota}{N_h^k} \left( \sigma_h^{\mathrm{A},k} - \left( \mu_h^{\mathrm{A},k} \right)^2 \right) } + \frac{H^2\iota}{N_h^k}, \ \forall (s,a,h,k) \right\}.$$

*(c) With probability at least $1 - \delta$, the following event holds:*

$$\mathcal{E}_3 = \left\{ \left| \sum_{n=1}^{N_h^k} \frac{\tilde{\eta}_n^{N_h^k}}{N_h^{k^n+1}} \sum_{i=1}^{N_h^{k^n+1}} \left( \mathbb{1}_{s_{h+1}^{k^i,j^i,m^i}} - \mathbb{P}_{s,a,h} \right) V_{h+1}^{\mathrm{R},k^i} \right| \right.$$
$$\left. \lesssim \sqrt{ \frac{\iota}{N_h^k} \left( \sigma_h^{\mathrm{R},k} - \left( \mu_h^{\mathrm{R},k} \right)^2 \right) } + \frac{H^2\iota}{N_h^k}, \ \forall (s,a,h,k) \right\}.$$

*(d)* ([113, Lemma C.3]) *With probability at least $1 - \delta$, the following event holds:*

$$\mathcal{E}_4 = \left\{ \sum_{n=1}^{N_h^k} \tilde{\eta}_n^{N_h^k} \left( \mathbb{1}_{s_{h+1}^{k^n,j^n,m^n}} - \mathbb{P}_{s,a,h} \right) V_{h+1}^\star \lesssim \sqrt{\frac{H^3\iota}{N_h^k}}, \ \forall (s,a,h,k) \right\}.$$

*(e) With probability at least $1 - \delta$, the following event holds:*

$$\mathcal{E}_5 = \left\{ \sum_{h=1}^H (e^{\frac{3}{H}})^{h-1} \sum_{k,j,m} \left( \mathbb{P}_{s_h^{k,j,m},a_h^{k,j,m},h} - \mathbb{1}_{s_{h+1}^{k,j,m}} \right) (V_{h+1}^\star - V_{h+1}^{\pi^k}) \leq 27\sqrt{2H^2 T_1 \iota} \right\}.$$

*(f) With probability at least $1 - \delta$, the following event holds:*

$$\mathcal{E}_6 = \left\{ \frac{\sum_{n=1}^{N_h^k} \left( \mathbb{1}_{s_{h+1}^{k^n,j^n,m^n}} - \mathbb{P}_{s,a,h} \right) V_{h+1}^\star}{N_h^k(s,a)} \leq H\sqrt{\frac{2\iota}{N_h^k(s,a)}}, \ \forall (s,a,h,k) \right\}.$$

*(g) With probability at least $1 - \delta$, the following event holds:*

$$\mathcal{E}_7 = \left\{ \frac{\sum_{n=1}^{N_h^k} \left( \mathbb{1}_{s_{h+1}^{k^n,j^n,m^n}} - \mathbb{P}_{s,a,h} \right) \left( V_{h+1}^\star \right)^2}{N_h^k(s,a)} \leq H^2\sqrt{\frac{2\iota}{N_h^k(s,a)}}, \ \forall (s,a,h,k) \right\}.$$

*(h)* ([113, Lemma E.6])*With probability at least $1 - \delta$, the following event holds:*

$$\mathcal{E}_8 = \left\{ \sum_{h=1}^H \sum_{k,j,m} \mathbb{V}_{s_h^{k,j,m},a_h^{k,j,m},h}(V_{h+1}^{\pi^k}) \lesssim HT_1 + H^3\iota \right\}.$$

*(i) With probability at least $1 - \delta$, the following event holds:*

$$\mathcal{E}_9 = \left\{ \sum_{h=1}^H \sum_{k,j,m} \left( \mathbb{P}_{s_h^{k,j,m},a_h^{k,j,m},h} - \mathbb{1}_{s_{h+1}^{k,j,m}} \right) \left( V_{h+1}^\star - V_{h+1}^{\pi^k} \right) \leq \sqrt{2H^2 T_1 \iota} \right\}.$$

*(j)* *With probability at least* $1 - \delta$, *the following event holds:*

$$\mathcal{E}_{10} = \left\{ \sum_{h=1}^{H} \sum_{k,j,m} \left( V_{h+1}^{\mathrm{U},k} - V_{h+1}^{\pi^k} \right) (s_{h+1}^{k,j,m}) \leq H^3 \sqrt{SAT_1 \iota} + MH^5 SA \sqrt{\iota} \right\}.$$

*(k)* *With probability at least* $1 - \delta$, *the following event holds:*

$$\mathcal{E}_{11} = \left\{ \sum_{h,k,j,m} \mathbb{P}_{s_h^{k,j,m}, a_h^{k,j,m}, h} \left\{ (V_{h+1}^k - V_{h+1}^{\mathrm{L},k})(s_{h+1}^{k,j,m}) \mathbb{I}[(V_{h+1}^k - V_{h+1}^{\mathrm{L},k})(s_{h+1}^{k,j,m}) > \beta] \right\} \right.$$

$$\left. \leq 3 \sum_{h=1}^{H} \sum_{k,j,m} \left( V_{h+1}^k - V_{h+1}^{\mathrm{L},k} \right)(s_{h+1}^{k,j,m}) \mathbb{I} \left[ \left( V_{h+1}^k - V_{h+1}^{\mathrm{L},k} \right)(s_{h+1}^{k,j,m}) > \beta \right] + H\iota \right\}.$$

*(l)* *With probability at least* $1 - \delta$, *the following event holds:*

$$\mathcal{E}_{12} = \left\{ \left| \frac{1}{N_h^{k^n+1}} \sum_{i=1}^{N_h^{k^n+1}} \left( \mathbb{1}_{s_{h+1}^{k^i,j^i,m^i}} - \mathbb{P}_{s,a,h} \right) \left( \hat{V}_{h+1}^{\mathrm{R},k^i} - V_{h+1}^{\star} \right) \right| \right.$$

$$\left. \leq \beta \sqrt{\frac{2\iota}{N_h^{k^n+1}}}, \ \forall (s,a,h,k,n) \right\}.$$

*(m)* *With probability at least* $1 - \delta$, *the following event holds:*

$$\mathcal{E}_{13} = \left\{ \left| \frac{1}{N_h^{k^n+1}} \sum_{i=1}^{N_h^{k^n+1}} \left( \mathbb{1}_{s_{h+1}^{k^i,j^i,m^i}} - \mathbb{P}_{s,a,h} \right) V_{h+1}^{\star} \right| \right.$$

$$\left. \leq 4 \sqrt{\frac{\mathbb{V}_{s,a,h}(V_{h+1}^{\star})\iota}{N_h^{k^n+1}}} + \frac{7H\iota}{N_h^{k^n+1}}, \ \forall (s,a,h,k,n) \right\}.$$

*(n)* *With probability at least* $1 - \delta$, *the following event holds:*

$$\mathcal{E}_{14} = \left\{ \left| \sum_{n=1}^{N_h^k} \tilde{\eta}_n^{N_h^k} \left( \mathbb{1}_{s_{h+1}^{k^n,j^n,m^n}} - \mathbb{P}_{s_h^{k,j,m}, a_h^{k,j,m}, h} \right) \left( V_{h+1}^{\star} - \hat{V}_{h+1}^{\mathrm{R},k^n} \right) \right| \right.$$

$$\left. \lesssim \beta \sqrt{\frac{H\iota}{N_h^k}} + \frac{\beta H\iota}{N_h^k}, \ \forall (s,a,h,k) \right\}.$$

*Proof.* (b) Using the Lemma F.5, with probability at least $1 - \delta/2$, we know that for $\forall (s,a,h,k)$:

$$\left| \sum_{n=1}^{N_h^k} \tilde{\eta}_n^{N_h^k} \left( \mathbb{1}_{s_{h+1}^{k^n,j^n,m^n}} - \mathbb{P}_{s,a,h} \right) \left( V_{h+1}^{k^n} - V_{h+1}^{\mathrm{R},k^n} \right) \right|$$

$$\lesssim \sqrt{\frac{H\iota}{N_h^k} \sum_{n=1}^{N_h^k} \tilde{\eta}_n^{N_h^k} \mathbb{V}_{s,a,h} \left( V_{h+1}^{k^n} - V_{h+1}^{\mathrm{R},k^n} \right)} + \frac{H^2\iota}{N_h^k}. \tag{33}$$

Next we will bound the difference

$$I_1 = \sum_{n=1}^{N_h^k} \tilde{\eta}_n^{N_h^k} \mathbb{V}_{s,a,h} \left( V_{h+1}^{k^n} - V_{h+1}^{\mathrm{R},k^n} \right) - \left( \sigma_h^{\mathrm{A},k} - (\mu_h^{\mathrm{A},k})^2 \right) \triangleq X_{\mathrm{A}} - \left( \sigma_h^{\mathrm{A},k} - (\mu_h^{\mathrm{A},k})^2 \right),$$

where

$$X_{\mathrm{A}} = \sum_{n=1}^{N_h^k} \tilde{\eta}_n^{N_h^k} \mathbb{V}_{s,a,h} \left( V_{h+1}^{k^n} - V_{h+1}^{\mathrm{R},k^n} \right).$$

Based on the update rule of Equation (8) and Equation (12), by recursion, we have:

$$\mu_h^{\text{A},k} = \sum_{n=1}^{N_h^k} \tilde{\eta}_n^{N_h^k} \left( V_{h+1}^{k^n} - V_{h+1}^{\text{R},k^n} \right) \left( s_{h+1}^{k^n,j^n,m^n} \right), \sigma_h^{\text{A},k} = \sum_{n=1}^{N_h^k} \tilde{\eta}_n^{N_h^k} \left( V_{h+1}^{k^n} - V_{h+1}^{\text{R},k^n} \right)^2 \left( s_{h+1}^{k^n,j^n,m^n} \right).$$

$$(34)$$

According to the definition of $\mathbb{V}_{s,a,h}$, we also have

$$\mathbb{V}_{s,a,h} \left( V_{h+1}^{k^n} - V_{h+1}^{\text{R},k^n} \right) = \mathbb{P}_{s,a,h} \left( V_{h+1}^{k^n} - V_{h+1}^{\text{R},k^n} \right)^2 - \left( \mathbb{P}_{s,a,h} \left( V_{h+1}^{k^n} - V_{h+1}^{\text{R},k^n} \right) \right)^2. \quad (35)$$

Combining the results of Equation (34) and Equation (35), we can decompose the difference $I_1$:

$$I_1 = \sum_{n=1}^{N_h^k} \tilde{\eta}_n^{N_h^k} \left( \mathbb{P}_{s,a,h} - \mathbb{1}_{s_{h+1}^{k^n,j^n,m^n}} \right) \left( V_{h+1}^{k^n} - V_{h+1}^{\text{R},k^n} \right)^2$$

$$+ \left[ \left( \sum_{n=1}^{N_h^k} \tilde{\eta}_n^{N_h^k} (V_{h+1}^{k^n} - V_{h+1}^{\text{R},k^n}) \left( s_{h+1}^{k^n,j^n,m^n} \right) \right)^2 - \sum_{n=1}^{N_h^k} \tilde{\eta}_n^{N_h^k} \left( \mathbb{P}_{s,a,h} (V_{h+1}^{k^n} - V_{h+1}^{\text{R},k^n}) \right)^2 \right]. \quad (36)$$

For the first term of Equation (36), with Lemma F.5, with probability at least $1 - \delta/4$, it holds for $\forall (s,a,h,k)$ that:

$$\left| \sum_{n=1}^{N_h^k} \tilde{\eta}_n^{N_h^k} \left( \mathbb{P}_{s,a,h} - \mathbb{1}_{s_{h+1}^{k^n,j^n,m^n}} \right) \left( V_{h+1}^{k^n} - V_{h+1}^{\text{R},k^n} \right)^2 \right|$$

$$\lesssim \sqrt{ \frac{H\iota}{N_h^k} \sum_{n=1}^{N_h^k} \tilde{\eta}_n^{N_h^k} \mathbb{V}_{s,a,h} \left( V_{h+1}^{k^n} - V_{h+1}^{\text{R},k^n} \right)^2 + \frac{H^3\iota}{N_h^k} }$$

$$\lesssim \sqrt{ \frac{H^3\iota}{N_h^k} \sum_{n=1}^{N_h^k} \tilde{\eta}_n^{N_h^k} \mathbb{V}_{s,a,h} \left( V_{h+1}^{k^n} - V_{h+1}^{\text{R},k^n} \right) + \frac{H^3\iota}{N_h^k} }$$

$$= \sqrt{ \frac{H^3\iota}{N_h^k} X_{\text{A}} } + \frac{H^3\iota}{N_h^k}. \quad (37)$$

The last inequality is by $\mathbb{V}_{s,a,h}(X^2) \leq 4C^2 \mathbb{V}_{s,a,h}(X)$ for $|X| \leq C$. For the second term of Equation (36), since $V_{h+1}^{k^n} \leq V_{h+1}^{\text{R},k^n}$, by Cauchy-Schwarz inequality, we reach

$$\left( \sum_{n=1}^{N_h^k} \tilde{\eta}_n^{N_h^k} \left( V_{h+1}^{k^n} - V_{h+1}^{\text{R},k^n} \right) \left( s_{h+1}^{k^n,j^n,m^n} \right) \right)^2 - \sum_{n=1}^{N_h^k} \tilde{\eta}_n^{N_h^k} \left( \mathbb{P}_{s,a,h} (V_{h+1}^{k^n} - V_{h+1}^{\text{R},k^n}) \right)^2$$

$$\leq \left( \sum_{n=1}^{N_h^k} \tilde{\eta}_n^{N_h^k} \left( V_{h+1}^{k^n} - V_{h+1}^{\text{R},k^n} \right) \left( s_{h+1}^{k^n,j^n,m^n} \right) \right)^2 - \left( \sum_{n=1}^{N_h^k} \tilde{\eta}_n^{N_h^k} \mathbb{P}_{s,a,h} \left( V_{h+1}^{k^n} - V_{h+1}^{\text{R},k^n} \right) \right)^2$$

$$\leq 2H \left| \sum_{n=1}^{N_h^k} \tilde{\eta}_n^{N_h^k} \left( \mathbb{1}_{s_{h+1}^{k^n,j^n,m^n}} - \mathbb{P}_{s,a,h} \right) \left( V_{h+1}^{k^n} - V_{h+1}^{\text{R},k^n} \right) \right|$$

$$\lesssim \sqrt{ \frac{H^3\iota}{N_h^k} X_{\text{A}} } + \frac{H^3\iota}{N_h^k}. \quad (38)$$

The last inequality holds for $\forall (s,a,h,k)$ with probability at least $1 - \delta/4$ by Lemma F.5. Combining Equation (37) and Equation (38), back to Equation (36), with probability at least $1 - \delta/2$, we know

$$I_1 = X_{\text{A}} - \left( \sigma_h^{\text{A},k} - (\mu_h^{\text{A},k})^2 \right) \lesssim \sqrt{ \frac{H^3\iota}{N_h^k} X_{\text{A}} } + \frac{H^3\iota}{N_h^k}.$$

Solving the inequality, with probability at least $1 - \delta/2$, we have

$$X_{\mathrm{A}} \lesssim \left(\sigma_h^{\mathrm{A},k} - (\mu_h^{\mathrm{A},k})^2\right) + \frac{H^3\iota}{N_h^k}.$$

Applying this inequality to Equation (33), with probability at least $1 - \delta$, the following relationship holds for $\forall(s, a, h, k)$:

$$\sum_{n=1}^{N_h^k} \tilde{\eta}_n^{N_h^k} \left(\mathbb{1}_{s_{h+1}^{k^n,j^n,m^n}} - \mathbb{P}_{s,a,h}\right) \left(V_{h+1}^{k^n} - V_{h+1}^{\mathrm{R},k^n}\right) \lesssim \sqrt{\frac{H\iota}{N_h^k}\left(\sigma_h^{\mathrm{A},k} - \left(\mu_h^{\mathrm{A},k}\right)^2\right)} + \frac{H^2\iota}{N_h^k}.$$

(c) To begin, note that

$$\sum_{n=1}^{N_h^k} \frac{\tilde{\eta}_n^{N_h^k}}{N_h^{k^n+1}} \sum_{i=1}^{N_h^{k^n+1}} \left(\mathbb{1}_{s_{h+1}^{k^i,j^i,m^i}} - \mathbb{P}_{s,a,h}\right) V_{h+1}^{\mathrm{R},k^i}$$

$$= \sum_{k'=1}^{k-1} \left(\sum_{n=N_h^{k'}+1}^{N_h^{k'+1}} \tilde{\eta}_i^{N_h^k}\right) \frac{1}{N_h^{k'+1}} \sum_{i=1}^{N_h^{k'+1}} \left(\mathbb{1}_{s_{h+1}^{k^i,j^i,m^i}} - \mathbb{P}_{s,a,h}\right) V_{h+1}^{\mathrm{R},k^i}$$

$$= \sum_{k'=1}^{k-1} \left(\sum_{n=N_h^{k'}+1}^{N_h^{k'+1}} \eta_i^{N_h^k}\right) \frac{1}{N_h^{k'+1}} \sum_{i=1}^{N_h^{k'+1}} \left(\mathbb{1}_{s_{h+1}^{k^i,j^i,m^i}} - \mathbb{P}_{s,a,h}\right) V_{h+1}^{\mathrm{R},k^i}$$

$$= \sum_{n=1}^{N_h^k} \frac{\eta_n^{N_h^k}}{N_h^{k^n+1}} \sum_{i=1}^{N_h^{k^n+1}} \left(\mathbb{1}_{s_{h+1}^{k^i,j^i,m^i}} - \mathbb{P}_{s,a,h}\right) V_{h+1}^{\mathrm{R},k^i}. \tag{39}$$

For any given $(s, a, h)$ and $N_h^{k^n+1} = N \in [T_1/H]$, using Lemma F.4 with $C_u(N_h^{k^n}) = 1/N_h^{k^n+1}$ and $C_w = 2H$, with probability at least $1 - \delta/2SAT_1$, it holds that:

$$\left|\frac{1}{N_h^{k^n+1}} \sum_{i=1}^{N_h^{k^n+1}} \left(\mathbb{1}_{s_{h+1}^{k^i,j^i,m^i}} - \mathbb{P}_{s,a,h}\right) V_{h+1}^{\mathrm{R},k^i}\right| \lesssim \sqrt{\frac{\iota}{N_h^{k^n+1}} \sum_{i=1}^{N_h^{k^n+1}} \frac{\mathbb{V}_{s,a,h}(V_{h+1}^{\mathrm{R},k^i})}{N_h^{k^n+1}}} + \frac{H\iota}{N_h^{k^n+1}}. \tag{40}$$

Consider all the possible combinations $(s, a, h, N) \in \mathcal{S} \times \mathcal{A} \times [H] \times [\frac{T_1}{H}]$, we know with probability at least $1 - \delta/2$, Equation (40) holds simultaneously for any $(n, s, a, h, k^n)$ and $N_h^{k^n+1} = N \in [T_1/H]$. Therefore, applying this inequality to Equation (39), with probability at least $1 - \delta/2$, we then have:

$$\left|\sum_{n=1}^{N_h^k} \frac{\tilde{\eta}_n^{N_h^k}}{N_h^{k^n+1}} \sum_{i=1}^{N_h^{k^n+1}} \left(\mathbb{1}_{s_{h+1}^{k^i,j^i,m^i}} - \mathbb{P}_{s,a,h}\right) V_{h+1}^{\mathrm{R},k^i}\right|$$

$$\lesssim \sum_{n=1}^{N_h^k} \eta_n^{N_h^k} \left(\sqrt{\frac{\iota}{N_h^{k^n+1}} \sum_{i=1}^{N_h^{k^n+1}} \frac{\mathbb{V}_{s,a,h}(V_{h+1}^{\mathrm{R},k^i})}{N_h^{k^n+1}}} + \frac{H\iota}{N_h^{k^n+1}}\right). \tag{41}$$

For the first term of Equation (41), by Cauchy-Schwarz inequality, it holds that:

$$\sum_{n=1}^{N_h^k} \eta_n^{N_h^k} \sqrt{\frac{\iota}{N_h^{k^n+1}} \sum_{i=1}^{N_h^{k^n+1}} \frac{\mathbb{V}_{s,a,h}(V_{h+1}^{\mathrm{R},k^i})}{N_h^{k^n+1}}} \leq \sqrt{\left(\sum_{n=1}^{N_h^k} \frac{\eta_n^{N_h^k}\iota}{N_h^{k^n+1}}\right) \left(\sum_{n=1}^{N_h^k} \eta_n^{N_h^k} \sum_{i=1}^{N_h^{k^n+1}} \frac{\mathbb{V}_{s,a,h}(V_{h+1}^{\mathrm{R},k^i})}{N_h^{k^n+1}}\right)}. \tag{42}$$

By the definition of $k^n$, we know $N_h^{k^n+1} \geq n$ and then by (a) of Lemma F.2 with $\alpha = 1$, we have

$$\sum_{n=1}^{N_h^k} \frac{\eta_n^{N_h^k}\iota}{N_h^{k^n+1}} \leq \sum_{n=1}^{N_h^k} \frac{\eta_n^{N_h^k}\iota}{n} \lesssim \frac{\iota}{N_h^k}, \tag{43}$$

and

$$\sum_{n=1}^{N_h^k} \eta_n^{N_h^k} \sum_{i=1}^{N_h^{k^n+1}} \frac{\mathbb{V}_{s,a,h}(V_{h+1}^{\mathrm{R},k^i})}{N_h^{k^n+1}} = \sum_{i=1}^{N_h^k} \sum_{C_n} \frac{\eta_n^{N_h^k}}{N_h^{k^n+1}} \mathbb{V}_{s,a,h}(V_{h+1}^{\mathrm{R},k^i}) \lesssim \sum_{i=1}^{N_h^k} \frac{1}{N_h^k} \mathbb{V}_{s,a,h}(V_{h+1}^{\mathrm{R},k^i}). \quad (44)$$

Here, $C_n = \{n : N_h^{k^n+1} \geq i, n \leq N_h^k\}$. Applying Equation (43) and Equation (44) to Equation (42), then we can bound the first term of Equation (41):

$$\sum_{n=1}^{N_h^k} \eta_n^{N_h^k} \sqrt{\frac{\iota}{N_h^{k^n+1}} \sum_{i=1}^{N_h^{k^n+1}} \frac{\mathbb{V}_{s,a,h}(V_{h+1}^{\mathrm{R},k^i})}{N_h^{k^n+1}}} \lesssim \sqrt{\frac{\iota}{N_h^k} \sum_{i=1}^{N_h^k} \frac{1}{N_h^k} \mathbb{V}_{s,a,h}(V_{h+1}^{\mathrm{R},k^i})}.$$

For the second term of Equation (41), same to Equation (43), we have:

$$\sum_{n=1}^{N_h^k} \eta_n^{N_h^k} \frac{H\iota}{N_h^{k^n+1}} \lesssim \frac{H\iota}{N_h^k}.$$

Applying these two upper bounds to Equation (41), with probability at least $1 - \delta/2$, we know that

$$\left| \sum_{n=1}^{N_h^k} \frac{\tilde{\eta}_n^{N_h^k}}{N_h^{k^n+1}} \sum_{i=1}^{N_h^{k^n+1}} \left( \mathbb{1}_{s_{h+1}^{k^i,j^i,m^i}} - \mathbb{P}_{s,a,h} \right) V_{h+1}^{\mathrm{R},k^i} \right| \lesssim \sqrt{\frac{\iota}{N_h^k} \sum_{n=1}^{N_h^k} \frac{1}{N_h^k} \mathbb{V}_{s,a,h}(V_{h+1}^{\mathrm{R},k^n})} + \frac{H\iota}{N_h^k}. \quad (45)$$

Next we will bound the difference

$$I_2 = \sum_{n=1}^{N_h^k} \frac{1}{N_h^k} \mathbb{V}_{s,a,h}(V_{h+1}^{\mathrm{R},k^n}) - \left( \sigma_h^{\mathrm{R},k} - (\mu_h^{\mathrm{R},k})^2 \right) \triangleq X_{\mathrm{R}} - \left( \sigma_h^{\mathrm{R},k} - (\mu_h^{\mathrm{R},k})^2 \right),$$

where

$$X_{\mathrm{R}} = \sum_{n=1}^{N_h^k} \frac{1}{N_h^k} \mathbb{V}_{s,a,h}(V_{h+1}^{\mathrm{R},k^n}).$$

Based on the update rule of Equation (6), by recursion, we have:

$$\mu_h^{\mathrm{R},k} = \sum_{n=1}^{N_h^k} \frac{1}{N_h^k} V_{h+1}^{\mathrm{R},k^n}\left( s_{h+1}^{k^n,j^n,m^n} \right), \quad \sigma_h^{\mathrm{R},k} = \sum_{n=1}^{N_h^k} \frac{1}{N_h^k} \left( V_{h+1}^{\mathrm{R},k^n} \right)^2 \left( s_{h+1}^{k^n,j^n,m^n} \right). \quad (46)$$

According to the definition of $\mathbb{V}_{s,a,h}$, we also have

$$\mathbb{V}_{s,a,h}\left( V_{h+1}^{\mathrm{R},k^n} \right) = \mathbb{P}_{s,a,h}\left( V_{h+1}^{\mathrm{R},k^n} \right)^2 - \left( \mathbb{P}_{s,a,h} V_{h+1}^{\mathrm{R},k^n} \right)^2. \quad (47)$$

Combining the results of Equation (46) and Equation (47), we can decompose the difference $I_2$:

$$I_2 = \sum_{n=1}^{N_h^k} \frac{1}{N_h^k} \left( \mathbb{P}_{s,a,h} - \mathbb{1}_{s_{h+1}^{k^n,j^n,m^n}} \right) \left( V_{h+1}^{\mathrm{R},k^n} \right)^2$$

$$+ \left[ \left( \sum_{n=1}^{N_h^k} \frac{1}{N_h^k} V_{h+1}^{\mathrm{R},k^n}\left( s_{h+1}^{k^n,j^n,m^n} \right) \right)^2 - \sum_{n=1}^{N_h^k} \frac{1}{N_h^k} \left( \mathbb{P}_{s,a,h} V_{h+1}^{\mathrm{R},k^n} \right)^2 \right]. \quad (48)$$

For the first term of Equation (48), with Lemma F.4, with probability at least $1 - \delta/4$, it holds for $\forall(s, a, h, k)$ that:

$$\left| \sum_{n=1}^{N_h^k} \frac{1}{N_h^k} \left( \mathbb{P}_{s,a,h} - \mathbb{1}_{s_{h+1}^{k^n,j^n,m^n}} \right) \left( V_{h+1}^{\mathrm{R},k^n} \right)^2 \right| \lesssim \sqrt{\frac{\iota}{N_h^k} \sum_{n=1}^{N_h^k} \frac{1}{N_h^k} \mathbb{V}_{s,a,h}\left( V_{h+1}^{\mathrm{R},k^n} \right)^2} + \frac{H^2\iota}{N_h^k}$$

$$\lesssim \sqrt{\frac{H^2\iota}{N_h^k} \sum_{n=1}^{N_h^k} \frac{1}{N_h^k} \mathbb{V}_{s,a,h}\left( V_{h+1}^{\mathrm{R},k^n} \right)} + \frac{H^2\iota}{N_h^k}$$

$$= \sqrt{\frac{H^2\iota}{N_h^k} X_{\mathrm{R}}} + \frac{H^2\iota}{N_h^k}. \quad (49)$$

The last inequality is by $\mathbb{V}_{s,a,h}(X^2) \le 4C^2 \mathbb{V}_{s,a,h}(X)$ for $|X| \le C$. For the second term of Equation (48), by Cauchy-Schwarz inequality, we reach

$$\left( \sum_{n=1}^{N_h^k} \frac{1}{N_h^k} V_{h+1}^{\mathrm{R},k^n} \left( s_{h+1}^{k^n,j^n,m^n} \right) \right)^2 - \sum_{n=1}^{N_h^k} \frac{1}{N_h^k} \left( \mathbb{P}_{s,a,h} V_{h+1}^{\mathrm{R},k^n} \right)^2$$

$$\le \left( \sum_{n=1}^{N_h^k} \frac{1}{N_h^k} V_{h+1}^{\mathrm{R},k^n} \left( s_{h+1}^{k^n,j^n,m^n} \right) \right)^2 - \left( \sum_{n=1}^{N_h^k} \frac{1}{N_h^k} \mathbb{P}_{s,a,h} V_{h+1}^{\mathrm{R},k^n} \right)^2$$

$$\le 2H \left| \sum_{n=1}^{N_h^k} \frac{1}{N_h^k} \left( \mathbb{1}_{s_{h+1}^{k^n,j^n,m^n}} - \mathbb{P}_{s,a,h} \right) V_{h+1}^{\mathrm{R},k^n} \right|$$

$$\lesssim \sqrt{\frac{H^2 \iota}{N_h^k} X_{\mathrm{R}}} + \frac{H^2 \iota}{N_h^k}. \tag{50}$$

The last inequality holds for $\forall (s, a, h, k)$ with probability at least $1 - \delta/4$ by Lemma F.4. Combining Equation (49) and Equation (50), back to Equation (48), with probability at least $1 - \delta/2$, we know

$$I_1 = X_{\mathrm{R}} - \left( \sigma_h^{\mathrm{R},k} - (\mu_h^{\mathrm{R},k})^2 \right) \lesssim \sqrt{\frac{H^2 \iota}{N_h^k} X_{\mathrm{A}}} + \frac{H^2 \iota}{N_h^k}.$$

Solving the inequality, with probability at least $1 - \delta/2$, we have

$$X_{\mathrm{R}} \lesssim \left( \sigma_h^{\mathrm{R},k} - (\mu_h^{\mathrm{R},k})^2 \right) + \frac{H^2 \iota}{N_h^k}.$$

Applying this inequality to Equation (45), with probability at least $1 - \delta$, the following relationship holds for $\forall (s, a, h, k)$:

$$\left| \sum_{n=1}^{N_h^k} \frac{\tilde{\eta}_n^{N_h^k}}{N_h^{k^n+1}} \sum_{i=1}^{N_h^{k^n+1}} \left( \mathbb{1}_{s_{h+1}^{k^i,j^i,m^i}} - \mathbb{P}_{s,a,h} \right) V_{h+1}^{\mathrm{R},k^i} \right| \lesssim \sqrt{\frac{H\iota}{N_h^k} \left( \sigma_h^{\mathrm{R},k} - \left( \mu_h^{\mathrm{R},k} \right)^2 \right)} + \frac{H^2 \iota}{N_h^k}.$$

(e) For $\mathcal{E}_5$, the sequence

$$\left\{ (e^{\frac{3}{H}})^{h-1} \left( \mathbb{P}_{s_h^{k,j,m}, a_h^{k,j,m}, h} - \mathbb{1}_{s_{h+1}^{k,j,m}} \right) \left( V_{h+1}^{\star} - V_{h+1}^{\pi^k} \right) \right\}_{k,j,h,m}$$

can be reordered to a martingale sequence based on the "round first, episode second, step third, agent fourth" rule. The absolute values of the sequence are bounded by $27H$. After appending multiple 0s to the summation such that there are $T_1$ terms, the sequence is still a martingale. According to Azuma-Hoeffding inequality, for any $\delta \in (0, 1)$, with probability at least $1 - \delta$, it holds that:

$$\sum_{h=1}^{H} (e^{\frac{3}{H}})^{h-1} \sum_{k,j,m} \left( \mathbb{P}_{s_h^{k,j,m}, a_h^{k,j,m}, h} - \mathbb{1}_{s_{h+1}^{k,j,m}} \right) \left( V_{h+1}^{\star} - V_{h+1}^{\pi^k} \right) \le 27 \sqrt{2H^2 T_1 \iota}.$$

(f)

$$\left\{ \left( \mathbb{1}_{s_{h+1}^{k^n,j^n,m^n}} - \mathbb{P}_{s,a,h} \right) V_{h+1}^{\star} \right\}_{n \in \mathbb{N}^+}$$

is a martingale sequence bounded by $H$. Then according to Azuma-Hoeffding inequality, for any $\delta \in (0, 1)$, with probability at least $1 - \delta/SAT_1$, it holds for a given $N_h^k(s, a) = N \in \mathbb{N}_+$ that:

$$\frac{1}{N} \left| \sum_{i=1}^{N} \left( \mathbb{1}_{s_{h+1}^{k^n,j^n,m^n}} - \mathbb{P}_{s,a,h} \right) V_{h+1}^{\star} \right| \le H \sqrt{\frac{2\iota}{N}}.$$

For any $k \in [K]$, we have $N_h^k(s, a) \in [\frac{T_1}{H}]$. Considering all the possible combinations $(s, a, h, N) \in \mathcal{S} \times \mathcal{A} \times [H] \times [\frac{T_1}{H}]$, with probability at least $1 - \delta$, it holds simultaneously for all $(s, a, h, k) \in \mathcal{S} \times \mathcal{A} \times [H] \times [K]$ that:

$$\frac{1}{N_h^k} \left| \sum_{i=1}^{N_h^k} \left( \mathbb{1}_{s_{h+1}^{k^n,j^n,m^n}} - \mathbb{P}_{s,a,h} \right) V_{h+1}^{\star} \right| \le H \sqrt{\frac{2\iota}{N_h^k(s, a)}}.$$

(g) The proof is similar to (f) with

$$\left\{\left(\mathbb{1}_{s_{h+1}^{k^n,j^n,m^n}} - \mathbb{P}_{s,a,h}\right)(V_{h+1}^\star)^2\right\}_{n\in\mathbb{N}^+}$$

being a martingale sequence bounded by $H^2$.

(i): The proof is similar to (e).

(j): The proof is provided in section C.3 of [113] when bounding the term $\sum_{k=1}^{K}\delta_1^k$.

(k) It follows by Theorem E.3 with $l = H$.

(l)

$$\left\{\left(\mathbb{1}_{s_{h+1}^{k^i,j^i,m^i}} - \mathbb{P}_{s,a,h}\right)\left(\hat{V}_{h+1}^{\text{R},k^i} - V_{h+1}^\star\right)\right\}_{n\in\mathbb{N}^+}$$

is a martingale sequence bounded by $\beta$ . Then according to Azuma-Hoeffding inequality, for any $\delta \in (0,1)$, with probability at least $1 - \delta/SAT_1$, it holds for a given $N_h^{k^n+1}(s,a) = N \in \mathbb{N}_+$ that:

$$\frac{1}{N}\left|\sum_{i=1}^{N}\left(\mathbb{1}_{s_{h+1}^{k^i,j^i,m^i}} - \mathbb{P}_{s,a,h}\right)\left(\hat{V}_{h+1}^{\text{R},k^i} - V_{h+1}^\star\right)\right| \le \beta\sqrt{\frac{2\iota}{N}}.$$

For any $k^n \in [K]$, we have $N_h^{k^n+1}(s,a) \in [\frac{T_1}{H}]$. Considering all the possible combinations $(s,a,h,N) \in \mathcal{S} \times \mathcal{A} \times [H] \times [\frac{T_1}{H}]$, with probability at least $1 - \delta$, it holds simultaneously for all $(s,a,h,k^n) \in \mathcal{S} \times \mathcal{A} \times [H] \times [K]$ that:

$$\left|\frac{1}{N_h^{k^n+1}}\sum_{i=1}^{N_h^{k^n+1}}\left(\mathbb{1}_{s_{h+1}^{k^i,j^i,m^i}} - \mathbb{P}_{s,a,h}\right)\left(\hat{V}_{h+1}^{\text{R},k^i} - V_{h+1}^\star\right)\right| \le \beta\sqrt{\frac{2\iota}{N_h^{k^n+1}}}.$$

(m)

$$\left\{\left(\mathbb{1}_{s_{h+1}^{k^i,j^i,m^i}} - \mathbb{P}_{s,a,h}\right)V_{h+1}^\star\right\}_{n\in\mathbb{N}^+}$$

is a martingale sequence bounded by $2H$. Then according to Freedman's inequality Theorem E.2, for any $\delta \in (0,1)$, with probability at least $1 - \delta/SAT_1$, it holds for a given $N_h^{k^n+1}(s,a) = N \in \mathbb{N}_+$ that:

$$\left|\frac{1}{N}\sum_{i=1}^{N}\left(\mathbb{1}_{s_{h+1}^{k^i,j^i,m^i}} - \mathbb{P}_{s,a,h}\right)V_{h+1}^\star\right| \le 4\sqrt{\frac{\mathbb{V}_{s,a,h}(V_{h+1}^\star)\iota}{N}} + \frac{7H\iota}{N}.$$

Here we set $W_N = N\mathbb{V}_{s,a,h}(V_{h+1}^\star)$, $R = H$, $m = \log_2(T_1)$ and $\sigma^2 = T_1 H$.

For any $k^n \in [K]$, we have $N_h^{k^n+1}(s,a) \in [\frac{T_1}{H}]$. Considering all the possible combinations $(s,a,h,N) \in \mathcal{S} \times \mathcal{A} \times [H] \times [\frac{T_1}{H}]$, with probability at least $1 - \delta$, it holds simultaneously for all $(s,a,h,k,n) \in \mathcal{S} \times \mathcal{A} \times [H] \times [K] \times [T_1]$ that:

$$\left|\frac{1}{N_h^{k^n+1}}\sum_{i=1}^{N_h^{k^n+1}}\left(\mathbb{1}_{s_{h+1}^{k^i,j^i,m^i}} - \mathbb{P}_{s,a,h}\right)V_{h+1}^\star\right| \le 4\sqrt{\frac{\mathbb{V}_{s,a,h}(V_{h+1}^\star)\iota}{N_h^{k^n+1}}} + \frac{7H\iota}{N_h^{k^n+1}}.$$

(n) By Lemma F.5, since $0 \le \hat{V}_{h+1}^{\text{R},k^n}(s) - V_{h+1}^\star(s) \le \beta$, then with probability at least $1 - \delta/SAT_1$, it holds for a given $N_h^k(s,a) = N \in [T_1/H]$ that

$$\sum_{n=1}^{N}\tilde{\eta}_n^N\left(\mathbb{1}_{s_{h+1}^{k^n,j^n,m^n}} - \mathbb{P}_{s,a,h}\right)\left(V_{h+1}^\star - \hat{V}_{h+1}^{\text{R},k^n}\right)$$

$$\lesssim \sqrt{\frac{H\iota}{N}\sum_{n=1}^{N}\eta_n^N\mathbb{V}_{s,a,h}(V_{h+1}^\star - \hat{V}_{h+1}^{\text{R},k^n})} + \frac{\beta H\iota}{N} \overset{(i)}{\le} \beta\sqrt{\frac{H\iota}{N}} + \frac{\beta H\iota}{N}.$$

Here, (i) is because $\mathbb{V}_{s_h^{k,j,m},a_h^{k,j,m},h}(V_{h+1}^\star - \hat{V}_{h+1}^{\text{R},k^n}) \le \beta^2$ and $\sum_{n=1}^{N_h^k}\eta_n^{N_h^k} \le 1$. Considering all the possible combinations $(s,a,h,N) \in \mathcal{S} \times \mathcal{A} \times [H] \times [\frac{T_1}{H}]$, we finish the proof. $\qquad\square$

# H  Key Properties of estimated $Q-$functions and $V-$functions

We first prove the optimism property of the estimated $Q-$function $Q_h^k(s,a)$.

**Lemma H.1.** *Under the event $\bigcap_{i=1}^3 \mathcal{E}_i$ in Lemma G.1, it holds that for any $(s,a,h,k) \in \mathcal{S} \times \mathcal{A} \times [H] \times [K]$:*

$$Q_h^k(s,a) \geq Q_h^\star(s,a) \text{ and } V_h^k(s) \geq V_h^\star(s).$$

*Proof.* We use mathematical induction on $k$ to prove $Q_h^k(s,a) \geq Q_h^\star(s,a)$ and $V_h^k(s) \geq V_h^\star(s)$ for any $(s,a,h,k) \in \mathcal{S} \times \mathcal{A} \times [H] \times [K]$.

For $k = 1$, $Q_h^1(s,a) = H \geq Q_h^\star(s,a)$ and $V_h^1(s) = H \geq V_h^\star(s)$ for any $(s,a,h) \in \mathcal{S} \times \mathcal{A} \times [H]$.

For $k \geq 2$, assume we already have $Q_h^{k'}(s,a) \geq Q_h^\star(s,a)$ for any $(s,a,h,k') \in \mathcal{S} \times \mathcal{A} \times [H] \times [k-1]$, then we will prove for any $(s,a,h) \in \mathcal{S} \times \mathcal{A} \times [H]$, $Q_h^k(s,a) \geq Q_h^\star(s,a)$.

It is sufficient to show that

$$\min\left\{ Q_h^{\mathrm{U},k}(s,a), Q_h^{\mathrm{R},k}(s,a) \right\} \geq Q_h^\star(s,a).$$

The event $\mathcal{E}_1$ in Lemma G.1 shows that $Q_h^{\mathrm{U},k}(s,a) \geq Q_h^\star(s,a)$. Then it is sufficient to prove that

$$Q_h^{\mathrm{R},k}(s,a) \geq Q_h^\star(s,a). \tag{51}$$

To begin with, according to the update rule Equation (11) and Equation (15), we obtain

$$Q_h^{\mathrm{R},k}(s,a) = \tilde{\eta}_0^{N_h^k} H + \sum_{n=1}^{N_h^k} \tilde{\eta}_n^{N_h^k} \left( r_h(s,a) + \left( V_{h+1}^{k^n} - V_{h+1}^{\mathrm{R},k^n} \right) \left( s_{h+1}^{k^n,j^n,m^n} \right) + \mu_h^{\mathrm{R},k^n+1} \right) + B_h^{\mathrm{R},k+1}$$

$$+ \sum_{n=1}^{N_h^k} \eta_n^{N_h^k} b_{h,n}^{\mathrm{R}}.$$

Since $\sum_{n=0}^{N_h^k} \tilde{\eta}_n^{N_h^k} = 1$ by (b) of Lemma F.3, it leads to

$$Q_h^{\mathrm{R},k}(s,a) - Q_h^\star(s,a) = \eta_0^{N_h^k} \left( H - Q_h^\star(s,a) \right) + \sum_{n=1}^{N_h^k} \eta_n^{N_h^k} b_{h,n}^{\mathrm{R}}$$

$$+ \sum_{n=1}^{N_h^k} \tilde{\eta}_n^{N_h^k} \left( r_h(s,a) + \left( V_{h+1}^{k^n} - V_{h+1}^{\mathrm{R},k^n} \right) \left( s_{h+1}^{k^n,j^n,m^n} \right) + \mu_h^{\mathrm{R},k^n+1} - Q_h^\star(s,a) \right). \tag{52}$$

To continue, invoking the Bellman optimality equation (1),

$$Q_h^\star(s,a) = r_h(s,a) + \mathbb{P}_{s,a,h} V_{h+1}^\star,$$

and using the update rule of $\mu_h^{\mathrm{R},k+1}$ in Equation (6), we reach

$$r_h(s,a) + \left( V_{h+1}^{k^n} - V_{h+1}^{\mathrm{R},k^n} \right) \left( s_{h+1}^{k^n,j^n,m^n} \right) + \mu_h^{\mathrm{R},k^n+1} - Q_h^\star(s,a)$$

$$= \left( V_{h+1}^{k^n} - V_{h+1}^{\mathrm{R},k^n} \right) \left( s_{h+1}^{k^n,j^n,m^n} \right) + \frac{\sum_{i=1}^{N_h^{k^n+1}} V_{h+1}^{\mathrm{R},k^i} \left( s_{h+1}^{k^i,j^i,m^i} \right)}{N_h^{k^n+1}} - \mathbb{P}_{s,a,h} V_{h+1}^\star$$

$$= \mathbb{P}_{s,a,h} \left( V_{h+1}^{k^n} - V_{h+1}^\star + \frac{\sum_{i=1}^{N_h^{k^n+1}} \left( V_{h+1}^{\mathrm{R},k^i} - V_{h+1}^{\mathrm{R},k^n} \right)}{N_h^{k^n+1}} \right) + \xi_h^k.$$

where we have introduced the following quantity

$$\xi_h^k := \left( \mathbb{1}_{s_{h+1}^{k^n,j^n,m^n}} - \mathbb{P}_{s,a,h} \right) \left( V_{h+1}^{k^n} - V_{h+1}^{\mathrm{R},k^n} \right) + \frac{1}{N_h^{k^n+1}} \sum_{i=1}^{N_h^{k^n+1}} \left( \mathbb{1}_{s_{h+1}^{k^i,j^i,m^i}} - \mathbb{P}_{s,a,h} \right) V_{h+1}^{\mathrm{R},k^i}.$$

Since $k^n \leq k-1$, we know $V_{h+1}^{k^n} - V_{h+1}^{\star} \geq 0$. We also have $V_{h+1}^{\mathrm{R},k^i} - V_{h+1}^{\mathrm{R},k^n} \geq 0$ for $k^i \leq k^n$ because the reference function $V_h^{\mathrm{R},k}(s)$ is monotonically non-increasing in view of the monotonicity of $V_h^k(s)$. Back to Equation (52), we know that

$$Q_h^{\mathrm{R},k}(s,a) - Q_h^{\star}(s,a) \leq \sum_{n=1}^{N_h^k} \tilde{\eta}_n^{N_h^k} \xi_h^k + \sum_{n=1}^{N_h^k} \eta_n^{N_h^k} b_{h,n}^{\mathrm{R}}.$$

Therefore, we only need to prove that

$$\left| \sum_{n=1}^{N_h^k} \tilde{\eta}_n^{N_h^k} \xi_h^k \right| \leq \sum_{n=1}^{N_h^k} \eta_n^{N_h^k} b_{h,n}^{\mathrm{R}}.$$

Let $\{k_1, k_2, ..., k_t\}$ be the collection of round indices that $n_h^{k_i}(s,a) > 0$ for any $i \in [t]$ and $k_1 < k_2 < ... < k_t < k$. Let $k_0 = 0$ and $k_{t+1} = k$, then for any $i \in [t+1]$, we have $\beta_h^{\mathrm{R},k_i}(s,a) = \beta_h^{\mathrm{R},k_{i-1}+1}(s,a)$ ($k_0 = 0$) and $N_h^{k_i}(s,a) = N_h^{k_{i-1}+1}(s,a)$ with $k_0$ since there is no visit to $(s,a,h)$ from round $k_{i-1}+1$ to round $k_i - 1$. Then it holds that (Here, $\eta^c(n+1,n) = 1$ for any $n \in \mathbb{N}_+$):

$$\sum_{n=1}^{N_h^k} \eta_n^{N_h^k} b_{h,n}^{\mathrm{R}} = \sum_{i=1}^t \sum_{n=N_h^{k_i}+1}^{N_h^{k_i+1}} \eta_n^{N_h^k} b_{h,n}^{\mathrm{R}}$$

$$= \sum_{i=1}^t \left\{ \left( \sum_{n=N_h^{k_i}+1}^{N_h^{k_i+1}-1} \eta_n^{N_h^k} + \frac{1-\eta_{N_h^{k_i+1}}}{\eta_{N_h^{k_i+1}}} \eta_{N_h^{k_i+1}}^{N_h^k} \right) \beta_h^{\mathrm{R},k_i} + \frac{\eta_{N_h^{k_i+1}}^{N_h^k}}{\eta_{N_h^{k_i+1}}} \beta_h^{\mathrm{R},k_i+1} \right\} + \sum_{n=1}^{N_h^k} \frac{c_b^{\mathrm{R},2} \eta_n^{N_h^k} H^2 \iota}{n}$$

$$= \sum_{i=1}^t \left( -\eta^c(N_h^{k_i}+1, N_h^k) \beta_h^{\mathrm{R},k_i} + \eta^c(N_h^{k_i+1}+1, N_h^k) \beta_h^{\mathrm{R},k_i+1} \right) + \sum_{n=1}^{N_h^k} \frac{c_b^{\mathrm{R},2} \eta_n^{N_h^k} H^2 \iota}{n}$$

$$= \sum_{i=1}^t \left( -\eta^c(N_h^{k_{i-1}+1}+1, N_h^k) \beta_h^{\mathrm{R},k_{i-1}+1} + \eta^c(N_h^{k_i+1}+1, N_h^k) \beta_h^{\mathrm{R},k_i+1} \right) + \sum_{n=1}^{N_h^k} \frac{c_b^{\mathrm{R},2} \eta_n^{N_h^k} H^2 \iota}{n}$$

$$= \beta_h^{\mathrm{R},k} + \sum_{n=1}^{N_h^k} \frac{c_b^{\mathrm{R},2} \eta_n^{N_h^k} H^2 \iota}{n} \geq \beta_h^{\mathrm{R},k} + \frac{c_b^{\mathrm{R},2} H^2 \iota}{N_h^k}. \tag{53}$$

The last inequality is because of the property (a) of Lemma F.2 with $\alpha = 1$. Under the event $\bigcap_{i=2}^3 \mathcal{E}_i$,

$$\left| \sum_{n=1}^{N_h^k} \tilde{\eta}_n^{N_h^k} \xi_h^k \right| \lesssim \sqrt{\frac{H\iota}{N_h^k} \left( \sigma_h^{\mathrm{A},k} - \left( \mu_h^{\mathrm{A},k} \right)^2 \right)} + \sqrt{\frac{\iota}{N_h^k} \left( \sigma_h^{\mathrm{R},k} - \left( \mu_h^{\mathrm{R},k} \right)^2 \right)} + \frac{H^2 \iota}{N_h^k}.$$

Then for some sufficiently large constant $c_b^{\mathrm{R}}, c_b^{\mathrm{R},2} > 0$, by Equation (53):

$$\left| \sum_{n=1}^{N_h^k} \tilde{\eta}_n^{N_h^k} \xi_h^k \right| \leq \beta_h^{\mathrm{R},k} + \frac{c_b^{\mathrm{R},2} H^2 \iota}{N_h^k} \leq \sum_{n=1}^{N_h^k} \eta_n^{N_h^k} b_{h,n}^{\mathrm{R}}.$$

Now we finish the proof of Equation (51) and thus $Q_h^k(s,a) \geq Q_h^{\star}(s,a)$. Then we can conclude that

$$V_h^k(s) = \max_a \{Q_h^k(s,a)\} \geq \max_a \{Q_h^{\star}(s,a)\} = V_h^{\star}(s).$$

We have thus concluded the proof of Lemma H.1. $\qquad\square$

Next, we will present the pessimism property of the $Q$−estimates $Q_h^{\mathrm{L},k}(s,a)$.

**Lemma H.2.** *Under the event $\mathcal{E}_4$ in Lemma G.1, it holds that for any $(s,a,h,k) \in \mathcal{S} \times \mathcal{A} \times [H] \times [K]$:*

$$Q_h^{\mathrm{L},k}(s,a) \leq Q_h^{\star}(s,a) \quad and \quad V_h^{\mathrm{L},k}(s) \leq V_h^{\star}(s).$$

*Proof.* We use mathematical induction on $k$ to prove $Q_h^{\mathrm{L},k}(s,a) \leq Q_h^{\star}(s,a)$ and $V_h^{\mathrm{L},k}(s) \leq V_h^{\star}(s)$ for any $(s,a,h,k) \in \mathcal{S} \times \mathcal{A} \times [H] \times [K]$.

For $k = 1$, $Q_h^{\mathrm{L},1}(s,a) = 0 \leq Q_h^{\star}(s,a)$ and $V_h^{\mathrm{L},1}(s) = 0 \geq V_h^{\star}(s)$ for any $(s,a,h) \in \mathcal{S} \times \mathcal{A} \times [H]$.

For $k \geq 2$, assume we already have $Q_h^{\mathrm{L},k'}(s,a) \leq Q_h^{\star}(s,a)$ and $V_h^{\mathrm{L},k'}(s) \leq V_h^{\star}(s)$ for any $(s,a,h,k') \in \mathcal{S} \times \mathcal{A} \times [H] \times [k-1]$, then we will prove for any $(s,a,h) \in \mathcal{S} \times \mathcal{A} \times [H]$, $Q_h^{\mathrm{L},k}(s,a) \leq Q_h^{\star}(s,a)$ and $V_h^{\mathrm{L},k}(s) \leq V_h^{\star}(s)$. Based on the updating rules Equation (10) and Equation (14), by recursion, since $Q_h^{\mathrm{L},1}(s,a) = 0$, we have:

$$Q_h^{\mathrm{L},k}(s,a) = \sum_{n=1}^{N_h^k} \tilde{\eta}_n^{N_h^k}\left(r_h(s,a) + V_{h+1}^{\mathrm{L},k^n}\right) - \sum_{n=1}^{N_h^k} \eta_n^{N_h^k} b_n.$$

To continue, by invoking the Bellman optimality equation Equation (1),

$$Q_h^{\star}(s,a) = r_h(s,a) + \mathbb{P}_{s,a,h} V_{h+1}^{\star},$$

we have

$$Q_h^{\mathrm{L},k}(s,a) - Q_h^{\star}(s,a)$$

$$\leq \sum_{n=1}^{N_h^k} \tilde{\eta}_n^{N_h^k}\left(V_{h+1}^{\mathrm{L},k^n}(s_{h+1}^{k^n,j^n,m^n}) - \mathbb{P}_{s,a,h} V_{h+1}^{\star}\right) - \sum_{n=1}^{N_h^k} \eta_n^{N_h^k} b_n$$

$$= \sum_{n=1}^{N_h^k} \tilde{\eta}_n^{N_h^k}\left(V_{h+1}^{\mathrm{L},k^n} - V_{h+1}^{\star}\right)(s_{h+1}^{k^n,j^n,m^n}) + \sum_{n=1}^{N_h^k} \tilde{\eta}_n^{N_h^k}\left(\mathbb{1}_{s_{h+1}^{k^n,j^n,m^n}} - \mathbb{P}_{s,a,h}\right) V_{h+1}^{\star} - \sum_{n=1}^{N_h^k} \eta_n^{N_h^k} b_n$$

$$\leq \sum_{n=1}^{N_h^k} \tilde{\eta}_n^{N_h^k}\left(\mathbb{1}_{s_{h+1}^{k^n,j^n,m^n}} - \mathbb{P}_{s,a,h}\right) V_{h+1}^{\star} - \sum_{n=1}^{N_h^k} \eta_n^{N_h^k} b_n \leq 0.$$

The last inequality is by event $\mathcal{E}_4$ and

$$\sum_{n=1}^{N_h^k} \eta_n^{N_h^k} b_n = c_b \sqrt{H^3 \iota} \sum_{n=1}^{N_h^k} \frac{\eta_n^{N_h^k}}{\sqrt{n}} \geq c_b \sqrt{\frac{H^3 \iota}{N_h^k}}$$

by (a) of Lemma F.2 with $\alpha = \frac{1}{2}$. Now we have proved that $Q_h^{\mathrm{L},k}(s,a) \leq Q_h^{\star}(s,a) \leq V_h^{\star}(s)$. Then by definition Equation (17) of $V_h^{\mathrm{L},k}(s)$, we know

$$V_h^{\mathrm{L},k}(s) = \max\left\{\max_{a' \in \mathcal{A}} Q_h^{\mathrm{L},k}(s,a'), V_h^{\mathrm{L},k-1}(s)\right\} \leq V_h^{\star}(s).$$

$\square$

In the following lemma, we will bound the error between two $Q-$estimates $Q_h^k(s,a)$ and $Q_h^{\mathrm{L},k}(s,a)$.

**Lemma H.3.** *Under the event* $\bigcap_{i=1}^4 \mathcal{E}_i$ *in Lemma G.1, for FedQ-EarlySettled-LowCost algorithm and any non-negative weight sequence* $\{\omega_h^{k,j,m}\}_{h,k,j,m}$*, it holds for any* $h \in [H]$ *that:*

$$\sum_{k,j,m} \omega_h^{k,j,m}\left(Q_h^k - Q_h^{\mathrm{L},k}\right)(s_h^{k,j,m}, a_h^{k,j,m})$$

$$\lesssim \sqrt{H^5 SA \|\omega\|_{\infty,h} \|\omega\|_{1,h} \iota} + \sum_{h'=h}^H \sum_{k,j,m} \omega_{h'}^{k,j,m}(h) Y_{h'}^{k,j,m},$$

*where for any* $h \leq h' \leq H - 1$

$$\omega_h^{k,j,m}(h) := \omega_h^{k,j,m},$$

$$\omega_{h'+1}^{k,j,m}(h) = \sum_{k',j',m'} \omega_{h'}^{k',j',m'}(h) \mathbb{I}\left[N_{h'}^{k'}(s,a)_{h'}^{k',j',m'} \geq i_0\right] \sum_{i=1}^{N_{h'}^{k'}} \tilde{\eta}_i^{N_{h'}^{k'}} \mathbb{I}\left[(k^i, j^i, m^i) = (k,j,m)\right],$$

*and*

$$Y_{h'}^{k,j,m} = \eta_0^{N_{h'}^k} H + H\mathbb{I}[0 < N_{h'}^k(s_{h'}^{k,j,m}, a_{h'}^{k,j,m}) < i_0] + \sqrt{\frac{H^3\iota}{N_{h'}^k}}\mathbb{I}[0 < N_{h'}^k(s_{h'}^{k,j,m}, a_{h'}^{k,j,m}) < M].$$

*Proof.* To begin with, according to the update rule Equation (9) and Equation (13), we obtain

$$Q_h^{\mathrm{U},k}(s,a) = \eta_0^{N_h^k} H + \sum_{i=1}^{N_h^k} \tilde{\eta}_i^{N_h^k}\left(r_h(s,a) + V_{h+1}^{k^i}(s_{h+1}^{k^i,j^i,m^i})\right) + \sum_{i=1}^{N_h^k} \eta_i^{N_h^k} b_i.$$

Similarly, according to the update rule Equation (10) and Equation (14), we obtain

$$Q_h^{\mathrm{L},k}(s,a) = \sum_{i=1}^{N_h^k} \tilde{\eta}_i^{N_h^k}\left(r_h(s,a) + V_{h+1}^{\mathrm{L},k^i}(s_{h+1}^{k^i,j^i,m^i})\right) - \sum_{i=1}^{N_h^k} \eta_i^{N_h^k} b_i.$$

and

$$\sum_{k,j,m} \omega_h^{k,j,m}\left(Q_h^k - Q_h^{\mathrm{L},k}\right)(s_h^{k,j,m}, a_h^{k,j,m})$$

$$\leq \sum_{k,j,m} \omega_h^{k,j,m}\left(Q_h^{\mathrm{U},k} - Q_h^{\mathrm{L},k}\right)(s_h^{k,j,m}, a_h^{k,j,m})$$

$$\leq \sum_{k,j,m} \omega_h^{k,j,m}\eta_0^{N_h^k} H + \sum_{k,j,m,N_h^k>0} \omega_h^{k,j,m} \sum_{i=1}^{N_h^k} \tilde{\eta}_i^{N_h^k}(V_{h+1}^{k^i} - V_{h+1}^{\mathrm{L},k^i})(s_{h+1}^{k^i,j^i,m^i})$$

$$+ 2 \sum_{k,j,m,N_h^k>0} \omega_h^{k,j,m} \sum_{i=1}^{N_h^k} \eta_i^{N_h^k} b_i. \tag{54}$$

**For the last term of Equation (54)**, by (a) of Lemma F.2, we have

$$\sum_{i=1}^{N_h^k} \eta_i^{N_h^k} b_i = \sum_{i=1}^{N_h^k} \eta_i^{N_h^k} c_b\sqrt{\frac{H^3\iota}{i}} \lesssim \sqrt{\frac{H^3\iota}{N_h^k}}.$$

Then by Lemma F.6, it holds that

$$\sum_{k,j,m,N_h^k>0} \omega_h^{k,j,m} \sum_{i=1}^{N_h^k} \eta_i^{N_h^k} b_i \lesssim \sqrt{H^3\iota} \sum_{k,j,m,N_h^k>0} \omega_h^{k,j,m}\sqrt{\frac{1}{N_h^k(s_h^{k,j,m}, a_h^{k,j,m})}}$$

$$\lesssim \sum_{k,j,m} \omega_h^{k,j,m}\sqrt{\frac{H^3\iota}{N_h^k(s_h^{k,j,m}, a_h^{k,j,m})}}\mathbb{I}\left[0 < N_h^k < M\right] + \sqrt{H^3 SA\|\omega\|_{\infty,h}\|\omega\|_{1,h}\iota}. \tag{55}$$

Next, we will bound the second term of Equation (54). We can decompose the term into two parts as

$$\sum_{k,j,m,N_h^k>0} \omega_h^{k,j,m} \sum_{i=1}^{N_h^k} \tilde{\eta}_i^{N_h^k}(V_{h+1}^{k^i} - V_{h+1}^{\mathrm{L},k^i})(s_{h+1}^{k^i,j^i,m^i})$$

$$= \sum_{k,j,m} \omega_h^{k,j,m} \sum_{i=1}^{N_h^k} \tilde{\eta}_i^{N_h^k}(V_{h+1}^{k^i} - V_{h+1}^{\mathrm{L},k^i})(s_{h+1}^{k^i,j^i,m^i})\left(\mathbb{I}\left[0 < N_h^k < i_0\right] + \mathbb{I}\left[N_h^k \geq i_0\right]\right).$$

**For the first part of the second term in Equation (54)**, because $\sum_{i=1}^{N_h^k} \tilde{\eta}_i^{N_h^k} \leq 1$ by (b) of Lemma F.3, we have

$$\sum_{k,j,m} \omega_h^{k,j,m} \sum_{i=1}^{N_h^k} \tilde{\eta}_i^{N_h^k}(V_{h+1}^{k^i} - V_{h+1}^{\mathrm{L},k^i})(s_{h+1}^{k^i,j^i,m^i})\mathbb{I}\left[0 < N_h^k(s_h^{k,j,m}, a_h^{k,j,m}) < i_0\right]$$

$$\leq H \sum_{k,j,m} \omega_h^{k,j,m}\mathbb{I}\left[0 < N_h^k(s_h^{k,j,m}, a_h^{k,j,m}) < i_0\right] \tag{56}$$

**For the second part of the second term in Equation (54)**, we regroup the summations as follows:

$$\sum_{k,j,m} \omega_h^{k,j,m} \sum_{i=1}^{N_h^k} \tilde{\eta}_i^{N_h^k} (V_{h+1}^{k^i} - V_{h+1}^{\mathrm{L},k^i})(s_{h+1}^{k^i,j^i,m^i}) \mathbb{I}\left[ N_h^k(s_h^{k,j,m}, a_h^{k,j,m}) \geq i_0 \right]$$
$$= \sum_{k',j',m'} \tilde{\omega}_h^{k',j',m'} \left( V_{h+1}^{k'} - V_{h+1}^{\mathrm{L},k'} \right)(s_{h+1}^{k',j',m'}), \tag{57}$$

where

$$\tilde{\omega}_h^{k',j',m'} = \sum_{k,j,m} \mathbb{I}\left[ N_h^k(s_h^{k,j,m}, a_h^{k,j,m}) \geq i_0 \right] \omega_h^{k,j,m} \sum_{i=1}^{N_h^k} \tilde{\eta}_i^{N_h^k} \mathbb{I}\left[ (k^i, j^i, m^i) = (k', j', m') \right].$$

Let $\|\tilde{\omega}\|_{\infty,h} = \max_{k,j,m}\{\tilde{\omega}_h^{k,j,m}\}$ and $\|\tilde{\omega}\|_{1,h} = \sum_{k,j,m} \tilde{\omega}_h^{k,j,m}$. Since $\sum_{i=1}^{N_h^k} \tilde{\eta}_i^{N_h^k} \leq 1$ by (b) of Lemma F.3, we have the following property:

$$\|\tilde{\omega}\|_{1,h} = \sum_{k',m',j'} \tilde{\omega}_h^{k',j',m'} \leq \sum_{k,j,m} \mathbb{I}\left[ N_h^k(s_h^{k,j,m}, a_h^{k,j,m}) \geq i_0 \right] \omega_h^{k,j,m} \leq \|\omega\|_{1,h}.$$

If we have proved that:

$$\|\tilde{\omega}\|_{\infty,h} \leq \exp(3/H)\|\omega\|_{\infty,h}, \tag{58}$$

**then combining the results of Equation (55), Equation (56) and Equation (57) together with Equation (54)**, we reach

$$\sum_{k,j,m} \omega_h^{k,j,m} \left( Q_h^k - Q_h^{\mathrm{L},k} \right)(s_h^{k,j,m}, a_h^{k,j,m})$$
$$\lesssim \sum_{k',j',m'} \tilde{\omega}_h^{k',j',m'} (V_{h+1}^{k'} - V_{h+1}^{\mathrm{L},k'})(s_{h+1}^{k',j',m'}) + \sqrt{H^3 SA \|\omega\|_{\infty,h}\|\omega\|_{1,h}\iota} + \sum_{k,j,m} \omega_h^{k,j,m} \eta_0^{N_h^k} H$$
$$+ \sum_{k,j,m} \omega_h^{k,j,m} H \mathbb{I}\left[ 0 < N_h^k < i_0 \right] + \sum_{k,j,m} \omega_h^{k,j,m} \sqrt{\frac{H^3\iota}{N_h^k}} \mathbb{I}\left[ 0 < N_h^k < M \right]$$
$$\lesssim \sum_{k',j',m'} \tilde{\omega}_h^{k',j',m'} (Q_{h+1}^{k'} - Q_{h+1}^{\mathrm{L},k'})(s_{h+1}^{k',j',m'}, a_{h+1}^{k',j',m'}) + \sqrt{H^3 SA \|\omega\|_{\infty,h}\|\omega\|_{1,h}\iota}$$
$$+ \sum_{k,j,m} \omega_h^{k,j,m} Y_h^{k,j,m}. \tag{59}$$

with $\|\tilde{\omega}\|_{1,h} \leq \|\omega\|_{1,h}$ and $\|\tilde{\omega}\|_{\infty,h} \leq \exp(3/H)\|\omega\|_{\infty,h}$. Here, the last inequality is because

$$V_{h+1}^{k'}(s_{h+1}^{k',j',m'}) = Q_{h+1}^{k'}(s_{h+1}^{k',j',m'}, a_{h+1}^{k',j',m'}) \text{ and } V_{h+1}^{\mathrm{L},K'}(s_{h+1}^{k',j',m'}) \geq Q_{h+1}^{\mathrm{L},k'}(s_{h+1}^{k',j',m'}, a_{h+1}^{k',j',m'}).$$

With Equation (59), we develop a recursive relationship for the weighted sum of $Q_h^k - Q_h^\star$ between step $h$ and step $h+1$. By recursions with regard to $h, h+1, ..., H$, we finish the proof.

**Proof of Equation (58):** Now we have

$$\tilde{\omega}_h^{k',j',m'} = \sum_{k,j,m} \mathbb{I}\left[ N_h^k(s_h^{k,j,m}, a_h^{k,j,m}) \geq i_0 \right] \omega_h^{k,j,m} \sum_{i=1}^{N_h^k} \tilde{\eta}_i^{N_h^k} \mathbb{I}\left[ (k^i, j^i, m^i) = (k', j', m') \right]$$

$$\leq \|\omega\|_{\infty,h} \sum_{k,j,m} \mathbb{I}\left[ N_h^k(s_h^{k,j,m}, a_h^{k,j,m}) \geq i_0 \right] \sum_{i=1}^{N_h^k} \tilde{\eta}_i^{N_h^k} \mathbb{I}\left[ (k^i, j^i, m^i) = (k', j', m') \right]$$

We only need to prove for any triple $(k', j', m')$ and any $h \in [H]$,

$$\sum_{k,j,m} \mathbb{I}\left[ N_h^k(s_h^{k,j,m}, a_h^{k,j,m}) \geq i_0 \right] \sum_{i=1}^{N_h^k} \tilde{\eta}_i^{N_h^k} \mathbb{I}\left[ (k^i, j^i, m^i) = (k', j', m') \right] \leq \exp(3/H). \tag{60}$$

By definition of $k^i$, $j^i$ and $m^i$, for any given triple $(k', j', m')$,

$$\sum_{i=1}^{N_h^k} \tilde{\eta}_i^{N_h^k} \mathbb{I}\left[(k^i, j^i, m^i) = (k', j', m')\right] > 0$$

if and only if

$$(s_h^{k,j,m}, a_h^{k,j,m}) = (s_h^{k',j',m'}, a_h^{k',j',m'}), k' < k \text{ and } i'(k', j', m') \leq N_h^k,$$

where $i'(k', j', m')$ is the global visiting number for $(s_h^{k',j',m'}, a_h^{k',j',m'})$ at $(k', m', j')$. When there is no ambiguity, we will use $i'$ for short. Therefore

$$\sum_{k,j,m} \mathbb{I}\left[N_h^k(s_h^{k,j,m}, a_h^{k,j,m}) \geq i_0\right] \sum_{i=1}^{N_h^k} \tilde{\eta}_i^{N_h^k} \mathbb{I}\left[(k^i, j^i, m^i) = (k', j', m')\right]$$

$$= \sum_{k=k'+1}^{K} \sum_{j,m} \mathbb{I}\left[N_h^k(s_h^{k',j',m'}, a_h^{k',j',m'}) \geq i_0, (s_h^{k,j,m}, a_h^{k,j,m}) = (s_h^{k',j',m'}, a_h^{k',j',m'})\right] \tilde{\eta}_{i'}^{N_h^k}. \quad (61)$$

Let $k' < k_1 < k_2 < ... < k_t \leq K$ be all the round index such that $n_h^{k_q}(s_h^{k',j',m'}, a_h^{k',j',m'}) > 0$ and $N_h^{k_q}(s_h^{k',j',m'}, a_h^{k',j',m'}) \geq i_0$ for any $q \in [t]$, then we can simplify Equation (61):

$$\sum_{k,j,m} \mathbb{I}\left[N_h^k(s_h^{k,j,m}, a_h^{k,j,m}) \geq i_0\right] \sum_{i=1}^{N_h^k} \tilde{\eta}_i^{N_h^k} \mathbb{I}\left[(k^i, j^i, m^i) = (k', j', m')\right]$$

$$= \sum_{q=1}^{t} \left(\sum_{j,m} \mathbb{I}\left[(s_h^{k_q,j,m}, a_h^{k_q,j,m}) = (s_h^{k',j',m'}, a_h^{k',j',m'})\right]\right) \tilde{\eta}_{i'}^{N_h^{k_q}}$$

$$\leq \sum_{q=1}^{t} n_h^{k_q}(s_h^{k',j',m'}, a_h^{k',j',m'}) \tilde{\eta}_{i'}^{N_h^{k_q}} \quad (62)$$

For any $q \in [t]$ and $p \in [n_h^{k_q}]$, by (e) of Lemma F.2, the following relationship holds

$$\frac{\eta_{i'}^{N_h^{k_q}}}{\eta_{i'}^{N_h^{k_q}+p}} \leq \exp(1/H). \quad (63)$$

Combining Equation (63) with the property (c) of Lemma F.3, for any $p \in [n_h^{k_q}]$, we have

$$\tilde{\eta}_{i'}^{N_h^{k_q}} \leq \exp(1/H)\eta_{i'}^{N_h^{k_q}} \leq \exp(2/H)\eta_{i'}^{N_h^{k_q}+p},$$

and thus

$$\sum_{q=1}^{t} n_h^{k_q}(s_h^{k',j',m'}, a_h^{k',j',m'})\tilde{\eta}_{i'}^{N_h^{k_q}} \leq e^{\frac{2}{H}} \sum_{q=1}^{t} \sum_{p=1}^{n_h^{k_q}} \eta_{i'}^{N_h^{k_q}+p} \overset{(i)}{\leq} e^{\frac{2}{H}} \sum_{r=i'}^{\infty} \eta_{i'}^{r} \leq \exp(3/H). \quad (64)$$

Here (i) is because $k_1 < k_2 < ... < k_t \leq K$ and $N_h^{k_1} \geq N_h^{k'+1} \geq i'$. The last inequality is by (c) of Lemma F.2. Applying this inequality to Equation (62), we complete the proof of Equation (60), and consequently, Equation (58). □

**Lemma H.4.** *Under the event* $\bigcap_{i=1}^{4} \mathcal{E}_i$ *in Lemma G.1, for all* $\epsilon \in (0, H)$, *we have the following two conclusions:*

$$\sum_{h=1}^{H} \sum_{k,j,m} \mathbb{I}\left[Q_h^k(s_h^{k,j,m}, a_h^{k,j,m}) - Q_h^{\mathrm{L},k}(s_h^{k,j,m}, a_h^{k,j,m}) > \epsilon\right] \lesssim \frac{H^6 S A \iota}{\epsilon^2} + \frac{MSAH^5\sqrt{\iota}}{\epsilon},$$

*and*

$$\sum_{h=1}^{H} \sum_{k,j,m} \left(Q_h^k - Q_h^{\mathrm{L},k}\right)(s_h^{k,j,m}, a_h^{k,j,m})\mathbb{I}\left[\left(Q_h^k - Q_h^{\mathrm{L},k}\right)(s_h^{k,j,m}, a_h^{k,j,m}) > \epsilon\right]$$

$$\lesssim \frac{H^6 S A \iota}{\epsilon} + MSAH^5\sqrt{\iota}.$$

*Proof.* Let $N = \lceil \log_2(H/\epsilon) \rceil$. For any $i \in [N-1]$, $k \in [K]$ and given $h \in [H]$, let:

$$\omega_{h,i}^{k,j,m} = \mathbb{I}\left[Q_h^k(s_h^{k,j,m}, a_h^{k,j,m}) - Q_h^{\mathrm{L},k}(s_h^{k,j,m}, a_h^{k,j,m}) \in \left[2^{i-1}\epsilon, 2^i\epsilon\right)\right],$$

and

$$\omega_{h,N}^{k,j,m} = \mathbb{I}\left[Q_h^k(s_h^{k,j,m}, a_h^{k,j,m}) - Q_h^{\mathrm{L},k}(s_h^{k,j,m}, a_h^{k,j,m}) \in \left[2^{N-1}\epsilon, H\right]\right].$$

Then

$$\|\omega\|_{\infty,h}^{(i)} = \max_{k,j,m} \omega_{h,i}^{k,j,m} \leq 1, \quad \|\omega\|_{1,h}^{(i)} = \sum_{k,j,m} \omega_{h,i}^{k,j,m}.$$

Now for any $i \in [N]$, we have the following relationship:

$$\sum_{k,j,m} \omega_{h,i}^{k,j,m}\left(Q_h^k - Q_h^{\mathrm{L},k}\right)(s_h^{k,j,m}, a_h^{k,j,m}) \geq 2^{i-1}\epsilon\|\omega\|_{1,h}^{(i)}. \tag{65}$$

Combining the results of Lemma H.3 and Equation (65), we have:

$$2^{i-1}\epsilon\|\omega\|_{1,h}^{(i)} \lesssim \sqrt{H^5 SA\|\omega\|_{1,h}^{(i)}\iota} + \sum_{h'=h}^{H}\sum_{k,j,m} \omega_{h',i}^{k,j,m}(h)Y_{h'}^{k,j,m}, \tag{66}$$

where for any $h \leq h' \leq H-1$,

$$\omega_{h,i}^{k,j,m}(h) := \omega_{h,i}^{k,j,m},$$

$$\omega_{h'+1,i}^{k,j,m}(h) = \sum_{k',j',m'} \omega_{h',i}^{k',j',m'}(h)\mathbb{I}\left[N_{h'}^{k'}(s,a)_{h'}^{k',j',m'} \geq i_0\right]\sum_{i=1}^{N_{h'}^{k'}} \tilde{\eta}_i^{N_{h'}^{k'}}\mathbb{I}\left[(k^i, j^i, m^i) = (k,j,m)\right],$$

Therefore, for any triple $(k,j,m)$ and $h \leq h' \leq H-1$, we have

$$\sum_{i=1}^{N} \omega_{h'+1,i}^{k,j,m}(h) = \sum_{k',j',m'}\left(\sum_{i=1}^{N}\omega_{h',i}^{k',j',m'}(h)\right)\mathbb{I}\left[N_{h'}^{k'} \geq i_0\right]\sum_{i=1}^{N_{h'}^{k'}} \tilde{\eta}_i^{N_{h'}^{k'}}\mathbb{I}\left[(k^i, j^i, m^i) = (k,j,m)\right]$$

Then by mathematical induction on $h' \in [h, H]$, it is straightforward to prove that for any $j \in [K]$,

$$\sum_{i=1}^{N} \omega_{h',i}^{k,j,m}(h) \leq (\exp(3/H))^{h'-h} < 27, \tag{67}$$

given Equation (60) and the base case $\sum_{i=1}^{N} \omega_{h,i}^{k,j,m}(h) = \sum_{i=1}^{N} \omega_{h,i}^{k,j,m} \leq 1$. Solving Equation (66), we can derive the following relationship:

$$\|\omega\|_{1,h}^{(i)} \lesssim \frac{H^5 SA\iota}{4^i\epsilon^2} + \frac{\sum_{h'=h}^{H}\sum_{k,j,m}\omega_{h',i}^{k,j,m}(h)Y_{h'}^{k,j,m}}{2^i\epsilon}. \tag{68}$$

We claim that

$$\sum_{h'=1}^{H}\sum_{k,j,m} Y_{h'}^{k,j,m} \lesssim MH^4 SA\sqrt{\iota}, \tag{69}$$

which will be proved later. Therefore, by

$$\mathbb{I}\left[\left(Q_h^k - Q_h^{\mathrm{L},k}\right)(s_h^{k,j,m}, a_h^{k,j,m}) \geq \epsilon\right] = \sum_{i=1}^{N} \omega_{h,i}^{k,j,m},$$

we have

$$\sum_{h=1}^{H}\sum_{k,j,m} \mathbb{I}\left[Q_h^k(s_h^{k,j,m}, a_h^{k,j,m}) - Q_h^{\mathrm{L},k}(s_h^{k,j,m}, a_h^{k,j,m}) \geq \epsilon\right] = \sum_{h=1}^{H}\sum_{i=1}^{N} \|\omega\|_{1,h}^{(i)}. \tag{70}$$

By Equation (68), it holds that

$$\sum_{i=1}^{N} \|\omega\|_{1,h}^{(i)} \lesssim \sum_{i=1}^{N} \frac{H^5 SA\iota}{4^i \epsilon^2} + \sum_{i=1}^{N} \frac{\sum_{h'=h}^{H} \sum_{k,j,m} \omega_{h',i}^{k,j,m}(h) Y_{h'}^{k,j,m}}{2^i \epsilon}$$

$$\lesssim \frac{H^5 SA\iota}{\epsilon^2} + \sum_{i=1}^{N} \frac{\sum_{h'=1}^{H} \sum_{k,j,m} Y_{h'}^{k,j,m}}{2^i \epsilon}$$

$$\lesssim \frac{H^5 SA\iota}{\epsilon^2} + \frac{MH^4 SA\sqrt{\iota}}{\epsilon}. \tag{71}$$

Here, the second inequality is because $0 \le \omega_{h',i}^{k,j,m}(h) < 27$ by Equation (67). The last inequality is because of Equation (69). Combing the results of Equation (70) and Equation (71), we reach

$$\sum_{h=1}^{H} \sum_{k,j,m} \mathbb{I}\left[ Q_h^k(s_h^{k,j,m}, a_h^{k,j,m}) - Q_h^{L,k}(s_h^{k,j,m}, a_h^{k,j,m}) \ge \epsilon \right] \lesssim \frac{H^6 SA\iota}{\epsilon^2} + \frac{MH^5 SA\sqrt{\iota}}{\epsilon}.$$

Now we finish the proof of the first conclusion. Further, noting that

$$\sum_{h=1}^{H} \sum_{k,j,m} \left( Q_h^k - Q_h^{L,k} \right)(s_h^{k,j,m}, a_h^{k,j,m}) \mathbb{I}\left[ \left( Q_h^k - Q_h^{L,k} \right)(s_h^{k,j,m}, a_h^{k,j,m}) \ge \epsilon \right]$$

$$\le \sum_{h=1}^{H} \sum_{i=1}^{N} 2^i \epsilon \|\omega\|_{1,h}^{(i)}$$

$$\lesssim \sum_{h=1}^{H} \sum_{i=1}^{N} \frac{H^5 SA\iota}{2^i \epsilon} + \sum_{h=1}^{H} \sum_{h'=h}^{H} \sum_{k,j,m} \left( \sum_{i=1}^{N} \omega_{h',i}^{k,j,m}(h) \right) Y_{h'}^{k,j,m}$$

$$\lesssim \frac{H^6 SA\iota}{\epsilon} + \sum_{h=1}^{H} \sum_{h'=h}^{H} \sum_{k,j,m} Y_{h'}^{k,j,m}$$

$$\lesssim \frac{H^6 SA\iota}{\epsilon} + MH^5 SA\sqrt{\iota}.$$

Here, the second inequality is by Equation (68). The second last inequality is by Equation (67), and the last inequality is because of Equation (69). Next, we only need to prove Equation (69).

**Proof of Equation (69):** By definition of $Y_{h'}^{k,m,j}$, we have the following equation

$$\sum_{k,j,m} Y_{h'}^{k,j,m} = \sum_{k,j,m} \eta_0^{N_{h'}^k} H + H \sum_{k,j,m} \mathbb{I}\left[ 0 < N_{h'}^k < i_0 \right] + \sum_{k,j,m} \sqrt{\frac{H^3 \iota}{N_{h'}^k}} \mathbb{I}\left[ 0 < N_{h'}^k < M \right]. \tag{72}$$

For the first term of Equation (72), we have

$$\sum_{k,j,m} \eta_0^{N_{h'}^k} H \le H \sum_{s,a} \sum_{k,j,m} \mathbb{I}[N_{h'}^k(s,a) = 0, (s_{h'}^{k,j,m}, a_{h'}^{k,j,m}) = (s,a)] \le MHSA. \tag{73}$$

The last inequality is because if we let $k_0(s,a)$ be the round index such that $N_{h'}^{k_0}(s,a) = 0$ and $N_{h'}^{k_0+1}(s,a) > 0$, then by (a) of Lemma F.1, it holds that

$$\sum_{k,j,m} \mathbb{I}[N_{h'}^k(s,a) = 0, (s_{h'}^{k,j,m}, a_{h'}^{k,j,m}) = (s,a)] = n_{h'}^{k_0}(s,a) \le M.$$

Let $k_1(s,a) = \max\{k \mid 1 \le k \le K, N_{h'}^k(s,a) < i_0\}$. Then for the second term of Equation (72)

$$\sum_{k,j,m} H\mathbb{I}\left[ 0 < N_{h'}^k(s_{h'}^{k,j,m}, a_{h'}^{k,j,m}) < i_0 \right]$$

$$= H \sum_{s,a} \sum_{k,j,m} \mathbb{I}\left[ 0 < N_{h'}^k(s,a) < i_0, (s_{h'}^{k,j,m}, a_{h'}^{k,j,m}) = (s,a) \right]$$

$$\le H \sum_{s,a} \sum_{k=1}^{k_1} \sum_{j,m} \mathbb{I}\left[ (s_{h'}^{k,j,m}, a_{h'}^{k,j,m}) = (s,a) \right] = H \sum_{s,a} N_{h'}^{k_1+1}(s,a) \lesssim MH^3 SA. \tag{74}$$

Here, the last inequality is because $N_{h'}^{k_1}(s,a) < i_0$ and thus $n_{h'}^{k_1}(s,a) \le 2M$ by (a) of Lemma F.1. Finally, for the last term of Equation (72), by Equation (24) with $\alpha = 1/2$ and $\omega_h^{k,j,m} = 1$, we have

$$\sum_{k,j,m} \sqrt{\frac{H^3\iota}{N_{h'}^k}} \mathbb{I}\left[0 < N_{h'}^k(s_{h'}^{k,j,m}, a_{h'}^{k,j,m}) < M\right] \le 2M\sqrt{H^3}SA\sqrt{\iota}. \tag{75}$$

By applying Equation (73), Equation (74) and Equation (75) to Equation (72), we finish the proof. $\quad\square$

In the next lemma, we will bound the difference between $V_h^{\mathrm{R},k}(s)$ and $V_h^k(s)$.

**Lemma H.5.** *Under $\bigcap_{i=1}^4 \mathcal{E}_i$ in Lemma G.1, it holds for any $(s,h,k) \in \mathcal{S} \times [H] \times [K]$ that:*
$$0 \le V_h^{\mathrm{R},k}(s) - V_h^k(s) \le \beta.$$

*Proof.* If for any $k \in [K+1]$, $V_h^k(s) - V_h^{\mathrm{L},k}(s) > \beta$, then according to the update rule of the reference function in Algorithm 1, we know $V_h^{\mathrm{R},k} - V_h^k(s) = 0$.

Otherwise we can assume that there exists $k \in [K+1]$ such that $V_h^k(s) - V_h^{\mathrm{L},k}(s) \le \beta$. Define
$$k_1 = \min\{k \mid V_h^k(s) - V_h^{\mathrm{L},k}(s) \le \beta\}.$$
Then for any $k < k_1$, it holds that $V_h^k(s) - V_h^{\mathrm{L},k}(s) > \beta$ and thus $V_h^{\mathrm{R},k} - V_h^k(s) = 0$.

We claim that $u_h^{\mathrm{R},k_1-1}(s) = \text{True}$. If $u_h^{\mathrm{R},k_1-1}(s) = \text{False}$, then there exists $k_0 \le k_1 - 1$ such that $u_h^{\mathrm{R},k_0-1}(s) = \text{True}$ and $u_h^{\mathrm{R},k_0}(s) = \text{False}$. Based on the update rule of the reference function, in this case, we have $V_h^{k_0}(s) - V_h^{\mathrm{L},k_0}(s) \le \beta$, which is contradictory to the minimality of $k_1$.

Since $V_h^{k_1}(s) - V_h^{\mathrm{L},k_1}(s) \le \beta$ and $u_h^{\mathrm{R},k_1-1}(s) = \text{True}$, we know for any $k \ge k_1$,
$$V_h^{\mathrm{R},k}(s) = V_h^{\mathrm{R},k_1}(s) = V_h^{k_1}(s) \le V_h^{\mathrm{L},k_1}(s) + \beta \le V_h^\star(s) + \beta \le V_h^k(s) + \beta.$$
The last two inequalities are because $V_h^{\mathrm{L},k_1}(s) \le V_h^\star(s) \le V_h^k(s)$ by Lemma H.2 and Lemma H.1. For any $k \ge k_1$, we also have $V_h^{\mathrm{R},k}(s) = V_h^{\mathrm{R},k_1}(s) = V_h^{k_1}(s) \ge V_h^k(s)$ and thus finish the proof. $\quad\square$

**Lemma H.6.** *Under the event $\bigcap_{i=1}^4 \mathcal{E}_i$ in Lemma G.1, for any $(s,h,k) \in \mathcal{S} \times [H] \times [K]$, we have the following two conclusions:*

- *If $V_{h+1}^k(s) - V_{h+1}^{\mathrm{L},k}(s) \le \beta$, then $V_{h+1}^{\mathrm{R},K+1}(s) = V_{h+1}^{\mathrm{R},k}(s) = \hat{V}_{h+1}^{\mathrm{R},k}(s)$.*

- *If $V_{h+1}^k(s) - V_{h+1}^{\mathrm{L},k}(s) > \beta$, then we have:*
$$0 \le V_{h+1}^{\mathrm{R},k}(s) - \hat{V}_{h+1}^{\mathrm{R},k}(s), \; |\hat{V}_{h+1}^{\mathrm{R},k}(s) - V_{h+1}^{\mathrm{R},K+1}(s)| \le V_{h+1}^k(s) - V_{h+1}^{\mathrm{L},k}(s).$$

*Proof.* 
- If for given $k \in [K]$, $V_{h+1}^k(s) - V_{h+1}^{\mathrm{L},k}(s) \le \beta$, then there exists $k_1 \in [K]$ such that:
$$k_1 = \min\left\{k : V_{h+1}^k(s) - V_{h+1}^{\mathrm{L},k}(s) \le \beta\right\}.$$
Then according the analysis in Lemma H.5, we have $u_{h+1}^{\mathrm{R},k_1-1}(s) = \text{True}$, or it is contradictory to the minimality of $k_1$. Therefore, in this case, we have:
$$V_{h+1}^{\mathrm{R},K+1}(s) = V_{h+1}^{\mathrm{R},k}(s) = V_{h+1}^{\mathrm{R},k_1}(s) = V_{h+1}^{k_1}(s) \le V_{h+1}^{\mathrm{L},k_1}(s) + \beta \le V_{h+1}^\star(s) + \beta,$$
and
$$V_{h+1}^{\mathrm{R},k}(s) = V_{h+1}^{\mathrm{R},k_1}(s) = V_{h+1}^{k_1}(s) \ge V_{h+1}^\star(s).$$
According to the definition of $\hat{V}_{h+1}^{\mathrm{R},k}(s)$, we have $\hat{V}_{h+1}^{\mathrm{R},k}(s) = V_{h+1}^{\mathrm{R},k}(s) = V_{h+1}^{\mathrm{R},K+1}(s)$.

- Moreover, if $V_{h+1}^k(s) - V_{h+1}^{\mathrm{L},k}(s) > \beta$, according to the algorithm, we have $V_{h+1}^{\mathrm{R},k}(s) = V_{h+1}^k(s)$ and then $0 \le V_{h+1}^{\mathrm{R},k}(s) - \hat{V}_{h+1}^{\mathrm{R},k}(s) \le V_{h+1}^k(s) - V_{h+1}^{\mathrm{L},k}(s)$.

In this case, we also have $V_{h+1}^{\mathrm{L},k}(s) \le V_{h+1}^\star(s) \le V_{h+1}^{\mathrm{R},K+1}(s) \le V_{h+1}^{\mathrm{R},k}(s) = V_{h+1}^k(s)$ and then $V_{h+1}^{\mathrm{L},k}(s) \le V_{h+1}^\star(s) \le \hat{V}_{h+1}^{\mathrm{R},k}(s) \le V_{h+1}^{\mathrm{R},k}(s) = V_{h+1}^k(s)$. These two inequalities imply that $|\hat{V}_{h+1}^{\mathrm{R},k}(s) - V_{h+1}^{\mathrm{R},K+1}(s)| \le V_{h+1}^k(s) - V_{h+1}^{\mathrm{L},k}(s)$.

$\quad\square$

# I Proof of the Worst-Case Regret (Theorem 4.1 and Theorem 4.3)

## I.1 Proof Sketch

In this section, we bound the worst-case regret under the event $\bigcap_{i=1}^{14} \mathcal{E}_i$ in Lemma G.1.

For $h \in [H+1]$, denote:

$$\delta_h^k = \sum_{j=1}^{n^{m,k}} \sum_{m=1}^{M} \left(V_h^k - V_h^\star\right)(s_h^{k,j,m}), \zeta_h^k = \sum_{j=1}^{n^{m,k}} \sum_{m=1}^{M} \left(V_h^k - V_h^{\pi^k}\right)(s_h^{k,j,m}).$$

Here, $\delta_{H+1}^k = \zeta_{H+1}^k = 0$. Because $V_h^\star(s) = \sup_\pi V_h^\pi(s)$, we have $\delta_h^k \leq \zeta_h^k$ for any $h \in [H+1]$. In addition, as $V_h^k(s) \geq V_h^\star(s)$ for all $(s,h,k) \in \mathcal{S} \times [H] \times [K]$, according to Lemma H.1, we have:

$$\text{Regret}(T) = \sum_{k,j,m} \left(V_1^\star(s_1^{k,j,m}) - V_1^{\pi^k}(s_1^{k,j,m})\right) \leq \sum_{k,j,m} \left(V_1^k(s_1^{k,j,m}) - V_1^{\pi^k}(s_1^{k,j,m})\right) = \sum_{k=1}^{K} \zeta_1^k.$$

Thus, we only need to bound $\sum_{k=1}^{K} \zeta_1^k$.

$$\begin{aligned}
\sum_{k=1}^{K} \zeta_h^k &= \sum_{k,j,m} (Q_h^k - Q_h^{\pi^k})(s_h^{k,j,m}, a_h^{k,j,m}) \\
&= \sum_{k,j,m} (Q_h^k - Q_h^\star)(s_h^{k,j,m}, a_h^{k,j,m}) + \sum_{k,j,m} (Q_h^\star - Q_h^{\pi^k})(s_h^{k,j,m}, a_h^{k,j,m}) \\
&\leq \sum_{k,j,m} (Q_h^{\text{R},k} - Q_h^\star)(s_h^{k,j,m}, a_h^{k,j,m}) + \sum_{k,j,m} \mathbb{P}_{s_h^{k,j,m}, a_h^{k,j,m}, h}(V_{h+1}^\star - V_{h+1}^{\pi^k}). \quad (76)
\end{aligned}$$

In the last inequality, we use $Q_h^k(s,a) \leq Q_h^{\text{R},k}(s,a)$ and Equation (1):

$$Q_h^\star(s,a) = r_h(s,a) + \mathbb{P}_{s,a,h} V_{h+1}^\star, \quad Q_h^{\pi^k}(s,a) = r_h(s,a) + \mathbb{P}_{s,a,h} V_{h+1}^{\pi^k}.$$

Next, we will bound the first term of Equation (76). Back to Equation (52), since $\hat{V}_{h+1}^{\text{R},k^n} \leq V_{h+1}^{\text{R},k^n}$, we have the following relationship (Here we use the shorthand $\mathbb{P} = \mathbb{P}_{s_h^{k,j,m}, a_h^{k,j,m}, h}$):

$$(Q_h^{\text{R},k} - Q_h^\star)(s_h^{k,j,m}, a_h^{k,j,m})$$

$$\leq \eta_0^{N_h^k} H + \sum_{n=1}^{N_h^k} \tilde{\eta}_n^{N_h^k} \left( (V_{h+1}^{k^n} - V_{h+1}^{\text{R},k^n})(s_{h+1}^{k^n,j^n,m^n}) + \mu_h^{\text{R},k^n+1} - \mathbb{P}V_{h+1}^\star \right) + \sum_{n=1}^{N_h^k} \eta_n^{N_h^k} b_{h,n}^{\text{R}}$$

$$\leq \eta_0^{N_h^k} H + \sum_{n=1}^{N_h^k} \tilde{\eta}_n^{N_h^k} \left( (V_{h+1}^{k^n} - \hat{V}_{h+1}^{\text{R},k^n})(s_{h+1}^{k^n,j^n,m^n}) + \mu_h^{\text{R},k^n+1} - \mathbb{P}V_{h+1}^\star \right) + \sum_{n=1}^{N_h^k} \eta_n^{N_h^k} b_{h,n}^{\text{R}}, \quad (77)$$

and thus

$$\sum_{k,j,m} (Q_h^{\text{R},k} - Q_h^\star)(s_h^{k,j,m}, a_h^{k,j,m})$$

$$\leq \sum_{k,j,m} \left\{ \eta_0^{N_h^k} H + \sum_{n=1}^{N_h^k} \tilde{\eta}_n^{N_h^k} \left( (V_{h+1}^{k^n} - \hat{V}_{h+1}^{\text{R},k^n})(s_{h+1}^{k^n,j^n,m^n}) + \mu_h^{\text{R},k^n+1} - \mathbb{P}V_{h+1}^\star \right) + \sum_{n=1}^{N_h^k} \eta_n^{N_h^k} b_{h,n}^{\text{R}} \right\}$$

$$\leq \sum_{k,j,m} \eta_0^{N_h^k} H + \sum_{k,j,m,N_h^k>0} \left( \beta_h^{\text{R},k}(s_h^{k,j,m}, a_h^{k,j,m}) + 2c_b^{\text{R},2} \frac{H^2 \iota}{N_h^k} \right)$$

$$+ \sum_{k,j,m,N_h^k>0} \sum_{n=1}^{N_h^k} \tilde{\eta}_n^{N_h^k} (V_{h+1}^{k^n} - V_{h+1}^\star)(s_{h+1}^{k^n,j^n,m^n})$$

$$+ \sum_{k,j,m,N_h^k>0} \sum_{n=1}^{N_h^k} \tilde{\eta}_n^{N_h^k} \left( (V_{h+1}^\star - \hat{V}_{h+1}^{\text{R},k^n})(s_{h+1}^{k^n,j^n,m^n}) + \mu_h^{\text{R},k^n+1} - \mathbb{P}_{s_h^{k,j,m}, a_h^{k,j,m}, h} V_{h+1}^\star \right). \quad (78)$$

The last inequality is because

$$\sum_{n=1}^{N_h^k} \eta_n^{N_h^k} b_{h,n}^{\text{R}} = \beta_h^{\text{R},k} + \sum_{n=1}^{N_h^k} \frac{c_b^{\text{R},2} \eta_n^{N_h^k} H^2 \iota}{n} \le \beta_h^{\text{R},k} + 2 c_b^{\text{R},2} \frac{H^2 \iota}{N_h^k}. \tag{79}$$

by Equation (53) and (a) of Lemma F.2. For the first term of Equation (78), similar to Equation (73), we have

$$\sum_{k,j,m} \eta_0^{N_h^k} H = H \sum_{s,a} \sum_{k,j,m} \mathbb{I}[N_h^k = 0, (s_h^{k,j,m}, a_h^{k,j,m}) = (s,a)] \le MHSA. \tag{80}$$

For the second term of Equation (78), by Lemma F.6, it holds that

$$\sum_{k,j,m,N_h^k>0} 2 c_b^{\text{R},2} \frac{H^2 \iota}{N_h^k} \le 2 c_b^{\text{R},2} MH^2 SA\iota + 2 c_b^{\text{R},2} H^2 SA\iota^2. \tag{81}$$

For the third term of Equation (78), similar to the proof of [113, Equation (25)], it holds that

$$\sum_{k,j,m,N_h^k>0} \sum_{n=1}^{N_h^k} \tilde{\eta}_n^{N_h^k} \left(V_{h+1}^{k^n} - V_{h+1}^\star\right)(s_{h+1}^{k^n,j^n,m^n}) \le \exp\left(\frac{3}{H}\right) \sum_{k=1}^K \delta_{h+1}^k + 2MH^3 SA. \tag{82}$$

Taking the above results Equation (80), Equation (81) and Equation (82) together with Equation (78), we can rearrange terms of Equation (76) to reach

$$\sum_{k=1}^K \zeta_h^k \le \exp\left(\frac{3}{H}\right) \sum_{k=1}^K \delta_{h+1}^k + (2 c_b^{\text{R},2} + 3)MH^3 SA\iota + 2 c_b^{\text{R},2} H^2 SA\iota^2$$

$$+ \sum_{k,j,m,N_h^k>0} \sum_{n=1}^{N_h^k} \tilde{\eta}_n^{N_h^k} \left( (V_{h+1}^\star - \hat{V}_{h+1}^{\text{R},k^n})(s_{h+1}^{k^n,j^n,m^n}) + \mu_h^{\text{R},k^n+1} - \mathbb{P}_{s_h^{k,j,m}, a_h^{k,j,m}, h} V_{h+1}^\star \right)$$

$$+ \sum_{k,j,m} \mathbb{P}_{s_h^{k,j,m}, a_h^{k,j,m}, h} \left( V_{h+1}^\star - V_{h+1}^{\pi^k} \right) + \sum_{k,j,m,N_h^k>0} \beta_h^{\text{R},k}(s_h^{k,j,m}, a_h^{k,j,m})$$

$$\le \exp\left(\frac{3}{H}\right) \sum_{k=1}^K \zeta_{h+1}^k + (2 c_b^{\text{R},2} + 3)MH^3 SA\iota + 2 c_b^{\text{R},2} H^2 SA\iota^2$$

$$+ \sum_{k,j,m,N_h^k>0} \sum_{n=1}^{N_h^k} \tilde{\eta}_n^{N_h^k} \left( \mathbb{1}_{s_{h+1}^{k^n,j^n,m^n}} - \mathbb{P}_{s_h^{k,j,m}, a_h^{k,j,m}, h} \right) \left( V_{h+1}^\star - \hat{V}_{h+1}^{\text{R},k^n} \right)$$

$$+ \sum_{k,j,m,N_h^k>0} \sum_{n=1}^{N_h^k} \tilde{\eta}_n^{N_h^k} \frac{1}{N_h^{k^n+1}} \sum_{i=1}^{N_h^{k^n+1}} \left( V_h^{\text{R},k^i}(s_{h+1}^{k^i,j^i,m^i}) - \mathbb{P}_{s_h^{k,j,m}, a_h^{k,j,m}, h} \hat{V}_{h+1}^{\text{R},k^n} \right)$$

$$+ \sum_{k,j,m} \left( \mathbb{P}_{s_h^{k,j,m}, a_h^{k,j,m}, h} - \mathbb{1}_{s_h^{k,j,m}} \right) \left( V_{h+1}^\star - V_{h+1}^{\pi^k} \right) + \sum_{k,j,m,N_h^k>0} \beta_h^{\text{R},k}(s_h^{k,j,m}, a_h^{k,j,m}).$$

By recursion on $h$, since $\zeta_{H+1}^k = 0$, we can get the following conclusion:

$$\text{Regret}(T) \le \sum_{k=1}^K \zeta_1^k \lesssim MH^4 SA\iota^2 + R_1 + R_2 + R_3 + R_4,$$

where

$$R_1 = \sum_{h=1}^H c_H^{h-1} \sum_{k,j,m,N_h^k>0} \beta_h^{\text{R},k}(s_h^{k,j,m}, a_h^{k,j,m}).$$

$$R_2 = \sum_{h=1}^H c_H^{h-1} \sum_{k,j,m,N_h^k>0} \sum_{n=1}^{N_h^k} \tilde{\eta}_n^{N_h^k} \left( \mathbb{1}_{s_{h+1}^{k^n,j^n,m^n}} - \mathbb{P}_{s_h^{k,j,m}, a_h^{k,j,m}, h} \right) \left( V_{h+1}^\star - \hat{V}_{h+1}^{\text{R},k^n} \right).$$

$$R_3 = \sum_{h=1}^{H} c_H^{h-1} \sum_{k,j,m,N_h^k>0} \sum_{n=1}^{N_h^k} \tilde{\eta}_n^{N_h^k} \frac{1}{N_h^{k^n+1}} \sum_{i=1}^{N_h^{k^n+1}} \left( V_h^{\mathrm{R},k^i}(s_{h+1}^{k^i,j^i,m^i}) - \mathbb{P}_{s_h^{k,j,m},a_h^{k,j,m},h} \hat{V}_{h+1}^{\mathrm{R},k^n} \right).$$

$$R_4 = \sum_{h=1}^{H} c_H^{h-1} \sum_{k,j,m,N_h^k>0} \left( \mathbb{P}_{s_h^{k,j,m},a_h^{k,j,m},h} - \mathbb{1}_{s_{h+1}^{k,j,m}} \right) \left( V_{h+1}^{\star} - V_{h+1}^{\pi^k} \right).$$

and $c_H = \exp(3/H)$. Under the event $\mathcal{E}_5$ in Lemma G.1, we can bound $R_4$ by $27\sqrt{2H^2T_1\iota}$. Then, we reach

$$\mathrm{Regret}(T) \lesssim \sqrt{H^2T_1\iota} + MH^4SA\iota^2 + R_1 + R_2 + R_3.$$

By Lemma I.1, Lemma I.2 and Lemma I.3, we have:

$$\mathrm{Regret}(T) \lesssim (1+\beta)\sqrt{H^2SAT_1\iota^2} + \frac{1}{\beta}H^6SA\iota^2 + MH^5SA\iota^2.$$

By (b) of Lemma F.1, we have

$$\iota = \log\left(\frac{2SAT_1}{\delta}\right) \leq O\left(\log\left(\frac{2SA\hat{T}}{\delta}\right) + \log\left(\frac{2MHSA}{\delta}\right)\right) = O\left(\log\left(\frac{MSAT}{\delta}\right)\right).$$

Let $p = \delta/14$ and $\iota_1 = \log\left(\frac{MSAT}{p}\right)$, then with probability at least $1 - p$, we have

$$\mathrm{Regret}(T) \leq O\left((1+\beta)\sqrt{MH^2SAT\iota_1^2} + \frac{1}{\beta}H^6SA\iota_1^2 + MH^5SA\iota_1^2\right).$$

The proof also holds for $M = 1$.

## I.2   Upper bounds of $R_1$, $R_2$, $R_3$

**Lemma I.1.** *Under the event $\bigcap_{i=1}^{14} \mathcal{E}_i$ in Lemma G.1, we have*

$$R_1 \lesssim (1+\beta)\sqrt{H^2SAT_1\iota} + \sqrt{\frac{1}{\beta}}H^4SA\iota^2 + MH^4SA\iota^2.$$

*Proof.* Since $\beta_h^{\mathrm{R},k}(s_h^{k,j,m}, a_h^{k,j,m}) \geq 0$, we have

$$R_1 \lesssim \sum_{h=1}^{H} \sum_{k,j,m,N_h^k>0} \beta_h^{\mathrm{R},k}(s_h^{k,j,m}, a_h^{k,j,m})$$

$$\lesssim \sum_{h=1}^{H} \sum_{k,j,m,N_h^k>0} \sqrt{\frac{\iota}{N_h^k}} \left( \sqrt{(\sigma_h^{\mathrm{R},k} - (\mu_h^{\mathrm{R},k})^2)} + \sqrt{H\left(\sigma_h^{\mathrm{A},k} - (\mu_h^{\mathrm{A},k})^2\right)} \right) \tag{83}$$

For the second term, we make the observation that

$$\sqrt{\frac{\sigma_h^{\mathrm{A},k} - (\mu_h^{\mathrm{A},k})^2}{N_h^k(s_h^{k,j,m}, a_h^{k,j,m})}} \leq \sqrt{\frac{\sigma_h^{\mathrm{A},k}}{N_h^k}} = \sqrt{\frac{\sum_{n=1}^{N_h^k} \tilde{\eta}_n^{N_h^k}(V_{h+1}^{k^n} - V_{h+1}^{\mathrm{R},k^n})^2(s_{h+1}^{k^n,j^n,m^n})}{N_h^k}} \leq \beta\sqrt{\frac{1}{N_h^k}}. \tag{84}$$

The last inequality is because $0 \leq V_{h+1}^{\mathrm{R},k^n} - V_{h+1}^{k^n} \leq \beta$ by Lemma H.5 and $\sum_{n=1}^{N_h^k} \tilde{\eta}_n^{N_h^k} \leq 1$ by (b) of Lemma F.3. Therefore, we can bound the second term by Lemma F.6 with $\omega_h^{k,m,j} = 1$ and $\alpha = \frac{1}{2}$:

$$\sum_{h=1}^{H} \sum_{k,j,m,N_h^k>0} \sqrt{\frac{\iota}{N_h^k}} \sqrt{H\left(\sigma_h^{\mathrm{A},k} - (\mu_h^{\mathrm{A},k})^2\right)}$$

$$\leq \sum_{h=1}^{H} \sum_{k,j,m,N_h^k>0} \beta\sqrt{\frac{H\iota}{N_h^k}}$$

$$\lesssim \beta MH^{\frac{3}{2}}SA\sqrt{\iota} + \beta\sqrt{H^2SAT_1\iota}. \tag{85}$$

Next, we will bound the first term of Equation (83). Since $V_{h+1}^{\mathrm{R},k}(s) \geq \hat{V}_{h+1}^{\mathrm{R},k}(s)$, we have

$$\sqrt{\frac{\sigma_h^{\mathrm{R},k} - \left(\mu_h^{\mathrm{R},k}\right)^2}{N_h^k(s_h^{k,j,m}, a_h^{k,j,m})}} \leq \sqrt{\frac{J_{1,h}^{k,j,m} + J_{2,h}^{k,j,m}}{N_h^k(s_h^{k,j,m}, a_h^{k,j,m})}}, \tag{86}$$

where:

$$J_{1,h}^{k,j,m} = \frac{\sum_{n=1}^{N_h^k} \left(V_{h+1}^{\mathrm{R},k^n}(s_{h+1}^{k^n,j^n,m^n})\right)^2}{N_h^k(s_h^{k,j,m}, a_h^{k,j,m})} - \frac{\sum_{n=1}^{N_h^k} \left(\hat{V}_{h+1}^{\mathrm{R},k^n}(s_{h+1}^{k^n,j^n,m^n})\right)^2}{N_h^k(s_h^{k,j,m}, a_h^{k,j,m})}$$

and

$$J_{2,h}^{k,j,m} = \frac{\sum_{n=1}^{N_h^k} \left(\hat{V}_{h+1}^{\mathrm{R},k^n}(s_{h+1}^{k^n,j^n,m^n})\right)^2}{N_h^k(s_h^{k,j,m}, a_h^{k,j,m})} - \left(\frac{\sum_{n=1}^{N_h^k} \hat{V}_{h+1}^{\mathrm{R},k^n}(s_{h+1}^{k^n,j^n,m^n})}{N_h^k(s_h^{k,j,m}, a_h^{k,j,m})}\right)^2.$$

Now we want to bound both $J_{1,h}^{k,j,m}$ and $J_{2,h}^{k,j,m}$. Note that

$$J_{1,h}^{k,j,m} = \frac{\sum_{n=1}^{N_h^k} \left(V_{h+1}^{\mathrm{R},k^n} + \hat{V}_{h+1}^{\mathrm{R},k^n}\right)\left(V_{h+1}^{\mathrm{R},k^n} - \hat{V}_{h+1}^{\mathrm{R},k^n}\right)(s_{h+1}^{k^n,j^n,m^n})}{N_h^k(s_h^{k,j,m}, a_h^{k,j,m})}$$

$$\leq \frac{2H\Psi_h^k(s_h^{k,j,m}, a_h^{k,j,m})}{N_h^k(s_h^{k,j,m}, a_h^{k,j,m})}. \tag{87}$$

where

$$\Psi_h^k(s_h^{k,j,m}, a_h^{k,j,m}) = \sum_{n=1}^{N_h^k} \left(V_{h+1}^{\mathrm{R},k^n}(s_{h+1}^{k^n,j^n,m^n}) - \hat{V}_{h+1}^{\mathrm{R},k^n}(s_{h+1}^{k^n,j^n,m^n})\right). \tag{88}$$

For the second term $J_{2,h}^{k,j,m}$, because of Cauchy's Inequality, we have:

$$J_{2,h}^{k,j,m} = \frac{\sum_{n=1}^{N_h^k} \left(\hat{V}_{h+1}^{\mathrm{R},k^n}(s_{h+1}^{k^n,j^n,m^n}) - \frac{\sum_{i=1}^{N_h^k} \hat{V}_{h+1}^{\mathrm{R},k^i}(s_{h+1}^{k^i,j^i,m^i})}{N_h^k(s_h^{k,j,m}, a_h^{k,j,m})}\right)^2}{N_h^k(s_h^{k,j,m}, a_h^{k,j,m})}$$

$$\leq 2\left(J_{2,h,1}^{k,j,m} + J_{2,h,2}^{k,j,m}\right),$$

where:

$$J_{2,h,1}^{k,j,m} = \frac{\sum_{n=1}^{N_h^k} \left(\left(\hat{V}_{h+1}^{\mathrm{R},k^n} - V_{h+1}^\star\right)(s_{h+1}^{k^n,j^n,m^n}) + \frac{\sum_{i=1}^{N_h^k} \left(V_{h+1}^\star - \hat{V}_{h+1}^{\mathrm{R},k^i}\right)(s_{h+1}^{k^i,j^i,m^i})}{N_h^k(s_h^{k,j,m}, a_h^{k,j,m})}\right)^2}{N_h^k(s_h^{k,j,m}, a_h^{k,j,m})},$$

and

$$J_{2,h,2}^{k,j,m} = \frac{\sum_{n=1}^{N_h^k} \left(V_{h+1}^\star(s_{h+1}^{k^n,j^n,m^n}) - \frac{\sum_{i=1}^{N_h^k} V_{h+1}^\star(s_{h+1}^{k^i,j^i,m^i})}{N_h^k(s_h^{k,j,m}, a_h^{k,j,m})}\right)^2}{N_h^k(s_h^{k,j,m}, a_h^{k,j,m})}$$

$$= \frac{\sum_{n=1}^{N_h^k} \left(V_{h+1}^\star(s_{h+1}^{k^n,j^n,m^n})\right)^2}{N_h^k(s_h^{k,j,m}, a_h^{k,j,m})} - \left(\frac{\sum_{n=1}^{N_h^k} V_{h+1}^\star(s_{h+1}^{k^n,j^n,m^n})}{N_h^k(s_h^{k,j,m}, a_h^{k,j,m})}\right)^2.$$

Since $V_{h+1}^\star(s) \leq \hat{V}_{h+1}^{\mathrm{R},k^n}(s) \leq V_{h+1}^\star(s) + \beta$, it holds that:

$$\left|\left(\hat{V}_{h+1}^{\mathrm{R},k^n} - V_{h+1}^\star\right)(s_{h+1}^{k^n,j^n,m^n}) + \frac{\sum_{i=1}^{N_h^k} \left(V_{h+1}^\star - \hat{V}_{h+1}^{\mathrm{R},k^i}\right)(s_{h+1}^{k^i,j^i,m^i})}{N_h^k(s_h^{k,j,m}, a_h^{k,j,m})}\right|$$

$$\leq \left|\left(\hat{V}_{h+1}^{\mathrm{R},k^n} - V_{h+1}^\star\right)(s_{h+1}^{k^n,j^n,m^n})\right| + \left|\frac{\sum_{i=1}^{N_h^k} \left(V_{h+1}^\star - \hat{V}_{h+1}^{\mathrm{R},k^i}\right)(s_{h+1}^{k^i,j^i,m^i})}{N_h^k(s_h^{k,j,m}, a_h^{k,j,m})}\right|$$

$$\leq 2\beta.$$

Therefore, applying this inequality to $J_{2,h,1}^{k,j,m}$, we have $J_{2,h,1}^{k,j,m} \leq 4\beta^2$.

Moreover, we claim that:

$$J_{2,h,2}^{k,j,m} \lesssim \mathbb{V}_{s_h^{k,j,m}, a_h^{k,j,m}, h}(V_{h+1}^\star) + H^2\sqrt{\frac{\iota}{N_h^k(s_h^{k,j,m}, a_h^{k,j,m})}}. \tag{89}$$

This is because (Here we use the shorthand $\mathbb{P} = \mathbb{P}_{s_h^{k,j,m}, a_h^{k,j,m}, h}$)

$$J_{2,h,2}^{k,j,m} - \mathbb{V}_{s_h^{k,j,m}, a_h^{k,j,m}, h}(V_{h+1}^\star)$$

$$= \frac{\sum_{n=1}^{N_h^k} \left(\mathbb{1}_{s_{h+1}^{k^n, j^n, m^n}} - \mathbb{P}\right)\left(V_{h+1}^\star\right)^2}{N_h^k(s_h^{k,j,m}, a_h^{k,j,m})} + \left(\mathbb{P}V_{h+1}^\star\right)^2 - \left(\frac{\sum_{n=1}^{N_h^k} V_{h+1}^\star(s_{h+1}^{k^n, j^n, m^n})}{N_h^k(s_h^{k,j,m}, a_h^{k,j,m})}\right)^2$$

$$\leq \frac{\sum_{n=1}^{N_h^k} \left(\mathbb{1}_{s_{h+1}^{k^n, j^n, m^n}} - \mathbb{P}\right)\left(V_{h+1}^\star\right)^2}{N_h^k(s_h^{k,j,m}, a_h^{k,j,m})} + 2H \left|\frac{\sum_{n=1}^{N_h^k} \left(\mathbb{1}_{s_{h+1}^{k^n, j^n, m^n}} - \mathbb{P}\right) V_{h+1}^\star}{N_h^k(s_h^{k,j,m}, a_h^{k,j,m})}\right|$$

$$\lesssim H^2\sqrt{\frac{\iota}{N_h^k(s_h^{k,j,m}, a_h^{k,j,m})}}.$$

The last inequality is because of the events $\mathcal{E}_6$ and $\mathcal{E}_7$ in Lemma G.1. Applying results of Equation (87) and Equation (89) with $J_{2,h,1}^{k,j,m} \leq 4\beta^2$ to Equation (86), we reach

$$\sqrt{\frac{\sigma_h^{\mathrm{R},k} - \left(\mu_h^{\mathrm{R},k}\right)^2}{N_h^k(s_h^{k,j,m}, a_h^{k,j,m})}} \lesssim \frac{\sqrt{H\Psi_h^k(s_h^{k,j,m}, a_h^{k,j,m})}}{N_h^k(s_h^{k,j,m}, a_h^{k,j,m})} + \sqrt{\frac{\mathbb{V}_{s_h^{k,j,m}, a_h^{k,j,m}, h}(V_{h+1}^\star)}{N_h^k(s_h^{k,j,m}, a_h^{k,j,m})}}$$

$$+ \sqrt{\frac{\beta^2}{N_h^k(s_h^{k,j,m}, a_h^{k,j,m})}} + \frac{H\iota^{\frac{1}{4}}}{(N_h^k(s_h^{k,j,m}, a_h^{k,j,m}))^{\frac{3}{4}}}. \tag{90}$$

Note that by Lemma F.6 with $\alpha = \frac{1}{2}$ and $\frac{3}{4}$, we have

$$\sum_{h=1}^H \sum_{k,j,m, N_h^k > 0} \sqrt{\frac{\beta^2}{N_h^k(s_h^{k,j,m}, a_h^{k,j,m})}} \lesssim \beta MHSA + \beta\sqrt{HSAT_1},$$

and

$$\sum_{h=1}^H \sum_{k,j,m, N_h^k > 0} \frac{H\iota^{\frac{1}{4}}}{(N_h^k(s_h^{k,j,m}, a_h^{k,j,m}))^{\frac{3}{4}}} \lesssim MH^2SA\iota^{\frac{1}{4}} + H^2(SA)^{\frac{3}{4}}(T_1\iota)^{\frac{1}{4}}.$$

Applying these two inequalities to Equation (90), it holds that

$$\sum_{h=1}^H \sum_{k,j,m, N_h^k > 0} \sqrt{\frac{\sigma_h^{\mathrm{R},k} - \left(\mu_h^{\mathrm{R},k}\right)^2}{N_h^k(s_h^{k,j,m}, a_h^{k,j,m})}}$$

$$\lesssim \sum_{h=1}^H \sum_{k,j,m, N_h^k > 0} \frac{\sqrt{H\Psi_h^k(s_h^{k,j,m}, a_h^{k,j,m})}}{N_h^k(s_h^{k,j,m}, a_h^{k,j,m})} + \sum_{h=1}^H \sum_{k,j,m, N_h^k > 0} \sqrt{\frac{\mathbb{V}_{s_h^{k,j,m}, a_h^{k,j,m}, h}(V_{h+1}^\star)}{N_h^k(s_h^{k,j,m}, a_h^{k,j,m})}}$$

$$+ MH^2SA\iota^{\frac{1}{4}} + H^2(SA)^{\frac{3}{4}}(T_1\iota)^{\frac{1}{4}} + \beta\sqrt{HSAT_1}. \tag{91}$$

Now we claim the following two conclusions

$$\sum_{h=1}^H \sum_{k,j,m, N_h^k > 0} \sqrt{\frac{\mathbb{V}_{s_h^{k,j,m}, a_h^{k,j,m}, h}(V_{h+1}^\star)}{N_h^k(s_h^{k,j,m}, a_h^{k,j,m})}} \leq \sqrt{H^2SAT_1} + MH^4SA\sqrt{\iota}, \tag{92}$$

and

$$\sum_{h=1}^{H} \sum_{k,j,m,N_h^k>0} \frac{\sqrt{H\Psi_h^k(s_h^{k,j,m}, a_h^{k,j,m})}}{N_h^k(s_h^{k,j,m}, a_h^{k,j,m})} \leq \sqrt{\frac{1}{\beta}}H^4 SA\iota^{\frac{3}{2}} + MH^{\frac{7}{2}}SA\iota^{\frac{5}{4}}, \tag{93}$$

which will be proved later. Combining the results of Equation (85), Equation (91), Equation (92) and Equation (93), we have

$$R_1 \lesssim (1+\beta)\sqrt{H^2 SAT_1\iota} + H^2(SA\iota)^{\frac{3}{4}}(T_1)^{\frac{1}{4}} + \sqrt{\frac{1}{\beta}}H^4 SA\iota^2 + MH^4 SA\iota^{\frac{7}{4}}$$

$$\lesssim (1+\beta)\sqrt{H^2 SAT_1\iota} + \sqrt{\frac{1}{\beta}}H^4 SA\iota^2 + MH^4 SA\iota^2.$$

The last inequality is because $H^2(SA\iota)^{\frac{3}{4}}(T_1)^{\frac{1}{4}} \leq \sqrt{H^2 SAT_1\iota} + H^3 SA\iota$ by AM-GM inequality.

Next, we will prove Equation (92) and Equation (93).

**Proof of Equation (92):**

Because of Equation (23) and Equation (29) with $\omega_h^{k,j,m} = 1$, $\alpha = \frac{1}{2}$, we know

$$\sum_{k,j,m,N_h^k>0} \frac{\mathbb{I}\left[(s_h^{k,j,m}, a_h^{k,j,m}) = (s,a)\right]}{\sqrt{N_h^k(s,a)}} \lesssim M + \sqrt{N_h^{K+1}(s,a)}. \tag{94}$$

Then we can derive the following relationship

$$\sum_{h=1}^{H} \sum_{k,j,m,N_h^k>0} \sqrt{\frac{\mathbb{V}_{s_h^{k,j,m}, a_h^{k,j,m}, h}(V_{h+1}^{\star})}{N_h^k(s_h^{k,j,m}, a_h^{k,j,m})}}$$

$$= \sum_{h=1}^{H} \sum_{s,a} \sum_{k,j,m,N_h^k>0} \sqrt{\frac{\mathbb{V}_{s,a,h}(V_{h+1}^{\star})}{N_h^k(s,a)}}\mathbb{I}\left[(s_h^{k,j,m}, a_h^{k,j,m}) = (s,a)\right]$$

$$\lesssim \sum_{h=1}^{H} \sum_{s,a} \sqrt{\mathbb{V}_{s,a,h}(V_{h+1}^{\star})}\left(M + \sqrt{N_h^{K+1}(s,a)}\right)$$

$$\lesssim MH^2 SA + \sum_{h=1}^{H} \sum_{s,a} \sqrt{N_h^{K+1}(s,a)\mathbb{V}_{s,a,h}(V_{h+1}^{\star})}$$

$$\leq MH^2 SA + \sqrt{HSA}\sqrt{\sum_{h=1}^{H} \sum_{k,j,m} \mathbb{V}_{s_h^{k,j,m}, a_h^{k,j,m}, h}(V_{h+1}^{\star})}. \tag{95}$$

The first inequality is because of Equation (94). The second inequality is by $\mathbb{V}_{s,a,h}(V_{h+1}^{\star}) \leq H^2$. The last inequality is by the Cauchy-Schwarz inequality. Next, we will bound the term in Equation (95).

$$\sum_{h=1}^{H} \sum_{k,j,m} \mathbb{V}_{s_h^{k,j,m}, a_h^{k,j,m}, h}(V_{h+1}^{\star})$$

$$\leq \sum_{h=1}^{H} \sum_{k,j,m} \mathbb{V}_{s_h^{k,j,m}, a_h^{k,j,m}, h}(V_{h+1}^{\pi^k}) + \sum_{h=1}^{H} \sum_{k,j,m} \left|\mathbb{V}_{s_h^{k,j,m}, a_h^{k,j,m}, h}(V_{h+1}^{\star}) - \mathbb{V}_{s_h^{k,j,m}, a_h^{k,j,m}, h}(V_{h+1}^{\pi^k})\right|$$

$$\lesssim HT_1 + H^3\iota + \sum_{h=1}^{H} \sum_{k,j,m} \left|\mathbb{V}_{s_h^{k,j,m}, a_h^{k,j,m}, h}(V_{h+1}^{\star}) - \mathbb{V}_{s_h^{k,j,m}, a_h^{k,j,m}, h}(V_{h+1}^{\pi^k})\right|. \tag{96}$$

The last inequality follows directly from $\mathcal{E}_8$ in Lemma G.1. The second term on the right-hand side of Equation (96) can be bounded as follows (Here we use the shorthand $\mathbb{P} = \mathbb{P}_{s_h^{k,j,m}, a_h^{k,j,m}, h}$)

$$\sum_{h=1}^{H} \sum_{k,j,m} \left| \mathbb{V}_{s_h^{k,j,m}, a_h^{k,j,m}, h}(V_{h+1}^\star) - \mathbb{V}_{s_h^{k,j,m}, a_h^{k,j,m}, h}(V_{h+1}^{\pi^k}) \right|$$

$$= \sum_{h=1}^{H} \sum_{k,j,m} \left| \mathbb{P}(V_{h+1}^\star)^2 - \mathbb{P}(V_{h+1}^{\pi^k})^2 - (\mathbb{P}V_{h+1}^\star)^2 + (\mathbb{P}V_{h+1}^{\pi^k})^2 \right|$$

$$\leq \sum_{h=1}^{H} \sum_{k,j,m} \left\{ \mathbb{P}\left( (V_{h+1}^\star - V_{h+1}^{\pi^k})(V_{h+1}^\star + V_{h+1}^{\pi^k}) \right) + \left| (\mathbb{P}V_{h+1}^\star)^2 - (\mathbb{P}V_{h+1}^{\pi^k})^2 \right| \right\}$$

$$\leq 4H \sum_{h=1}^{H} \sum_{k,j,m} \mathbb{P}\left( V_{h+1}^\star - V_{h+1}^{\pi^k} \right)$$

$$= 4H \sum_{h=1}^{H} \sum_{k,j,m} \left\{ V_{h+1}^\star(s_{h+1}^{k,j,m}) - V_{h+1}^{\pi^k}(s_{h+1}^{k,j,m}) + \left( \mathbb{P} - \mathbb{1}_{s_{h+1}^{k,j,m}} \right) (V_{h+1}^\star - V_{h+1}^{\pi^k}) \right\}$$

$$\overset{(i)}{\lesssim} H \sum_{h=1}^{H} \sum_{k,j,m} \left( V_{h+1}^{\mathrm{U},k} - V_{h+1}^{\pi^k} \right)(s_{h+1}^{k,j,m}) + H^2 \sqrt{T_1 \iota}$$

$$\overset{(ii)}{\lesssim} H^4 \sqrt{SAT_1 \iota} + MH^6 SA \sqrt{\iota}, \tag{97}$$

Here, (i) is because $V_{h+1}^{\mathrm{U},k} \geq V_{h+1}^k \geq V_{h+1}^\star$ by Lemma H.1 and the event $\mathcal{E}_9$ in Lemma G.1. (ii) follows the event $\mathcal{E}_{10}$ in Lemma G.1. Therefore, applying Equation (97) to Equation (96), we reach

$$\sum_{h=1}^{H} \sum_{k,j,m} \mathbb{V}_{s_h^{k,j,m}, a_h^{k,j,m}, h}(V_{h+1}^\star) \lesssim HT_1 + MH^7 SA \iota. \tag{98}$$

Here we use $H^4 \sqrt{SAT_1 \iota} \leq HT_1 + MH^7 SA \iota$ by AM-GM inequality. Applying Equation (98) to Equation (95), we finish the proof of Equation (92):

$$\sum_{h=1}^{H} \sum_{k,j,m,N_h^k > 0} \sqrt{\frac{\mathbb{V}_{s_h^{k,j,m}, a_h^{k,j,m}, h}(V_{h+1}^\star)}{N_h^k(s_h^{k,j,m}, a_h^{k,j,m})}} \lesssim MH^2 SA + \sqrt{HSA}\sqrt{HT_1 + MH^7 SA \iota}$$

$$\leq \sqrt{H^2 SAT_1} + MH^4 SA \sqrt{\iota}.$$

**Proof of Equation (93):**

First, by Equation (24) with $\omega_h^{k,j,m} = 1$ and $\alpha = \frac{1}{2}$ and Equation (32), we know

$$\sum_{s,a} \sum_{k=1}^{K} \frac{n_h^k(s,a)}{\sqrt{N_h^k(s,a)}} \mathbb{I}\left[ 0 < N_h^k(s,a) < M \right] \lesssim MSA. \tag{99}$$

$$\sum_{k=1}^{K} \frac{n_h^k(s,a)}{N_h^k(s,a)} \mathbb{I}\left[ N_h^k(s,a) \geq M \right] \lesssim \log(T_1). \tag{100}$$

Because $\Psi_h^k(s,a) \leq HN_h^k(s,a)$, it holds that

$$\sum_{h=1}^{H} \sum_{k,j,m,N_h^k > 0} \frac{\sqrt{H\Psi_h^k(s_h^{k,j,m}, a_h^{k,j,m})}}{N_h^k(s_h^{k,j,m}, a_h^{k,j,m})} \mathbb{I}\left[ 0 < N_h^k(s_h^{k,j,m}, a_h^{k,j,m}) < M \right]$$

$$\leq H \sum_{h=1}^{H} \sum_{s,a} \sum_{k=1}^{K} \frac{n_h^k(s,a)}{\sqrt{N_h^k(s,a)}} \mathbb{I}\left[ 0 < N_h^k(s,a) < M \right] \lesssim MH^2 SA. \tag{101}$$

The last inequality is because of Equation (99). By Equation (100), We also have

$$
\sum_{h=1}^{H} \sum_{k,j,m,N_h^k>0} \frac{\sqrt{H\Psi_h^k(s_h^{k,j,m}, a_h^{k,j,m})}}{N_h^k(s_h^{k,j,m}, a_h^{k,j,m})} \mathbb{I}\left[N_h^k(s_h^{k,j,m}, a_h^{k,j,m}) \geq M\right]
$$

$$
= \sum_{h=1}^{H} \sum_{s,a} \sum_{k,j,m,N_h^k>0} \frac{\sqrt{H\Psi_h^k(s,a)}}{N_h^k(s,a)} \mathbb{I}\left[N_h^k(s,a) \geq M, (s_h^{k,j,m}, a_h^{k,j,m}) = (s,a)\right]
$$

$$
\leq \sum_{h=1}^{H} \sum_{s,a} \sqrt{H\Psi_h^K(s,a)} \sum_{k=1}^{K} \frac{n_h^k(s,a)}{N_h^k(s,a)} \mathbb{I}\left[N_h^k(s,a) \geq M\right]
$$

$$
\lesssim \log(T_1) \sum_{h=1}^{H} \sum_{s,a} \sqrt{H\Psi_h^K(s,a)}
$$

$$
\leq \iota \sqrt{SAH^2 \sum_{h=1}^{H} \sum_{s,a} \Psi_h^K(s,a)}. \tag{102}
$$

Here, the first inequality is because of the monotonically increasing property of $\Psi_h^k(s,a)$ with respect to $k$ (see its definition in Equation (88)) as guaranteed by $V_{h+1}^{\mathrm{R},k}(s) \geq \hat{V}_{h+1}^{\mathrm{R},k}(s)$. The last inequality is by the Cauchy-Schwarz inequality. To continue, by Lemma H.6, we reach

$$
\sum_{h=1}^{H} \sum_{s,a} \Psi_h^K(s,a)
$$

$$
= \sum_{h=1}^{H} \sum_{k,j,m} \left(V_{h+1}^{\mathrm{R},k} - \hat{V}_{h+1}^{\mathrm{R},k}\right)(s_{h+1}^{k,j,m})
$$

$$
\leq \sum_{h=1}^{H} \sum_{k,j,m} \left(V_{h+1}^{k} - V_{h+1}^{\mathrm{L},k}\right)(s_{h+1}^{k,j,m}) \cdot \mathbb{I}\left[\left(V_{h+1}^{k} - V_{h+1}^{\mathrm{L},k}\right)(s_{h+1}^{k,j,m}) > \beta\right]
$$

$$
\overset{(i)}{\lesssim} \sum_{h=1}^{H} \sum_{k,j,m} \left(Q_{h+1}^{k} - Q_{h+1}^{\mathrm{L},k}\right)(s_{h+1}^{k,j,m}, a_{h+1}^{k,j,m}) \cdot \mathbb{I}\left[\left(Q_{h+1}^{k} - Q_{h+1}^{\mathrm{L},k}\right)(s_{h+1}^{k,j,m}, a_{h+1}^{k,j,m}) > \beta\right]
$$

$$
\lesssim \frac{H^6 SA\iota}{\beta} + MH^5 SA\sqrt{\iota}. \tag{103}
$$

Here, (i) is because

$$
Q_{h+1}^{k}(s_{h+1}^{k,j,m}, a_{h+1}^{k,j,m}) = V_{h+1}^{k}(s_{h+1}^{k,j,m}), \ Q_{h+1}^{\mathrm{L},k}(s_{h+1}^{k,j,m}, a_{h+1}^{k,j,m}) \leq V_{h+1}^{\mathrm{L},k}(s_{h+1}^{k,j,m}).
$$

The last inequality is by Lemma H.4.

By applying Equation (103) to Equation (102), and combining the result of Equation (101), we complete the proof of Equation (93). $\qquad\square$

**Lemma I.2.** *Under the event $\bigcap_{i=1}^{14} \mathcal{E}_i$ in Lemma G.1, we have*

$$
R_2 \lesssim \beta\sqrt{H^2 SAT_1\iota} + MH^3 SA\iota^2.
$$

*Proof.* By event $\mathcal{E}_{14}$ in Lemma G.1, we have

$$
R_2 \lesssim \beta \sum_{h=1}^{H} \sum_{k,j,m,N_h^k>0} \sqrt{\frac{H\iota}{N_h^k}} + \sum_{h=1}^{H} \sum_{k,j,m,N_h^k>0} \frac{\beta H\iota}{N_h^k}
$$

$$
\lesssim \beta\sqrt{H\iota}(MHSA + \sqrt{HSAT_1}) + \beta H\iota(MHSA + HSA\iota)
$$

$$
\lesssim \beta\sqrt{H^2 SAT_1\iota} + MH^3 SA\iota^2.
$$

The second inequality is by Lemma F.6 with $\alpha = \frac{1}{2}$ and $\alpha = 1$, $\qquad\square$

**Lemma I.3.** *Under the event $\bigcap_{i=1}^{14} \mathcal{E}_i$ in Lemma G.1, we have*

$$R_3 \lesssim (1+\beta)\sqrt{H^2 SAT_1 \iota^2} + \frac{H^6 SA\iota^2}{\beta} + MH^5 SA\iota^2.$$

*Proof.* We can decompose $R_3$ into five terms:

$$R_3 = \sum_{h=1}^{H} c_H^{h-1} \sum_{k,j,m,N_h^k>0} \sum_{n=1}^{N_h^k} \tilde{\eta}_n^{N_h^k} \frac{1}{N_h^{k^n+1}} \sum_{i=1}^{N_h^{k^n+1}} \left( V_h^{\mathrm{R},k^i}(s_{h+1}^{k^i,j^i,m^i}) - \mathbb{P}_{s_h^{k,j,m},a_h^{k,j,m},h} \hat{V}_{h+1}^{\mathrm{R},k^n} \right)$$

$$= R_{3,1} + R_{3,2} + R_{3,3} + R_{3,4} + R_{3,5},$$

where

$$R_{3,1} = \sum_{h=1}^{H} c_H^{h-1} \sum_{k,j,m,N_h^k>0} \sum_{n=1}^{N_h^k} \tilde{\eta}_n^{N_h^k} \frac{1}{N_h^{k^n+1}} \sum_{i=1}^{N_h^{k^n+1}} \left( \mathbb{1}_{s_{h+1}^{k^i,j^i,m^i}} - \mathbb{P}_{s_h^{k,j,m},a_h^{k,j,m},h} \right)(\hat{V}_{h+1}^{\mathrm{R},k^i} - V_{h+1}^{\star}),$$

$$R_{3,2} = \sum_{h=1}^{H} c_H^{h-1} \sum_{k,j,m,N_h^k>0} \sum_{n=1}^{N_h^k} \tilde{\eta}_n^{N_h^k} \frac{1}{N_h^{k^n+1}} \sum_{i=1}^{N_h^{k^n+1}} \left( \mathbb{1}_{s_{h+1}^{k^i,j^i,m^i}} - \mathbb{P}_{s_h^{k,j,m},a_h^{k,j,m},h} \right) V_{h+1}^{\star},$$

$$R_{3,3} = \sum_{h=1}^{H} c_H^{h-1} \sum_{k,j,m,N_h^k>0} \sum_{n=1}^{N_h^k} \tilde{\eta}_n^{N_h^k} \frac{1}{N_h^{k^n+1}} \sum_{i=1}^{N_h^{k^n+1}} \left( V_{h+1}^{\mathrm{R},k^i}(s_{h+1}^{k^i,j^i,m^i}) - \hat{V}_{h+1}^{\mathrm{R},k^i}(s_{h+1}^{k^i,j^i,m^i}) \right),$$

$$R_{3,4} = \sum_{h=1}^{H} c_H^{h-1} \sum_{k,j,m,N_h^k>0} \sum_{n=1}^{N_h^k} \tilde{\eta}_n^{N_h^k} \frac{1}{N_h^{k^n+1}} \sum_{i=1}^{N_h^{k^n+1}} \mathbb{P}_{s_h^{k,j,m},a_h^{k,j,m},h} \left( \hat{V}_{h+1}^{\mathrm{R},k^i} - V_{h+1}^{\mathrm{R},K+1} \right),$$

and

$$R_{3,5} = \sum_{h=1}^{H} c_H^{h-1} \sum_{k,j,m,N_h^k>0} \sum_{n=1}^{N_h^k} \tilde{\eta}_n^{N_h^k} \mathbb{P}_{s_h^{k,j,m},a_h^{k,j,m},h} \left( V_{h+1}^{\mathrm{R},K+1} - \hat{V}_{h+1}^{\mathrm{R},k^n} \right).$$

Next, we will bound these five terms respectively:

**Upper bound of $R_{3,1}$:**

According to the event $\mathcal{E}_{12}$ in Lemma G.1, we know

$$\sum_{n=1}^{N_h^k} \tilde{\eta}_n^{N_h^k} \frac{1}{N_h^{k^n+1}} \sum_{i=1}^{N_h^{k^n+1}} \left( \mathbb{1}_{s_{h+1}^{k^i,j^i,m^i}} - \mathbb{P}_{s_h^{k,j,m},a_h^{k,j,m},h} \right) \left( \hat{V}_{h+1}^{\mathrm{R},k^i} - V_{h+1}^{\star} \right)$$

$$\leq \sum_{n=1}^{N_h^k} \frac{\beta \tilde{\eta}_n^{N_h^k} \sqrt{2\iota}}{\sqrt{N_h^{k^n+1}}}$$

$$\overset{(i)}{\lesssim} \beta\sqrt{\iota} \sum_{n=1}^{N_h^k} \frac{\eta_n^{N_h^k}}{\sqrt{n}} \overset{(ii)}{\lesssim} \frac{\beta\sqrt{\iota}}{\sqrt{N_h^k}}. \tag{104}$$

Here, (i) is because $\tilde{\eta}_n^{N_h^k} \leq \exp(1/H)\eta_n^{N_h^k}$ by (c) of Lemma F.3 and $N_h^{k^n+1} \geq n$ by the definition of $k^n$. (ii) directly follows (a) in Lemma F.2 with $\alpha = \frac{1}{2}$. Therefore, by Lemma F.6 with $\alpha = \frac{1}{2}$,

$$R_{3,1} \lesssim \beta\sqrt{\iota} \sum_{h=1}^{H} \sum_{k,j,m,N_h^k>0} \frac{1}{\sqrt{N_h^k}} \leq \beta MHSA\sqrt{\iota} + \beta\sqrt{HSAT_1\iota}. \tag{105}$$

**Upper bound of $R_{3,2}$:**

According to the event $\mathcal{E}_{13}$ in Lemma G.1, we know

$$\sum_{n=1}^{N_h^k} \tilde{\eta}_n^{N_h^k} \frac{1}{N_h^{k^n+1}} \sum_{i=1}^{N_h^{k^n+1}} \left( \mathbb{1}_{s_{h+1}^{k^i,j^i,m^i}} - \mathbb{P}_{s_h^{k,j,m},a_h^{k,j,m},h} \right) V_{h+1}^\star$$

$$\leq \sum_{n=1}^{N_h^k} \tilde{\eta}_n^{N_h^k} \left( 4\sqrt{\frac{\mathbb{V}_{s_h^{k,j,m},a_h^{k,j,m},h}(V_{h+1}^\star)\iota}{N_h^{k^n+1}}} + \frac{7H\iota}{N_h^{k^n+1}} \right)$$

$$\overset{(i)}{\lesssim} \left( \sum_{n=1}^{N_h^k} \frac{\eta_n^{N_h^k}}{\sqrt{n}} \right) \sqrt{\mathbb{V}_{s_h^{k,j,m},a_h^{k,j,m},h}(V_{h+1}^\star)\iota} + H\iota \sum_{n=1}^{N_h^k} \frac{\eta_n^{N_h^k}}{n}$$

$$\overset{(ii)}{\lesssim} \sqrt{\frac{\mathbb{V}_{s_h^{k,j,m},a_h^{k,j,m},h}(V_{h+1}^\star)\iota}{N_h^k}} + \frac{H\iota}{N_h^k}. \tag{106}$$

Here, (i) is because $\tilde{\eta}_n^{N_h^k} \leq \exp(1/H)\eta_n^{N_h^k}$ by (c) of Lemma F.3, $N_h^{k^n+1} \geq n$ by the definition of $k^n$. (ii) directly follows (a) in Lemma F.2 with $\alpha = \frac{1}{2}$ and $\alpha = 1$. Then according to the definition of $R_{3,2}$, we have

$$R_{3,2} \lesssim \sum_{h=1}^{H} \sum_{k,j,m,N_h^k>0} \sqrt{\frac{\mathbb{V}_{s_h^{k,j,m},a_h^{k,j,m},h}(V_{h+1}^\star)\iota}{N_h^k}} + H\iota \sum_{h=1}^{H} \sum_{k,j,m,N_h^k>0} \frac{1}{N_h^k}. \tag{107}$$

For the first term in Equation (112), by Cauchy-Schwarz inequality, we have

$$\sum_{h=1}^{H} \sum_{k,j,m,N_h^k>0} \sqrt{\frac{\mathbb{V}_{s_h^{k,j,m},a_h^{k,j,m},h}(V_{h+1}^\star)\iota}{N_h^k}}$$

$$= \sum_{h=1}^{H} \sum_{k,j,m} \sqrt{\frac{\mathbb{V}_{s_h^{k,j,m},a_h^{k,j,m},h}(V_{h+1}^\star)\iota}{N_h^k}} \left( \mathbb{I}\left[0 < N_h^k < M\right] + \mathbb{I}\left[N_h^k \geq M\right] \right)$$

$$\leq \sqrt{\sum_{h=1}^{H} \sum_{k,j,m} \frac{\mathbb{I}\left[0 < N_h^k < M\right]}{N_h^k}} \cdot \sqrt{\sum_{h=1}^{H} \sum_{k,j,m} \mathbb{V}_{s_h^{k,j,m},a_h^{k,j,m},h}(V_{h+1}^\star)\iota\mathbb{I}\left[0 < N_h^k < M\right]}$$

$$+ \sqrt{\sum_{h=1}^{H} \sum_{k,j,m} \frac{\mathbb{I}\left[N_h^k \geq M\right]}{N_h^k}} \cdot \sqrt{\sum_{h=1}^{H} \sum_{k,j,m} \mathbb{V}_{s_h^{k,j,m},a_h^{k,j,m},h}(V_{h+1}^\star)\iota\mathbb{I}\left[N_h^k \geq M\right]}$$

$$\lesssim \sqrt{MHSA} \cdot \sqrt{H^2\iota \sum_{h=1}^{H} \sum_{k,j,m} \mathbb{I}\left[0 < N_h^k < M\right]} + \sqrt{HSA\iota} \cdot \sqrt{HT_1\iota + MH^7SA\iota^2}. \tag{108}$$

Here, the last inequality is because $\mathbb{V}_{s_h^{k,j,m},a_h^{k,j,m},h}(V_{h+1}^\star) \leq H^2$,

$$\sum_{h=1}^{H} \sum_{k,j,m} \frac{\mathbb{I}\left[0 < N_h^k < M\right]}{N_h^k} \lesssim MHSA, \quad \sum_{h=1}^{H} \sum_{k,j,m} \frac{\mathbb{I}\left[N_h^k \geq M\right]}{N_h^k} \lesssim HSA\iota$$

by Equation (24) and Equation (32), and

$$\sum_{h=1}^{H} \sum_{k,j,m} \mathbb{V}_{s_h^{k,j,m},a_h^{k,j,m},h}(V_{h+1}^\star) \lesssim HT_1 + MH^7SA\iota.$$

by Equation (98). Next, we will bound the term $\sum_{h=1}^{H} \sum_{k,j,m} \mathbb{I}\left[0 < N_h^k < M\right]$. Let $k_0(s,a) = \max\{k \mid 1 \le k \le K, N_h^k(s,a) < M\}$. Then it holds that

$$
\begin{aligned}
&\sum_{h=1}^{H} \sum_{k,j,m} \mathbb{I}\left[0 < N_h^k < M\right] \\
&= \sum_{h=1}^{H} \sum_{s,a} \sum_{k,j,m} \mathbb{I}\left[0 < N_h^k(s,a) < M, (s_h^{k,j,m}, a_h^{k,j,m}) = (s,a)\right] \\
&= \sum_{h=1}^{H} \sum_{s,a} \sum_{k=1}^{k_0(s,a)} \sum_{j,m} \mathbb{I}\left[(s_h^{k,j,m}, a_h^{k,j,m}) = (s,a)\right] \\
&= \sum_{h=1}^{H} \sum_{s,a} N_h^{k_0+1}(s,a) \lesssim MHSA
\end{aligned}
\tag{109}
$$

The last inequality is because $N_h^{k_0}(s,a) \le M$ and $n_h^{k_0}(s,a) \le M$ by (a) of Lemma F.1. Applying this inequality to Equation (108), we can bound the first term in Equation (107):

$$
\sum_{h=1}^{H} \sum_{k,j,m,N_h^k>0} \sqrt{\frac{\mathbb{V}_{s_h^{k,j,m},a_h^{k,j,m},h}(V_{h+1}^\star)\iota}{N_h^k}} \lesssim \sqrt{H^2 SAT_1 \iota^2} + MH^4 SA\iota^2.
\tag{110}
$$

For the second term in Equation (107), according to Lemma F.6 with $\alpha = 1$, we reach

$$
H\iota \sum_{h=1}^{H} \sum_{k,j,m,N_h^k>0} \frac{1}{N_h^k} \lesssim MH^2 SA\iota + H^2 SA\iota^2 \lesssim MH^4 SA\iota^2.
\tag{111}
$$

Applying the results of Equation (110) and Equation (111) to Equation (107), we can bound $R_{3,2}$:

$$
R_{3,2} \lesssim \sqrt{H^2 SAT_1 \iota^2} + MH^4 SA\iota^2.
\tag{112}
$$

**Upper bound of $R_{3,3}$ and $R_{3,4}$:**

We first substitute the terms $(V_{h+1}^{\mathrm{R},k^i} - \hat{V}_{h+1}^{\mathrm{R},k^i})(s_{h+1}^{k^i,j^i,m^i})$ in $R_{3,3}$ or $\mathbb{P}_{s_h^{k,j,m},a_h^{k,j,m},h}\left|\hat{V}_{h+1}^{\mathrm{R},k^i} - V_{h+1}^{\mathrm{R},K+1}\right|$ in $R_{3,4}$ by $W_{h+1}^{k^i}(s_h^{k,j,m}, a_h^{k,j,m})$ and deal with the following general structure:

$$
\sum_{h=1}^{H} c_H^{h-1} \sum_{k,j,m,N_h^k>0} \sum_{n=1}^{N_h^k} \tilde{\eta}_n^{N_h^k} \frac{1}{N_h^{k^n+1}} \sum_{i=1}^{N_h^{k^n+1}} W_{h+1}^{k^i}(s_h^{k,j,m}, a_h^{k,j,m}).
$$

Here, $0 \le W_{h+1}^{k^i}(s_h^{k,j,m}, a_h^{k,j,m}) \le H$ because $H \ge V_{h+1}^{\mathrm{R},k^i}(s_{h+1}^{k^i,j^i,m^i}) \ge \hat{V}_{h+1}^{\mathrm{R},k^i}(s_{h+1}^{k^i,j^i,m^i})$. Since $W_{h+1}^{k^i}(s_h^{k,j,m}, a_h^{k,j,m}) \le H$ and $\sum_{n=1}^{N_h^k} \tilde{\eta}_n^{N_h^k} \le 1$ by (b) of Lemma F.3, we first note that when $0 < N_h^k(s_h^{k,j,m}, a_h^{k,j,m}) < M$,

$$
\begin{aligned}
&\sum_{h=1}^{H} c_H^{h-1} \sum_{k,j,m} \sum_{n=1}^{N_h^k} \tilde{\eta}_n^{N_h^k} \frac{1}{N_h^{k^n+1}} \sum_{i=1}^{N_h^{k^n+1}} W_{h+1}^{k^i}(s_h^{k,j,m}, a_h^{k,j,m}) \mathbb{I}\left[0 < N_h^k < M\right] \\
&\le \sum_{h=1}^{H} c_H^{h-1} \sum_{k,j,m} \sum_{n=1}^{N_h^k} \tilde{\eta}_n^{N_h^k} \frac{1}{N_h^{k^n+1}} \sum_{i=1}^{N_h^{k^n+1}} H\mathbb{I}\left[0 < N_h^k < M\right] \\
&\lesssim \sum_{h=1}^{H} \sum_{k,j,m} H\mathbb{I}\left[0 < N_h^k < M\right] \lesssim MH^2 SA.
\end{aligned}
\tag{113}
$$

The last inequality is by Equation (109). Next, we consider the case when $N_h^k(s_h^{k,j,m}, a_h^{k,j,m}) \geq M$:

$$\sum_{h=1}^{H} c_H^{h-1} \sum_{k,j,m} \sum_{n=1}^{N_h^k} \tilde{\eta}_n^{N_h^k} \frac{1}{N_h^{k^n+1}} \sum_{i=1}^{N_h^{k^n+1}} W_{h+1}^{k^i}(s_h^{k,j,m}, a_h^{k,j,m}) \mathbb{I}\left[N_h^k \geq M\right]$$

$$= \sum_{h=1}^{H} \sum_{k',j',m'} W_{h+1}^{k'}(s_h^{k,j,m}, a_h^{k,j,m}) \times$$

$$\left( c_H^{h-1} \sum_{k,j,m} \sum_{n=1}^{N_h^k} \tilde{\eta}_n^{N_h^k} \frac{\mathbb{I}\left[N_h^k \geq M\right]}{N_h^{k^n+1}} \sum_{i=1}^{N_h^{k^n+1}} \mathbb{I}\left[(k^i, j^i, m^i) = (k', j', m')\right] \right). \qquad (114)$$

By the definitions of $k^i$, $j^i$ and $m^i$, for any given triple $(k', j', m')$,

$$\sum_{i=1}^{N_h^{k^n+1}} \mathbb{I}\left[(k^i, j^i, m^i) = (k', j', m')\right] = 1$$

if and only if

$$(s_h^{k,j,m}, a_h^{k,j,m}) = (s_h^{k',j',m'}, a_h^{k',j',m'}) \text{ and } N_h^{k^n+1} \geq i'(k', j', m'),$$

where $i'(k', j', m')$ is the global visiting number for $(s_h^{k',j',m'}, a_h^{k',j',m'}, h)$ at $(k', j', m')$ with the order "round first, episode second, agent third". When there is no ambiguity, we will use $i'$ for short. Therefore, for the coefficient of $W_{h+1}^{k'}(s_h^{k,j,m}, a_h^{k,j,m})$ in Equation (114), we can derive the following upper bound:

$$c_H^{h-1} \sum_{k,j,m} \sum_{n=1}^{N_h^k} \tilde{\eta}_n^{N_h^k} \frac{\mathbb{I}\left[N_h^k \geq M\right]}{N_h^{k^n+1}} \sum_{i=1}^{N_h^{k^n+1}} \mathbb{I}\left[(k^i, j^i, m^i) = (k', j', m')\right]$$

$$\leq c_H^{h-1} \sum_{k,j,m} \mathbb{I}\left[(s_h^{k,j,m}, a_h^{k,j,m}) = (s_h^{k',j',m'}, a_h^{k',j',m'}), N_h^k \geq M\right] \sum_{n=1}^{N_h^k} \tilde{\eta}_n^{N_h^k} \frac{1}{N_h^{k^n+1}}$$

$$\overset{(i)}{\lesssim} \sum_{k,j,m} \mathbb{I}\left[(s_h^{k,j,m}, a_h^{k,j,m}) = (s_h^{k',j',m'}, a_h^{k',j',m'}), N_h^k \geq M\right] \frac{1}{N_h^k}$$

$$= \sum_{k=1}^{K} \frac{n_h^k(s_h^{k',j',m'}, a_h^{k',j',m'})}{N_h^k(s_h^{k',j',m'}, a_h^{k',j',m'})} \mathbb{I}\left[N_h^k(s_h^{k',j',m'}, a_h^{k',j',m'}) \geq M\right]$$

$$\overset{(ii)}{\lesssim} \iota. \qquad (115)$$

Here, (i) is because

$$\sum_{n=1}^{N_h^k} \tilde{\eta}_n^{N_h^k} \frac{1}{N_h^{k^n+1}} \lesssim \sum_{n=1}^{N_h^k} \frac{\eta_n^{N_h^k}}{n} \lesssim \frac{1}{N_h^k},$$

where the first inequality is because $\tilde{\eta}_n^{N_h^k} \lesssim \eta_n^{N_h^k}$ by (c) of Lemma F.3 and the last inequality is by (a) of Lemma F.2. (ii) is because of Equation (32). When the coefficient of $W_{h+1}^{k'}(s_h^{k,j,m}, a_h^{k,j,m})$ in Equation (114) is non-zero, we have $(s_h^{k,j,m}, a_h^{k,j,m}) = (s_h^{k',j',m'}, a_h^{k',j',m'})$ and thus $W_{h+1}^{k'}(s_h^{k,j,m}, a_h^{k,j,m}) = W_{h+1}^{k'}(s_h^{k',j',m'}, a_h^{k',j',m'})$. Therefore, by applying Equation (115) to Equation (114), we know that

$$\sum_{h=1}^{H} c_H^{h-1} \sum_{k,j,m} \sum_{n=1}^{N_h^k} \tilde{\eta}_n^{N_h^k} \frac{1}{N_h^{k^n+1}} \sum_{i=1}^{N_h^{k^n+1}} W_{h+1}^{k^i}(s_h^{k,j,m}, a_h^{k,j,m}) \mathbb{I}\left[N_h^k \geq M\right]$$

$$\lesssim \iota \sum_{h=1}^{H} \sum_{k',j',m'} W_{h+1}^{k'}(s_h^{k',j',m'}, a_h^{k',j',m'}) \qquad (116)$$

Next, we will bound the term $\sum_{h=1}^{H} \sum_{k,j,m} W_{h+1}^{k}(s_h^{k,j,m}, a_h^{k,j,m})$. By Lemma H.6, we have

$$\sum_{h=1}^{H} \sum_{k,j,m} \left( V_{h+1}^{\mathrm{R},k}(s_{h+1}^{k,j,m}) - \hat{V}_{h+1}^{\mathrm{R},k}(s_{h+1}^{k,j,m}) \right)$$

$$\leq \sum_{h=1}^{H} \sum_{k,j,m} (V_{h+1}^{k}(s_{h+1}^{k,j,m}) - V_{h+1}^{\mathrm{L},k}(s_{h+1}^{k,j,m})) \mathbb{I}\left[ V_{h+1}^{k}(s_{h+1}^{k,j,m}) - V_{h+1}^{\mathrm{L},k}(s_{h+1}^{k,j,m}) > \beta \right]$$

$$\stackrel{(i)}{\leq} \sum_{h=1}^{H} \sum_{k,j,m} (Q_{h+1}^{k} - Q_{h+1}^{\mathrm{L},k})(s_{h+1}^{k,j,m}, a_{h+1}^{k,j,m}) \mathbb{I}\left[ (Q_{h+1}^{k} - Q_{h+1}^{\mathrm{L},k})(s_{h+1}^{k,j,m}, a_{h+1}^{k,j,m}) > \beta \right]$$

$$\lesssim \frac{H^6 S A \iota}{\beta} + M H^5 S A \sqrt{\iota}. \tag{117}$$

Here, (i) is because

$$Q_{h+1}^{k}(s_{h+1}^{k,j,m}, a_{h+1}^{k,j,m}) = V_{h+1}^{k}(s_{h+1}^{k,j,m}), \ Q_{h+1}^{\mathrm{L},k}(s_{h+1}^{k,j,m}, a_{h+1}^{k,j,m}) \leq V_{h+1}^{\mathrm{L},k}(s_{h+1}^{k,j,m}).$$

The last inequality is by Lemma H.4. Similarly, by Lemma H.6, we also have

$$\sum_{h=1}^{H} \sum_{k,j,m} \mathbb{P}_{s_h^{k,j,m}, a_h^{k,j,m}, h} \left| \hat{V}_{h+1}^{\mathrm{R},k} - V_{h+1}^{\mathrm{R},K+1} \right|$$

$$\leq \sum_{h=1}^{H} \sum_{k,j,m} \mathbb{P}_{s_h^{k,j,m}, a_h^{k,j,m}, h} \left\{ (V_{h+1}^{k} - V_{h+1}^{\mathrm{L},k})(s_{h+1}^{k,j,m}) \mathbb{I}\left[ V_{h+1}^{k}(s_{h+1}^{k,j,m}) - V_{h+1}^{\mathrm{L},k}(s_{h+1}^{k,j,m}) > \beta \right] \right\}$$

$$\stackrel{(i)}{\leq} \sum_{h=1}^{H} \sum_{k,j,m} (V_{h+1}^{k}(s_{h+1}^{k,j,m}) - V_{h+1}^{\mathrm{L},k}(s_{h+1}^{k,j,m})) \mathbb{I}\left[ V_{h+1}^{k}(s_{h+1}^{k,j,m}) - V_{h+1}^{\mathrm{L},k}(s_{h+1}^{k,j,m}) > \beta \right] + H\iota$$

$$\leq \sum_{h=1}^{H} \sum_{k,j,m} (Q_{h+1}^{k} - Q_{h+1}^{\mathrm{L},k})(s_{h+1}^{k,j,m}, a_{h+1}^{k,j,m}) \mathbb{I}\left[ (Q_{h+1}^{k} - Q_{h+1}^{\mathrm{L},k})(s_{h+1}^{k,j,m}, a_{h+1}^{k,j,m}) > \beta \right] + H\iota$$

$$\lesssim \frac{H^6 S A \iota}{\beta} + M H^5 S A \sqrt{\iota}. \tag{118}$$

Here, (i) is because of the event $\mathcal{E}_{11}$ of Lemma G.1. Combining the results of Equation (117) and Equation (118), we conclude that

$$\sum_{h=1}^{H} \sum_{k,j,m} W_{h+1}^{k}(s_h^{k,j,m}, a_h^{k,j,m}) \lesssim \frac{H^6 S A \iota}{\beta} + M H^5 S A \sqrt{\iota}.$$

Back to Equation (116), we have the following conclusion:

$$\sum_{h=1}^{H} c_H^{h-1} \sum_{k,j,m} \sum_{n=1}^{N_h^k} \tilde{\eta}_n^{N_h^k} \frac{1}{N_h^{k^n+1}} \sum_{i=1}^{N_h^{k^n+1}} W_{h+1}^{k^i}(s_h^{k,j,m}, a_h^{k,j,m}) \mathbb{I}\left[ N_h^k \geq M \right]$$

$$\lesssim \frac{H^6 S A \iota^2}{\beta} + M H^5 S A \iota^2.$$

Together with Equation (113), we reach that:

$$\sum_{h=1}^{H} c_H^{h-1} \sum_{k,j,m,N_h^k>0} \sum_{n=1}^{N_h^k} \frac{\tilde{\eta}_n^{N_h^k}}{N_h^{k^n+1}} \sum_{i=1}^{N_h^{k^n+1}} W_{h+1}^{k^i}(s_h^{k,j,m}, a_h^{k,j,m}) \lesssim \frac{H^6 S A \iota^2}{\beta} + M H^5 S A \iota^2 \tag{119}$$

Therefore, we can bound $R_{3,3}, R_{3,4}$:

$$R_{3,3}, R_{3,4} \lesssim \frac{H^6 S A \iota^2}{\beta} + M H^5 S A \iota^2. \tag{120}$$

**Upper bound of $R_{3,5}$:**

$$R_{3,5} = \sum_{h=1}^{H} c_H^{h-1} \sum_{k,j,m,N_h^k>0} \sum_{n=1}^{N_h^k} \tilde{\eta}_n^{N_h^k} \mathbb{P}_{s_h^{k,j,m},a_h^{k,j,m},h} \left( V_{h+1}^{R,K+1} - \hat{V}_{h+1}^{R,k^n} \right)$$

$$\leq \sum_{h=1}^{H} c_H^{h-1} \sum_{k,j,m,N_h^k>0} \sum_{n=1}^{N_h^k} \tilde{\eta}_n^{N_h^k} \mathbb{P}_{s_h^{k,j,m},a_h^{k,j,m},h} \left| V_{h+1}^{R,K+1} - \hat{V}_{h+1}^{R,k^n} \right|.$$

Since $\left| V_{h+1}^{R,K+1}(s) - \hat{V}_{h+1}^{R,k^n}(s) \right| \leq H$ and $\sum_{n=1}^{N_h^k} \tilde{\eta}_n^{N_h^k} \leq 1$ by (b) of Lemma F.3, we first note that when $0 < N_h^k(s_h^{k,j,m}, a_h^{k,j,m}) < i_0$,

$$\sum_{h=1}^{H} c_H^{h-1} \sum_{k,j,m} \sum_{n=1}^{N_h^k} \tilde{\eta}_n^{N_h^k} \mathbb{P}_{s_h^{k,j,m},a_h^{k,j,m},h} \left| V_{h+1}^{R,K+1} - \hat{V}_{h+1}^{R,k^n} \right| \mathbb{I}\left[ 0 < N_h^k < i_0 \right]$$

$$\lesssim \sum_{h=1}^{H} \sum_{k,j,m} H \mathbb{I}\left[ 0 < N_h^k < i_0 \right]$$

$$\lesssim M H^4 S A. \tag{121}$$

The last inequality is because

$$\sum_{k,j,m} \mathbb{I}\left[ 0 < N_h^k(s_h^{k,j,m}, a_h^{k,j,m}) < i_0 \right] \lesssim M H^2 S A, \tag{122}$$

by Equation (74). When $N_h^k(s_h^{k,j,m}, a_h^{k,j,m}) \geq i_0$, we have:

$$\sum_{h=1}^{H} c_H^{h-1} \sum_{k,j,m} \sum_{n=1}^{N_h^k} \tilde{\eta}_n^{N_h^k} \mathbb{P}_{s_h^{k,j,m},a_h^{k,j,m},h} \left| V_{h+1}^{R,K+1} - \hat{V}_{h+1}^{R,k^n} \right| \mathbb{I}\left[ N_h^k \geq i_0 \right]$$

$$= \sum_{h=1}^{H} \sum_{k',j',m'} \mathbb{P}_{s_h^{k,j,m},a_h^{k,j,m},h} \left| V_{h+1}^{R,K+1} - \hat{V}_{h+1}^{R,k'} \right| \times$$

$$\left( c_H^{h-1} \sum_{k,j,m} \sum_{n=1}^{N_h^k} \tilde{\eta}_n^{N_h^k} \mathbb{I}\left[ N_h^k \geq i_0 \right] \mathbb{I}\left[ (k^n, j^n, m^n) = (k', j', m') \right] \right)$$

$$\lesssim \sum_{h=1}^{H} \sum_{k',j',m'} \mathbb{P}_{s_h^{k',j',m'},a_h^{k',j',m'},h} \left| V_{h+1}^{R,K+1} - \hat{V}_{h+1}^{R,k'} \right|$$

$$\lesssim \frac{H^6 S A \iota}{\beta} + M H^5 S A \sqrt{\iota}. \tag{123}$$

The first inequality is by Equation (60) and the last inequality is because of Equation (118). Together with Equation (121), we can bound $R_{3,5}$:

$$R_{3,5} \leq \sum_{h=1}^{H} c_H^{h-1} \sum_{k,j,m,N_h^k>0} \sum_{n=1}^{N_h^k} \tilde{\eta}_n^{N_h^k} \mathbb{P}_{s_h^{k,j,m},a_h^{k,j,m},h} \left| V_{h+1}^{R,K+1} - \hat{V}_{h+1}^{R,k^n} \right|$$

$$\lesssim \frac{H^6 S A \iota}{\beta} + M H^5 S A \sqrt{\iota}. \tag{124}$$

Combining Equations (105), (112), (120) and (124), since $\beta \leq H$, we can bound $R_3$:

$$R_3 \lesssim (1+\beta)\sqrt{H^2 S A T_1 \iota^2} + \frac{H^6 S A \iota^2}{\beta} + M H^5 S A \iota^2.$$

$\square$

## J  Proof of Worst-Case Switching/Communication Cost (Theorem 4.2 and Theorem 4.4)

*Proof.* We first prove the Theorem 4.4. When $T = \hat{T}/M \leq 3H^2(H+1)SA$, the number of rounds $k$ is no more than $\hat{T} \leq 3MH^2(H+1)SA$. Next, we consider the case when $T \geq 3H^2(H+1)SA$.

Since each round terminates when some triple $(s, a, h)$ satisfies the trigger condition (see Equation (3)), the total number of communication rounds is upper bounded by the total number of times the trigger condition is satisfied across all triples $(s, a, h)$. In the following proof, we provide an upper bound on the total number of such trigger events.

Let $t_h(s, a)$ denote the number of times the trigger condition is satisfied by the triple $(s, a, h)$. According to Equation (3), each time the trigger condition is satisfied during any round $k$, the number of visits to $(s, a, h)$ increases by at least 1 when $N_h^k < MH(H+1) \stackrel{\triangle}{=} i_0'$, and increases by at least a factor of $1 + \frac{1}{2MH(H+1)}$ when $N_h^k \geq i_0'$. Define the set $\mathcal{C} = \{(s, a, h) \mid t_h(s, a) \geq i_0'\}$. Then for every $(s, a, h) \in \mathcal{C}$, after the trigger condition has been satisfied $i_0'$ times, the number of visits to $(s, a, h)$ is at least $i_0'$. Therefore, for $(s, a, h) \in \mathcal{C}$, we have:

$$\left(1 + \frac{1}{2MH(H+1)}\right)^{t_h(s,a) - i_0'} \leq \frac{N_h^{K+1}(s,a)}{i_0'}.$$

and thus

$$t_h(s,a) \leq \frac{\log(N_h^{K+1}(s,a)/i_0')}{\log\left(1 + \frac{1}{2MH(H+1)}\right)} + i_0'.$$

If $\mathcal{C} = \emptyset$, then

$$\sum_{h=1}^{H} \sum_{s,a} t_h(s,a) = \sum_{(s,a,h) \notin \mathcal{C}} t_h(s,a) \leq HSAi_0' \leq MH^2(H+1)SA.$$

Otherwise, the total number of trigger times is no more than

$$\sum_{h=1}^{H} \sum_{s,a} t_h(s,a) = \sum_{(s,a,h) \notin \mathcal{C}} t_h(s,a) + \sum_{(s,a,h) \in \mathcal{C}} t_h(s,a)$$

$$\leq 2MH^2(H+1)SA + \sum_{(s,a,h) \in \mathcal{C}} \frac{\log(N_h^{K+1}(s,a)/i_0)}{\log\left(1 + \frac{1}{2MH(H+1)}\right)}$$

$$\leq 2MH^2(H+1)SA + |\mathcal{C}| \frac{\log(\frac{T}{H(H+1)|\mathcal{C}|})}{\log\left(1 + \frac{1}{2MH(H+1)}\right)}$$

$$\leq 2MH^2(H+1)SA + HSA \frac{\log\left(\frac{T}{H^2(H+1)SA}\right)}{\log\left(1 + \frac{1}{2MH(H+1)}\right)}$$

$$\leq 2MH^2(H+1)SA + 4MH^2(H+1)SA \log\left(\frac{T}{H^2(H+1)SA}\right).$$

The first inequality is because for $(s, a, h) \notin \mathcal{C}$, $t_h(s, a) \leq i_0'$. The second inequality follows from Jensen's inequality. The second last inequality is because the term in the previous line increases with $|\mathcal{C}|$ when $T \geq 3H^2(H+1)SA$ and $1 \leq |\mathcal{C}| \leq HSA$. The last inequality is because $\log(1+x) \geq x/2$ for $x \in (0, 1)$, which applies to $x = \frac{1}{2MH(H+1)}$.

For $M = 1$, the switching cost is no more than the number of communication rounds. Then we also finish the proof of Theorem 4.2. $\qquad\square$

## K  Proof of Gap-Dependent Regret (Theorem 4.5 and Theorem 4.8)

### K.1  Proof Sketch

We now prove the gap-dependent regret upper bound under the event $\bigcap_{i=1}^{14} \mathcal{E}_i$ in Lemma G.1.

To begin, we point out that

$$\left(V_1^\star - V_1^{\pi^k}\right)(s_1^{k,j,m}) = V_1^\star(s_1^{k,j,m}) - Q_1^\star(s_1^{k,j,m}, a_1^{k,j,m}) + \left(Q_1^\star - Q_1^{\pi^k}\right)(s_1^{k,j,m}, a_1^{k,j,m})$$

$$\stackrel{(i)}{=} \Delta_1(s_1^{k,j,m}, a_1^{k,j,m}) + \mathbb{E}\left[\left(V_2^\star - V_2^{\pi^k}\right)(s_2^{k,j,m}) \mid s_2^{k,j,m} \sim P_1(\cdot \mid s_1^{k,j,m}, a_1^{k,j,m})\right]$$

$$= \mathbb{E}\left[\Delta_1(s_1^{k,j,m}, a_1^{k,j,m}) + \Delta_2(s_2^{k,j,m}, a_2^{k,j,m}) \mid s_2^{k,j,m} \sim P_1(\cdot \mid s_1^{k,j,m}, a_1^{k,j,m})\right]$$

$$+ \mathbb{E}\left[\left(Q_2^\star - Q_2^{\pi^k}\right)(s_2^{k,j,m}, a_2^{k,j,m}) \mid s_2^{k,j,m} \sim P_1(\cdot \mid s_1^{k,j,m}, a_1^{k,j,m})\right]$$

$$= \cdots = \mathbb{E}\left[\sum_{h=1}^H \Delta_h\left(s_h^{k,j,m}, a_h^{k,j,m}\right) \middle| s_{h+1}^{k,j,m} \sim P_h(\cdot \mid s_h^{k,j,m}, a_h^{k,j,m}), h \in [H-1]\right].$$

Here (i) is by the Bellman Equation and Bellman Optimality Equation (1). Therefore, we can derive the following expression of the expected regret

$$\mathbb{E}\left(\text{Regret}(T)\right) = \mathbb{E}\left[\sum_{k,j,m}\left(V_1^\star - V_1^{\pi^k}\right)(s_1^{k,j,m})\right] = \mathbb{E}\left[\sum_{k,j,m}\sum_{h=1}^H \Delta_h(s_h^{k,j,m}, a_h^{k,j,m})\right].$$

Note that we have

$$Q_h^k(s_h^{k,j,m}, a_h^{k,j,m}) = \max_a\{Q_h^k(s_h^{k,j,m}, a)\} \geq \max_a\{Q_h^\star(s_h^{k,j,m}, a)\} = V_h^\star(s_h^{k,j,m}).$$

Thus, if we define $\text{clip}[x \mid y] := x \cdot \mathbb{I}[x \geq y]$, then for any round-step pair $(k, h)$,

$$\Delta_h(s_h^{k,j,m}, a_h^{k,j,m}) = \text{clip}\left[V_h^\star(s_h^{k,j,m}) - Q_h^\star(s_h^{k,j,m}, a_h^{k,j,m}) \mid \Delta_{\min}\right]$$

$$\leq \text{clip}\left[\left(Q_h^k - Q_h^\star\right)(s_h^{k,j,m}, a_h^{k,j,m}) \mid \Delta_{\min}\right].$$

which further implies

$$\mathbb{E}\left(\text{Regret}(T)\right) \leq \mathbb{E}\left[\sum_{h=1}^H \sum_{k,j,m} \text{clip}\left[\left(Q_h^k - Q_h^\star\right)(s_h^{k,j,m}, a_h^{k,j,m}) \mid \Delta_{\min}\right]\right].$$

Let $\mathcal{E} = \bigcap_{i=1}^{14} \mathcal{E}_i$ and $\delta = 1/14T_1$, we have:

$$\mathbb{E}\left(\text{Regret}(T)\right) \leq \mathbb{E}\left[\sum_{h=1}^H \sum_{k,j,m} \text{clip}\left[\left(Q_h^k - Q_h^\star\right)(s_h^{k,j,m}, a_h^{k,j,m}) \mid \Delta_{\min}\right] \middle| \mathcal{E}\right] \mathbb{P}(\mathcal{E})$$

$$+ \mathbb{E}\left[\sum_{h=1}^H \sum_{k,j,m} \text{clip}\left[\left(Q_h^k - Q_h^\star\right)(s_h^{k,j,m}, a_h^{k,j,m}) \mid \Delta_{\min}\right] \middle| \mathcal{E}^c\right] \mathbb{P}(\mathcal{E}^c)$$

$$\leq O\left(\frac{(\mathbb{Q}^\star + \beta^2 H) H^3 SA\iota}{\Delta_{\min}} + \frac{H^7 SA\iota^2}{\beta} + MH^6 SA\iota^2\right).$$

The last inequality is because under the event $\mathcal{E}$, by Lemma K.2, we have

$$\sum_{h=1}^H \sum_{k,j,m} \text{clip}\left[\left(Q_h^k - Q_h^\star\right)(s_h^{k,j,m}, a_h^{k,j,m}) \mid \Delta_{\min}\right]$$

$$\leq O\left(\frac{(\mathbb{Q}^\star + \beta^2 H) H^3 SA\iota}{\Delta_{\min}} + \frac{H^7 SA\iota^2}{\beta} + MH^6 SA\iota^2\right).$$

and under the event $\mathcal{E}^c$, we have

$$\sum_{h=1}^H \sum_{k,j,m} \text{clip}\left[\left(Q_h^k - Q_h^\star\right)(s_h^{k,j,m}, a_h^{k,j,m}) \mid \Delta_{\min}\right] \leq HT_1.$$

Since $\iota = \log(2SAT_1/\delta) = \log(2SAT_1^2)$, by (b) of Lemma F.1, we have $\iota \leq O\left(\log(MSAT)\right)$. Therefore, let $\iota_2 = \log(MSAT)$, we have

$$\mathbb{E}\left(\text{Regret}(T)\right) \leq O\left(\frac{(\mathbb{Q}^\star + \beta^2 H) H^3 SA\iota_2}{\Delta_{\min}} + \frac{H^7 SA\iota_2^2}{\beta} + MH^6 SA\iota_2^2\right).$$

## K.2 Auxiliary Lemmas

**Lemma K.1.** *Under the event $\bigcap_{i=1}^{14} \mathcal{E}_i$ in Lemma G.1, for FedQ-EarlySettled-LowCost algorithm and any non-negative weight sequence $\{\omega_h^{k,j,m}\}_{h,k,j,m}$, it holds for any $h \in [H]$ that:*

$$\sum_{k,j,m} \omega_h^{k,j,m} \left(Q_h^k - Q_h^\star\right)(s_h^{k,j,m}, a_h^{k,j,m}) \lesssim H\sqrt{(\mathbb{Q}^\star + \beta^2 H)SA\|\omega\|_{\infty,h}\|\omega\|_{1,h}\iota}$$

$$+ H^2(SA\|\omega\|_{\infty,h}\iota)^{\frac{3}{4}}(\|\omega\|_{1,h})^{\frac{1}{4}} + \sum_{h'=h}^{H} \sum_{k,j,m} \omega_{h'}^{k,j,m}(h)Z_{h'}^{k,j,m},$$

*where for any $h \leq h' \leq H - 1$*

$$\omega_h^{k,j,m}(h) := \omega_h^{k,j,m},$$

$$\omega_{h'+1}^{k,j,m}(h) = \sum_{k',j',m'} \omega_{h'}^{k',j',m'}(h)\mathbb{I}\left[N_{h'}^{k'} \geq i_0\right] \sum_{i=1}^{N_{h'}^{k'}} \tilde{\eta}_i^{N_{h'}^{k'}} \mathbb{I}\left[(k^i,j^i,m^i) = (k,j,m)\right],$$

*and*

$$Z_{h'}^{k,j,m} = \eta_0^{N_{h'}^k} H + \frac{H^2\iota + \sqrt{H\Psi_{h'}^k\iota}}{N_{h'}^k} + H\mathbb{I}\left[0 < N_{h'}^k < i_0\right]$$

$$+ \left(\sqrt{\frac{(\mathbb{Q}^\star + \beta^2 H)\iota}{N_{h'}^k}} + \frac{H\iota^{\frac{3}{4}}}{(N_{h'}^k)^{\frac{3}{4}}}\right)\mathbb{I}\left[0 < N_{h'}^k < M\right]$$

$$+ \sum_{n=1}^{N_{h'}^k} \frac{\tilde{\eta}_n^{N_{h'}^k}}{N_{h'}^{k^n+1}} \sum_{i=1}^{N_{h'}^{k^n+1}} \left(\left(V_{h'+1}^{R,k^i} - \hat{V}_{h'+1}^{R,k^i}\right)(s_{h'+1}^{k^i,j^i,m^i}) + \mathbb{P}_{s_{h'}^{k,j,m}, a_{h'}^{k,j,m}, h'}\left(\hat{V}_{h'+1}^{R,k^i} - \hat{V}_{h'+1}^{R,k^n}\right)\right). \quad (125)$$

*Here $N_{h'}^{k'}, N_{h'}^k$ is the abbreviation for $N_{h'}^{k'}(s_{h'}^{k',j',m'}, a_{h'}^{k',j',m'}), N_{h'}^k(s_{h'}^{k,j,m}, a_{h'}^{k,j,m})$, respectively.*

*Proof.* To begin with, since $Q_h^k(s_h^{k,j,m}, a_h^{k,j,m}) \leq Q_h^{R,k}(s_h^{k,j,m}, a_h^{k,j,m})$, by Equation (77), we have (Here we use the shorthand $\mathbb{P} = \mathbb{P}_{s_h^{k,j,m}, a_h^{k,j,m}, h}$)

$$(Q_h^k - Q_h^\star)(s_h^{k,j,m}, a_h^{k,j,m}) \leq (Q_h^{R,k} - Q_h^\star)(s_h^{k,j,m}, a_h^{k,j,m})$$

$$\leq \eta_0^{N_h^k} H + \sum_{n=1}^{N_h^k} \tilde{\eta}_n^{N_h^k} \left(\left(V_{h+1}^{k^n} - \hat{V}_{h+1}^{R,k^n}\right)(s_{h+1}^{k^n,j^n,m^n}) + \mu_h^{R,k^n+1} - \mathbb{P}V_{h+1}^\star\right) + \sum_{n=1}^{N_h^k} \eta_n^{N_h^k} b_{h,n}^R. \quad (126)$$

$$\leq \eta_0^{N_h^k} H + \left(\beta_h^{R,k} + 2c_b^{R,2}\frac{H^2\iota}{N_h^k}\right) + \sum_{n=1}^{N_h^k} \tilde{\eta}_n^{N_h^k} \left(V_{h+1}^{k^n} - V_{h+1}^\star\right)(s_{h+1}^{k^n,j^n,m^n})$$

$$+ \sum_{n=1}^{N_h^k} \tilde{\eta}_n^{N_h^k} \left(\mathbb{1}_{s_{h+1}^{k,j,m}} - \mathbb{P}\right)\left(V_{h+1}^\star - \hat{V}_{h+1}^{R,k^n}\right) + \sum_{n=1}^{N_h^k} \tilde{\eta}_n^{N_h^k} \frac{1}{N_h^{k^n+1}} \sum_{i=1}^{N_h^{k^n+1}} \left(\mathbb{1}_{s_{h+1}^{k^i,j^i,m^i}} - \mathbb{P}\right) V_{h+1}^\star$$

$$+ \sum_{n=1}^{N_h^k} \tilde{\eta}_n^{N_h^k} \frac{1}{N_h^{k^n+1}} \sum_{i=1}^{N_h^{k^n+1}} \left(\mathbb{1}_{s_{h+1}^{k^i,j^i,m^i}} - \mathbb{P}\right)\left(\hat{V}_{h+1}^{R,k^i} - V_{h+1}^\star\right)$$

$$+ \sum_{n=1}^{N_h^k} \tilde{\eta}_n^{N_h^k} \frac{1}{N_h^{k^n+1}} \sum_{i=1}^{N_h^{k^n+1}} \left(\left(V_{h+1}^{R,k^i} - \hat{V}_{h+1}^{R,k^i}\right)(s_{h+1}^{k^i,j^i,m^i}) + \mathbb{P}\left(\hat{V}_{h+1}^{R,k^i} - \hat{V}_{h+1}^{R,k^n}\right)\right). \quad (127)$$

The last inequality is by Equation (79). In the last inequality, we decompose the second term in Equation (126) into the last five terms in Equation (127). We then claim the following five conclusions:

$$\sum_{n=1}^{N_h^k} \tilde{\eta}_n^{N_h^k} \left(V_{h+1}^{k^n} - V_{h+1}^\star\right)(s_{h+1}^{k^n,j^n,m^n})\mathbb{I}\left[0 < N_h^k < i_0\right] \leq H\mathbb{I}\left[0 < N_h^k < i_0\right],$$

$$\beta_h^{\mathrm{R},k}(s_h^{k,j,m}, a_h^{k,j,m}) \lesssim \beta\sqrt{\frac{H\iota}{N_h^k}} + \sqrt{\frac{(\mathbb{Q}^\star + \beta^2)\iota}{N_h^k}} + \frac{H\iota^{\frac{3}{4}}}{(N_h^k)^{\frac{3}{4}}} + \frac{\sqrt{H\Psi_h^k\iota}}{N_h^k},$$

$$\sum_{n=1}^{N_h^k} \tilde{\eta}_n^{N_h^k}\left(\mathbb{1}_{s_{h+1}^{k,j,m}} - \mathbb{P}_{s_h^{k,j,m},a_h^{k,j,m},h}\right)\left(V_{h+1}^\star - \hat{V}_{h+1}^{\mathrm{R},k^n}\right) \lesssim \beta\sqrt{\frac{H\iota}{N_h^k}} + \frac{\beta H\iota}{N_h^k},$$

$$\sum_{n=1}^{N_h^k} \tilde{\eta}_n^{N_h^k} \frac{1}{N_h^{k^n+1}} \sum_{i=1}^{N_h^{k^n+1}} \left(\mathbb{1}_{s_{h+1}^{k^i,j^i,m^i}} - \mathbb{P}_{s_h^{k,j,m},a_h^{k,j,m},h}\right)\left(\hat{V}_{h+1}^{\mathrm{R},k^i} - V_{h+1}^\star\right) \lesssim \beta\sqrt{\frac{\iota}{N_h^k}},$$

$$\sum_{n=1}^{N_h^k} \tilde{\eta}_n^{N_h^k} \frac{1}{N_h^{k^n+1}} \sum_{i=1}^{N_h^{k^n+1}} \left(\mathbb{1}_{s_{h+1}^{k^i,j^i,m^i}} - \mathbb{P}_{s_h^{k,j,m},a_h^{k,j,m},h}\right) V_{h+1}^\star \lesssim \sqrt{\frac{\mathbb{Q}^\star\iota}{N_h^k}} + \frac{H\iota}{N_h^k}.$$

Here, the first conclusion is because $0 \leq V_{h+1}^{k^n}(s) - V_{h+1}^\star(s) \leq H$ and $\sum_{n=1}^{N_h^k} \tilde{\eta}_n^{N_h^k} \leq 1$ by (b) of Lemma F.3. The second conclusion is proved by combining Equation (84) and Equation (90) and using $\mathbb{V}_{s_h^{k,j,m},a_h^{k,j,m},h}(V_{h+1}^\star) \leq \mathbb{Q}^\star$. The third and fourth conclusions follow directly from $\mathcal{E}_{14}$ in Lemma G.1 and Equation (104), respectively. The last one is proved in Equation (106) with $\mathbb{V}_{s_h^{k,j,m},a_h^{k,j,m},h}(V_{h+1}^\star) \leq \mathbb{Q}^\star$. Applying these five conclusions to Equation (127), we then reach:

$$(Q_h^k - Q_h^\star)(s_h^{k,j,m}, a_h^{k,j,m}) \lesssim \sum_{n=1}^{N_h^k} \tilde{\eta}_n^{N_h^k}\left(V_{h+1}^{k^n} - V_{h+1}^\star\right)(s_{h+1}^{k^n,j^n,m^n})\mathbb{I}\left[N_h^k \geq i_0\right]$$

$$+ \left(\sqrt{\frac{(\mathbb{Q}^\star + \beta^2 H)\iota}{N_h^k}} + \frac{H\iota^{\frac{3}{4}}}{(N_h^k)^{\frac{3}{4}}}\right)\mathbb{I}\left[N_h^k \geq M\right] + Z_h^{k,j,m},$$

and thus

$$\sum_{k,j,m} \omega_h^{k,j,m}(Q_h^k - Q_h^\star)(s_h^{k,j,m}, a_h^{k,j,m})$$

$$\lesssim \sum_{k,j,m} \omega_h^{k,j,m} \sum_{n=1}^{N_h^k} \tilde{\eta}_n^{N_h^k}\left(V_{h+1}^{k^n} - V_{h+1}^\star\right)(s_{h+1}^{k^n,j^n,m^n})\mathbb{I}\left[N_h^k \geq i_0\right]$$

$$+ \sum_{k,j,m} \omega_h^{k,j,m}\left(\sqrt{\frac{(\mathbb{Q}^\star + \beta^2 H)\iota}{N_h^k}} + \frac{H\iota^{\frac{3}{4}}}{(N_h^k)^{\frac{3}{4}}}\right)\mathbb{I}\left[N_h^k \geq M\right] + \sum_{k,j,m} \omega_h^{k,j,m} Z_h^{k,j,m}. \qquad (128)$$

Similar to Equation (57), we can prove:

$$\sum_{k,j,m} \omega_h^{k,j,m} \sum_{n=1}^{N_h^k} \tilde{\eta}_n^{N_h^k}\left(V_{h+1}^{k^n} - V_{h+1}^\star\right)(s_{h+1}^{k^n,j^n,m^n})\mathbb{I}\left[N_h^k \geq i_0\right]$$

$$= \sum_{k',j',m'} \tilde{\omega}_h^{k',j',m'}\left(V_{h+1}^{k'} - V_{h+1}^\star\right)(s_{h+1}^{k',j',m'}), \qquad (129)$$

where

$$\tilde{\omega}_h^{k',j',m'} = \sum_{k,j,m} \mathbb{I}\left[N_h^k(s_h^{k,j,m}, a_h^{k,j,m}) \geq i_0\right] \omega_h^{k,j,m} \sum_{i=1}^{N_h^k} \tilde{\eta}_i^{N_h^k} \mathbb{I}\left[(k^i, j^i, m^i) = (k', j', m')\right].$$

By Equation (30) in Lemma F.6, it holds that:

$$\sum_{k,j,m} \omega_h^{k,j,m}\left(\sqrt{\frac{(\mathbb{Q}^\star + \beta^2 H)\iota}{N_h^k}} + \frac{H\iota^{\frac{3}{4}}}{(N_h^k)^{\frac{3}{4}}}\right)\mathbb{I}\left[N_h^k \geq M\right]$$

$$\lesssim \sqrt{(\mathbb{Q}^\star + \beta^2 H)SA\|\omega\|_{\infty,h}\|\omega\|_{1,h}\iota} + H(SA\|\omega\|_{\infty,h}\iota)^{\frac{3}{4}}(\|\omega\|_{1,h})^{\frac{1}{4}}. \qquad (130)$$

Applying Equation (129) and Equation (130) to Equation (128), we reach:

$$\sum_{k,j,m} \omega_h^{k,j,m}(Q_h^k - Q_h^\star)(s_h^{k,j,m}, a_h^{k,j,m})$$

$$\lesssim \sum_{k',j',m'} \tilde{\omega}_h^{k',j',m'} \left( V_{h+1}^{k'} - V_{h+1}^\star \right)(s_{h+1}^{k',j',m'}) + \sqrt{(\mathbb{Q}^\star + \beta^2 H)SA \|\omega\|_{\infty,h} \|\omega\|_{1,h} \iota}$$

$$+ H(SA\|\omega\|_{\infty,h}\iota)^{\frac{3}{4}}(\|\omega\|_{1,h})^{\frac{1}{4}} + \sum_{k,j,m} \omega_h^{k,j,m} Z_h^{k,j,m}$$

with $\|\tilde{\omega}\|_{1,h} \leq \|\omega\|_{1,h}$ and $\|\tilde{\omega}\|_{\infty,h} \leq \exp(3/H)\|\omega\|_{\infty,h}$. Here, the last inequality is because

$$V_{h+1}^{k'}(s_{h+1}^{k',j',m'}) = Q_{h+1}^{k'}(s_{h+1}^{k',j',m'}, a_{h+1}^{k',j',m'}) \text{ and } V_{h+1}^\star(s_{h+1}^{k',j',m'}) \geq Q_{h+1}^\star(s_{h+1}^{k',j',m'}, a_{h+1}^{k',j',m'}).$$

Now we have developed a recursive relationship for the weighted sum of $Q_h^k - Q_h^\star$ between step $h$ and step $h+1$. By recursions with regard to $h, h+1, ..., H$, we finish the proof. $\qquad\square$

**Lemma K.2.** *Under the event $\bigcap_{i=1}^{14} \mathcal{E}_i$ in Lemma G.1, for all $\epsilon \in (0, H)$, we have the following conclusion:*

$$\sum_{h=1}^{H} \sum_{k,j,m} \mathbb{I}\left[ Q_h^k(s_h^{k,j,m}, a_h^{k,j,m}) - Q_h^\star(s_h^{k,j,m}, a_h^{k,j,m}) > \epsilon \right]$$

$$\lesssim \frac{(\mathbb{Q}^\star + \beta^2 H) H^3 SA\iota}{\epsilon^2} + \frac{H^7 SA\iota^2}{\beta\epsilon} + \frac{MH^6 SA\iota^2}{\epsilon},$$

*and*

$$\sum_{h=1}^{H} \sum_{k,j,m} \left( Q_h^k - Q_h^\star \right)(s_h^{k,j,m}, a_h^{k,j,m}) \mathbb{I}\left[ \left( Q_h^k - Q_h^\star \right)(s_h^{k,j,m}, a_h^{k,j,m}) > \epsilon \right]$$

$$\lesssim \frac{(\mathbb{Q}^\star + \beta^2 H) H^3 SA\iota}{\epsilon} + \frac{H^7 SA\iota^2}{\beta} + MH^6 SA\iota^2.$$

*Proof.* Let $N = \lceil \log_2(H/\epsilon) \rceil$. For any $i < N$, $k \in [K]$ and given $h \in [H]$, let:

$$\omega_{h,i}^{k,j,m} = \mathbb{I}\left[ Q_h^k(s_h^{k,j,m}, a_h^{k,j,m}) - Q_h^\star(s_h^{k,j,m}, a_h^{k,j,m}) \in \left[ 2^{i-1}\epsilon, 2^i\epsilon \right) \right],$$

and

$$\omega_{h,N}^{k,j,m} = \mathbb{I}\left[ Q_h^k(s_h^{k,j,m}, a_h^{k,j,m}) - Q_h^\star(s_h^{k,j,m}, a_h^{k,j,m}) \in \left[ 2^{N-1}\epsilon, H \right] \right].$$

Then

$$\|\omega\|_{\infty,h}^{(i)} = \max_{k,j,m} \omega_{h,i}^{k,j,m} \leq 1, \ \|\omega\|_{1,h}^{(i)} = \sum_{k,j,m} \omega_{h,i}^{k,j,m}.$$

Now for any $i \in [N]$, we have the following relationship:

$$\sum_{k,j,m} \omega_{h,i}^{k,j,m} \left( Q_h^k - Q_h^\star \right)(s_h^{k,j,m}, a_h^{k,j,m}) \geq 2^{i-1}\epsilon\|\omega\|_{1,h}^{(i)}. \tag{131}$$

Combining the results of Lemma K.1 and Equation (131), we have:

$$2^{i-1}\epsilon\|\omega\|_{1,h}^{(i)} \lesssim H\sqrt{(\mathbb{Q}^\star + \beta^2 H)SA\|\omega\|_{\infty,h}^{(i)}\|\omega\|_{1,h}^{(i)}\iota} + H^2(SA\|\omega\|_{\infty,h}^{(i)}\iota)^{\frac{3}{4}}(\|\omega\|_{1,h})^{\frac{1}{4}}$$

$$+ \sum_{h'=h}^{H} \sum_{k,j,m} \omega_{h',i}^{k,j,m}(h) Z_{h'}^{k,j,m}$$

$$\leq H\sqrt{(\mathbb{Q}^\star + \beta^2 H)SA\|\omega\|_{1,h}^{(i)}\iota} + H^2(SA\iota)^{\frac{3}{4}}(\|\omega\|_{1,h}^{(i)})^{\frac{1}{4}} + \sum_{h'=h}^{H} \sum_{k,j,m} \omega_{h',i}^{k,j,m}(h) Z_{h'}^{k,j,m}, \tag{132}$$

where for any $h \leq h' \leq H - 1$

$$\omega_{h,i}^{k,j,m}(h) := \omega_{h,i}^{k,j,m},$$

$$\omega_{h'+1,i}^{k,j,m}(h) = \sum_{k',j',m'} \omega_{h',i}^{k',j',m'}(h) \mathbb{I}\left[N_{h'}^{k'} \geq i_0\right] \sum_{i=1}^{N_{h'}^{k'}} \tilde{\eta}_i^{N_{h'}^{k'}} \mathbb{I}\left[(k^i, j^i, m^i) = (k, j, m)\right].$$

The definition of these coefficients is the same as that in Lemma H.4. By Equation (132), at least one of the following three inequalities holds:

$$2^{i-1}\epsilon \|\omega\|_{1,h}^{(i)} \lesssim H\sqrt{(\mathbb{Q}^\star + \beta^2 H)SA\|\omega\|_{1,h}^{(i)}\iota}$$

$$2^{i-1}\epsilon \|\omega\|_{1,h}^{(i)} \lesssim H^2(SA\iota)^{\frac{3}{4}}(\|\omega\|_{1,h}^{(i)})^{\frac{1}{4}},$$

$$2^{i-1}\epsilon \|\omega\|_{1,h}^{(i)} \lesssim \sum_{h'=h}^{H}\sum_{k,j,m} \omega_{h',i}^{k,j,m}(h) Z_{h'}^{k,j,m}.$$

Solving these three inequalities, we know that:

$$\|\omega\|_{1,h}^{(i)} \lesssim \max\left\{\frac{(\mathbb{Q}^\star + \beta^2 H)H^2 SA\iota}{4^{i-2}\epsilon^2}, \frac{H^3 SA\iota}{(2^{i-1}\epsilon)^{\frac{4}{3}}}, \frac{\sum_{h'=h}^{H}\sum_{k,j,m} \omega_{h',i}^{k,j,m}(h) Z_{h'}^{k,j,m}}{2^{i-1}\epsilon}\right\}$$

$$\lesssim \frac{(\mathbb{Q}^\star + \beta^2 H)H^2 SA\iota}{4^{i-2}\epsilon^2} + \frac{H^3 SA\iota}{(2^{i-1}\epsilon)^{\frac{4}{3}}} + \frac{\sum_{h'=h}^{H}\sum_{k,j,m} \omega_{h',i}^{k,j,m}(h) Z_{h'}^{k,j,m}}{2^{i-1}\epsilon}. \qquad (133)$$

We claim that

$$\sum_{h'=1}^{H}\sum_{k,j,m} Z_{h'}^{k,j,m} \lesssim \frac{H^6 SA\iota^2}{\beta} + MH^5 SA\iota^2, \qquad (134)$$

which will be proved later. Therefore, by

$$\mathbb{I}\left[(Q_h^k - Q_h^\star)(s_h^{k,j,m}, a_h^{k,j,m}) \geq \epsilon\right] = \sum_{i=1}^{N} \omega_{h,i}^{k,j,m},$$

we have

$$\sum_{h=1}^{H}\sum_{k,j,m} \mathbb{I}\left[Q_h^k(s_h^{k,j,m}, a_h^{k,j,m}) - Q_h^\star(s_h^{k,j,m}, a_h^{k,j,m}) \geq \epsilon\right] = \sum_{h=1}^{H}\sum_{i=1}^{N} \|\omega\|_{1,h}^{(i)}. \qquad (135)$$

By Equation (133), it holds that

$$\sum_{i=1}^{N} \|\omega\|_{1,h}^{(i)} \lesssim \sum_{i=1}^{N} \frac{(\mathbb{Q}^\star + \beta^2 H)H^2 SA\iota}{4^{i-2}\epsilon^2} + \sum_{i=1}^{N} \frac{H^3 SA\iota}{(2^{i-1}\epsilon)^{\frac{4}{3}}} + \sum_{i=1}^{N} \frac{\sum_{h'=h}^{H}\sum_{k,j,m} \omega_{h',i}^{k,j,m}(h) Z_{h'}^{k,j,m}}{2^{i-1}\epsilon}$$

$$\overset{(i)}{\lesssim} \frac{(\mathbb{Q}^\star + \beta^2 H)H^2 SA\iota}{\epsilon^2} + \frac{H^3 SA}{\epsilon^{\frac{4}{3}}} + \sum_{i=1}^{N} \frac{\sum_{h'=1}^{H}\sum_{k,j,m} Z_{h'}^{k,j,m}}{2^i\epsilon}$$

$$\overset{(ii)}{\lesssim} \frac{(\mathbb{Q}^\star + \beta^2 H)H^2 SA\iota}{\epsilon^2} + \frac{H^3 SA\iota}{\epsilon^{\frac{4}{3}}} + \frac{H^6 SA\iota^2}{\beta\epsilon} + \frac{MH^5 SA\iota^2}{\epsilon}$$

$$\lesssim \frac{(\mathbb{Q}^\star + \beta^2 H)H^2 SA\iota}{\epsilon^2} + \frac{H^6 SA\iota^2}{\beta\epsilon} + \frac{MH^5 SA\iota^2}{\epsilon}. \qquad (136)$$

Here, (i) is because $0 \leq \omega_{h',i}^{k,j,m}(h) < 27$ by Equation (67) and $Z_{h'}^{k,j,m} \geq 0$. (ii) is because of Equation (134). The last inequality is because

$$\frac{H^3 SA\iota}{\epsilon^{\frac{4}{3}}} \leq \frac{\beta^2 H^3 SA\iota}{\epsilon^2} + \frac{H^3 SA\iota}{\beta\epsilon} + \frac{H^3 SA\iota}{\beta\epsilon} \lesssim \frac{\beta^2 H^3 SA\iota}{\epsilon^2} + \frac{H^6 SA\iota^2}{\beta\epsilon}$$

by AM-GM inequality. Combing the results of Equation (135) and Equation (136), we finish the proof of the first conclusion. Further, we can prove the second conclusion by noting that

$$
\sum_{h=1}^{H} \sum_{k,j,m} \left(Q_h^k - Q_h^\star\right)(s_h^{k,j,m}, a_h^{k,j,m}) \mathbb{I}\left[\left(Q_h^k - Q_h^\star\right)(s_h^{k,j,m}, a_h^{k,j,m}) \geq \epsilon\right] \leq \sum_{h=1}^{H} \sum_{i=1}^{N} 2^i \epsilon \|\omega\|_{1,h}^{(i)}
$$

$$
\overset{(i)}{\lesssim} \sum_{h=1}^{H} \sum_{i=1}^{N} \frac{\left(\mathbb{Q}^\star + \beta^2 H\right) H^2 SA\iota}{2^i \epsilon} + \sum_{h=1}^{H} \sum_{i=1}^{N} \frac{H^3 SA\iota}{(2^{i-1}\epsilon)^{\frac{1}{3}}} + \sum_{h=1}^{H} \sum_{h'=h}^{H} \sum_{k,j,m} \left(\sum_{i=1}^{N} \omega_{h',i}^{k,j,m}(h)\right) Z_{h'}^{k,j,m}
$$

$$
\overset{(ii)}{\lesssim} \frac{\left(\mathbb{Q}^\star + \beta^2 H\right) H^3 SA\iota}{\epsilon} + \frac{H^4 SA\iota}{\epsilon^{\frac{1}{3}}} + \sum_{h=1}^{H} \sum_{h'=h}^{H} \sum_{k,j,m} Z_{h'}^{k,j,m}
$$

$$
\lesssim \frac{\left(\mathbb{Q}^\star + \beta^2 H\right) H^3 SA\iota}{\epsilon} + \frac{H^7 SA\iota^2}{\beta} + MH^6 SA\iota^2.
$$

Here, (i) is by Equation (133) and (ii) is by Equation (67). The last inequality is because of Equation (134) and

$$
\frac{H^4 SA\iota}{\epsilon^{\frac{1}{3}}} \leq \frac{\beta^2 H^4 SA\iota}{\epsilon} + \frac{H^4 SA\iota}{\beta} + \frac{H^4 SA\iota}{\beta} \lesssim \frac{\beta^2 H^4 SA\iota}{\epsilon} + \frac{H^7 SA\iota^2}{\beta}
$$

by AM-GM inequality. Next, we only need to prove Equation (134).

**Proof of Equation (134):**

By definition of $Z_{h'}^{k,m,j}$ (see Equation (125) in Lemma K.1), we have the following relationship

$$
Z_{h'}^{k,j,m} = \eta_0^{N_{h'}^k} H + \frac{H^2 \iota + \sqrt{H \Psi_{h'}^k \iota}}{N_{h'}^k} + H \mathbb{I}\left[0 < N_{h'}^k < i_0\right]
$$

$$
+ \left(\sqrt{\frac{(\mathbb{Q}^\star + \beta^2 H)\iota}{N_{h'}^k}} + \frac{H\iota^{\frac{3}{4}}}{(N_{h'}^k)^{\frac{3}{4}}}\right) \mathbb{I}\left[0 < N_{h'}^k < M\right]
$$

$$
+ \sum_{n=1}^{N_{h'}^k} \frac{\tilde{\eta}_n^{N_{h'}^k}}{N_{h'}^{k^{n+1}}} \sum_{i=1}^{N_{h'}^{k^{n+1}}} \left((V_{h'+1}^{\mathrm{R},k^i} - \hat{V}_{h'+1}^{\mathrm{R},k^i})(s_{h'+1}^{k^i,j^i,m^i}) + \mathbb{P}_{s_{h'}^{k,j,m}, a_{h'}^{k,j,m}, h'}(\hat{V}_{h'+1}^{\mathrm{R},k^i} - \hat{V}_{h'+1}^{\mathrm{R},k^n})\right). \quad (137)
$$

**For the first term of Equation (137)**, by Equation (73), we have

$$
\sum_{h'=1}^{H} \sum_{k,j,m} \eta_0^{N_{h'}^k} H \lesssim MH^2 SA. \quad (138)
$$

**For the second term of Equation (137)**, by Lemma F.6 with $\alpha = 1$, we know

$$
\sum_{h'=1}^{H} \sum_{k,j,m,N_{h'}^k>0} \frac{H^2 \iota}{N_{h'}^k(s_{h'}^{k,j,m}, a_{h'}^{k,j,m})} \lesssim MH^3 SA\iota + H^3 SA\iota^2. \quad (139)
$$

By Equation (93), since $\beta \leq H$, it holds that:

$$
\sum_{h'=1}^{H} \sum_{k,j,m,N_{h'}^k>0} \frac{\sqrt{H \Psi_{h'}^k(s_{h'}^{k,j,m}, a_{h'}^{k,j,m})\iota}}{N_{h'}^k(s_{h'}^{k,j,m}, a_{h'}^{k,j,m})} \lesssim \frac{H^5 SA\iota^2}{\beta} + MH^4 SA\iota^2. \quad (140)
$$

Combining the results of Equation (139) and Equation (140), we can bound the second term:

$$
\sum_{h'=1}^{H} \sum_{k,j,m,N_{h'}^k>0} \frac{H^2 \iota + \sqrt{H \Psi_{h'}^k(s_{h'}^{k,j,m}, a_{h'}^{k,j,m})\iota}}{N_{h'}^k(s_{h'}^{k,j,m}, a_{h'}^{k,j,m})} \lesssim \frac{H^5 SA\iota^2}{\beta} + MH^4 SA\iota^2. \quad (141)
$$

**For the third term of Equation (137)**, by Equation (122), we know

$$\sum_{h'=1}^{H} \sum_{k,j,m} H \mathbb{I}\left[0 < N_{h'}^{k}(s_{h'}^{k,j,m}, a_{h'}^{k,j,m}) < i_0\right] \lesssim MH^4 SA. \tag{142}$$

**For the fourth term of Equation (137)**, by Equation (24) in Lemma F.6 with $\omega_h^{k,j,m} = 1$ and $\alpha = \frac{1}{2}$, we have

$$\sum_{h'=1}^{H} \sum_{k,j,m} \left(\sqrt{\frac{(\mathbb{Q}^\star + \beta^2 H)\iota}{N_{h'}^{k}(s_{h'}^{k,j,m}, a_{h'}^{k,j,m})}} + \frac{H\iota^{\frac{1}{4}}}{N_{h'}^{k}(s_{h'}^{k,j,m}, a_{h'}^{k,j,m})^{\frac{3}{4}}}\right) \mathbb{I}\left[0 < N_{h'}^{k}(s_{h'}^{k,j,m}, a_{h'}^{k,j,m}) < M\right]$$

$$\lesssim \sum_{h'=1}^{H} \sum_{k,j,m} \sqrt{H^3 \iota} \mathbb{I}\left[0 < N_{h'}^{k}(s_{h'}^{k,j,m}, a_{h'}^{k,j,m}) < M\right]$$

$$\lesssim MH^3 SA\iota. \tag{143}$$

Here the first inequality is because $\mathbb{Q}^\star \leq H^2$ and $0 \leq \beta \leq H$.

**For the last term of Equation (137)**, by the triangle inequality, we can bound it with

$$\sum_{h'=1}^{H} \sum_{k,j,m,N_{h'}^k>0} \sum_{n=1}^{N_{h'}^k} \tilde{\eta}_n^{N_{h'}^k} \frac{1}{N_{h'}^{k^n+1}} \sum_{i=1}^{N_{h'}^{k^n+1}} (V_{h'+1}^{\mathrm{R},k^i} - \hat{V}_{h'+1}^{\mathrm{R},k^i})(s_{h'+1}^{k^i,j^i,m^i})$$

$$+ \sum_{h'=1}^{H} \sum_{k,j,m,N_{h'}^k>0} \sum_{n=1}^{N_{h'}^k} \tilde{\eta}_n^{N_{h'}^k} \frac{1}{N_{h'}^{k^n+1}} \sum_{i=1}^{N_{h'}^{k^n+1}} \mathbb{P}_{s_{h'}^{k,j,m}, a_{h'}^{k,j,m}, h'} \left|\hat{V}_{h'+1}^{\mathrm{R},k^i} - V_{h'+1}^{\mathrm{R},K+1}\right|$$

$$+ \sum_{h'=1}^{H} \sum_{k,j,m,N_{h'}^k>0} \sum_{n=1}^{N_{h'}^k} \tilde{\eta}_n^{N_{h'}^k} \mathbb{P}_{s_{h'}^{k,j,m}, a_{h'}^{k,j,m}, h'} \left|V_{h'+1}^{\mathrm{R},K+1} - \hat{V}_{h'+1}^{\mathrm{R},k^n}\right|$$

$$\lesssim \frac{H^6 SA\iota^2}{\beta} + MH^5 SA\iota^2. \tag{144}$$

The last inequality is by Equation (119) and Equation (124). Combining the results of Equation (138), Equation (141), Equation (142), Equation (143) and Equation (144), we completed the proof of Equation (134). $\qquad\square$

## L    Proof of Gap-Dependent Switching/Communication Cost (Theorem 4.7 and Theorem 4.9)

### L.1    Auxiliary Lemmas

**Lemma L.1.** *We have the following conclusions:*

*(a) Under the event $\bigcap_{i=1}^{14} \mathcal{E}_i$, the following event holds for some sufficiently large constant $c_0 > 1$:*

$$\mathcal{E}_{15} = \left\{ \sum_{h=1}^{H} \sum_{k,j,m} \mathbb{I}\left[(Q_h^k - Q_h^\star)(s_h^{k,m,j}, a_h^{k,m,j}) \geq \Delta_{\min}\right]\right.$$

$$\left. \leq C_{\min} \stackrel{\triangle}{=} c_0 \left(\frac{(\mathbb{Q}^\star + \beta^2 H) H^3 SA\iota}{\Delta_{\min}^2} + \frac{H^7 SA\iota^2}{\beta\Delta_{\min}} + \frac{MH^6 SA\iota^2}{\Delta_{\min}}\right), \forall h \in [H]\right\}.$$

*(b) For any deterministic optimal policy $\pi^\star$, with probability at least $1 - \delta$, the following event holds:*

$$\mathcal{E}_{16} = \left\{ \sum_{k=1}^{k'} \sum_{j,m} \mathbb{P}\left(a_h^{k,j,m} \neq \pi_h^\star(s_h^{k,j,m}) \mid \pi^k\right)\right.$$

$$\left. \leq 3 \sum_{k=1}^{k'} \sum_{j,m} \mathbb{I}\left[a_h^{k,j,m} \neq \pi_h^\star(s_h^{k,j,m})\right] + 2\iota, \forall h \in [H], k' \in [K]\right\}.$$

*(c) For any $k' \in [K]$, let $R_{k'} = \sum_{k=1}^{k'} \sum_{j,m} 1$, which is the total number of episodes in the first $k'$ rounds. Then with probability at least $1 - \delta$, the following event holds:*

$$\mathcal{E}_{17} = \left\{ \left| \sum_{k=1}^{k'} \sum_{j,m} \left\{ \mathbb{I} \left[ s_h^{k,j,m} = s \right] - \mathbb{P} \left( s_h^{k,j,m} = s | \pi^k \right) \right\} \right| \right.$$

$$\left. \leq \sqrt{24 \left( \sum_{k=1}^{k'} \sum_{j,m} \mathbb{P} \left( s_h^{k,j,m} = s | \pi^k \right) \right) \iota + 9\iota}, \; \forall s \in \mathcal{S}, h \in [H], k' \in [K] \right\}.$$

*(d) With probability at least $1 - \delta$, the following event holds:*

$$\mathcal{E}_{18} = \left\{ \left| \sum_{j=1}^{J_k} \left\{ \mathbb{I} \left[ s_h^{k,j,m} = s \right] - \mathbb{P} \left( s_h^{k,j,m} = s | \pi^k \right) \right\} \right| \right.$$

$$\left. \leq \sqrt{32 \left( \sum_{j=1}^{J_k} \mathbb{P} \left( s_h^{k,j,m} = s | \pi^k \right) \right) \iota + 11\iota}, \; \forall (s, h, k, m) \right\}.$$

*Here, under the full synchronization assumption, we can assume that in the $k-$th round, each agent will generate $J_k$ episodes. Note that given the round $k$ and the policy $\pi^k$, under random initialization assumption, the probability $\mathbb{P}(s_h^{k,j,m} = s | \pi^k)$ is independent of the index $m, j$. Let $\mathbb{P}_{s,h}^k = \mathbb{P}(s_h^{k,j,m} = s | \pi^k)$, then $\mathcal{E}_{18}$ can be written as*

$$\mathcal{E}_{18} = \left\{ \left| \sum_{j=1}^{J_k} \mathbb{I} \left[ s_h^{k,j,m} = s \right] - J_k \mathbb{P}_{s,h}^k \right| \leq \sqrt{32 J_k \mathbb{P}_{s,h}^k \iota + 11\iota}, \; \forall (s, h, k, m) \right\}.$$

*Proof.* (a) It is proved by Lemma K.2.

(b) We order all the episodes in the sequence following the rule: first by round index, second by episode index, and third by agent index. Let $n(k, j, m)$ denote the position of the $j-$th episode of the $m-$th agent in the $k-$th round of the sequence. The filtration $\mathcal{F}_{n(k,j,m)}$ is the $\sigma-$field generated by all the random variables until the first $n(k, j, m) - 1$ episodes. When there is no ambiguity, we will abbreviate $n(k, j, m)$ as $n$ and $\mathcal{F}_{n(k,j,m)}$ as $\mathcal{F}_n$. Then we have:

$$\mathbb{P} \left( a_{h'}^{k,j,m} \neq \pi_{h'}^{\star}(s_{h'}^{k,j,m}) \mid \pi^k \right) = \mathbb{P} \left( a_{h'}^{k,j,m} \neq \pi_{h'}^{\star}(s_{h'}^{k,j,m}) \mid \mathcal{F}_n \right).$$

According to Theorem E.3, with probability at least $1 - \delta/T_1^2$, the following inequality holds for any given $h = h' \in [H]$, $k' = k'_0 \in [\frac{T_1}{H}]$ and $R_{k'_0} = \sum_{k=1}^{k'_0} \sum_{j,m} 1 \in [T_1]$ :

$$\sum_{k=1}^{k'_0} \sum_{j,m} \mathbb{P} \left( a_{h'}^{k,j,m} \neq \pi_{h'}^{\star}(s_{h'}^{k,j,m}) \mid \pi^k \right) \leq 3 \sum_{k=1}^{k'_0} \sum_{j,m} \mathbb{I} \left[ a_{h'}^{k,j,m} \neq \pi_{h'}^{\star}(s_{h'}^{k,j,m}) \right] + 2\iota, \; \forall k' \in [K].$$

Considering all the possible values of $h = h' \in [H]$, $k' = k'_0 \in [\frac{T_1}{H}]$ and $R_{k'_0} = \sum_{k=1}^{k'_0} \sum_{j,m} 1 \in [T_1]$, with probability at least $1 - \delta$, it holds simultaneously for all $h \in [H]$, $k' \in [\frac{T_1}{H}]$ and $R_{k'} = \sum_{k=1}^{k'} \sum_{j,m} 1 \in [T_1]$ that

$$\sum_{k=1}^{k'} \sum_{j,m} \mathbb{P} \left( a_h^{k,j,m} \neq \pi_h^{\star}(s_h^{k,j,m}) \mid \pi^k \right) \leq 3 \sum_{k=1}^{k'} \sum_{j,m} \mathbb{I} \left[ a_h^{k,j,m} \neq \pi_h^{\star}(s_h^{k,j,m}) \right] + 2\iota.$$

(c) According to Theorem E.2, with probability at least $1 - \delta/ST_1^2$, the following inequality holds for any given $s' \in \mathcal{S}$, $h = h' \in [H]$, $k' = k'_0 \in [\frac{T_1}{H}]$ and $R_{k'_0} = \sum_{k=1}^{k'_0} \sum_{j,m} 1 \in [T_1]$ :

$$\left| \sum_{k=1}^{k'_0} \sum_{j,m} \left\{ \mathbb{I} \left[ s_{h'}^{k,j,m} = s' \right] - \mathbb{P} \left( s_{h'}^{k,j,m} = s' | \pi^k \right) \right\} \right| \leq \sqrt{24 \left( \sum_{k=1}^{k'_0} \sum_{j,m} \mathbb{P} \left( s_{h'}^{k,j,m} = s' | \pi^k \right) \right) \iota + 9\iota}.$$

Here, we let $\sigma^2 = T_1$, $m = \lceil \log_2(T_1) \rceil$ in Theorem E.2 and

$$W_n = \sum_{k=1}^{k_0'} \sum_{j,m} \mathbb{P}\left(s_{h'}^{k,j,m} = s'|\pi^k\right)\left(1 - \mathbb{P}\left(s_{h'}^{k,j,m} = s'|\pi^k\right)\right) \le \sum_{k=1}^{k_0'} \sum_{j,m} \mathbb{P}\left(s_{h'}^{k,j,m} = s'|\pi^k\right).$$

Considering all the possible values of $s = s' \in \mathcal{S}$, $h = h' \in [H]$, $k' = k_0' \in [\frac{T_1}{H}]$, $\hat{T} = T' \in [T_1]$, with probability at least $1 - \delta$, it holds simultaneously for all $s \in \mathcal{S}$, $h \in [H]$, $k' \in [\frac{T_1}{H}]$ and $\hat{T} \in [T_1]$

$$\left| \sum_{k=1}^{k'} \sum_{j,m} \left\{ \mathbb{I}\left[s_h^{k,j,m} = s\right] - \mathbb{P}\left(s_h^{k,j,m} = s|\pi^k\right)\right\} \right| \le \sqrt{24\left(\sum_{k=1}^{k'}\sum_{j,m} \mathbb{P}\left(s_h^{k,j,m} = s|\pi^k\right)\right)\iota} + 9\iota.$$

(d) The proof is similar to (c), considering all the combinations of $(s, h, m, k, R_k) \in \mathcal{S} \times [H] \times [M] \times [\frac{T_1}{H}] \times [T_1]$. $\qquad\square$

**Lemma L.2.** *Under the event $\bigcap_{i=1}^{18} \mathcal{E}_i$, for any given deterministic optimal policy $\pi^\star$, we have*

$$\sum_{h=1}^{H} \sum_{k,j,m} \mathbb{I}\left[a_h^{k,j,m} \notin \mathcal{A}_h^\star(s_h^{k,j,m})\right] \le C_{\min},$$

$$\sum_{k,j,m} \mathbb{P}\left(a_h^{k,j,m} \ne \pi_h^\star(s_h^{k,j,m}) \mid \pi^k\right) \le 4C_{\min}.$$

*Here $C_{\min}$ is the upper bound in the right-hand side of $\mathcal{E}_{15}$ in Lemma L.1 .*

*Proof.* For any $h \in [H]$, let the set $D_h$ be all triples of $(s, a, h)$ such that $a \notin \mathcal{A}_h^\star(s)$, that is $D_h = \{(s, a, h)|a \notin \mathcal{A}_h^\star(s)\}$. We also let the set $D = \bigcup_{h=1}^{H} D_h$ and the set $D_{\mathrm{opt}} = \{(s, a, h)|a \in \mathcal{A}_h^\star(s)\}$. Then we have $|D| + |D_{\mathrm{opt}}| = SAH$.

If for given $(h, k, j, m)$, $(s_h^{k,m,j}, a_h^{k,m,j}, h) \in D_h$, we have $\Delta_h(s_h^{k,m,j}, a_h^{k,m,j}) \ge \Delta_{\min}$. By Lemma H.1, we have

$$Q_h^k(s_h^{k,j,m}, a_h^{k,j,m}) = \max_a\{Q_h^k(s_h^{k,j,m}, a)\} \ge \max_a\{Q_h^\star(s_h^{k,j,m}, a)\} = V_h^\star(s_h^{k,j,m}).$$

Therefore, it holds that

$$Q_h^k(s_h^{k,m,j}, a_h^{k,m,j}) - Q_h^\star(s_h^{k,m,j}, a_h^{k,m,j}) \ge \Delta_h(s_h^{k,m,j}, a_h^{k,m,j}) \ge \Delta_{\min}.$$

Then we have

$$\mathbb{I}\left[a_h^{k,j,m} \notin \mathcal{A}_h^\star(s_h^{k,j,m})\right] = \mathbb{I}\left[(s_h^{k,m,j}, a_h^{k,m,j}, h) \in D_h\right]$$
$$\le \mathbb{I}\left[Q_h^k(s_h^{k,m,j}, a_h^{k,m,j}) - Q_h^\star(s_h^{k,m,j}, a_h^{k,m,j}) \ge \Delta_{\min}\right],$$

and thus by the event $\mathcal{E}_{15}$ in Lemma L.1, it holds that

$$\sum_{h=1}^{H} \sum_{k,j,m} \mathbb{I}\left[a_h^{k,j,m} \notin \mathcal{A}_h^\star(s_h^{k,j,m})\right]$$
$$\le \sum_{h=1}^{H} \sum_{k,j,m} \mathbb{I}\left[Q_h^k(s_h^{k,m,j}, a_h^{k,m,j}) - Q_h^\star(s_h^{k,m,j}, a_h^{k,m,j}) \ge \Delta_{\min}\right] \le C_{\min}. \qquad (145)$$

Next, we prove the second conclusion. Let $\mathcal{S}_h^0 = \{s \mid \mathbb{P}_{s,h}^\star = 0\}$. For any given deterministic optimal policy $\pi^\star$, we have

$$\mathbb{I}\left[a_h^{k,j,m} \ne \pi_h^\star(s_h^{k,j,m})\right] = \mathbb{I}\left[a_h^{k,j,m} \ne \pi_h^\star(s_h^{k,j,m}), s_h^{k,j,m} \notin \mathcal{S}_h^0\right]$$
$$+ \mathbb{I}\left[a_h^{k,j,m} \ne \pi_h^\star(s_h^{k,j,m}), s_h^{k,j,m} \in \mathcal{S}_h^0\right]. \qquad (146)$$

For $s_h^{k,j,m} \notin \mathcal{S}_h^0$, we have $\mathbb{P}^{\star}_{s_h^{k,j,m},h} > 0$ and $|\mathcal{A}_h^{\star}(s_h^{k,j,m})| = 1$ by condition (b) of Definition 4.6. This means $\pi_h^{\star}(s_h^{k,j,m})$ is the only element in $\mathcal{A}_h^{\star}(s_h^{k,j,m})$. Therefore, we have

$$\mathbb{I}\left[a_h^{k,j,m} \neq \pi_h^{\star}(s_h^{k,j,m}), s_h^{k,j,m} \notin \mathcal{S}_h^0\right] \leq \mathbb{I}\left[a_h^{k,j,m} \notin \mathcal{A}_h^{\star}(s_h^{k,j,m})\right]. \tag{147}$$

For the second term in Equation (146), if $h = 1$, because of the randomness of the selection of $s_1^{k,j,m}$, we have $\mathbb{P}(s_1 = s_1^{k,j,m} | \pi^{\star}) = \mathbb{P}(s_1 = s_1^{k,j,m}) > 0$ and then

$$\mathbb{I}\left[a_1^{k,j,m} \neq \pi_1^{\star}(s_1^{k,j,m}), s_1^{k,j,m} \in \mathcal{S}_1^0\right] = 0. \tag{148}$$

To bound the second term in Equation (146) for $h > 1$, we first prove a lemma.

**Lemma L.3.** *For any $h \in [H]$ and the trajectory $\{(s_h^{k,j,m}, a_h^{k,j,m}, r_h^{k,j,m})\}_{h=1}^H$ in $j-$th episode of agent $m$ in round $k$, if $\mathbb{P}^{\star}_{s_h^{k,j,m},h} > 0$ and $a_h^{k,j,m}$ is the unique optimal action for state $s_h^{k,j,m}$ at step $h$, then we have $\mathbb{P}^{\star}_{s_{h+1}^{k,j,m},h+1} > 0$*

*Proof.* For any given optimal policy $\pi^{\star}$, it holds that

$$\begin{aligned}
\mathbb{P}^{\star}_{s_{h+1}^{k,j,m},h+1} &= \mathbb{P}\left(s_{h+1} = s_{h+1}^{k,j,m} \mid \pi^{\star}\right) \\
&\geq \mathbb{P}\left(s_{h+1} = s_{h+1}^{k,j,m} \mid s_h = s_h^{k,j,m}, a_h = a_h^{k,j,m}, \pi^{\star}\right) \times \mathbb{P}\left(s_h = s_h^{k,j,m}, a_h = a_h^{k,j,m} \mid \pi^{\star}\right) \\
&\overset{(I)}{=} \mathbb{P}\left(s_{h+1} = s_{h+1}^{k,j,m} \mid s_h = s_h^{k,j,m}, a_h = a_h^{k,j,m}\right) \times \mathbb{P}^{\star}_{s_h^{k,j,m},h} > 0
\end{aligned}$$

The equation (I) is by Markov property and

$$\mathbb{P}(s_h = s_h^{k,j,m}, a_h = a_h^{k,j,m} \mid \pi^{\star}) = \mathbb{P}(s_h = s_h^{k,j,m} \mid \pi^{\star}) = \mathbb{P}^{\star}_{s_h^{k,j,m},h}.$$

$\square$

For every initial state $s_1^{k,j,m}$, we know $\mathbb{P}^{\star}_{s_1^{k,j,m},1} > 0$. Therefore, if for $h > 1$, $\mathbb{P}^{\star}_{s_h^{k,j,m},h} = 0$ and $s_h^{k,j,m} \in \mathcal{S}_h^0$, by Lemma L.3, we know there exists $h' < h$ such that $a_{h'}^{k,m,j}$ is not an optimal action for state $s_{h'}^{k,m,j}$ at step $h'$, otherwise we have $\mathbb{P}^{\star}_{s_h^{k,j,m},h} > 0$. Therefore, for the second term in Equation (146), we have

$$\mathbb{I}\left[a_h^{k,j,m} \neq \pi_h^{\star}(s_h^{k,j,m}), s_h^{k,j,m} \in \mathcal{S}_h^0\right] \leq \mathbb{I}\left[s_h^{k,j,m} \in \mathcal{S}_h^0\right] \leq \sum_{h'=1}^{h-1} \mathbb{I}\left[a_{h'}^{k,j,m} \notin \mathcal{A}_{h'}^{\star}(s_{h'}^{k,j,m})\right]. \tag{149}$$

By combining the results of Equation (147), Equation (148) and Equation (149), we can bound the Equation (146) as follows:

$$\mathbb{I}\left[a_h^{k,j,m} \neq \pi_h^{\star}(s_h^{k,j,m})\right] \leq \sum_{h'=1}^{h} \mathbb{I}\left[a_{h'}^{k,j,m} \notin \mathcal{A}_{h'}^{\star}(s_{h'}^{k,j,m})\right] \leq \sum_{h'=1}^{H} \mathbb{I}\left[a_{h'}^{k,j,m} \notin \mathcal{A}_{h'}^{\star}(s_{h'}^{k,j,m})\right].$$

Therefore, using Equation (145), we reach

$$\sum_{k,j,m} \mathbb{I}\left[a_h^{k,j,m} \neq \pi_h^{\star}(s_h^{k,j,m})\right] \leq \sum_{k,j,m} \sum_{h'=1}^{H} \mathbb{I}\left[a_{h'}^{k,j,m} \notin \mathcal{A}_{h'}^{\star}(s_{h'}^{k,j,m})\right] \leq C_{\min}.$$

Combined with the event $\mathcal{E}_{16}$ in Lemma L.1, we can conclude that for any $h \in [H]$ and $k' \in [K]$,

$$\sum_{k=1}^{k'} \sum_{j,m} \mathbb{P}\left(a_h^{k,j,m} \neq \pi_h^{\star}(s_h^{k,j,m}) \mid \pi^k\right) \leq 4C_{\min}.$$

$\square$

Let $i_1 = 300MH(H+1)\iota$, $i_2 = 6500H^3 C_{\min}/C_{st}$, and $\tilde{C} = 1/(H(H+1))$. For any $(s, h) \in \mathcal{S} \times [H]$ such that $\mathbb{P}_{s,h}^\star > 0$, we use $\pi_h^\star(s)$ to denote its unique optimal action. Before proceeding, we present two lemmas. Lemma L.4 shows agent-wise simultaneous sufficient increase of visits for the triple $(s, a, h)$ that satisfies the trigger condition in round $k$ when $N_h^k(s, a) > i_1$.

**Lemma L.4.** *Under the event $\bigcap_{i=1}^{18} \mathcal{E}_i$, for any $(s, a, h, k) \in \mathcal{S} \times \mathcal{A} \times [H] \times [K]$ such that $N_h^k(s, a) > i_1$ and the triple $(s, a, h)$ satisfies the trigger condition* (3) *in round $k$, it holds that*

$$N_h^{k+1}(s, a) \geq (1 + \tilde{C}/3) N_h^k(s, a).$$

*Proof.* If the trigger condition is met by the triple $(s, a, h)$ in round $k$, then we have $a = \pi_h^k(s)$. For such $(s, a, h)$, by $\mathcal{E}_{18}$ in Lemma L.1, it holds for any $s \in \mathcal{S}$, $h \in [H]$, $k \in [K]$ and $m \in [M]$ that

$$\sum_{j=1}^{J_k} \mathbb{I}\left[s_h^{k,j,m} = s, a_h^{k,j,m} = a\right] = \sum_{j=1}^{J_k} \mathbb{I}\left[s_h^{k,j,m} = s\right]$$

$$\in \left[J_k \mathbb{P}_{s,h}^k - \sqrt{32 J_k \mathbb{P}_{s,h}^k \iota} - 11\iota, \; J_k \mathbb{P}_{s,h}^k + \sqrt{32 J_k \mathbb{P}_{s,h}^k \iota} + 11\iota\right]. \tag{150}$$

Since $(s, a, h)$ satisfies the trigger condition in round $k$, there exists an agent $m_0$ such that $n_h^{k,m_0}(s, a) = c_h^k(s, a)$. Then we reach

$$J_k \mathbb{P}_{s,h}^k + \sqrt{32 J_k \mathbb{P}_{s,h}^k \iota} + 11\iota \overset{(i)}{\geq} \frac{N_h^k(s, a)}{MH(H+1)} - 1 \overset{\triangle}{=} C_N > 299\iota.$$

The last inequality is because $N_h^k(s, a) > i_1$. Solving the inequality (i), it holds that

$$\sqrt{J_k \mathbb{P}_{s,h}^k} \geq \sqrt{C_N - 3\iota} - \sqrt{8\iota}.$$

Then by Equation (150), for any other agent $m$,

$$\sum_{j=1}^{J_k} \mathbb{I}\left[s_h^{k,j,m} = s, a_h^{k,j,m} = a\right] \geq J_k \mathbb{P}_{s,h}^k - \sqrt{32 J_k \mathbb{P}_{s,h}^k \iota} - 11\iota = \left(\sqrt{J_k \mathbb{P}_{s,h}^k} - \sqrt{8\iota}\right)^2 - 19\iota$$

$$\geq \left(\sqrt{C_N - 3\iota} - 2\sqrt{8\iota}\right)^2 - 19\iota \geq \frac{C_N + 1}{3}.$$

The last inequality is because $C_N > 299\iota$. Therefore, we have

$$n_h^k(s, a) = \sum_{m=1}^M n_h^{m,k}(s, a) = \sum_{m=1}^M \sum_{j=1}^{J_k} \mathbb{I}\left[s_h^{k,j,m} = s, a_h^{k,j,m} = a\right] \geq \frac{M(C_N + 1)}{3} = \frac{N_h^k(s, a)}{3H(H+1)},$$

and thus

$$N_h^{k+1}(s, a) = N_h^k(s, a) + n_h^k(s, a) \geq \left(1 + \frac{1}{3H(H+1)}\right) N_h^k(s, a).$$

$\square$

Lemma L.5 shows the state-wise simultaneous sufficient increase of visits for states with unique optimal action.

**Lemma L.5.** *Under the event $\bigcap_{i=1}^{18} \mathcal{E}_i$, if there exists $(s_0, a_0, h_0) \in \mathcal{S} \times \mathcal{A} \times [H]$, such that it satisfies the trigger condition in round $k$ and $N_{h_0}^k(s_0, a_0) > i_1 + i_2$, then $a_0 \in \mathcal{A}_{h_0}^\star(s_0)$. Moreover, if such $(s_0, a_0, h_0)$ also satisfies that $\mathbb{P}_{s_0, h_0}^\star > 0$, then*

$$N_{h'}^{k+1}(s', \pi_{h'}^\star(s')) \geq (1 + \tilde{C}/6) N_{h'}^k(s', \pi_{h'}^\star(s'))$$

*holds for any $(s', h') \in \mathcal{S} \times [H]$ such that $\mathbb{P}_{s', h'}^\star > 0$.*

*Proof.* Because $N_h^k(s_0, a_0) > i_1 + i_2 > C_{\min}$, by Lemma L.2, we know $a_0 \in \mathcal{A}_h^\star(s_0)$. Next, we prove the second conclusion. Using the law of total probability, for any $0 \le h \le H - 1$, $s \in \mathcal{S}$, and any given deterministic optimal policy $\pi^\star$, we have the following relationship

$$
\mathbb{P}\left(s_{h+1}^{k,j,m} = s \mid \pi^k\right)
$$
$$
= \sum_{s'} \mathbb{P}\left(s_{h+1}^{k,j,m} = s | s_h^{k,j,m} = s', a_h^{k,j,m} = \pi_h^\star(s'), \pi^k\right) \mathbb{P}\left(s_h^{k,j,m} = s', a_h^{k,j,m} = \pi_h^\star(s') \mid \pi^k\right)
$$
$$
+ \mathbb{P}\left(s_{h+1}^{k,j,m} = s, a_h^{k,j,m} \neq \pi_h^\star(s_h^{k,j,m}) \mid \pi^k\right)
$$
$$
= \sum_{s'} \mathbb{P}_{s,s',h}^{k,j,m} \cdot \mathbb{P}\left(s_h^{k,j,m} = s', a_h^{k,j,m} = \pi_h^\star(s') \mid \pi^k\right) + \mathbb{P}\left(s_{h+1}^{k,j,m} = s, a_h^{k,j,m} \neq \pi_h^\star(s_h^{k,j,m}) \mid \pi^k\right),
$$
$$(151)$$

where

$$
\mathbb{P}_{s,s',h}^{k,j,m} = \mathbb{P}\left(s_{h+1}^{k,j,m} = s | s_h^{k,j,m} = s', a_h^{k,j,m} = \pi_h^\star(s'), \pi^k\right)
$$
$$
= \mathbb{P}\left(s_{h+1}^{k,j,m} = s | s_h^{k,j,m} = s', a_h^{k,j,m} = \pi_h^\star(s')\right).
$$

The last equality is because of the Markov property. We also have

$$
\mathbb{P}\left(s_{h+1}^{k,j,m} = s | \pi^\star\right) = \sum_{s'} \mathbb{P}\left(s_{h+1}^{k,j,m} = s | s_h^{k,j,m} = s', \pi^\star\right) \mathbb{P}\left(s_h^{k,j,m} = s' | \pi^\star\right)
$$
$$
= \sum_{s'} \mathbb{P}_{s,s',h}^{k,j,m} \cdot \mathbb{P}\left(s_h^{k,j,m} = s' | \pi^\star\right),
$$
$$(152)$$

where the last equation is because

$$
\mathbb{P}(s_{h+1}^{k,j,m} = s | s_h^{k,j,m} = s', \pi^\star) = \mathbb{P}\left(s_{h+1}^{k,j,m} = s | s_h^{k,j,m} = s', a_h^{k,j,m} = \pi_h^\star(s')\right) = \mathbb{P}_{s,s',h}^{k,j,m}.
$$

Combining the results of Equation (151) and Equation (152), then it holds

$$
\mathbb{P}\left(s_{h+1}^{k,j,m} = s | \pi^k\right) - \mathbb{P}\left(s_{h+1}^{k,j,m} = s | \pi^\star\right)
$$
$$
= \sum_{s'} \mathbb{P}_{s,s',h}^{k,j,m} \left[\mathbb{P}\left(s_h^{k,j,m} = s', a_h^{k,j,m} = \pi_h^\star(s') | \pi^k\right) - \mathbb{P}\left(s_h^{k,j,m} = s' | \pi^\star\right)\right]
$$
$$
+ \mathbb{P}\left(s_{h+1}^{k,j,m} = s, a_h^{k,j,m} \neq \pi_h^\star(s_h^{k,j,m}) | \pi^k\right)
$$
$$
= \sum_{s'} \mathbb{P}_{s,s',h}^{k,j,m} \left[\mathbb{P}\left(s_h^{k,j,m} = s' \mid \pi^k\right) - \mathbb{P}\left(s_h^{k,j,m} = s' | \pi^\star\right)\right]
$$
$$
- \sum_{s'} \mathbb{P}_{s,s',h}^{k,j,m} \mathbb{P}\left(s_h^{k,j,m} = s', a_h^{k,j,m} \neq \pi_h^\star(s') \mid \pi^k\right) + \mathbb{P}\left(s_{h+1}^{k,j,m} = s, a_h^{k,j,m} \neq \pi_h^\star(s_h^{k,j,m}) \mid \pi^k\right).
$$

Therefore for any $0 \le h \le H - 1$ and $s \in \mathcal{S}$, by the triangle inequality, it holds that

$$
\left|\mathbb{P}\left(s_{h+1}^{k,j,m} = s \mid \pi^k\right) - \mathbb{P}\left(s_{h+1}^{k,j,m} = s | \pi^\star\right)\right|
$$
$$
\le \sum_{s'} \mathbb{P}_{s,s',h}^{k,j,m} \left|\mathbb{P}\left(s_h^{k,j,m} = s' \mid \pi^k\right) - \mathbb{P}\left(s_h^{k,j,m} = s' | \pi^\star\right)\right|
$$
$$
+ \sum_{s'} \mathbb{P}_{s,s',h}^{k,j,m} \mathbb{P}\left(s_h^{k,j,m} = s', a_h^{k,j,m} \neq \pi_h^\star(s') \mid \pi^k\right) + \mathbb{P}\left(s_{h+1}^{k,j,m} = s, a_h^{k,j,m} \neq \pi_h^\star(s_h^{k,j,m}) \mid \pi^k\right).
$$

Summing the above inequality for all $s \in \mathcal{S}$, since $\sum_s \mathbb{P}_{s,s',h} = 1$, then we can derive that:

$$
\sum_s \left|\mathbb{P}\left(s_{h+1}^{k,j,m} = s \mid \pi^k\right) - \mathbb{P}\left(s_{h+1}^{k,j,m} = s | \pi^\star\right)\right|
$$
$$
\le \sum_{s'} \left|\mathbb{P}\left(s_h^{k,j,m} = s' \mid \pi^k\right) - \mathbb{P}\left(s_h^{k,j,m} = s' | \pi^\star\right)\right| + 2\mathbb{P}\left(a_h^{k,j,m} \neq \pi_h^\star(s_h^{k,j,m}) \mid \pi^k\right).
$$

Since $\mathbb{P}(s_1^{k,j,m} = s \mid \pi^k) - \mathbb{P}(s_1^{k,j,m} = s|\pi^\star) = 0$, by recursion, for any $h' \in [H]$ we have

$$\sum_s \left| \mathbb{P}\left(s_{h'}^{k,j,m} = s \mid \pi^k\right) - \mathbb{P}\left(s_{h'}^{k,j,m} = s|\pi^\star\right) \right| \leq 2 \sum_{h=1}^{h'-1} \mathbb{P}\left(a_h^{k,j,m} \neq \pi_h^\star(s_h^{k,j,m}) \mid \pi^k\right)$$

$$\leq 2 \sum_{h=1}^{H} \mathbb{P}\left(a_h^{k,j,m} \neq \pi_h^\star(s_h^{k,j,m}) \mid \pi^k\right).$$

Using Lemma L.2, then for any $h \in [H]$ and $k' \in [K]$, it holds that:

$$\sum_s \sum_{k=1}^{k'} \sum_{j,m} \left| \mathbb{P}\left(s_h^{k,j,m} = s \mid \pi^k\right) - \mathbb{P}\left(s_h^{k,j,m} = s|\pi^\star\right) \right|$$

$$\leq 2 \sum_{h=1}^{H} \sum_{k=1}^{k'} \sum_{j,m} \mathbb{P}\left(a_h^{k,j,m} \neq \pi_h^\star(s_h^{k,j,m}) \mid \pi^k\right) \leq 8HC_{\min}.$$

Note that based on the property (b) of Definition 4.6, we have $\mathbb{P}(s_h^{k,j,m} = s|\pi^\star) = \mathbb{P}_{s,h}^\star$, then for any $s \in \mathcal{S}$, $h \in [H]$ and $k' \in [K]$, by the triangle inequality, we also have

$$\sum_{k=1}^{k'} \sum_{j,m} \mathbb{P}\left(s_h^{k,j,m} = s \mid \pi^k\right)$$

$$\leq R_{k'}\mathbb{P}_{s,h}^\star + \sum_{k=1}^{k'} \sum_{j,m} \left| \mathbb{P}\left(s_h^{k,j,m} = s \mid \pi^k\right) - \mathbb{P}\left(s_h^{k,j,m} = s|\pi^\star\right) \right| \leq R_{k'}\mathbb{P}_{s,h}^\star + 8HC_{\min}. \quad (153)$$

Applying Equation (153) to $\mathcal{E}_{17}$ in Lemma L.1, for any $s \in \mathcal{S}$, $h \in [H]$ and $k' \in [K]$, we have

$$\left| \sum_{k=1}^{k'} \sum_{j,m} \left\{ \mathbb{I}\left[s_h^{k,j,m} = s\right] - \mathbb{P}\left(s_h^{k,j,m} = s|\pi^k\right) \right\} \right| \leq \sqrt{24 \left( \sum_{k=1}^{k'} \sum_{j,m} \mathbb{P}\left(s_h^{k,j,m} = s|\pi^k\right) \right) \iota + 9\iota}$$

$$\leq \sqrt{24 \left( R_{k'}\mathbb{P}_{s,h}^\star + 8HC_{\min} \right) \iota + 9\iota}$$

$$\leq 5\sqrt{R_{k'}\mathbb{P}_{s,h}^\star \iota} + 23HC_{\min}. \quad (154)$$

Combining the results of Equation (153) and Equation (154), by triangle inequality, we can derive the following relationship for any $s \in \mathcal{S}$, $h \in [H]$, and $k' \in [K]$

$$\left| \sum_{k=1}^{k'} \sum_{j,m} \mathbb{I}\left[s_h^{k,j,m} = s\right] - R_{k'}\mathbb{P}_{s,h}^\star \right| \leq 5\sqrt{R_{k'}\mathbb{P}_{s,h}^\star \iota} + 31HC_{\min}. \quad (155)$$

Then by triangle inequality, it holds for any $s \in \mathcal{S}$, $h \in [H]$ and $k' \in [K]$ that

$$\left| \sum_{k=1}^{k'} \sum_{j,m} \mathbb{I}\left[s_h^{k,j,m} = s, a_h^{k,j,m} \in \mathcal{A}_h^\star(s)\right] - R_{k'}\mathbb{P}_{s,h}^\star \right|$$

$$\leq \left| \sum_{k=1}^{k'} \sum_{j,m} \mathbb{I}\left[s_h^{k,j,m} = s\right] - R_{k'}\mathbb{P}_{s,h}^\star \right| + \sum_{k=1}^{k'} \sum_{j,m} \mathbb{I}\left[s_h^{k,j,m} = s, a_h^{k,j,m} \notin \mathcal{A}_h^\star(s)\right]$$

$$\leq 5\sqrt{R_{k'}\mathbb{P}_{s,h}^\star \iota} + 32HC_{\min}. \quad (156)$$

Here, the last inequality is by Equation (155), together with the fact from Lemma L.2 that

$$\sum_{k=1}^{k'} \sum_{j,m} \mathbb{I}\left[s_h^{k,j,m} = s, a_h^{k,j,m} \notin \mathcal{A}_h^\star(s)\right] \leq \sum_{k=1}^{k'} \sum_{j,m} \mathbb{I}\left[a_h^{k,j,m} \notin \mathcal{A}_h^\star(s_h^{k,j,m})\right] \leq C_{\min}.$$

For $P^\star_{s,h} > 0$, the optimal action is unique. Then for any $(s, a, h)$ such that $a = \pi^\star_h(s)$ and $P^\star_{s,h} > 0$, we can simplify the results of Equation (156) as follows:

$$R_{k'}\mathbb{P}^\star_{s,h} - 5\sqrt{R_{k'}\mathbb{P}^\star_{s,h}\iota} - 32HC_{\min} \leq N^{k'+1}_h(s, a) \leq R_{k'}\mathbb{P}^\star_{s,h} + 5\sqrt{R_{k'}\mathbb{P}^\star_{s,h}\iota} + 32HC_{\min}.$$

(157)

By Equation (157), for any $s' \in \mathcal{S}$ and $h' \in [H]$ such that $\mathbb{P}^\star_{s',h'} > 0$, we have

$$\frac{R_k\mathbb{P}^\star_{s',h'} - 5\sqrt{R_k\mathbb{P}^\star_{s',h'}\iota} - 32HC_{\min}}{R_{k-1}\mathbb{P}^\star_{s',h'} + 5\sqrt{R_{k-1}\mathbb{P}^\star_{s',h'}\iota} + 32HC_{\min}} \leq \frac{N^{k+1}_{h'}(s', \pi^\star_{h'}(s'))}{N^k_{h'}(s', \pi^\star_{h'}(s'))}.$$

For the second conclusion, we only need to prove that, for any $(s', h') \in \mathcal{S} \times [H]$ such that $\mathbb{P}^\star_{s',h'} > 0$,

$$\frac{R_k\mathbb{P}^\star_{s',h'} - 5\sqrt{R_k\mathbb{P}^\star_{s',h'}\iota} - 32HC_{\min}}{R_{k-1}\mathbb{P}^\star_{s',h'} + 5\sqrt{R_{k-1}\mathbb{P}^\star_{s',h'}\iota} + 32HC_{\min}} \geq 1 + \frac{1}{6H(H+1)}.$$

(158)

Next, we will prove the Equation (158). For the triple $(s_0, a_0, h_0)$, by Equation (157), we know that

$$\frac{6500H^3C_{\min}}{C_{st}} < N^k_{h_0}(s_0, a_0) \leq R_{k-1}\mathbb{P}^\star_{s_0,h_0} + 5\sqrt{R_{k-1}\mathbb{P}^\star_{s_0,h_0}\iota} + 32HC_{\min}.$$

Solving the inequality, we have

$$\sqrt{R_k\mathbb{P}^\star_{s_0,h_0}} > \sqrt{R_{k-1}\mathbb{P}^\star_{s_0,h_0}} > \sqrt{\frac{6500H^3C_{\min}}{C_{st}} - 32HC_{\min} + \frac{25\iota}{4}} - \frac{5\sqrt{\iota}}{2}$$

$$> \sqrt{\frac{6468H^3C_{\min}}{C_{st}}} - \sqrt{\frac{H^3C_{\min}}{C_{st}}} > 79\sqrt{\frac{H^3C_{\min}}{C_{st}}}.$$

(159)

Therefore, for any $s' \in \mathcal{S}$ and $h' \in [H]$ such that $\mathbb{P}^\star_{s',h'}$, by Equation (159), we have

$$\sqrt{R_k\mathbb{P}^\star_{s',h'}} > \sqrt{R_{k-1}\mathbb{P}^\star_{s',h'}} \geq \sqrt{R_{k-1}\mathbb{P}^\star_{s_0,h_0}} \cdot \sqrt{C_{st}} > 79\sqrt{H^3C_{\min}}.$$

(160)

For $X > 6241H^3C_{\min} = 79^2H^3C_{\min}$, note that

$$5\sqrt{X\iota} + 32HC_{\min} \leq \sqrt{\frac{C_{\min}X}{H}} + 32HC_{\min} \leq \frac{X}{56H^2}.$$

(161)

Here, the first inequality is because $5\sqrt{\iota} < \sqrt{\frac{C_{\min}}{H}}$ for $H \geq 2$. Therefore, based on Equation (159) and Equation (160), we can apply Equation (161) to $R_{k-1}\mathbb{P}^\star_{s_0,h}$, $R_k\mathbb{P}^\star_{s_0,h}$, $R_{k-1}\mathbb{P}^\star_{s',h}$ and $R_k\mathbb{P}^\star_{s',h}$:

$$5\sqrt{R_{k-1}\mathbb{P}^\star_{s_0,h_0}\iota} + 32HC_{\min} \leq \frac{R_{k-1}\mathbb{P}^\star_{s_0,h_0}}{56H^2}, \ 5\sqrt{R_k\mathbb{P}^\star_{s_0,h_0}\iota} + 32HC_{\min} \leq \frac{R_k\mathbb{P}^\star_{s_0,h_0}}{56H^2}.$$

(162)

and

$$5\sqrt{R_{k-1}\mathbb{P}^\star_{s',h'}\iota} + 32HC_{\min} \leq \frac{R_{k-1}\mathbb{P}^\star_{s',h'}}{56H^2}, \ 5\sqrt{R_k\mathbb{P}^\star_{s',h'}\iota} + 32HC_{\min} \leq \frac{R_k\mathbb{P}^\star_{s',h'}}{56H^2}$$

(163)

Since $N^k_h(s_0, a_0) > i_1$ and the trigger condition is satisfied by $(s, a, h)$ in round $k$, by Lemma L.4:

$$\frac{N^{k+1}_{h_0}(s_0, a_0)}{N^k_{h_0}(s_0, a_0)} \geq 1 + \frac{1}{3H(H+1)}.$$

Together with Equation (157), it holds that

$$\frac{R_k\mathbb{P}^\star_{s_0,h_0} + 5\sqrt{R_k\mathbb{P}^\star_{s_0,h_0}\iota} + 32HC_{\min}}{R_{k-1}\mathbb{P}^\star_{s_0,h_0} - 5\sqrt{R_{k-1}\mathbb{P}^\star_{s_0,h_0}\iota} - 32HC_{\min}} \geq \frac{N^{k+1}_{h_0}(s_0, a_0)}{N^k_{h_0}(s_0, a_0)} \geq 1 + \frac{1}{3H(H+1)}.$$

(164)

Applying Equation (162) to Equation (164), we have

$$1 + \frac{1}{3H(H+1)} \le \frac{R_k \mathbb{P}^\star_{s_0,h_0} + 5\sqrt{R_k \mathbb{P}^\star_{s_0,h_0}\iota} + 32HC_{\min}}{R_{k-1}\mathbb{P}^\star_{s_0,h_0} - 5\sqrt{R_{k-1}\mathbb{P}^\star_{s_0,h_0}\iota} - 32HC_{\min}} \le \frac{(1 + \frac{1}{56H^2})R_k}{(1 - \frac{1}{56H^2})R_{k-1}}.$$

Therefore, using Equation (163), we have

$$\frac{R_k \mathbb{P}^\star_{s',h'} - 5\sqrt{R_k \mathbb{P}^\star_{s',h'}\iota} - 32HC_{\min}}{R_{k-1}\mathbb{P}^\star_{s',h'} + 5\sqrt{R_{k-1}\mathbb{P}^\star_{s',h'}\iota} + 32HC_{\min}} \ge \frac{(1 - \frac{1}{56H^2})R_k}{(1 + \frac{1}{56H^2})R_{k-1}}$$

$$\ge \left(1 + \frac{1}{3H(H+1)}\right)\left(\frac{1 - \frac{1}{56H^2}}{1 + \frac{1}{56H^2}}\right)^2. \quad (165)$$

Let

$$c = \frac{1}{6H^2 + 6H + 2} \cdot \left(\frac{1}{1 + \sqrt{\frac{6H^2+6H+1}{6H^2+6H+2}}}\right)^2 > \frac{1}{4(6H^2 + 6H + 2)} \ge \frac{1}{56H^2},$$

and thus

$$\frac{1 + \frac{1}{6H(H+1)}}{1 + \frac{1}{3H(H+1)}} = \frac{6H^2 + 6H + 1}{6H^2 + 6H + 2} = \left(\frac{1-c}{1+c}\right)^2 \le \left(\frac{1 - \frac{1}{56H^2}}{1 + \frac{1}{56H^2}}\right)^2.$$

Applying this inequality to Equation (165) completes the proof of Equation (158), thereby proving the second conclusion. □

## L.2  Details of Final Discussion

In this section, we will discuss the number of communication rounds in four different situations:

**Situation 1:** In round $k$, the trigger condition is satisfied by $(s, a, h)$ when $N_h^k(s,a) \le i_1$. We will refer to this as a Type-I trigger.

For each time the trigger condition is met for $(s, a, h)$, the number of visits to $(s, a, h)$ increases by at least $1/(2MH(H+1))$ times by Equation (3). Therefore, the maximum number of Type-I triggers for each triple $(s, a, h)$, denoted $t_2(s, a, h)$, satisfies

$$\left(1 + \frac{1}{2MH(H+1)}\right)^{t_1(s,a,h)-2} \le i_1.$$

Therefore, we have $t_1(s, a, h) = O(MH^2 \log(i_1))$ and thus the number of rounds with Type-I triggers is bounded by

$$\sum_{s,a,h} t_1(s, a, h) \le O\left(MH^3 SA \log(i_1)\right).$$

**Situation 2:** In round $k$, the triple $(s, a, h)$ satisfies the trigger condition when $i_1 < N_h^k(s,a) < i_1 + i_2$. We will refer to this as a Type-II trigger if $a \notin \mathcal{A}_h^\star(s)$ or $a \in \mathcal{A}_h^\star(s)$ and $\mathbb{P}^\star_{s,h} = 0$, and as a Type-III trigger if $a \in \mathcal{A}_h^\star(s)$ and $\mathbb{P}^\star_{s,h} > 0$.

By Lemma L.4, for each time the trigger condition is satisfied by $(s, a, h)$, the number of visits to $(s, a, h)$ increases by at least $1/3H(H+1)$ times.

For $(s, a, h)$ satisfying the type-II trigger, by Lemma L.2 and Equation (156), we know that the maximum visit number to $(s, a, h)$ is $32HC_{\min}$. Therefore, the maximum number of Type-II triggers for each triple $(s, \pi_h^\star(s), h)$, denoted $t_2(s, a, h)$, satisfies

$$\left(1 + \frac{1}{3H(H+1)}\right)^{t_2(s,a,h)-1} \le \frac{32HC_{\min}}{i_1}.$$

Therefore, we have $t_2(s, a, h) \leq O(H^2 \log(32HC_{\min}/i_1))$ and thus the number of rounds with Type-II triggers is bounded by

$$\sum_{s,a,h} t_2(s, a, h) \leq O\left(H^3 SA \log\left(\frac{32HC_{\min}}{i_1}\right)\right).$$

**Situation 3:** By (b) of Definition 4.6, we know $a = \pi_h^\star(s)$ for a Type-III trigger. Therefore, the maximum number of Type-III triggers for each triple $(s, \pi_h^\star(s), h)$, denoted $t_3(s, \pi_h^\star(s), h)$, satisfies

$$\left(1 + \frac{1}{3H(H+1)}\right)^{t_3(s, \pi_h^\star(s), h) - 1} \leq \frac{i_1 + i_2}{i_1} \leq i_2.$$

Therefore, we have $t_3(s, \pi_h^\star(s), h) \leq O(H^2 \log(i_2))$. Because we only have $HS$ triples of $(s, \pi_h^\star(s), h)$ such that $\mathbb{P}_{s,h}^\star > 0$, the number of rounds with Type-III triggers is no more than

$$\sum_{s, \mathbb{P}_{s,h}^\star > 0} t_3(s, \pi_h^\star(s), h) \leq O\left(H^3 S \log(i_2)\right).$$

**Situation 4:** The trigger condition is satisfied by $(s, a, h)$ in round $k$ when $N_h^k(s, a) > i_1 + i_2$. We refer to this type of trigger as a Type-IV trigger.

By Lemma L.4, in this case, for each time the trigger condition is satisfied by $(s, a, h)$, we have $a \in \mathcal{A}_h^\star(s)$. we will first prove $\mathbb{P}_{s,h}^\star = 0$ in this case.

Let $\mathcal{S}_0 = \{(s, a, h) \mid a \in \mathcal{A}_h^\star(s),\ \mathbb{P}_{s,h}^\star = 0\}$. By Equation (156), we know for $(s, a, h) \in \mathcal{S}_0$, $N_h^{K+1}(s, a) \leq 32HC_{\min} < i_1 + i_2$. However, when the trigger condition is satisfied by $(s, a, h)$ in round $k$, we have $N_h^k(s, a) > i_1 + i_2$, which is contradicts the fact that $N_h^{K+1}(s, a) < i_1 + i_2$. Therefore the triple $(s, a, h)$ satisfies that $\mathbb{P}_{s,h}^\star > 0$. By Lemma L.5, for any $s' \in \mathcal{S}$ and $h' \in [H]$ such that $\mathbb{P}_{s',h'}^\star > 0$, it holds that

$$N_{h'}^{k+1}(s', \pi_{h'}^\star(s')) \geq \left(1 + \frac{1}{6H(H+1)}\right) N_{h'}^k(s', \pi_{h'}^\star(s')),$$

Therefore, the maximum number of Type-IV triggers, denoted $t_4$, satisfies

$$\left(1 + \frac{1}{6H(H+1)}\right)^{t_4} \leq \frac{\hat{T}}{i_1 + i_2} \leq \frac{T}{HSA}.$$

Then the number of rounds with Type-III triggers is bounded by

$$t_4 \leq \frac{\log(\frac{T}{HSA})}{\log\left(1 + \frac{1}{6H(H+1)}\right)} = O\left(H^2 \log\left(\frac{T}{HSA}\right)\right).$$

Combining these four cases, the number of rounds is no more than

$$\sum_{s,a,h} t_1(s, a, h) + \sum_{s,a,h} t_2(s, a, h) + \sum_{s, \mathbb{P}_{s,h}^\star > 0} t_3(s, \pi_h^\star(s), h) + t_4$$

$$\leq O\left(MH^3 SA \log(i_1) + H^3 SA \log\left(\frac{32HC_{\min}}{i_1}\right) + H^3 S \log(i_2) + H^2 \log\left(\frac{T}{HSA}\right)\right)$$

$$\leq O\left(MH^3 SA \log(MH\iota) + H^3 SA \log\left(\frac{H^4 SA}{\beta \Delta_{\min}^2}\right) + H^3 S \log\left(\frac{1}{C_{st}}\right) + H^2 \log\left(\frac{T}{HSA}\right)\right).$$

When $M = 1$, the number of communication rounds is an upper bound for switching cost.

