# OpenReview forum: "Regret-Optimal Q-Learning with Low Cost for Single-Agent and Federated Reinforcement Learning"
_NeurIPS.cc/2025/Conference — NeurIPS 2025 poster_

### Official Review · Reviewer_wTVm · 2025-06-08

**Clarity:** 3
**Significance:** 3
**Originality:** 2
**Rating:** 5
**Confidence:** 4

**Summary:**

This paper addresses the challenge of designing model-free reinforcement learning (RL) algorithms that are both sample-efficient and communication-efficient. In both single-agent and federated RL settings, existing methods either require large burn-in costs or incur frequent policy switching or communication. To mitigate this issue, this paper proposes two new algorithms: Q-EarlySettled-LowCost for single-agent RL and FedQ-EarlySettled-LowCost for federated RL. These are the first model-free methods to simultaneously achieve near-optimal regret, burn-in costs that scale linearly with the number of states and actions, and only logarithmic switching or communication costs.

The key idea seems to be the combination of a round-based learning structure with the classic LCB techniques, enabling infrequent updates without sacrificing learning efficiency. The algorithms outperform existing state-of-the-art methods by reducing the switching or communication overhead while maintaining low regret. In the theoretical aspect, the paper gives strong gap-dependent performance guarantees. In terms of technical novelty, the paper introduces new analytical tools to handle technical challenges in regret analysis. Overall, the paper provides practical and theoretically reliable solutions for efficient RL in settings where data collection or communication is costly.

**Questions:**

Do you think this result can be extend to general cases with continuous state and action space (such as MDP with function approximation)?

**Ethical Concerns:**

["NO or VERY MINOR ethics concerns only"]

**Final Justification:**

I thank the authors for their initial rebuttal and follow-ups.

1. The authors have addressed my concern by  incorporated suggested refinements and discussed extensions to the more general setting of continuous state and action space.

2. Regarding the points raised by other reviewers, I think the authors have done a good job at (i) clarifying the comparison with existing algorithm, (ii) emphasis on the technical novelty of the paper.

Given these information, I think this is a good paper and thus I maintain my score at 5 and recommend for acceptance.

Good luck!

**Limitations:**

yes

**Paper Formatting Concerns:**

No concern.

**Quality:**

4

**Strengths And Weaknesses:**

**Quality**

Strengths:

1, Theoretically Rigorous: The paper provides tight worst-case and gap-dependent regret bounds for both single-agent and federated reinforcement learning (FRL), matching or improving over existing results. The analysis incorporates nontrivial techniques (e.g., surrogate reference functions, LCB-based early settlement). In terms of technical novelty, the analysis successfully handles double non-adaptiveness, a nontrivial technical barrier in FRL.

2, Algorithmic Soundness: The proposed algorithms—Q-EarlySettled-LowCost and FedQ-EarlySettled-LowCost—are well-motivated and robustly designed. Their use of refined bonuses and structured policy updates yields practical and theoretical benefits. The numerical experiments confirm the regret and cost improvements over strong baselines, including UCB-Advantage and FedQ-Advantage.

**Clarity**

Strengths:

1, I think this paper has good clarity in general. The problem is clearly defined. The similarity and difference from existing works in federated RL is well described in detail. Technical novelty is explicitly mentioned.
I also really appreciate the illustration in Appendix D, which makes it easier for the reader to understand the federated RL framework which is quite complicated to be honest.
Other good practices include: Table 1 and 2 provide a clear comparison with existing methods.

The only issue is that the notation is a bit heavy (for example, page 6). But I think this is just a minor issue and given the theoretical nature and complexity of the federated RL problem setting, I do not think there is a significantly better way to present the theoretical results.

**Significance**

Strengths:

1, Addresses a Key Challenge in FRL: Communication and synchronization costs are one of the key problems in Federated RL. The proposed algorithms directly address these issues with provable logarithmic cost.


**Originality**

1, I think the major novelty of the paper is its technical improvement. Specifically, I think the successful adaptation of the existing surrogate reference functions to worst-case regret framework is one of the major originalities. This is an advancement in the field of statistical methodology, which can potentially have broader applicability in not only federated learning but other problem settings as well.

**Minor Weaknesses**
On the empirical side, only very simple numerical experiments have been conducted.
Notation is a bit heavy.
A lack of proof sketch in the main text (there is, though, in Appendix K).

---

> ### Author Rebuttal · Authors · 2025-07-31
>
> We thank you for your careful reading and thoughtful comments. Below are our point-by-point responses to your questions, and we hope our responses address your concerns.
>
> >### **Weakness 1: Simple numerical experiments**
>
> We have conducted additional experiments to demonstrate the reduced communication cost and low burn-in cost compared to other baselines.
>
> **Additional Experiment 1: Comparison of switching cost with UCB-Advantage [1] and the number of communication rounds with FedQ-Advantage [2]**
>
> To evaluate the switching/communication cost of different algorithms, we plot the average regret per episode, defined as
> $\text{Regret}(T)/(MT)$, against the policy switching cost in single-agent RL or the number of rounds in FRL.
>
> Specifically, to show our results, we choose a set of average regret thresholds $\\{0.5,0.3,0.1,0.05,0.03,0.01\\}.$ For each algorithm, we conduct 10 independent trials and report the median policy switching cost for single-agent RL or median number of communication rounds for FRL required for average regret to fall below each threshold. All evaluations are conducted in the two synthetic environments described in Appendix B, denoted by Settings I-II for single-agent RL and III-IV for FRL. The table below summarizes the results (rounded to the nearest integer), where “None” indicates that the algorithm did not reach the target average regret within the allotted number of episodes.
>
> **Table 1: Median policy switching cost (settiing I and II) or number of rounds (settiing III and IV) required to reach each average regret.**
> |Setting|Algorithm|0.5|0.3|0.1|0.05|0.03|0.01|
> |-|-|-|-|-|-|-|-|
> |I|UCB-Advantage|50|243|608|734|801|902|
> | |Ours|32|149|471|576|633|702|
> |II|UCB-Advantage|10345|13954|15773|17131|None|None|
> | |Ours|8767|11650|12711|13300|13661|None|
> |III|FedQ-Advantage|1063|1230|1608|1704|1768|1867|
> | |Ours|919|1000|1105|1152|1181|1229|
> |IV|FedQ-Advantage|13986|17059|20586|21908|22603|23395|
> | |Ours|10516|12316|14231|15098|15566|16357|
>
> These results demonstrate that, to achieve any given average regret, our algorithm incurs a lower policy switching cost than UCB-Advantage in single-agent RL, and requires fewer communication rounds than FedQ-Advantage in FRL.
>
> **Additional Experiment 2: Burn-in cost comparison**
>
> We also conducted additional experiments for burn-in cost under the two synthetic environments described in Appendix B, denoted by Settings I-II for single-agent RL and III-IV for FRL. To evaluate the burn-in cost of different algorithms, we plot the normalized regret curves
> $$\frac{\text{regret}(t)/\log(t+1)}{\text{regret}(N)/\log(N+2)},$$
> where $N = T/H$ denotes the total number of episodes per agent. By examining these curves, we observe that, our algorithm consistently requires the smaller number of episodes than all baseline methods to achieve the same normalized regret level.
>
> To show the results, we select a set of normalized regret thresholds $\\{0.5,0.8,0.85,0.9,0.95,0.99\\}.$ For each algorithm, we conduct 10 independent trials and report the median number of episodes per agent needed to reach or exceed each threshold. As shown in the following table, our method achieves lower burn-in cost and superior sample efficiency.
>
> **Table 2: Median number of episodes per agent required to reach each normalized regret.**
> |Setting|Algorithm|0.5|0.8|0.85|0.9|0.95|0.99|
> |-|-|-|-|-|-|-|-|
> |I|UCB-Advantage|3467|14459|20045|28541|44776|89882|
> | |Q-EarlySettled-Advantage|4130|20649|28355|43044|72082|139783|
> | |Ours|1453|7241|9821|13687|23122|44861|
> |II|UCB-Advantage|47905|395867|574937|848694|1285514|1628711|
> | |Q-EarlySettled-Advantage|93015|587867|786596|1063962|1478185|1884091|
> | |Ours|38335|131755|181790|272172|505089|841384|
> |III|FedQ-Advantage|420|2266|3074|3646|5786|10078|
> | |Ours|151|837|1099|1551|2336|3770|
> |IV|FedQ-Advantage|13694|64988|85668|112999|149730|229123|
> | |Ours|4389|18868|26872|41243|74978|132901|
>
>
> >### **Weakness 2: Notation is a bit heavy**
>
> Thank you for the suggestion. To improve clarity and ease of reference, we have included two notation summary tables for our algorithms in Appendix D.2 of the original submission.
>
> >### **Weakness 3: A lack of proof sketch in the main text**
>
> Thank you for the suggestion. Due to space constraints, we did not include the proof sketch in the main text of the original submission. However, we will incorporate a key proof sketch into the main text using the additional page provided for the revised manuscript.
>
> >### **Question: Extension to general cases with continuous state and action space**
>
> Thank you for the insightful question. Extending FedQ-EarlySettled-LowCost to non-tabular settings, such as those involving continuous state/action spaces or linear function approximation, is a promising but nontrivial direction for future work.
>
> Recent works have explored related ideas in the offline RL setting. For example, [3] proposes a federated offline RL algorithm using confidence-bound methods, while [4, 5] apply reference-advantage decomposition techniques in offline RL with linear function approximation. These works demonstrate the feasibility of using confidence bounds and reference-advantage decomposition techniques in non-tabular environments.
>
> However, adapting our event-triggered termination mechanism to non-tabular settings presents significant challenges. For environments with continuous state spaces, it becomes difficult to quantify visitation counts, the design of stopping conditions for exploration becomes ambiguous, and policy updates based on event-triggered signals must be carefully coordinated across agents become complex, especially under communication cost constraints. These challenges in the online RL and online FRL settings require new algorithmic and theoretical tools. We agree with you and view this as an important direction for future work.
>
> [1] Zihan Zhang, Yuan Zhou and Xiangyang Ji. Almost optimal model-free reinforcement learning via reference-advantage decomposition. NeurIPS 2020.
>
> [2] Zhong Zheng, Haochen Zhang and Lingzhou Xue. Federated Q-learning with reference advantage decomposition: Almost optimal regret and logarithmic communication cost. ICLR 2025.
>
> [3] Doudou Zhang, Yufeng Zhang, Aaron Sonabend-W, Zhaoran Wang, Junwei Lu and Tianxi Cai. Federated offline reinforcement learning. Journal of the American Statistical Association, 2024.
>
> [4] Wei Xiong, Han Zhong, Chengshuai Shi, Cong Shen, Liwei Wang and Tong Zhang. Nearly Minimax Optimal Offline Reinforcement Learning with Linear Function Approximation: Single-Agent MDP and Markov Game. ICLR, 2023.
>
> [5] Qiwei Di, Heyang Zhao, Jiafan He and Quanquan Gu. Pessimistic Nonlinear Least-Squares Value Iteration for Offline Reinforcement Learning. ICLR, 2024

---

> ### Comment · Reviewer_wTVm · 2025-08-04
> **Re rebuttal**
>
> I thank the authors for their effort in writing the detailed rebuttal. They addressed my questions well. I maintain my original scoring  for 'accept'.

---

### Official Review · Reviewer_Rdno · 2025-06-26

**Clarity:** 1
**Significance:** 2
**Originality:** 2
**Rating:** 5
**Confidence:** 3

**Summary:**

This paper aims to simultaneously reduce the **burn-in costs** and **policy switching costs** (for the single-agent RL setting)/**communication costs** (for the federated RL setting). The proposed algorithm improves over [1] by a logarithmic factor in the regret bound and a polynomial factor in the burn-in costs.

[1] Zheng, Z., Zhang, H., & Xue, L. (2025). Federated Q-learning with reference-advantage decomposition: almost optimal regret and logarithmic communication cost.

**Questions:**

1. I think the notation $\Omega(\cdot)$, instead of $O(\cdot)$, should be used in the lower bounds in Lines 24 and 50.
2. In the numerical experiments, how do the proposed algorithms compare to FedQ-Advantage regarding communication costs?
3. Can you explain how FedQ-EarlySettled-Advantage reduces the burn-in costs in the FedQ-Advantage algorithm [1]?
4. I think visualizing the burn-in costs in Figures 1 and 3 also helps highlight the improvements over the previous algorithms, for example, comparing the number of iterations where those curves "settle down".

**Ethical Concerns:**

["NO or VERY MINOR ethics concerns only"]

**Final Justification:**

After reading the authors' response, I appreciate their efforts to address my problems. I maintain my recommendation for acceptance as this paper provides a comprehensive analysis of federated RL. On the other hand, I think adding the table and the numerical results on the burn-in cost can improve the readability.

**Limitations:**

yes

**Quality:**

2

**Strengths And Weaknesses:**

* **Strengths:**
    1. The improvements in the burn-in costs are considerable (i.e., polynomial to $S,A,$ and $H$), which could be helpful in the applications;
    2. This paper presents a comprehensive regret analysis of the proposed algorithms and derives both the high probability and gap-dependent regret bounds.

* **Weaknesses:**
    1. The presentation is a bit unclear. For example, I think it would be better to put the exact bounds in Tables 1 and 2 to make comparisons between different algorithms. On the other hand, in the Numerical Experiments Section in Appendix B, the algorithm does not compare to [1] in terms of switching/communication costs;
    2. The techniques utilized in this paper, including the reference-advantage decomposition and the combination of UCB and LCB methods, have been discussed in several references [1, 2], and their combinations and applications to a federated setup seem to be rather direct.

[1] Zheng, Z., Zhang, H., & Xue, L. (2025). Federated Q-learning with reference-advantage decomposition: almost optimal regret and logarithmic communication cost.

[2] Li, G., Shi, L., Chen, Y., Gu, Y., & Chi, Y. (2021). Breaking the sample complexity barrier to regret-optimal model-free reinforcement learning.

---

> ### Author Rebuttal · Authors · 2025-07-31
>
> We thank you for your careful reading and thoughtful comments. Below are our point-by-point responses, and we hope our responses address your concerns.
>
> >### **Weakness 1 part 1: Exact bounds in Tables**
>
> We follow this helpful suggestion and provide updated tables with explicit bounds to facilitate clearer comparisons.
>
> We first define the following notations: $\iota=\log(SAT /p)$, where $p\in (0,1)$ is the failure probability; $C_g=SA\log(T)/\Delta_{\min}$; $\mathbb{Q}^\star:=\max_{s,a,h}\{\mathbb{V}\_{s,a,h}(V_{h+1}^\star)\}\in[0,H^2]$; $\beta\in (0,H]$ is an arbitrarily predefined hyper-parameter; and $O^\star$ only captures the main $\log T$ term. The following tables show that our algorithm achieves state-of-the-art performance in both regret and switching/communication costs (Our detailed switching/communication cost results can be found in Theorems 4.6 and 4.8).
>
> **Table 1: Comparison of model-free online single-agent RL algorithms.**
> |Algorithm|Worst-case regret|Burn-in cost|Gap-dependent regret|Switching cost|
> |-|-|-|-|-|
> |UCB-Hoeffding|$O(\sqrt{H^4SAT\iota})$|$\infty$|$O(H^6C_g)$|$O(T/H)$|
> |UCB-Bernstein|$O(\sqrt{H^3SAT\iota}+\sqrt{H^9S^3A^3\iota^4})$|$\infty$|❌|$O(T/H)$|
> |UCB2-Hoeffding|$O(\sqrt{H^4SAT\iota})$|$\infty$|❌|$O(H^3SA\log T)$|
> |UCB2-Bernstein|$O(\sqrt{H^3SAT\iota}+\sqrt{H^9S^3A^3\iota^4})$|$\infty$|❌|$O(H^3SA\log T)$|
> |UCB-Advantage|$O(\sqrt{H^2SAT\iota^3}+H^8S^2A^{\frac{3}{2}}T^{\frac{1}{4}})$|$[S^6A^4H^{28},\infty)$|$O^\star((\mathbb{Q}^\star+\beta^2H)H^3C_g)$|$O^\star(H^2S\log T)$|
> |Q-EarlySettled-Advantage|$O(\sqrt{H^2SAT\iota^4}+H^6SA\iota^3)$|$[SAH^{10},\infty)$|$O^\star((\mathbb{Q}^\star+\beta^2H)H^3C_g)$|$O(T/H)$|
> |Ours|$O(\sqrt{H^2SAT\iota^2}+H^6SA\iota^2)$|$[SAH^{10},\infty)$|$O^\star((\mathbb{Q}^\star+\beta^2H)H^3C_g)$|$O^\star(H^2\log T)$|
>
>
> **Table 2: Comparison of model-free online FRL algorithms.**
> |Algorithm|Worst-case regret|Burn-in cost|Gap-dependent regret|Communication cost|
> |-|-|-|-|-|
> |FedQ-Hoeffding|$O(\sqrt{MH^4SAT\iota}+MH^4SA\iota^2)$|$\infty$|$O^\star(H^6C_g)$|$O^\star(H^2\log T)$|
> |FedQ-Bernstein|$O(\sqrt{MH^3SAT\iota^2}+M\sqrt{H^9S^3A^3}\iota^2)$|$\infty$|$O^\star(H^6C_g)$|$O^\star(H^2\log T)$|
> |FedQ-Advantage|$O(\sqrt{MH^2SAT\iota^3}+MH^{7}S^{2}A^{\frac{3}{2}}\iota^2)$|$[MS^3A^2H^{12},\infty)$|❌|$O(MH^2SA\log T)$|
> |Ours|$O(\sqrt{MH^2SAT\iota^2}+MH^{6}SA\iota^2)$|$[MSAH^{10},\infty)$|$O^\star((\mathbb{Q}^\star+\beta^2H)H^3C_g)$|$O^\star(H^2\log T)$|
>
> >### **Weakness 1 part 2 and Question 2:  Additional comparison with FedQ-Advantage**
>
> Theoretically, for sufficiently large $T$, our algorithm achieves a switching/communication cost of $O(H^2\log T)$, which improves upon the $O(H^2S\log T)$ cost of UCB-Advantage [106] and the $O(MH^2SA\log T)$ cost of FedQ-Advantage [109].
>
> Empirically, to evaluate the switching/communication cost of different algorithms, we plot the average regret per episode, defined as
> $\text{Regret}(T)/(MT)$, against the policy switching cost in single-agent RL or the number of rounds in FRL.
>
> Specifically, to show our results, we choose a set of average regret thresholds $\\{0.5,0.3,0.1,0.05,0.03,0.01\\}.$ For each algorithm, we conduct 10 independent trials and report the median policy switching cost for single-agent RL or median number of communication rounds for FRL required for average regret to fall below each threshold. All evaluations are conducted in the two synthetic environments described in Appendix B, denoted by Settings I-II for single-agent RL and III-IV for FRL. The table below summarizes the results (rounded to the nearest integer), where “None” indicates that the algorithm did not reach the target average regret within the allotted number of episodes.
>
> **Table 3: Median policy switching cost (settiing I and II) or number of rounds (settiing III and IV) required to reach each average regret.**
> |Setting|Algorithm|0.5|0.3|0.1|0.05|0.03|0.01|
> |-|-|-|-|-|-|-|-|
> |I|UCB-Advantage|50|243|608|734|801|902|
> | |Ours|32|149|471|576|633|702|
> |II|UCB-Advantage|10345|13954|15773|17131|None|None|
> | |Ours|8767|11650|12711|13300|13661|None|
> |III|FedQ-Advantage|1063|1230|1608|1704|1768|1867|
> | |Ours|919|1000|1105|1152|1181|1229|
> |IV|FedQ-Advantage|13986|17059|20586|21908|22603|23395|
> | |Ours|10516|12316|14231|15098|15566|16357|
>
> These results demonstrate that, to achieve any given average regret, our algorithm incurs a lower policy switching cost than UCB-Advantage in single-agent RL, and requires fewer communication rounds than FedQ-Advantage in FRL.
>
> >### **Weakness 2: Technical novelty**
>
> We appreciate the opportunity to clarify our novelty. Our work does not simply combine prior techniques. Instead, we have developed novel analytical tools that go beyond prior work. Appendix C provides a detailed discussion of the challenges and our corresponding contributions. Our novelty and contribution lie in addressing non-trivial theoretical and algorithmic challenges that arise when combining their respective strengths.
>
> The reference-advantage decomposition, suffers from non-adaptiveness in both weights and reference functions when applied in a round-based setting, especially under communication constraints. We propose a new decomposition that uses **surrogate reference functions**, enabling tighter worst-case regret bounds while avoiding reliance on empirical process techniques that result in an additional factor of $\log T$.
>
> We also introduce a **new two-sided variance bound** to tighten regret under delayed updates in the proof of the optimism property ($Q_h^k\geq Q_h^\star$, Lemma H.1). Specifically, we derive both upper and lower bounds on the variance estimator (Eq. 33, 37, 38), whereas prior works (such as [48]) only use the lower bound. This refinement leads to more accurate confidence intervals in parts (b) and (c) of Lemma G.1 and allows us to improve the bonus term $\beta_h^R$ in the reference-advantage decomposition, reducing the cumulative bonus and yielding both tighter regret and better empirical performance.
>
> Further, we leverage the **Lower confidence bound** to trigger early reference settling. Unlike the single-agent setting, our method adopts a different design in the early and late rounds to integrate the LCB-based strategy with round-based synchronization, ensuring low switching and communication costs. This approach is formalized in Equations (10) and (14), which help bound the reference settling error in Lemma H.4 and contribute to reducing the burn-in cost—a benefit that prior FRL methods (e.g., FedQ-Advantage) cannot achieve.
>
> Finally, in the FRL setting, integrating these techniques involves new challenges in synchronizing learning progress across agents, managing reference settling in a distributed and delayed-update setting, and maintaining both low communication cost and low switching cost.
>
> >### **Question 1: Notation**
>
> Thank you for the advice! We will correct this notation accordingly in the revised manuscript.
>
> >### **Question 3: Key to reduce burn-in cost**
>
> Our work improves upon FedQ-Advantage by addressing key sources of estimation error that contribute to the burn-in cost. Specifically:
>
> **LCB-Based Reference Settling.** Our algorithm introduces an LCB-type estimate $Q_h^{\text{L},k}$ to control the reference settling error more tightly in Lemma H.4. This reduces dependence on $H,S,A$, leading to a lower burn-in cost.
>
> **Two-Sided Inequality for Tighter Bonuses.** As discussed in our response to Weakness 1 part 2, we use a new two-sided inequality to refine the reference-advantage bonus, yielding tighter confidence bounds and more accurate $Q$-estimates, which directly contribute to improved regret and reduced burn-in cost.
>
> **Surrogate Reference Functions.** We further reduce the burn-in by introducing surrogate reference functions, which address the non-adaptiveness of both the weights and the reference functions in the regret decomposition. This innovation improves the tightness of the analysis and avoids the inefficiencies associated with empirical process techniques used in earlier works.
>
> Together, these contributions enable our algorithm to achieve lower burn-in costs while maintaining strong regret and cost guarantees, marking a significant improvement over FedQ-Advantage.
>
> >### **Question 4: Visualizing the burn-in costs**
>
> Thank you for the helpful suggestion. We conducted additional experiments under the two synthetic environments described in Appendix B, denoted by Settings I-II for single-agent RL and III-IV for FRL. To evaluate the burn-in cost of different algorithms, we plot the normalized regret curves
> $$\frac{\text{regret}(t)/\log(t+1)}{\text{regret}(N)/\log(N+2)},$$where $N = T/H$ denotes the total number of episodes per agent. By examining these curves, we observe that, our algorithm consistently requires the smaller number of episodes than all baseline methods to achieve the same normalized regret level.
>
> To show the results, we select a set of normalized regret thresholds $\\{0.5,0.8,0.85,0.9,0.95,0.99\\}.$ For each algorithm, we conduct 10 independent trials and report the median number of episodes per agent needed to reach or exceed each threshold. As shown in the following table, our method achieves lower burn-in cost and superior sample efficiency.
>
> **Table 4: Median number of episodes per agent required to reach each normalized regret.**
> |Setting|Algorithm|0.5|0.8|0.85|0.9|0.95|0.99|
> |-|-|-|-|-|-|-|-|
> |I|UCB-Advantage|3467|14459|20045|28541|44776|89882|
> | |Q-EarlySettled-Advantage|4130|20649|28355|43044|72082|139783|
> | |Ours|1453|7241|9821|13687|23122|44861|
> |II|UCB-Advantage|47905|395867|574937|848694|1285514|1628711|
> | |Q-EarlySettled-Advantage|93015|587867|786596|1063962|1478185|1884091|
> | |Ours|38335|131755|181790|272172|505089|841384|
> |III|FedQ-Advantage|420|2266|3074|3646|5786|10078|
> | |Ours|151|837|1099|1551|2336|3770|
> |IV|FedQ-Advantage|13694|64988|85668|112999|149730|229123|
> | |Ours|4389|18868|26872|41243|74978|132901|
>
> (All reference numbers are from our paper.)

---

> > ### Comment · Reviewer_Rdno · 2025-08-04
> >
> > Thank you for your response. Your response has addressed my concern. I have increased my rating accordingly.

---

### Official Review · Reviewer_a3AW · 2025-06-26

**Clarity:** 2
**Significance:** 2
**Originality:** 2
**Rating:** 3
**Confidence:** 4

**Summary:**

This paper proposes two novel model-free reinforcement learning algorithms, Q-EarlySettled-LowCost for single-agent RL and FedQ-EarlySettled-LowCost for federated RL (FRL). The algorithms achieve near-optimal regret bounds while simultaneously ensuring low burn-in costs (i.e., the number of samples required to reach near-optimality) and logarithmic policy switching or communication costs. The design combines upper confidence bound (UCB) techniques with a reference-advantage decomposition and incorporates lower confidence bound techniques with an early-settling mechanism for reference functions in estimation. Meanwhile, the so-called ``Event-Triggered Termination of Exploration'' technique is also used in the proposed algorithm. Theoretical guarantees are provided in both worst-case and gap-dependent settings. Numerical results demonstrate superior regret performance compared to existing model-free baselines.

**Questions:**

The author(s) may wish to more clearly explain the technical challenges and their technical contribution --- especially which analysis, besides Lemma I.3, is new.

**Ethical Concerns:**

["NO or VERY MINOR ethics concerns only"]

**Final Justification:**

I thank the author(s) for their considerable efforts in preparing the paper and for addressing the questions during the rebuttal period. However, I find that both the problem studied and the proofs are essentially direct combinations of existing works. Furthermore, several parts of the analysis appear incremental rather than representing a substantial technical contribution. Therefore, I maintain my score.

**Limitations:**

Yes.

**Paper Formatting Concerns:**

None.

**Quality:**

2

**Strengths And Weaknesses:**

Strengths

+ The paper proposes the first algorithm that simultaneously achieve near-optimal regret, linear burn-in cost in $S$ and $A$, and logarithmic switching/communication cost under both single agent RL or FRL setting.

+ The authors provide improved gap-dependent regret bounds and establish the first gap-dependent switching cost guarantee for LCB-based algorithms.

+ Theoretical results are supported by experiments that verify the improved gap-dependent performance in practice.

+ The algorithm is well-engineered: by carefully tuning the bonus terms, it smoothly integrates existing methods such as Q-Early Settled-Advantage and Event-Triggered Termination of Exploration.



Weaknesses

- The technical contribution seems incremental. The algorithm essentially combines multiple components from prior work under the same setting. Key techniques such as Reference-Advantage Decomposition, LCB-based Early Settlement, and Event-Triggered Termination have all appeared in previous studies. Furthermore, the main analytical challenge, namely the non-adaptiveness of weights and reference functions, is addressed through direct application of surrogate function constructions from earlier studies, rather than by introducing novel analytical tools. While the author(s) have clearly invested substantial effort in assembling these components into a comprehensive (over eighty-page) manuscript, it is debatable whether such a narrowly scoped variation of the tabular RL problem warrants this level of complexity, particularly given the method’s modest empirical performance, as discussed below.

- Empirical performance seems not very practically useful: under $(H, S, A) = (7, 10, 5)$, the regret and switching cost grow with $\ln T$ at a scale of roughly 1000, indicating the constants hidden in the theory are large and the algorithm may not be practical for moderate horizons.

---

> ### Author Rebuttal · Authors · 2025-07-31
>
> We appreciate your careful reading and thoughtful comments. Below are our point-by-point responses, and we hope our responses address your concerns.
>
> >### **Weakness 1 and Question: Technical contributions**
>
> We respectfully disagree with the view that our method simply combines prior techniques. Instead, we have developed novel analytical tools that go beyond prior work. Appendix C provides a detailed discussion of the challenges and our corresponding contributions. Below, we further clarify why our contributions go beyond incremental advances.
>
> **Challenge 1: Non-adaptiveness of weights and reference functions**
>
> Previous works only addressed isolated aspects of non-adaptiveness: [1] applies round-wise approximation for non-adaptive weights, and [2] uses the empirical process for settled reference functions. However, **no prior work simultaneously resolves both forms of non-adaptiveness**, and their direct combination introduces substantial technical difficulties. Notably, the empirical process in [2] incurs an extra $\log T$ factor due to $\epsilon$-net construction over non-adaptive reference function classes. When combined with round-wise approximations, the existence of nonadaptive weights can increase the effective dimension of the $\epsilon$-nets, leading to looser regret bounds. It would require a new proof idea to solve this non-trivial challenge.
>
> **Contribution 1: Unified treatment via surrogate reference functions**
>
> We address both forms of non-adaptiveness by integrating the **surrogate reference function** technique from [3], originally developed for gap-dependent analysis, into our **worst-case regret framework**. This enables a new decomposition that improves both theoretical bounds and generality. Specifically, our framework can be applied to existing reference-advantage-based RL methods [2, 4, 5], tightening their regret bounds. In model-free RL or FRL, worst-case regret is typically bounded by relating regret to the estimation error of the optimal $Q$-function, that is $\sum_{k,j,m}(Q_h^k-Q_h^\star)(s_h^{k,j,m},a_h^{k,j,m}).$ Prior methods upper-bound each term via $Q_h^{\text{R}}-Q_h^\star$ and then bound the sum recursively over $h$. Specifically, based on the reference-advantage update rules (Eq. 11, 15), with high probability, $Q_h^{\text{R},k}-Q_h^\star\leq\mathcal{G}_0$, where $\mathcal{G}_0$ is defined in Appendix C.
>
> Following the structure of the reference-advantage decomposition, and using the **surrogate reference function** $\hat{V}\_h^{\text{R},k}$, $\mathcal{G}\_0$ can be upper bounded by the summations with respect to $C_B,\hat{\mathcal{G}}\_1,\hat{\mathcal{G}}\_2,\hat{\mathcal{G}}\_3,\hat{\mathcal{G}}\_4$, where $C_B$ collects some constants and the cumulative bonuses, $\hat{\mathcal{G}}\_1$ reflects the estimation error $V_{h+1}^{k}-V_{h+1}^\star$, $\hat{\mathcal{G}}\_3$ reflects the reference settling error $V_{h+1}^{\text{R},k}-\hat{V}\_{h+1}^{\text{R},k}$, and $\hat{\mathcal{G}}\_2,\hat{\mathcal{G}}\_4$ reflect the empirical estimation error for the settled advantage function $V_{h+1}^\star-\hat{V}\_{h+1}^{\text{R},k}$ and surrogate reference function $\hat{V}\_{h+1}^{\text{R},k}$.
>
> In contrast to the original decomposition from [2], where non-adaptive weights and non-adaptive reference functions together result in looser regret bounds, our surrogate reference function is **adaptive**, mitigating both issues. The adaptive design ensures that non-adaptive weights, which reflect delayed policy switching, is incorporated without introducing additional error via empirical process tools. As a result, we achieve a $\log(SAT/p)$ improvement in the regret bound over Q-EarlySettled-Advantage [2], marking the **first application of surrogate reference functions in worst-case RL** and significantly expanding the analytical tools for RL that overcome limitations in [2, 4, 5].
>
> **Challenge 2: Balancing low switching cost and regret**
>
> In online RL, achieving a logarithmic policy switching cost typically relies on delayed policy updates, as in [6, 7]. However, delayed policy updates introduce extra regret, since the same policy is reused across episodes before any state-action-step triple reaches a predefined threshold. This trade-off explains why, the low-switching-cost variants of UCB-Hoeffding and UCB-Bernstein ([6]) underperform their standard counterparts, as shown in Figure 1. How to achieve better regret at a low switching cost with delayed policy updates remains a significant challenge.
>
> **Contribution 2: Two-sided confidence bounds for improved bonus design**
>
> We introduce a **new two-sided variance bound** to tighten regret under delayed updates in the proof of the optimism property ($Q_h^k\geq Q_h^\star$, Lemma H.1). Specifically, we derive both upper and lower bounds on the variance estimator (Eq. 33, 37, 38), whereas prior works such as [2] only use the lower bound. This refinement leads to more accurate confidence intervals in parts (b) and (c) of Lemma G.1 and allows us to improve the bonus term $\beta_h^R$ in the reference-advantage decomposition, reducing the cumulative bonus $C_B$ and yielding both tighter regret and better empirical performance.
>
> Additionally, to integrate this with a round-based design, we adapt the variance estimator by using one estimator in early rounds and another in later rounds, as detailed in Eq. 8 and 12. This adaptive approach ensures consistent accuracy of the bonus term across different phases of learning.
>
> **Challenge 3: LCB with round-nased design**
>
> FedQ-Advantage [5] settles reference functions only after a state-step pair has been visited sufficiently often, which introduces a polynomial dependence on $S,A$ in the bound for $V_h^{\text{R},K}-V_h^K$, resulting in high burn-in cost. This dependence limits the practicality in large-scale environments.
>
> **Contribution 3: LCB-triggered reference settling for reduced burn-in cost**
>
> We are the first in the literature to apply the LCB technique in FRL. In our design, $V_h^{k}$ and LCB-estimates $V_h^{\text{L},k}$ serves as upper and lower bounds for $V_h^\star$. The reference function is settled once $V_h^k-V_h^{\text{L},k}\leq\beta$, ensuring $V_h^\star\in[V_h^{\text{L},k},V_h^k]$. This allows us to settle the reference earlier and bound the settling error $\hat{\mathcal{G}}_4$ more tightly, reducing dependence on $H,S,A$.
>
> To accommodate the round-based design of FRL, we modify the standard LCB strategy used in single-agent RL: early rounds adopt the same design as in the single-agent setting, while late rounds use an equal-weight estimator (Eq. 14), balancing accuracy with communication efficiency.
>
> >### **Weakness 2: Empirical performance**
>
> Notably, the $HSA$ term in the coefficient for both regret and switching/communication cost is **unavoidable**. In fact, [8] provides a lower bound of $\tilde{\Omega}(HSA/\Delta_{\min})$ for single-agent RL, demonstrating that any algorithm must incur regret with a coefficient that scales linearly with $HSA$. Similarly, [7] proves a lower bound $\tilde{\Omega}(HSA)$ on switching cost for any algorithm with $\tilde{O}(\sqrt{T})$ regret, highlighting that the coefficient of switching/communication costs must also grow linearly with $HSA$.
>
> Importantly, model-free RL algorithms with larger theoretical regret constants and $O(T)$ switching/communication cost are already used in practice. For example, in single-agent RL, [9] proposes Action-Robust Q-learning based on UCB-Hoeffding, which has a regret bound of $\tilde{O}(\sqrt{H^4SAT})$, corresponding to a coefficient larger than 10,000 in our setting (see Figure 1, UCB-Hoeffding), and incurs a switching cost scales as $O(T)$. Nevertheless, the algorithm performs well in Cliff Walking, a text-based environment with $(H,S,A)=(100,48,4)$, and Inverted Pendulum, a control task simulated in MuJoCo with $H=100$. Similarly, [10] extends UCB-Hoeffding to a multi-agent setting for cooperative navigation with $10^4$ states. Despite a regret bound of $O(H^6SA\log(T)/\Delta_{\min})$ with a coefficient over 10,000 (see Figure 3, FedQ-Hoeffding) and a communication cost scales as $O(MHSAT)$ under full communication, the algorithm still achieves strong empirical results.
>
> Therefore, large constants (especially those polynomial in $H,S,A$) do not preclude practical effectiveness. In comparison, our algorithm achieves smaller coefficient than other baselines (Figures 1–4), demonstrating practical utility even in moderate-horizon settings.
>
> [1] Zhong Zheng, Fengyu Gao, Lingzhou Xue and Jing Yang. Federated Q-learning: Linear regret speedup with low communication cost. ICLR 2024.
>
> [2] Gen Li, Laixi Shi, Yuxin Chen and Yuejie Chi. Breaking the sample complexity barrier to regret-optimal model-free reinforcement learning. Information and Inference, 2023.
>
> [3] Zhong Zheng, Haochen Zhang and Lingzhou Xue. Gap-dependent bounds for Q-learning using reference-advantage decomposition. ICLR 2025.
>
> [4] Zihan Zhang, Yuan Zhou and Xiangyang Ji. Almost optimal model-free reinforcement learning via reference-advantage decomposition. NeurIPS 2020.
>
> [5] Zhong Zheng, Haochen Zhang and Lingzhou Xue. Federated Q-learning with reference advantage decomposition: Almost optimal regret and logarithmic communication cost. ICLR 2025.
>
> [6] Yu Bai, Tengyang Xie, Nan Jiang and Yu-Xiang Wang. Provably efficient Q-learning with low switching cost. NeurIPS 2019.
>
> [7] Dan Qiao, Ming Yin, Ming Min and Yu-Xiang Wang. Sample-efficient reinforcement learning with loglog(t) switching cost. ICML 2022.
>
> [8] Haike Xu, Tengyu Ma and Simon Du. Fine-grained gap-dependent bounds for tabular mdps via adaptive multi-step bootstrap. ICML 2021.
>
> [9] Guanlin Liu, Zhihan Zhou, Han Liu and Lifeng Lai. Efficient Action Robust Reinforcement Learning with Probabilistic Policy Execution Uncertainty. TMLR 2024.
>
> [10] Justin Lidard, Udari Madhushani and Naomi Leonard. Provably efficient multi-agent reinforcement learning with fully decentralized communication. ACC 2022.

---

> > ### Comment · Reviewer_a3AW · 2025-08-05
> >
> > Thank you for your response. However, I respectfully disagree with the arguments presented.
> >
> > Regarding the practical value of the paper, I find the justification based on citing prior work with similar experimental settings to be insufficient. The experiments are conducted in a relatively small state/action space, which raises concerns about the scalability of the proposed algorithm. Moreover, it is unclear in what real-world scenarios Federated Learning would be necessary or appropriate for such a small-scale problem setting.
> >
> > From a theoretical perspective, I certainly appreciate the value of foundational research. However, I find the theoretical contribution of the current paper to be somewhat incremental. While your response reiterates many of the points already discussed in Appendix C, the core problem formulation, model-free learning with low switching/communication costs and low burning cost, appears to be a rather artificial synthesis of several well-studied components in theoretical RL. As such, the scope of the problem seems narrow and constructed. In line with another reviewer's observation, the proposed solution largely appears to be a composition of existing techniques. Although I acknowledge the technical effort required to integrate these elements into a coherent proof, I do not see significant methodological innovation or conceptual breakthroughs that would meet the bar of a top-tier venue like NeurIPS, particularly given that the optimal communication cost for this setting remains unresolved.

---

> > > ### Author Response · Authors · 2025-08-07
> > > **Follow-Up on Reviewer a3AW's Response to our Rebuttal (Part I)**
> > >
> > > Thank you for your response and for sharing your perspective. While we appreciate your feedback, we respectfully disagree with several points raised and would like to clarify both the motivation and the contributions of our work.
> > >
> > > **To summarize, our algorithms achieve state-of-the-art performance in terms of regret and switching cost in the single-agent model-free RL setting, as well as in terms of regret and communication cost in the model-free FRL setting, from both theoretical and empirical perspectives.** Below, we address your concerns by clarifying the motivation and technical contributions point by point. Our response is structured in four parts.
> > >
> > > In the first part, we aim to clarify the **practical value and relevance** of our research problem.
> > >
> > > Our problem formulation is not an artificial synthesis but is motivated by real-world settings where **both data collection and policy deployment are costly**, such as autonomous driving [1] and clinical trials [2]. These practical concerns motivate the **joint consideration of burn-in cost and switching cost** in online RL, and these two challenges have rarely been studied together. Prior work typically addresses only one: e.g., [3,4] focus on switching cost, while [5] considers burn-in cost. To our knowledge, our work is the first to address both in a unified, model-free RL framework.
> > >
> > > Furthermore, we extend our algorithm to online FRL, where communication efficiency is essential. Our method achieves logarithmic communication complexity, which is especially relevant for privacy-aware or resource-constrained applications (e.g., healthcare systems [6,7], finance [8], IoT networks [9]).
> > >
> > > In summary, our work addresses a practically motivated and theoretically novel direction in online RL and FRL that has not been explored in prior work.

---

> ### Author Response · Authors · 2025-08-05
> **Summary and Open to Further Discussion**
>
> Dear Reviewer a3AW,
>
> Thank you again for your thoughtful review. In our rebuttal, we have further clarified our technical contributions and provided additional explanation regarding the practical performance of our algorithm. We hope our responses have addressed your questions and concerns. Please feel free to let us know if there is anything else we can clarify. We would be happy to provide further information.
>
> Thank you for your time and consideration.

---

> ### Author Response · Authors · 2025-08-07
> **Follow-Up on Reviewer a3AW's Response to our Rebuttal (Part II)**
>
> Thank you once again for your response and for sharing your insights. In the second part of our response, we would like to further clarify the **contributions and technical novelties** of our work.
>
> We respectfully disagree that the contributions are incremental. While our work use some existing theoretical tools, directly combining them is non-trivial and insufficient to achieve our results:
>
> (1) Combining techniques such as the empirical process [5] to address non-adaptive weights and settled reference functions leads to an additional dependence on $T$ and results in looser regret bounds.
>
> (2) Delayed policy updates are a common strategy for achieving low switching cost [4, 10], but they incur extra regret due to outdated policies. Balancing low switching cost with strong regret performance remains non-trivial and requires new techniques.
>
> (3) Round-based designs for reference functions, such as FedQ-Advantage [11], suffer from high burn-in costs due to polynomial dependence on $S, A$.
>
> Beyond synthesizing known ideas, our paper introduces at least three following key technical innovations:
>
> **Novel decomposition for worst-case regret:** The reference-advantage decomposition, commonly used to achieve optimal regret in online RL, faces key limitations in our round-based setting, particularly under communication constraints, due to its non-adaptiveness in both the weighting scheme and the settled reference functions. To address this issue, we propose a novel decomposition for worst-case regret (detailed in Appendix C and also our initial response), which leverages surrogate reference functions to construct a tighter upper bound on worst-case regret.
>
> While surrogate reference functions were originally introduced in [11], our work is the first to apply them in the worst-case RL setting and within the federated RL framework, using only their core definition and without relying on auxiliary techniques previously developed for gap-dependent analysis. This decomposition represents a new and general contribution to the online RL literature. Furthermore, our decomposition is general and can be used to strengthen the analysis in several existing works [3, 5, 12].
>
> **Two-sided variance bound for improved bonuses:** Bonus design plays a crucial role in reinforcement learning. For instance, UCB-Bernstein outperforms UCB-Hoeffding by leveraging its tighter bonus derived from the Bernstein inequality [13]. Similarly, [14] achieves the information-theoretic lower bound using the same algorithmic structure as [15], with a single but crucial change: replacing the original bonus with a Bernstein-type bonus that incorporates a more accurate variance estimator. Such improvements rely on nontrivial analytical advances.
>
> In our work, rather than directly combining existing techniques, we develop a novel two-sided variance bound that yields a tighter bonus. This refinement is key to establishing the optimism property (Lemma H.1) and leads to both improved theoretical regret bounds and enhanced empirical performance. Our variance bound is general and can potentially benefit bonus design in prior works such as [3, 5, 11, 12].
>
> **Lower confidence bound (LCB) with round-based design:** To our knowledge, we are the first in the literature to deploy the LCB technique in the online FRL setting. Unlike its traditional use in single-agent scenarios, we adapt the classical LCB framework to a round-based paradigm that emphasizes low switching and communication costs.
>
> Our method introduces a novel hybrid design that adjusts strategies between early and late rounds, effectively combining LCB-based exploration with round-based synchronization. This adaptive mechanism enables more efficient learning under communication constraints and may serve as a blueprint for using LCB in similar federated settings.

---

> > ### Author Response · Authors · 2025-08-07
> > **Follow-Up on Reviewer a3AW's Response to our Rebuttal (Part III)**
> >
> > In the third part of our response, we address your concern regarding the optimality of the communication cost and provide additional large-scale experiments to demonstrate the effectiveness of our algorithms.
> >
> > >### **Regarding Optimal Communication Cost**
> >
> > There is a trade-off between regret and communication cost, which can be observed in two cases. The first case involves agents not communicating with each other and updating their policies independently. This corresponds to the situation of $M$ parallel single-agent mechanisms. In this case, the regret does not benefit from the speedup gained through agent collaboration. The second case involves agents communicating at all times, which can reduce the regret but results in high communication costs that scale linearly with $T$. As a result, finding the optimal communication cost is challenging, and to date, no prior work in online FRL has even established a lower bound for it.
> >
> > Although there is no known lower bound, our communication cost result in this work remains the best in model-free online FRL. The previous state-of-the-art was FedQ-Advantage, and our experiments in the next section also demonstrate that our method outperforms FedQ-Advantage.
> >
> > >### **Experimental Design**
> >
> > Our experiments serve to validate the theoretical claims under controlled conditions. While the state/action space is small, the settings are aligned with related foundational work (e.g., [11, 12, 16]). Scalability remains an important future direction, but the current results already provide theoretical and empirical support for our claims.
> >
> > To further underscore the practical potential of our approach, we also provide simulation results in a large-scale environment, which, to the best of our knowledge, is larger than most of prior theoretical RL studies.
> >
> > We consider the experimental setting with parameters $(H, S, A) = (10, 100, 50)$ and $50$ agents. We conduct 10 independent trials with the same hyper-parameter used in our work and generate $10^6$ episodes per agent, resulting in a total of $5*10^7$ episodes. These experiments go significantly beyond the typical scale used in theoretical studies and demonstrate that our method remains effective and stable in high-dimensional, multi-agent environments, reinforcing both the theoretical insights and the potential for broader real-world deployment.
> >
> > For comparison, we evaluate our algorithm against the prior state-of-the-art method, FedQ-Advantage, in terms of both regret and communication cost. Following the evaluation protocol in our work, we plot the **adjusted regret** $\text{Regret}(n) / \log(n+1)$ against the episode number $n$ for the regret metric, and $\text{Number of rounds} / \log(n+1)$ versus $n$ for communication cost.
> >
> > The results show that in both large-scale settings, our algorithm outperforms FedQ-Advantage with respect to both regret and communication efficiency. To quantitatively present our findings, we select a subset of episodes and report the median adjusted regret $\text{Regret}(n) / \log(n+1)$ over the 10 trials. The detailed results (round to nearest integer) are shown in the table below.
> >
> > **Table 1: Median adjusted regret at episode number n.**
> > |Algorithm|50000|250000|500000|2500000|5000000|25000000|50000000|
> > |-|-|-|-|-|-|-|-|
> > |FedQ-Advantage|10730 | 36657 | 60110 | 156690 | 199935 | 190909 | 183774
> > |Ours |4123 | 10621 | 13982 | 14859 | 14609 | 13787 | 13397 |
> >
> > **Table 2: Median adjusted number of rounds at episode number n.**
> > |Algorithm|50000|250000|500000|2500000|5000000|25000000|50000000|
> > |-|-|-|-|-|-|-|-|
> > |FedQ-Advantage|   112  |  453  |  943  |  3894  |  6483  | 14803  | 15610  |
> > |Ours | 92 | 402|  762  | 3385  | 5532  | 6756  | 6712  |
> >
> > From the results, we observe that our method also significantly reduces both regret and communication cost compared to FedQ-Advantage in large-scale settings.

---

> > > ### Author Response · Authors · 2025-08-07
> > > **Follow-Up on Reviewer a3AW's Response to our Rebuttal (Part IV)**
> > >
> > > In the final part, we include the references cited in our follow-up response. We appreciate your time and consideration.
> > >
> > > [1] Kiran, B. Ravi, Ibrahim Sobh, Victor Talpaert, Patrick Mannion, Ahmad A. Al Sallab, Senthil Yogamani, and Patrick Pérez. "Deep reinforcement learning for autonomous driving: A survey." IEEE Transactions on Intelligent Transportation Systems 23, no. 6 (2021): 4909-4926.
> > >
> > > [2] Komorowski, Matthieu, Leo A. Celi, Omar Badawi, Anthony C. Gordon, and A. Aldo Faisal. "The artificial intelligence clinician learns optimal treatment strategies for sepsis in intensive care." Nature Medicine 24, no. 11 (2018): 1716-1720.
> > >
> > > [3] Zihan Zhang, Yuan Zhou and Xiangyang Ji. Almost optimal model-free reinforcement learning via reference-advantage decomposition. NeurIPS 2020.
> > >
> > > [4] Yu Bai, Tengyang Xie, Nan Jiang and Yu-Xiang Wang. Provably efficient Q-learning with low switching cost. NeurIPS 2019.
> > >
> > > [5] Gen Li, Laixi Shi, Yuxin Chen and Yuejie Chi. Breaking the sample complexity barrier to regret-optimal model-free reinforcement learning. Information and Inference, 2023.
> > >
> > > [6] Brisimi, Theodora S., Ruidi Chen, Theofanie Mela, Alex Olshevsky, Ioannis Ch Paschalidis, and Wei Shi. "Federated learning of predictive models from federated electronic health records." International Journal of Medical Informatics 112 (2018): 59-67.
> > >
> > > [7] Xu, Jie, Benjamin S. Glicksberg, Chang Su, Peter Walker, Jiang Bian, and Fei Wang. "Federated learning for healthcare informatics." Journal of Healthcare Informatics Research 5, no. 1 (2021): 1-19.
> > >
> > > [8] Suzumura, Toyotaro, Yi Zhou, Natahalie Baracaldo, Guangnan Ye, Keith Houck, Ryo Kawahara, Ali Anwar et al. "Towards federated graph learning for collaborative financial crimes detection." arXiv:1909.12946.
> > >
> > > [9] Zhou, Zhi, Xu Chen, En Li, Liekang Zeng, Ke Luo, and Junshan Zhang. "Edge intelligence: Paving the last mile of artificial intelligence with edge computing." Proceedings of the IEEE 107, no. 8 (2019): 1738-1762.
> > >
> > > [10] Dan Qiao, Ming Yin, Ming Min and Yu-Xiang Wang. Sample-efficient reinforcement learning with loglog(t) switching cost. ICML 2022.
> > >
> > > [11] Zhong Zheng, Haochen Zhang and Lingzhou Xue. Gap-dependent bounds for Q-learning using reference-advantage decomposition. ICLR 2025.
> > >
> > > [12] Zhong Zheng, Haochen Zhang and Lingzhou Xue. Federated Q-learning with reference advantage decomposition: Almost optimal regret and logarithmic communication cost. ICLR 2025.
> > >
> > > [13] Chi Jin, Zeyuan Allen-Zhu, Sebastien Bubeck, and Michael I Jordan. Is q-learning provably efficient? NeurIPS, 2018.
> > >
> > > [14] Gen Li, Laixi Shi, Yuxin Chen, Yuejie Chi, and Yuting Wei. Settling the sample complexity of model-based offline reinforcement learning. The Annals of Statistics 52, no. 1 (2024): 233-260.
> > >
> > > [15] Tengyang Xie, Nan Jiang, Huan Wang, Caiming Xiong, and Yu Bai. Policy finetuning: Bridging sample-efficient offline and online reinforcement learning. NeurIPS 2021.
> > >
> > > [16] Labbi, S., Tiapkin, D., Mancini, L., Mangold, P., and Moulines, E. (2024). Federated UCBVI: Communication-Efficient Federated Regret Minimization with Heterogeneous Agents. arXiv:2410.22908.

---

### Official Review · Reviewer_CTk9 · 2025-07-01

**Clarity:** 3
**Significance:** 3
**Originality:** 3
**Rating:** 4
**Confidence:** 3

**Summary:**

This paper proposes a novel variant of the Q learning algorithm, and its federated counterpart, achieving both low switching cost and lower burn-in cost.
These algorithm, Q-EarlySettled-LowCost and FedQ-EarlySettled-LowCost, combine an upper confidence and a lower-confidence approach, together with the construction of a reference estimator, to reduce the burn-in cost.
Both methods are studied in terms of worst case dependence and gap-dependent bounds, achieving low switching cost in all cases.
In particular, the federated method achieves regret that grows in $\sqrt{M}$, where $M$ is the number of agents.

**Questions:**

1. Is there hope to reduce the number of communications, so that it does not scale linearly with $M$ anymore? Could the ideas presented in [1] be applied to this setting?
2. In the experiments presented in appendix, it seems that the proposed algorithm outperforms all other flavor of Q learning. How does this compare with model-based methods? Overall, I think it would make sense to include these experiments in the main text.
3. The obtained burn-in cost remains quite large, although it depends only linearly on S, A. Is there hope for further improvements?

Labbi, S., Tiapkin, D., Mancini, L., Mangold, P., & Moulines, E. (2024). Federated UCBVI: Communication-Efficient Federated Regret Minimization with Heterogeneous Agents. arXiv preprint arXiv:2410.22908.

**Ethical Concerns:**

["NO or VERY MINOR ethics concerns only"]

**Final Justification:**

After the authors' response, I remain positive about this paper. I appreciate that the authors considered adding additional experiments with model-based methods.

I remain skeptical about their notion of heterogeneity on the "exploration speed", which, to my knowledge, is not defined neither in the authors' manuscript nor in [1], nor in the existing literature. In particular, the reviewers claim that the number of communications has to increase with the number of agents $M$, which is not clear and contradicts other existing work. This is not detrimental to acceptance, but I would appreciate if the authors could discuss this in more detail in the revised manuscript.

Regarding the non-homogeneous case, the authors claim that all existing results exhibit a regret that scales linearly in T when agents are not homogeneous, which is true. Nonetheless, these results prove that the method works in slightly heterogeneous settings, up to a small additional error: in the results presented in the current manuscript, no such analysis is proposed, and this is an important limitation in the paper, since heterogeneity is one of the key factors of federated learning.

Despite these remarks, which I would appreciate if the authors can comment on

**Limitations:**

The discussion on the cost of communication is missing, although it can quickly become a problem in large scale federated RL cases.
Similarly, the relevance of the homogeneity assumption should be discussed.

**Paper Formatting Concerns:**

No issues.

**Quality:**

3

**Strengths And Weaknesses:**

**Strengths**
1. The proposed Q-learning methods are the first methods to achieve both low switching cost and low burn-in period. Reducing the burn-in period is crucial to achieve meaningful guarantees in RL, while low switching cost is essential in federated RL, in order to keep clients synchronized on the same policy without requiring a prohibive number of communications. Providing a method that achieves both is a significant open problem.
2. The federated algorithm indeed benefits from collaboration with other agents, with a summed regred scaling in $\sqrt{M}$ (i.e., averaged regret scaling in $\sqrt{1/M}$).
3. The proposed construction is a clever combination of upper and lower confidene bounds, allowing for early settlement of the reference function, which in turn significantly reduces the burn-in period.

**Weaknesses**
1. The proposed method unfortunately requires important number of communications, and grows in $M \log(T)$ for worst-case bounds and $M \log(\log(T))$ for gap-dependent bounds. This may quickly become prohibitive in large-scale federated RL scenarios.
2. The FL scenario that is considered is the fully heterogeneous scenario, which is not the most common in federated learning. Although I think the paper already presents interesting results, it would be interesting to discuss possible extensions to the non-homogeneous case.

The following reference, which studies Fed-UCBVI is missing from the discussion of related work, and may provide insights for possible extensions reducing the dependence on M of the communication :

Labbi, S., Tiapkin, D., Mancini, L., Mangold, P., & Moulines, E. (2024). Federated UCBVI: Communication-Efficient Federated Regret Minimization with Heterogeneous Agents. arXiv preprint arXiv:2410.22908.

---

> ### Author Rebuttal · Authors · 2025-07-31
>
> We thank you for your careful reading and thoughtful comments. Below are our point-by-point responses, and we hope our responses address your concerns.
>
> >### **Weakness 1 part 1 and Question 1: The linear dependency on $M$ in the number of communications**
>
> Thank you for raising this important point. We also appreciate your mention of the reference [1, Labbi], which we will cite in the revised manuscript.
>
> [1] proposes a model-based online FRL algorithm, Fed-UCBVI, and analyzes its communication cost. For comparison: in worst-case setting, our work provides the bound $O(M\log T)$, and both [1] and our gap-dependent analysis provide the bound $O(M\log\log T)$ under a homogeneous exploration speed assumption. Notably, we can refine our worst-case bound to match this improved communication rate under the same assumption. However, **no existing method eliminates the linear dependence on $M$ in general**, and the techniques in [1], which share similarities with our gap-dependent analysis, do not overcome this either. We elaborate below.
>
> The linear $M-$dependence in our method arises from the event-triggered termination condition (Eq. 3), where the threshold $c_h^k$ is scaled by $1/M$ to ensure that the total number of visits to a state-action-step triple across all agents is controlled in each round. In our worst-case analysis, we do not assume that the agents explore the environment at the same speed, so that each agent can generate different numbers of episodes in a given round. As such, the uniform threshold $c_h^k$ ensures robustness to heterogeneous agent exploration speeds, but results in a communication bound of $O(M\log T)$.
>
> In contrast, under the **homogeneous exploration speed assumption** used in both [1] and our gap-dependent analysis, all agents generate the same number of episodes per round. Since they share a common policy, the visits are i.i.d., and the law of large numbers implies that once one agent meets the threshold for a triple, the others are likely close to meeting it as well. In Lemma L.4, we have shown that when the visit count exceeds $i_1=\tilde{\Theta}(M)$, this concentration effect enables a global synchronization that helps remove the $M-$dependency of the number of communication rounds in the late stage with high probability. This reduces the overall $M-$dependency to $O(M\log\log T)$, which is caused by the early stage and remains unavoidable even under this assumption.
>
> In summary, although our method inherits a linear $M$-dependence in communication cost, this is a fundamental limitation shared by most existing FRL algorithms aiming for optimal regret guarantees. We can refine the bound in our worst-case analysis if we assume a homogeneous exploration speed as in [1] and our gap-dependent analysis. However, such an idea cannot remove the linear dependency on $M$ due to the early stage. We also remark that the bound $O(M\log T)$ holds deterministically, but the bound $O(M\log \log T)$ holds with high probability, due to the concentration arguments.
>
> >### **Weakness 1 part 2: The large-scale FRL scenarios**
>
> While removing the linear $M$-dependence remains an open problem in general FRL, we note that, in large-scale FRL scenarios (i.e., when $M$ is large), it is possible to remove the $M-$dependency **by allowing waiting time**, similar to the strategy used in FedQ-Advantage [109].
>
> Specifically, we can modify our algorithms so that each local agent terminates exploration in a round only when it independently triggers the synchronization condition, without relying on signals from other agents. In this situation, each agent, rather than just one in our current algorithm, ensures it collects a sufficient number of new visits in a round for at least one triple before proceeding. This modified design removes the need to coordinate exploration speeds across agents.
>
> We formalize this in the following result that will be added to the revised manuscript:
>
> **A New Theorem.** For the modified FedQ-EarlySettled-LowCost (Algorithms 1 and 2 with $\beta\in(0,H]$ and the modified synchronization condition), define $\tilde{C} = H^2(H+1)SA$. Then, the number of rounds $K$ is bounded by $4\tilde{C}+ 8\tilde{C} \max\\{\log(T/\tilde{C}),1\\}.$
>
> This new bound eliminates the linear dependence on $M$. This improvement comes at the cost of waiting time, as agents who complete exploration early must wait until slower agents also reach the synchronization condition. For more general scenarios, especially under heterogeneous agent behaviors, further research is required into new synchronization strategies and regret-communication trade-offs.
>
> >### **Weakness 2: The non-homogeneous case**
>
> We appreciate the reviewer’s suggestions for possible extensions to the non-homogeneous case. The literature on online FRL discusses two key forms of heterogeneity: (a) Heterogeneous exploration speed, where agents generate different numbers of episodes per round, as discussed in [108, 109] and this paper, and (b) Heterogeneous reward functions and transition kernels across agents, as discussed in Fed-UCBVI. While our current paper addresses (a), we agree that (b) is an important and challenging direction, and we elaborate on it below.
>
> Fed-UCBVI provides theoretical guarantees under heterogeneity (b) by leveraging centralized value functions and Bellman equations. In principle, a similar analysis could be extended to model-free FRL algorithms, including ours. However, the resulting regret bounds for both Fed-UCBVI and hypothetical model-free extensions inevitably lead to the **linear dependence on $T$**, highlighting a key limitation: none of these methods fully mitigates the bias introduced by heterogeneity (b).
>
> The main technical challenge arises from **heterogeneous visiting probabilities** for any given $(s,a,h)$ across agents. Even when agents share the same policy, heterogeneous environments result in different numbers of visits across agents. This causes an imbalance in sample contributions across agents, making the **standard equal-weight aggregation** used for the Q-update in all existing online FRL algorithms inherently biased.
>
> To overcome this issue, a promising direction may involve **weighted aggregation schemes** based on agent-specific visitation counts, as explored in some offline FRL papers [88, 89]. Adapting such techniques to the online setting could help address the bias introduced by heterogeneity, though doing so would require a careful redesign of both the algorithm and its theoretical analysis. We view this as an important direction for future work.
>
>
> >### **Question 2: Comparison with model-based method**
>
> Thank you for your suggestion and we will include these experiments to the main text.
>
> To address the comparison with model-based methods, we include an evaluation against Fed-UCBVI, the only known online model-based FRL algorithm with near-optimal regret guarantees. We conduct experiments on the same two environments described in Appendix B, with $(H,S,A) = (5,3,2)$ and $(7,10,5)$, using $M=10$ agents. For each method, we report both regret and communication round performance over 10 independent runs, using the same hyperparameter settings for Fed-UCBVI as specified in [1]. The results show that our algorithm achieves consistently lower regret than Fed-UCBVI, while Fed-UCBVI demonstrates lower communication rounds due to its use of transition kernel estimations. In terms of memory, Fed-UCBVI requires $\Theta(S^2)$, while our method requires only $\Theta(S)$, offering a substantial advantage in scalability. We summarize the regret results (rounded to the nearest integer) in two tables below.
>
> **Table 1: Median regret at different episode number for $(H,S,A)=(5,3,2)$.**
>
> |Algorithm|10000|50000|100000|150000|200000|250000|300000|
> |-|-|-|-|-|-|-|-|
> |Fed-UCBVI|917|965|971|975|977|978|978|
> |Ours|444|623|676|699|716|730|735|
>
> **Table 2: Median regret at different episode number for $(H,S,A)=(7,10,5)$.**
>
> |Algorithm|50000|100000|500000|1000000|1500000|2000000|
> |-|-|-|-|-|-|-|
> |Fed-UCBVI|18398|19092|23045|28767|32038|35477|
> |Ours|9405|11961|18777|21920|23749|25046|
>
> >### **Question 3: The burn-in cost**
>
> Thank you for raising this important point. Theorem 4.3 with $\beta=\Theta(1)$ provides the worst-case regret guarantee of FedQ-EarlySettled-LowCost as $\tilde{O}(\sqrt{MH^2SAT}+H^6SA+MH^5SA)$, where the additional terms $\tilde{O}(H^6SA + MH^5SA)$ lead to the burn-in cost of $\tilde{O}(MSAH^{10})$. Next, we discuss below the nature of the dependencies on $M,S,A,H$.
>
> The **linear dependence on $M$** is unavoidable in online FRL because each local agent must explore the environment using suboptimal policies. Such dependency also appears in [1, 108, 109].
>
> The **linear dependence on $SA$** is also unavoidable in online RL because the agent must explore the entire environment using suboptimal policies. Our method enjoys a lower burn-in cost than FedQ-Advantage and Fed-UCBVI, both of which show super-linear dependence on $SA$. In fact, this linear dependence is supported by the information lower bound in [102], which shows that even single-agent RL incurs a burn-in cost of $\Theta(H^2SA)$.
>
> The **super-linear dependence on $H$** is better than that of FedQ-Advantage, while model-based methods such as Fed-UCBVI exhibit better dependence on $H$ by estimating the transition kernels, but at the cost of higher memory requirements $\Theta(S^2)$ vs. our $\Theta(S)$. Notably, there is room for improving our $H$-dependence by possibly enhancing the Hoeffding-type Q-update (Eq.9 and 13) using more advanced techniques, such as Q-learning with momentum ([2]), which may accelerate reference settling and improve convergence. We view it as a promising direction for future work.
>
> [2] Pierre Ménard, Omar Darwiche Domingues, Xuedong Shang, and Michal Valko. "Ucb momentum q-learning: Correcting the bias without forgetting." ICLR 2021.
>
> (All reference numbers in this response, except for [1] and [2], are from our paper.)

---

> ### Comment · Reviewer_CTk9 · 2025-08-03
>
> I thank the authors for taking the time to answer my concerns. After the authors' response, I remain positive about this paper.
>
> I appreciate that the authors considered adding additional experiments with model-based methods.
>
> The term "exploration speed" remains quite confusing, and is not defined neither in the authors' manuscript nor in [1]; I would appreciate if the author can give more details on what they mean by this, especially since I don't believe [1] makes any assumption on such type of "exploration speed".
> In particular, [1] uses a local estimate of the global exploration to decide when communication should happen: based on this local counter and on an adequate "doubling trick", number of communication can be reduced by a factor $M$. Isn't the same idea possible to apply here?
>
> Regarding the non-homogeneous case, I agree that all existing results exhibit a regret that scales linearly in T when agents are not homogeneous. Nonetheless, these results *prove* that the method works in slightly heterogeneous settings, up to a small additional error: in the results presented in the current manuscript, no such analysis is proposed, and this is an important limitation in the paper, since heterogeneity is one of the key factors of federated learning.

---

> ### Author Response · Authors · 2025-08-04
> **Follow-Up on Reviewer CTk9's Response to our Rebuttal (Part I)**
>
> Thank you for the thoughtful follow-up. We provide the first part of our responses to each point below.
>
> >### **Clarifying “Exploration Speed"**
>
> In our initial response, exploration speed refers to the number of episodes generated by each agent per round, which may vary across agents.
>
> In our **worst-case analysis**, we denote the number of episodes generated by agent $m$ in round $k$ as $n^{m,k}$, and $n^{m,k}$ can vary across agents. Our algorithm proceeds in rounds, and each round terminates when the synchronization condition (Equation (3) in our work) is met for one agent. This setup eliminates waiting time for each agent and imposes no constraints on the number of episodes generated per agent, thereby allowing heterogeneous exploration speeds across agents.
>
> Accordingly, our regret is defined as:
>
> $$
> \text{Regret}(T) = \sum_{m=1}^{M} \sum_{k=1}^{K} \sum_{t=1}^{n^{m,k}} \left(V_1^{\star}(s_1^{m,k,t}) - V_1^{\pi_{k,t}}(s_1^{m,k,t})\right),
> $$
>
> where $s_1^{m,k,t}$ denotes the initial state of agent $m$ in $t$-th episode of round $k$, and $\pi_{k,t}$ is the policy executed in that episode. This formulation naturally accommodates heterogeneous exploration speeds.
>
> In contrast, [1] assumes that all agents generate the same number of episodes in each round—an assumption that, although not explicitly stated, is reflected in both the algorithm design and the regret formulation. [1] uses a global episode index $t \in [T]$ in the Algorithm 1, which implicitly assumes that, without any waiting time, all agents are synchronized at the same episode and generate the same number of episodes in each round. The assumption is also implied by its regret definition (see Eq. 4 of [1]):
>
> $$\text{Regret}(T) = \sum_{m=1}^{M} \sum_{t=1}^{T} \left(V_1^{\star}(s_1^{m,t}) - V_1^{\pi_t}(s_1^{m,t})\right),$$
>
> with homogeneous transition kernels and reward functions, where $s_1^{m,t}$ is the initial state of agent $m$ in episode $t$, and $\pi_t$ is the policy executed in episode $t$. This expression further implies that each agent generates $T$ episodes during the learning process, whereas our work does not make such an assumption and is closer to the spirit of first two online FRL works [2,3].
>
> We will revise the manuscript to distinguish between homogeneous and heterogeneous exploration speed settings more clearly, relate it to existing literature: [2, 3] and our work allow heterogeneous exploration speeds, while [1, 4] focus on the homogeneous exploration speed setting to improve communication efficiency.
>
> Moreover, in our work, the homogeneous exploration speed assumption is only used in the **gap-dependent analysis** and implies that all agents generate the same number of episodes per round. We formally refer to this as the \textbf{full synchronization} assumption, which is described in detail in Appendix L (line 1335) of the original submission. Specifically, under this assumption, the number of episodes generated by agent $m$ in round $k$ satisfies $n^{m,k} = n^k$, meaning the episode count is independent of the agent index $m$. Both [1] and our gap-dependent analysis leverage this assumption to eliminate the $O(M \log T)$ communication overhead, reducing it to an $O(M \log\log T)$ term and matching the optimal rate.

---

> ### Author Response · Authors · 2025-08-04
> **Follow-Up on Reviewer CTk9's Response to our Rebuttal (Part II)**
>
> Thank you again for the thoughtful follow-up. Below, we provide the second part of our response.
>
> >### **On the "Doubling Trick"**
>
> For definitions that are shared between [1] and our work, we adopt our notation instead of that used in [1], in order to maintain consistency with the submission. Let $n_h^{m,k}(s,a)$ denote the number of visits to the state-action-step triple $(s, a, h)$ by agent $m$ in round $k$, and define the global visit count to $(s, a, h)$ prior to round $k$ as
> $$N_h^k(s,a) = \sum_{r=1}^{k-1} \sum_{m=1}^{M} n_h^{m,r}(s,a).$$
>
> In [1], for agent $m$, a local estimate $\hat{N}\_{(k,t),h}^m(s,a)$ of the global visit count is maintained during the $t$-th episode of round $k$. A synchronization signal is sent when this local estimate exceeds twice the global count prior to round $k$, i.e., $$\hat{N}\_{(k,t),h}^m(s,a) > 2 N_h^k(s,a).\quad(1)$$
> According to the design in Algorithm 1 of [1], when agent $m$ sends the synchronization signal in the $t$-th episode of round $k$, it holds that
> $$\hat{N}\_{(k,t),h}^m(s,a)=N_h^k(s,a) + Mn_h^{m,k}(s,a).$$
> Substituting this into the synchronization condition (1) yields:
> $$N_h^k(s,a)+Mn_h^{m,k}(s,a)>2N_h^k(s,a),$$
> which is equivalent to
> $$n_h^{m,k}(s,a)>\frac{1}{M}N_h^k(s,a).$$
> This means in [1], agent $m$ send the synchronization signal if
> $$n_h^{m,k}(s,a)>\frac{1}{M}N_h^k(s,a).$$
> Our method adopts a similar idea but uses a more conservative threshold. Specifically, in round $k$, agent $m$ sends a synchronization signal if
> $$n_h^{m,k}(s,a)\geq\frac{1}{MH(H+1)}N_h^k(s,a).\quad(2)$$
> While the **doubling trick** is commonly used in model-based settings like [1,6], model-free approaches, including ours and [2,3,7], must use a more conservative rate of $\frac{1}{MH(H+1)}$ due to the lack of transition kernel estimates under the model-free framework.
>
> Furthermore, under the homogeneous exploration speed assumption (used in both [1] and our **gap-dependent analysis**), our **worst-case analysis** can also eliminate the $M \log T$ dependency. Since all agents share a common policy in each round, their visits to each state-action-step triple are i.i.d.. By the law of large numbers, once one agent meets the synchronization threshold for a given triple, the others are likely to be close as well. This phenomenon is theoretically established in Lemma L.4 (line 1396), where we show that for any triple satisfying the synchronization condition (Equation (2)) in round $k$, once the visit count prior to round $k$ exceeds $i_1 \tilde{\Theta}(M)$, the total number of visits will subsequently grow at a rate of $\Omega\left(\frac{1}{H(H+1)}\right)$—a rate that is independent of $M$. As a result, the overall $M$-dependency is reduced to $O(M\log\log T)$, which arises from the early stage when the visit count is still below $i_1$ and remains unavoidable even under this assumption. We will include these discussions in the revised manuscript.
>
> >### **On the Non-Homogeneous Case**
>
> Our work focuses on addressing **system heterogeneity**, a core challenge in federated learning highlighted in [5]. As discussed in [5], this type of heterogeneity arises from variations in system characteristics across the network. Devices can differ widely in terms of hardware capabilities, network connectivity, and energy constraints, which often leads to issues such as stragglers and partial participation. Our algorithm is designed to handle such heterogeneity without incurring any additional error that grows linearly with $T$, ensuring robustness in real-world federated environments.
>
> In contrast, [1] focuses primarily on **statistical heterogeneity**, which refers to highly non-identically distributed data across agents (as also noted in [5]). We agree this is an important challenge. While our current analysis does not cover this form of heterogeneity, extending our framework to such settings is a key direction for future work. We will add this discussion to the revised manuscript.
>
> [1] Labbi, S., Tiapkin, D., Mancini, L., Mangold, P., and Moulines, E. (2024). Federated UCBVI: Communication-Efficient Federated Regret Minimization with Heterogeneous Agents. arXiv preprint arXiv:2410.22908.
>
> [2] Zheng, Z., Gao, F., Xue, L., and Yang, J. Federated Q-learning: Linear regret speedup with low communication cost. ICLR, 2024.
>
> [3] Zheng, Z., Zhang, H., and Xue, L. Federated Q-learning with reference advantage decomposition: Almost optimal regret and logarithmic communication cost. ICLR, 2025.
>
> [4] Zhang, H., Zheng, Z., and Xue, L. Gap-Dependent Bounds for Federated Q-Learning. ICML, 2025
>
> [5] Li, T., Sahu, A. K., Talwalkar, A., and Smith, V. Federated learning: Challenges, methods, and future directions. IEEE Signal Processing Magazine 37, no. 3 (2020): 50-60.
>
> [6] Zhang, H., Chen, Y., Lee, J. D., and Du, S. S. Settling the sample complexity of online reinforcement learning. COLT, 2024.
>
> [7] Bai, Y., Xie, T., Jiang, N., and Wang, Y.-X. Provably efficient q-learning with low switching cost. NeurIPS, 2019.

---

> > ### Comment · Reviewer_CTk9 · 2025-08-06
> >
> > Thank you for clarifying the notion of "exploration speed", which appears to refer to agents collecting different numbers of episodes at each global time step. I believe including this discussion more explicitly in the manuscript would help readers better understand the setting under consideration.
> >
> > However, I am not fully convinced that this constitutes a particularly significant type of heterogeneity. It seems one could scale all estimates according to the number of local episodes collected by each agent, and adapt the results accordingly based on these scaled estimates.
> >
> > Regarding statistical heterogeneity, I understand that the setting considered in the paper is fully homogeneous in this respect. This is an important limitation, as such homogeneity assumptions are now quite uncommon in federated learning.
> >
> > I also agree with other reviewers' concerns that the contribution may be somewhat incremental, in that it essentially combines analyses from prior work, although the technical contribution is still significant. Overall, I remain positive about the paper.

---

> ### Author Response · Authors · 2025-08-07
> **Response to Second-Round Comment of Reviewer CTk9 (Part I)**
>
> Thank you for your response and for sharing your perspective. We will revise the manuscript to include a more explicit discussion to help readers better understand the setting related to exploration speed.
>
> We would also like to take this opportunity to clarify several key points regarding your concerns. Our response is structured into three parts.
>
> >### **Regarding the rescaling approach**
>
> Our work indeed adopts a similar idea, where the central server aggregates local means to update the $Q$-function. However, as noted in our initial response to Weakness 1, without the assumption of homogeneous exploration speed, applying such rescaling leads to a communication cost with an $O(M\log T)$ dependence. This highlights a trade-off between exploration heterogeneity and communication efficiency.
>
> >### **Regarding the heterogeneity**
>
> Our main focus is on addressing two central challenges in online federated RL: achieving low policy switching cost and low burn-in cost. These challenges are critical for improving learning efficiency and scalability and represent an important and underexplored gap in current research.
>
> We fully agree that statistical heterogeneity is another meaningful and realistic challenge in federated learning. Extending our framework to handle both statistical and system-level heterogeneity is a promising direction for future work, and we appreciate this valuable suggestion.

---

> > ### Author Response · Authors · 2025-08-07
> > **Response to Second-Round Comment of Reviewer CTk9 (Part II)**
> >
> > Thank you once again for your response and for sharing your insights. In the second part of our response, we would like to further clarify the contributions and novelties of our work.
> >
> > **In our work, we demonstrate that our algorithms achieve the best regret and switching cost performance in the single-agent model-free RL setting, as well as the best regret and communication cost performance in the model-free FRL setting, from both theoretical and empirical perspectives.** While our work uses some existing theoretical tools, directly combining them is non-trivial and insufficient to achieve our results:
> >
> > (1) Combining techniques such as the empirical process [1] to address non-adaptive weights and settled reference functions leads to an additional dependence on $T$ and results in looser regret bounds.
> >
> > (2) Delayed policy updates are a common strategy for achieving low switching cost [2,3,4], but they incur extra regret due to outdated policies. Balancing low switching cost with strong regret performance remains non-trivial and requires new techniques.
> >
> > (3) Round-based designs for reference functions, such as FedQ-Advantage [5], suffer from high burn-in costs due to polynomial dependence on $S, A$.
> >
> > Beyond synthesizing known ideas, our paper introduces at least three key innovations:
> >
> > **Novel decomposition for worst-case regret**
> >
> > The reference-advantage decomposition, commonly used to achieve optimal regret in online RL, faces key limitations in our round-based setting, particularly under communication constraints, due to its non-adaptiveness in both the weighting scheme and the settled reference functions. To address this issue, we propose a novel decomposition for worst-case regret (detailed in Appendix C and also our initial response), which leverages surrogate reference functions to construct a tighter upper bound on worst-case regret.
> >
> > While surrogate reference functions were originally introduced in [6], our work is the first to apply them in the worst-case RL setting and within the federated RL framework, using only their core definition and without relying on auxiliary techniques previously developed for gap-dependent analysis. This decomposition represents a new and general contribution to the online RL literature. Furthermore, our decomposition is general and can be used to strengthen the analysis in several existing works [1,4,5].
> >
> > **Two-sided variance bound for improved bonuses**
> >
> > Bonus design plays a crucial role in reinforcement learning. For instance, UCB-Bernstein outperforms UCB-Hoeffding by leveraging its tighter bonus derived from the Bernstein inequality [7]. Similarly, [8] achieves the information-theoretic lower bound using the same algorithmic structure as [9], with a single but crucial change: replacing the original bonus with a Bernstein-type bonus that incorporates a more accurate variance estimator. Such improvements rely on nontrivial analytical advances.
> >
> > In our work, rather than directly combining existing techniques, we develop a novel two-sided variance bound that yields a tighter bonus. This refinement is key to establishing the optimism property (Lemma H.1) and leads to both improved theoretical regret bounds and enhanced empirical performance. Our variance bound is general and can potentially benefit bonus design in prior works such as [1,4,5,6].
> >
> > **Lower confidence bound (LCB) with round-based design**
> >
> > To our knowledge, we are the first in the literature to deploy the LCB technique in the online FRL setting. Unlike its traditional use in single-agent scenarios, we adapt the classical LCB framework to a round-based paradigm that emphasizes low switching and communication costs.
> >
> > Our method introduces a novel hybrid design that adjusts strategies between early and late rounds, effectively combining LCB-based exploration with round-based synchronization. This adaptive mechanism enables more efficient learning under communication constraints and may serve as a blueprint for using LCB in similar federated settings.

---

> > > ### Author Response · Authors · 2025-08-07
> > > **Response to Second-Round Comment of Reviewer CTk9 (Part III)**
> > >
> > > In the final part, we include the references cited in our second-round response. We appreciate your time and consideration.
> > >
> > > [1] Gen Li, Laixi Shi, Yuxin Chen and Yuejie Chi. Breaking the sample complexity barrier to regret-optimal model-free reinforcement learning. Information and Inference, 2023.
> > >
> > > [2] Yu Bai, Tengyang Xie, Nan Jiang and Yu-Xiang Wang. Provably efficient Q-learning with low switching cost. NeurIPS 2019.
> > >
> > > [3] Dan Qiao, Ming Yin, Ming Min and Yu-Xiang Wang. Sample-efficient reinforcement learning with loglog(t) switching cost. ICML 2022.
> > >
> > > [4] Zihan Zhang, Yuan Zhou and Xiangyang Ji. Almost optimal model-free reinforcement learning via reference-advantage decomposition. NeurIPS 2020.
> > >
> > > [5] Zhong Zheng, Haochen Zhang and Lingzhou Xue. Federated Q-learning with reference advantage decomposition: Almost optimal regret and logarithmic communication cost. ICLR 2025.
> > >
> > > [6] Zhong Zheng, Haochen Zhang and Lingzhou Xue. Gap-dependent bounds for Q-learning using reference-advantage decomposition. ICLR 2025.
> > >
> > > [7] Chi Jin, Zeyuan Allen-Zhu, Sebastien Bubeck, and Michael I Jordan. Is q-learning provably efficient? NeurIPS, 2018.
> > >
> > > [8] Gen Li, Laixi Shi, Yuxin Chen, Yuejie Chi, and Yuting Wei. Settling the sample complexity of model-based offline reinforcement learning. The Annals of Statistics 52, no. 1 (2024): 233-260.
> > >
> > > [9] Tengyang Xie, Nan Jiang, Huan Wang, Caiming Xiong, and Yu Bai. Policy finetuning: Bridging sample-efficient offline and online reinforcement learning. NeurIPS 2021.

---

### Note · Authors · 2025-08-11

We thank all reviewers and Area Chair for your time and efforts. Below, we first highlight the key contributions and then summarize our responses to each reviewer.

>### **Summary of Contributions**

Our algorithms achieve state-of-the-art regret and switching cost in model-free online RL, and the best regret and communication cost in model-free online FRL, supported by both theoretical guarantees and empirical evidence. To achieve this, we introduce several key innovations:

**Novel decomposition for worst-case regret:** We propose a new decomposition using surrogate reference functions to derive tighter worst-case regret bounds in online FRL. Unlike prior works, we avoid gap-dependent tools and provide a more adaptive and general framework for worst-case RL analysis.

**Two-sided variance bound for improved bonuses:** We develop a novel two-sided variance bound that yields tighter bonuses, improving both theoretical regret and empirical performance, and can be applied to enhance bonus design in other RL algorithms.

**Lower confidence bound (LCB) with round-based design:** We present the first LCB-based approach in online FRL, adapted to a round-based paradigm that reduces switching and communication costs. Our hybrid design balances early exploration with communication efficiency, offering a new framework for efficient learning under communication constraints.

>### **Reviewer-Specific Responses**

**Reviewer CTk9:** We discussed how communication and burn-in costs depend on MDP parameters and agent count, compared our results with model-based FRL methods, and elaborated on different types of heterogeneity addressed by our and prior works.

**Reviewer a3AW:** We expanded the practical relevance of our research problem, provided detailed technical innovations, and added large-scale experiments demonstrating practical advantages over the prior best method.

**Reviewer Rdno:** We clarified detailed comparison with prior algorithms, highlighted key innovations behind improved burn-in cost, and conducted additional experiments on switching and communication cost versus the existing best approaches. We also presented numerical burn-in cost results to support our learning efficiency claims.

**Reviewer wTVm:** We incorporated suggested refinements and discussed potential extensions to continuous state and action space.

We believe our responses address the reviewers’ concerns and better convey the novelty, practical value, and theoretical depth of our work.

---

### Decision · Program_Chairs · 2025-09-17

**Decision:**

Accept (poster)

**Comment:**

Online RL (and federated RL) algorithms are designed with near-optimal regret, low (linear in S, A) burn-in costs and logarithmic policy switching/communication cost. The main technical challenge lies in achieving the latter two simultaneously.
The result is a lengthy analysis that requires combining and adapting several advanced (existing) techniques with great care. The reviewers agree that the paper is sound and clear. The crucial question is whether it is incremental in nature.
The discussion was not enough to dismiss the impression that the problem the authors set themselves to solve is rather artificial. Experiments were appreciated (with some doubts regarding scalability). However, neither these, nor the appeals to motivating practical settings, nor the reiteration of the fine technical solutions needed to achieve all of these theoretical guarantees simultaneously, seemed to budge the reviewers from their initial impression of incrementality.
However, given the intrinsic value of the theoretical analysis I confirm the positive evaluation maintained by most reviewers.